# The importance of diabatic processes for the dynamics of synoptic-scale extratropical weather systems—a review

Heini Wernli[1] and Suzanne L. Gray[2]

[1]Institute for Atmospheric and Climate Science, ETH Zurich, Zurich, Switzerland
[2]Department of Meteorology, University of Reading, Reading, UK

**Correspondence:** Heini Wernli (heini.wernli@env.ethz.ch)

**Abstract.** Many fundamental concepts of synoptic-scale extratropical dynamics are based on the quasi-geostrophic equations of a dry atmosphere. This "dry dynamics" provides the essential understanding of, e.g., the formation of extratropical cyclones and the propagation of Rossby waves, and makes potential vorticity (PV) a materially conserved quantity. Classically, for extratropical weather systems, the importance of so-called "diabatic effects", e.g. surface fluxes, phase changes of water in clouds, and radiation, has been regarded as secondary compared to the dry dynamical processes. As outlined in this review article, research during the last decades has modified this view of the role of diabatic processes. The combination of complementary research approaches revealed that the non-linear dynamics of extratropical cyclones and upper-tropospheric Rossby waves is—in some cases strongly—affected by diabatic processes. Despite the violation of material PV conservation in the presence of diabatic processes, the concept of PV has been of utmost importance to identify and quantify the role of diabatic processes, and to integrate their effects into the classical understanding based on dry dynamics. This review first summarizes the theoretical concepts of diabatic PV modification and moist PV and slantwise moist convection, and provides a concise overview on early research on diabatic effects until the late 1970s. Two poorly predicted high-impact cyclones affecting eastern North America then triggered an impressive diversity of efforts to investigate the role of diabatic processes for rapid cyclone intensification in the last two decades of the 20th century. These research activities, including the development of sophisticated diagnostics, growing applications of the Lagrangian perspective, real case and idealised numerical experiments, and dedicated field experiments, are reviewed in detail. This historical perspective provides insight about how societal relevance, international collaboration, technical development, and creative science contributed to establishing this important theme of dynamical meteorology. The second part of the review then more selectively outlines important achievements in the last two decades of how diabatic effects, in particular those related to cloud microphysics, affect the structure, dynamics, and predictability of different types of extratropical cyclones and their mesoscale substructures, upper-tropospheric blocks, Rossby waves and their interactions. A novel aspect is the relevance of research on diabatic processes for climate change research. The review closes by highlighting important implications of investigating diabatic processes in extratropical weather systems for the broader field of weather and climate dynamics, its fundamentals and representation in numerical models.

# Contents

# 1 Introduction

The fundamental theories of Rossby wave propagation and of the formation of extratropical cyclones and anticyclones via baroclinic instability are based on the dynamics of a dry atmosphere (e.g., Rossby and Collaborators, 1937; Eady, 1949; Holton and Hakim, 2013). Rossby waves can be understood by considering the quasi-geostrophic[1] vorticity equation in an atmosphere with a background gradient of planetary vorticity. This typically meridional vorticity gradient is often strongly enhanced within a relatively narrow band (Davies and Rossa, 1998) and has been referred to as the midlatitude waveguide (Massacand and Davies, 2001; Schwierz et al., 2004). The essence of baroclinic instability is that in a baroclinic atmosphere, i.e., an atmosphere with a meridional temperature gradient, vertically deep synoptic-scale disturbances with a wavelength of about 4000 km can grow exponentially at the expense of reducing the atmosphere's available potential energy. In this framework of "dry dynamics", potential temperature ($\theta$) and potential vorticity (PV, denoted as $Q$) are both materially conserved quantities (Ertel, 1942; Kleinschmidt, 1950a; Hoskins et al., 1985), i.e., $\theta$ and $Q$ of an air parcel do not change along the flow, where PV is defined in the usual way as

$$Q = \rho^{-1}\, \boldsymbol{\omega} \cdot \nabla\theta, \tag{1}$$

where $\rho$ denotes air density and $\boldsymbol{\omega}$ the absolute vorticity vector. Even for the highly idealised limit of an atmosphere with uniform PV, this framework provided elegant theories for the formation of fronts (Hoskins, 1982), frontal instability (Schär and Davies, 1990), and the variability of cyclone life cycles (Hoskins and West, 1979; Davies et al., 1991). However, as discussed in detail in this article, observations, (re-)analyses, and model simulations clearly show that many important extratropical weather systems are characterised by strongly non-uniform PV structures in the lower and upper troposphere, that these structures are important for the weather systems' dynamics, and that they result from the interaction of dry dynamics with so-called diabatic processes, in particular those related to clouds.

The response to the question of how clouds affect surface weather conditions has two obvious components: clouds can produce surface precipitation and clouds can reduce surface solar radiation. The first effect is at the heart of numerical weather prediction (NWP, e.g., Fritsch and Carbone, 2004), and the second one is essential for understanding the Earth's global energy budget (e.g., Wild et al., 2013), which is also directly relevant on longer climate timescales. In addition, there is a third effect, which is comparatively indirect: phase transitions associated with the formation and dissolution of cloud particles in and below clouds release or consume latent heat. As discussed in detail in this review article, this latent heating (and cooling) can influence the atmospheric flow on scales up to several 1000 km. An earlier review by Stewart et al. (1998) on midlatitude cyclones, clouds, and climate emphasised the role of cyclones for producing complex cloud systems with associated radiation perturbations. In this review article, we focus on the third effect and emphasise the relevance of two theoretical concepts to investigate how diabatic processes influence the structure, evolution, and predictability of extratropical weather systems. These concepts, introduced in Sect. 2, are (i) PV and its diabatic modification and (ii) slantwise moist convection.

---

[1]The concept of quasi-geostrophic flow is a cornerstone for studying synoptic and larger-scale atmospheric dynamics in the extratropics (Davies and Wernli, 2015). It essentially consists of a prognostic equation for the evolution of the geostrophic flow, and of a diagnostic equation for the ageostrophic flow forced by the primary geostrophic flow, also referred to as the "omega equation".

In Sect. 3, the history of research on diabatic effects on extratropical dynamics is summarised until the late 1970s, when a series of severe extratropical cyclones, in particular the Queen Elizabeth II storm in September 1978 and the Presidents' Day cyclone in February 1979, both occurring along the North American east coast, acted as a wake-up call for this research field. Triggered by these poorly predicted cyclones, research on the role of diabatic effects on atmospheric dynamics was substantially intensified and several observational field experiments, which were, at least partially, addressing diabatic effects, were realised. The progress in this research area during the last decades is then discussed in Sects. 4 and 5—enabled via the combination of field campaigns, numerical model experiments, the availability of reanalyses, and the development of specific theoretical concepts and diagnostics. Section 4 comprehensively covers the historical development of the field in the late 20th century, whereas the summary of studies in Sect. 5, which covers the period after the year 2000, is more selective. Separate sections discuss diabatic effects on extratropical cyclones, on their embedded mesoscale substructures, and on the upper-tropospheric flow. A specific section is also dedicated to the emerging linkage to climate change research, given the increase of atmospheric humidity in a warming climate. The concluding Sect. 6 summarises key elements of the historical evolution of the field and highlights current opportunities and challenges, with the intention to stimulate further research activities in this important field in the coming years. Where appropriate, reference will be made to other recent review articles on related topics, in particular the reviews about extratropical cyclones (Schultz et al., 2019), the extratropical transition of tropical cyclones (Evans et al., 2017; Keller et al., 2019), cyclone clustering (Dacre and Pinto, 2020), Mediterranean cyclones (Flaounas et al., 2022), atmospheric blocking (Kautz et al., 2022), Rossby wave packets (Wirth et al., 2018), and sting jets (Clark and Gray, 2018). In contrast to these excellent overview articles about specific dynamical phenomena, this review has at its centre the question about the relevance of diabatic processes for the dynamics of extratropical weather systems.

In addition to outlining the central focus of our review it is important to mention its limitations. We consider the impact of diabatic processes on NWP but mainly through the specific route of the impact of diabatic outflows on the Rossby waveguide (Sect. 5.5.3). The structure of this review is presented schematically in Fig. 1. As the research presented in the main body of this review is partitioned into three time periods, here we highlight where different meteorological phenomena can be found within the review. A third thread that runs through the review (but not explicitly addressed in the schematic) is the different methodological approaches that have been used. To some extent the approaches have evolved with time from, e.g., case studies and simple dry vs. moist physics sensitivity experiments in the early years to the use of more advanced approaches such as "PV tracers" more recently. Some sections of this review are focused on specific methodological aspects such as Sect. 4.3 on idealised numerical simulations and Sect. 5.4 on novel diagnostics of diabatic processes. However, for the most part the methodological approaches are described where the associated studies are reviewed. To aid the reader, a glossary of key terms and a list of acronyms are presented in Appendices A and B, respectively. Short summaries are also included at the end of Sect. 3 and each subsection of Sects. 4 and 5.

More specifically, the main three aims of this review article are (i) to provide evidence that our understanding of how diabatic processes affect extratropical weather systems has grown considerably since the review article on PV by Hoskins et al. (1985) and the comprehensive book chapter on the rapid intensification of extratropical cyclones by Uccellini (1990), (ii) to portray in detail the historical evolution of a specific research field over several decades and thereby to exemplify how scientific progress

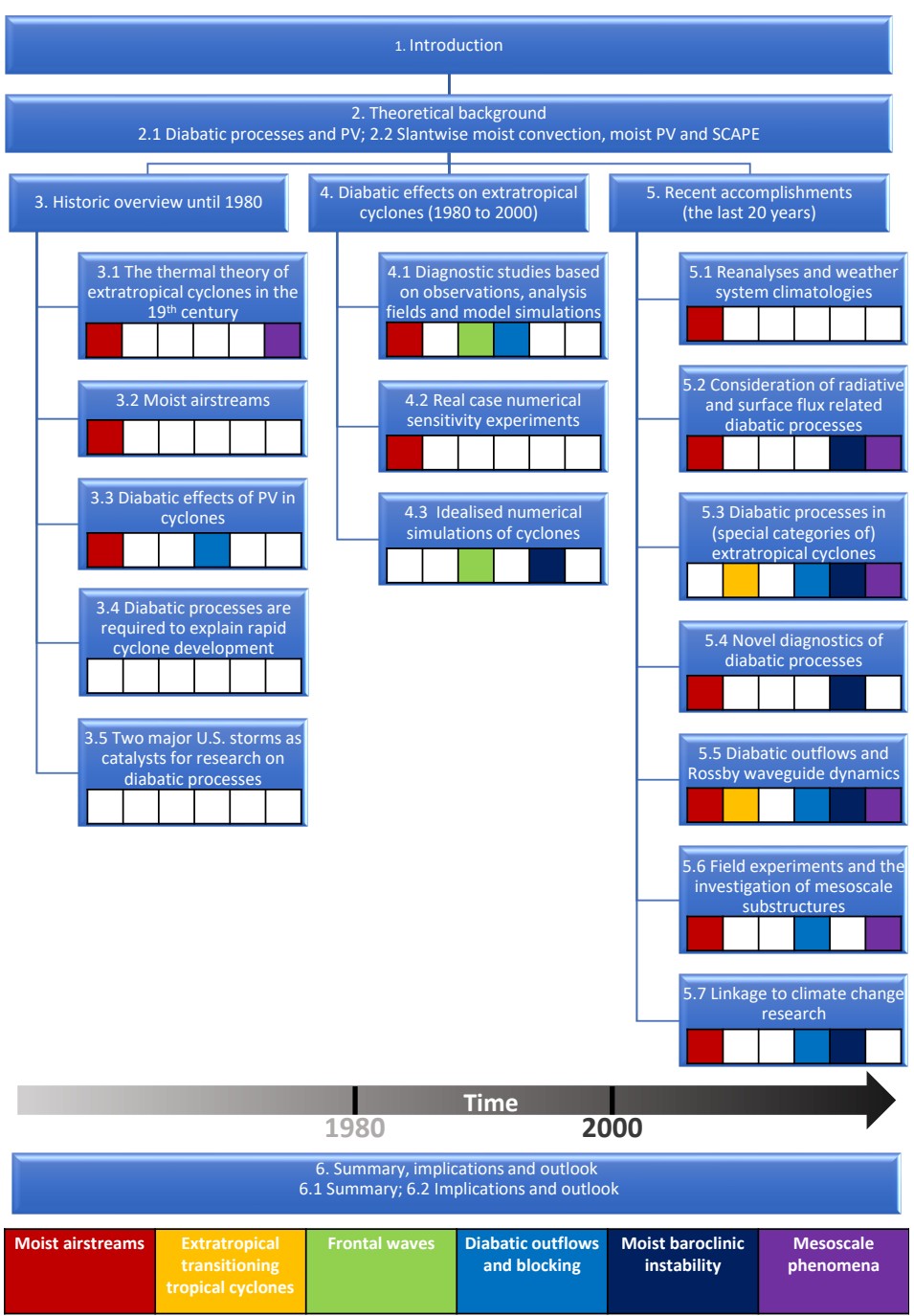

**Figure 1.** Schematic illustrating the structure of this review article and, in particular, where significant review material related to specific atmospheric phenomena can be found.

results from the combination and integration of complementary research approaches, and (iii) to promote the relevance of this research area in dynamical meteorology. As will be discussed, this relevance is at least fourfold:

1. research on diabatic effects adds essentially to our basic physical understanding of weather systems (e.g., Gray and Wernli, 2015);

2. it is at the heart of practical forecasting challenges and predictability issues (e.g., Rodwell et al., 2013, 2018);

3. it is of utmost importance for the evolution of several types of high-impact weather events (e.g., Jones and Golding, 2014; Ducrocq et al., 2016); and

4. it is essential for understanding aspects of climate variability and change related to storm track dynamics (e.g., Bony et al., 2015; Shaw et al., 2016).

## 2 Theoretical background

### 2.1 Diabatic processes and PV

We define diabatic processes as all processes associated with the release of heat in an air parcel or a transfer of heat, mass, or momentum across the air parcel boundary. The main categories of diabatic processes are surface fluxes of sensible and latent heat, turbulence, radiation, and phase changes of water species associated with clouds and precipitation. These processes typically lead to a material change of an air parcel's $\theta$ and/or vapour mass mixing ratio $q$, i.e.,

$$180 \quad \frac{D\theta}{Dt} = \sum_i S_{\theta,i}, \tag{2a}$$

$$\frac{Dq}{Dt} = \sum_i S_{q,i}, \tag{2b}$$

where $D/Dt$ denotes the material derivative ($= \frac{\partial}{\partial t} + u\frac{\partial}{\partial x} + v\frac{\partial}{\partial y} + w\frac{\partial}{\partial z}$) and $S_{\theta,i}$ and $S_{q,i}$ denote the diabatic sources and sinks, due to the $i$-th process, of $\theta$ and $q$, respectively. For convenience, $D\theta/Dt$ is often just written as $\dot{\theta}$. These processes, e.g., the ocean evaporation of water molecules or the freezing of cloud droplets, occur essentially on scales that are much smaller than

185 the numerical models' grid spacing and are therefore parameterised. The complexity of the underlying physics is approximated and for instance cloud particles are represented typically by a few categories, for instance, mass mixing ratios for cloud liquid water $q_c$, cloud ice $q_i$, rain $q_r$, and snow $q_s$. Some parameterizations additionally have categories for graupel and hail. An important difference between $q_c$ and $q_i$ and the other categories is that $q_c$ and $q_i$ represent very small cloud particles and are treated as not or only weakly sedimenting. Therefore, these categories occur in clouds, whereas, e.g., $q_r$ and $q_s$ sediment from

190 clouds and can be involved in below-cloud evaporation, melting and sublimation. Phase changes between water vapour, liquid and solid cloud particles are associated with latent heat release or consumption, and therefore lead to material changes of $\theta$ and mass mixing ratios. Phase changes between vapour and liquid affect $\theta$ more strongly than phase changes between liquid and solid, because of the latent heat of condensation ($L_c = 2.5 \cdot 10^6 \, \mathrm{J\,kg^{-1}}$) being larger by almost a factor of seven compared

to the latent heat of fusion ($L_f = 3.34 \cdot 10^5 \, \mathrm{J\,kg^{-1}}$) at 0°C. The term 'latent' comes from Latin and means 'lying hidden'; the term 'latent heat' was introduced around 1750 by Joseph Black (West, 2014), a professor first at the University of Glasgow and later Edinburgh. Black is also known for the discovery of carbon dioxide.

Because diabatic processes materially change $\theta$, they also have the potential to change an air parcel's PV. Following Eliassen and Kleinschmidt (1957) and Hoskins et al. (1985), the equation for the material change of $Q$ can be written as follows:

$$\frac{DQ}{Dt} = \rho^{-1} \left( \boldsymbol{\omega} \cdot \nabla \dot{\theta} + \nabla \times \mathbf{F} \cdot \nabla \theta \right). \tag{3}$$

The second term on the r.h.s. is the frictional term and involves the scalar product of $\nabla \theta$ and the curl of the non-conservative force $\mathbf{F}$. In a numerical model, $\mathbf{F}$ constitutes the sum of the parameterised momentum tendencies (e.g., from the boundary layer turbulence and the convection schemes). From the first term on the r.h.s., it follows that the PV of an air parcel increases if the gradient of latent heating has a component that points in the direction of the absolute vorticity vector, and PV decreases if this gradient has a component that points in the direction opposite to the absolute vorticity vector. Considering latent heating in extratropical cyclones and neglecting the frictional term, then as a reasonable first order simplification, the scalar product can be approximated by its third term, i.e.,

$$\frac{DQ}{Dt} \simeq \rho^{-1} \left( f + \zeta \right) \frac{\partial \dot{\theta}}{\partial z}, \tag{4}$$

where $f$ denotes the Coriolis parameter and $f + \zeta$ the vertical component of $\boldsymbol{\omega}$. Equation 4 indicates that PV production occurs below the maximum of latent heating in a cloud (e.g., due to condensation of water vapour to cloud droplets, Fig. 2a) and PV destruction above this level of maximum diabatic heating (given that $f + \zeta$ is typically positive in weather systems in the Northern Hemisphere extratropics[2]). In contrast, for a local maximum of latent cooling (e.g., due to evaporation of rain), PV production and destruction occur above and below the cooling maximum, respectively (Fig. 2b). Near fronts, where the horizontal gradient of $\theta$ and the horizontal vorticity components (related to strong vertical wind shear) can be large, Eq. 4 is only a rough approximation. But, even in this approximate form, the interpretation and application of this equation is not trivial, mainly for three reasons: (i) the material change of $Q$ is not proportional to the amplitude of the diabatic heating itself, but to its gradient; (ii) $\zeta$ tends to be highly variable in regions where diabatic processes occur (e.g., near fronts) such that a similar gradient of $\dot{\theta}$ in two regions can have a different impact on the material change of $Q$; and (iii) $\dot{\theta}$ itself results from a combination of different processes (see Eq. 2a), e.g., near the top of a cloud from radiation, turbulence, and phase changes of water. Note, for instance, that (i) implies that PV does not change in an air parcel in which $\dot{\theta}$ has a local maximum.

The first order interpretation of Eq. 4 is to view a cloud as a region of latent heating due to condensation with a maximum at a certain level, leading to instantaneous PV production in the lower and PV destruction in the upper part of the cloud (Hoskins et al., 1985; Thorpe and Emanuel, 1985; Haynes and McIntyre, 1987, see also earlier studies summarised in section 3.3 and the schematics in Fig. 2). When considering a quasi-steady state situation with, e.g., constantly rising motion along a front as

---

[2]With our terminology, we adopt a Northern Hemisphere perspective. For instance, we say that positive PV anomalies are associated with a cyclonic circulation, which is counterclockwise. In the Southern Hemisphere, because of the negative sign of the Coriolis parameter, negative PV anomalies are associated with a cyclonic circulation, which is clockwise.

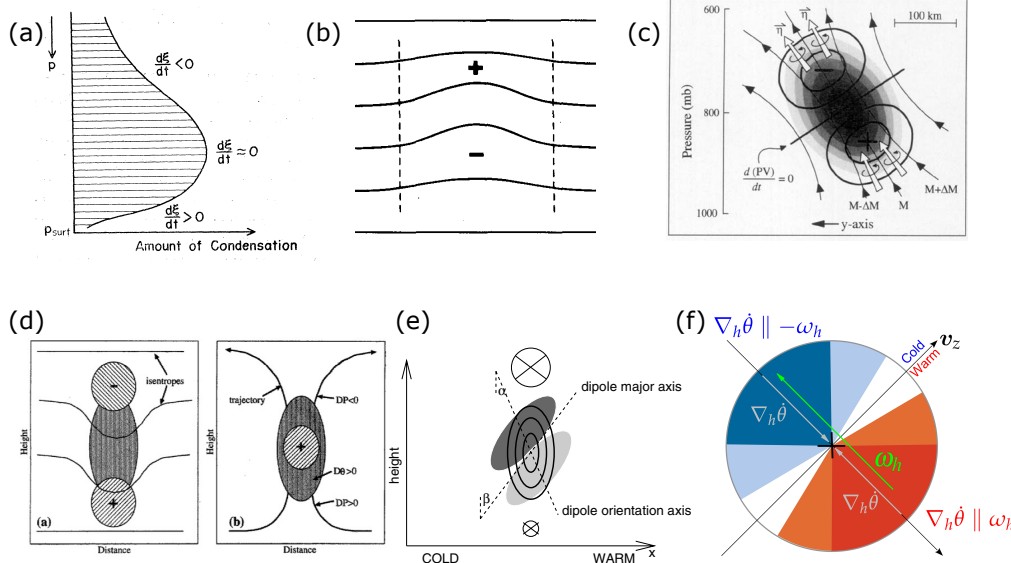

**Figure 2.** Schematics that serve to illustrate different aspects of the interpretation of Eq. 3 and 4. **(a)** from Manabe (1956, their Fig. 1), an idealised latent heating profile and the associated material PV tendencies according to Eq. 4 ($\xi$ denotes PV); **(b)** from Haynes and McIntyre (1987, their Fig. 2), isentropes in a vertical cross section after instantaneous localised latent cooling, which, according to Eq. 4 increases PV above the cooling ($+$ sign) and decreases PV below ($-$ sign)—they, however, interpreted the diabatic PV changes in terms of dilution and concentration of PV substance; **(c)** from Stoelinga (1996, their Fig. 9), diabatic PV modification dipole (thick contours) according to Eq. 3 due to sustained latent heating in a two-dimensional warm-frontal zone (warmer air to the right) along the direction of the absolute vorticity vector ($\eta$, white arrows), thin arrows show streamlines (which are also contours of absolute momentum, $M$) and gray shading is proportional to latent heating and the resulting PV (with maxima of PV generation and depletion indicated by $+$ and $-$ symbols); **(d)** from Wernli and Davies (1997, their Fig. 4), considers again Eq. 4 and shows on the left the same as (b) but for instantaneous heating (Eulerian perspective) and on the right the same as (c) for sustained heating (Lagrangian perspective)—here $P$ denotes PV and $D$ the material derivative, shading indicates the region with diabatic heating and hatching with a plus or minus sign the diabatically produced positive and negative PV anomalies; **(e)** from Chagnon and Gray (2009, their Fig. 1b), similar to (c), the orientation of the diabatically produced PV tendency dipole (dark shading negative and light shading positive tendencies; angle $\alpha$ defines the axis passing through the peak tendency amplitudes, given by $\arctan(\Lambda/f)$ where $\Lambda$ is the vertical wind shear, and $\beta$ defines the axis where linearised PV tendency is zero) arising from heating applied against a baroclinic environment containing vertical wind shear (directed into the page), emphasising that the stronger the shear the more the dipole tilts horizontally; and **(f)** from Oertel et al. (2020, their Fig. 1), horizontal view on the diabatic PV modification by a convective updraft embedded in a WCB (marked by $+$ and ascending out of page) in an environment with background horizontal vorticity (vertical shear, as in (e)), emphasising the horizontal components of the scalar product in Eq. 3—colours show PV tendencies (red for positive and blue for negative values), $\boldsymbol{v}_z$ denotes the vertical wind shear vector and $\boldsymbol{\omega}_h$ the horizontal vorticity vector. All figures used with permission: (b,c) from the American Meteorological Society and (d,e) from Wiley (figures reproduced from open access journals are not listed explicitly, here and in subsequent figures).

in a warm conveyor belt (WCB, see Sect. 3.2 and 4.1.5) and adopting a Lagrangian viewpoint, then including also the effect of
vertical advection leads to a modified picture with a maximum of diabatically produced PV at the level of maximum heating
(Thorpe and Clough, 1991; Stoelinga, 1996; Wernli and Davies, 1997). On the mesoscale, in particular when considering latent
heating in deep convective clouds in the presence of vertical shear, Eq. 4 is no longer a valid approximation. In such situations,
both the horizontal components of absolute vorticity and of the gradient of latent heating can be large and therefore, the full
scalar product in Eq. 3 becomes relevant for understanding the resulting pattern of diabatic PV modification (Chagnon and
Gray, 2009; Weijenborg et al., 2015, 2017; Harvey et al., 2020; Oertel et al., 2020). An important insight from these studies is
that PV in an air parcel cannot change sign when considering the PV equation in the simplified form (Eq. 4). PV can only turn
from positive to negative in the presence of vertical wind shear and when considering also the horizontal contributions to the
scalar product. For the situation of a localised heating maximum in a WCB due to embedded convection, negative PV values
can then appear on the poleward side of the upper-tropospheric jet stream close to the jet core (Harvey et al., 2020; Oertel et al.,
2020, see also Fig. 2f).

Regions where PV values deviate strongly from climatology (or another suitable reference PV) are referred to as "PV
anomalies". As outlined in detail in the seminal review on PV by Hoskins et al. (1985), PV anomalies are the essential building
blocks of the PV framework because of their far-field effect on temperature and the balanced flow, and their ability to interact
with each other (vortex-vortex interaction) and with the waveguide, i.e., with regions characterised by a strong isentropic PV
gradient (wave-vortex interaction). The quantitative analysis of the action at a distance of PV anomalies and their interactions
requires a so-called "PV inversion", which can be performed using balance conditions of different complexity (e.g., Hoskins
et al., 1985; Davis and Emanuel, 1991). Three illustrations of the temperature and horizontal circulations of inverted isolated
PV anomalies are shown in Fig. 3a,b and d. Figures 3b,d show schematically and from calculation, respectively, the fields
associated with an idealised isolated "ball" of uniform PV anomaly, whereas Fig. 3a shows the fields associated with a PV
anomaly resulting from a tropopause depression. The circulation associated with boundary temperature anomalies can be
inferred from considering boundary temperature anomalies to be equivalent to appropriately signed PV anomalies concentrated
at the boundary (Fig. 3c). In reality, PV anomalies rarely occur in isolation and Fig. 3e illustrates the effect of so-called PV
shielding. In the left panel of Fig. 3e (at an initial time) the negative PV anomalies shield the upper and lower parts of the
model domain from the positive PV anomaly in the centre such that the associated meridional wind field is locally confined.
At a later time (middle panel) the advection of the PV anomalies by a zonal wind shear causes unshielding of the positive PV
anomaly and consequently the meridional flow extends throughout the domain.

PV inversion has been used as an important diagnostic in many studies at the synoptic scale, and several of them will
be discussed in this review. When studying smaller-scale systems, the validity of the balanced flow assumption required for
agreement between the flow field obtained from PV inversion and the actual flow field becomes more questionable. However,
there is evidence to support that the PV concept is still useful at the mesoscale. Davis and Weisman (1994) showed that
mesoscale convective vortices evolving from mesoscale convective systems (with horizontal scales of 100-200 km) are nearly
balanced, although their formation depends on unbalanced motions. Weijenborg et al. (2017) concluded that the statistically
significant flow anomalies associated with PV anomalies resulting from cells of summertime deep moist convection imply

that these anomalies might be invertible in a statistical way and discussed possible routes to inverting PV at the convective-weather scale. This study, and those of Weijenborg et al. (2015) and Chagnon and Gray (2009), also found that PV dipoles can have longer lifetimes than the convective updraught that initiated them, increasing the likelihood that balanced circulations exist. Finally, individual mesoscale PV anomalies can aggregate to form larger anomalies that are associated with coherent larger-scale horizontal circulation anomalies, implying the qualitative validity of PV inversion at this scale.

The concept of PV inversion points to a fundamental distinction of how diabatic processes can influence flow dynamics: this influence can be direct in situations where an air parcel's PV is modified by the presence of, e.g., a cloud or a region of turbulence (Eq. 3), or it can be indirect in situations where diabatic PV anomalies created "in the vicinity" of a region of interest influence this region via their induced far-field effect. Note that in a quasi-geostrophic framework, PV inversion determines the geostrophic flow, and the ageostrophic flow, required to obtain the full three-dimensional flow response of a PV anomaly, is determined by the omega equation. This equation relates vertical motion (denoted in pressure coordinates by $\omega$) to the geostrophic flow and latent heating [see the textbook by Holton and Hakim (2013), Sect. 6.5, and the review by Davies (2015) for a description of the omega[3] equation].

We conclude this background section about PV by mentioning important theoretical considerations that went beyond the direct analysis of the material PV tendency equation (Eq. 3). Haynes and McIntyre (1987, 1990) wrote the PV equation in flux form

$$\frac{\partial}{\partial t}(\rho Q) + \nabla \cdot \mathbf{J} = 0, \tag{5}$$

where $\mathbf{J}$ is the total PV flux, which can be split into advective and non-advective parts,

$$\mathbf{J} = \mathbf{u}\rho Q + \mathbf{J}_N. \tag{6}$$

They studied integral conservation properties, which led them to conclude that there can be no net transport of PV across any isentropic surface and that PV cannot be created/destroyed in a layer bounded by two isentropes. As a consequence, the hydrostatic, isentropic coordinate expression for the non-advective PV flux,

$$\mathbf{J}_N = (\dot{\theta}\partial v/\partial\theta - F_y, -\dot{\theta}\partial u/\partial\theta + F_x, 0), \tag{7}$$

is parallel to the isentropic surface. Local PV modification by diabatic effects is then interpreted as a dilution or concentration of "PV substance", $\rho Q$, for which isentropes are impermeable. Methven (2015) and Saffin et al. (2021) used this alternative PV framework to study the PV evolution along WCBs, as discussed in Sect. 5.4. For steady state conditions, the total PV flux $\mathbf{J}$ can be written elegantly as the vector product of the gradients of $\theta$ and the Bernoulli function $B$, which indicates that also in the presence of diabatic and frictional processes, the intersections of surfaces of constant $\theta$ and $B$ are flux lines of PV transport (Schär, 1993). Within the same general framework, Névir (2004) introduced the so-called dynamical state index,

$$\text{DSI} = \rho^{-1}\, \nabla Q \cdot (\nabla\theta \times \nabla B), \tag{8}$$

---

[3]Note that $\omega$ here is unrelated to the absolute vorticity vector $\boldsymbol{\omega}$.

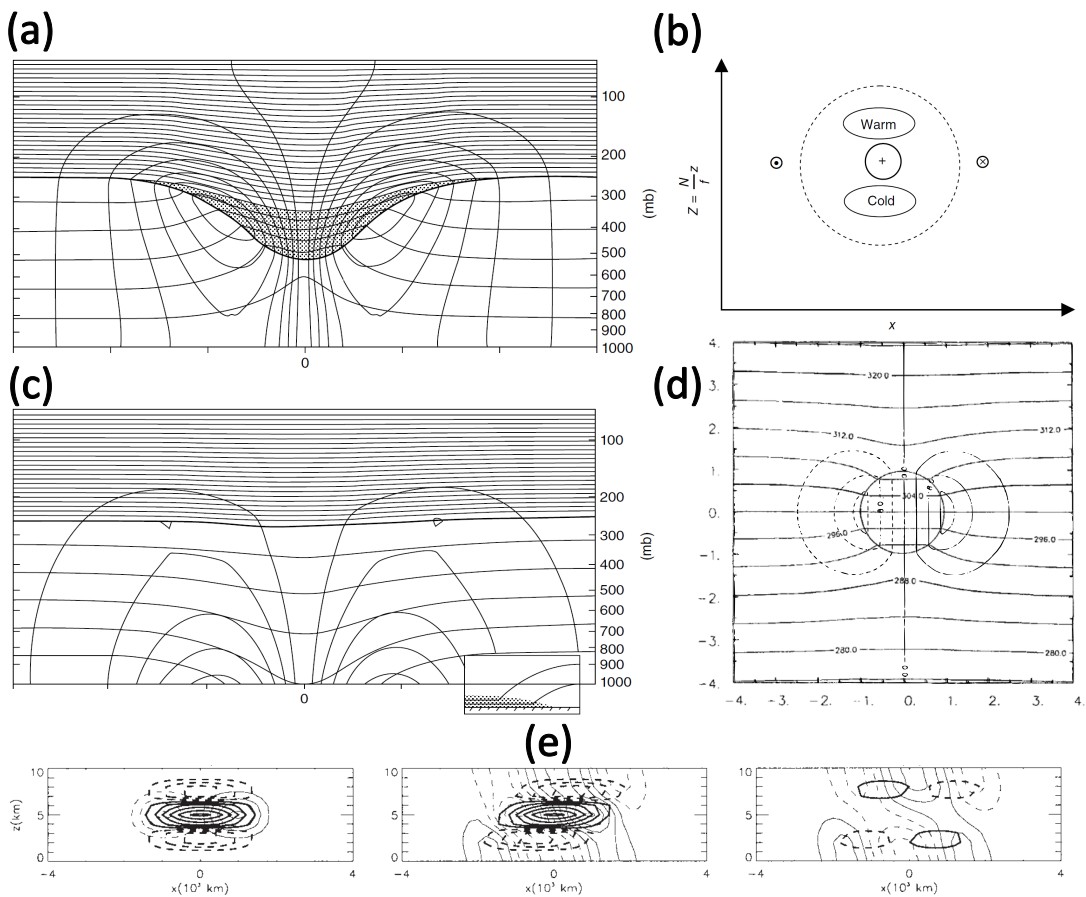

**Figure 3.** PV inversion and shielding. **(a)** from Hoskins et al. (1985, their Fig. 15a), inversion of a PV anomaly (stippled) associated with a tropopause depression to give $\theta$ (approximately horizontal contours) and horizontal wind fields (the other contours are the magnitude of the transverse wind, which is cyclonic; zero contour omitted); **(b)** from Hoskins and James (2014, their Fig. 14.17), schematic illustration of the wind and temperature anomalies associated with a point positive PV anomaly (indicated by +) in a deep fluid of uniform PV (dashed line indicates associated geostrophic streamfunction); **(c)** from Hoskins et al. (1985, their Fig. 16a), as for (a) but showing the effect of a warm anomaly on the lower boundary with the insert illustrating that a warm $\theta$ anomaly is equivalent to a positive PV anomaly concentrated at the surface (shown by stippling); **(d)** from Thorpe and Bishop (1995, superposition of the left and right panels of their Fig. 1a), vertical section showing the inverted $\theta$ (approximately horizontal contours) and normal velocity fields (cyclonic circulation) for a uniform semi-geostrophic PV anomaly of a specified magnitude with a spherical shape in the stretched coordinate used (axes marked with distance divided by the arbitrary anomaly radius); and **(e)** from Badger and Hoskins (2001, their Fig. 3), PV unshielding for a zonal shear flow showing PV (thick contours) and meridional wind perturbations (thin contours) at an initial and later time (left and middle panels, respectively) and their difference (right panel) where solid contours imply positive values, and dashed negative. Note that (a,c) were taken from Hoskins and James (2014) as the reproductions there are better quality than in the original paper. All figures used with permission: (a-d) from Wiley and (e) from the American Meteorological Society.

as a measure of the "non-stationarity" of the flow. The DSI has been used, for instance, to investigate the intensification and decay of cyclones (Weber and Névir, 2008) and the formation of blocks (Müller and Névir, 2019). Weber and Névir (2008) showed that strongly diabatic cyclones are associated with DSI dipoles in the lower troposphere, and that, evaluated on the seasonal timescale, this diagnostic can therefore serve as an alternative measure for storm track activity. Finally, it is mentioned that different forms of 'moist PV' have been introduced to study dynamical processes, as outlined in the next subsection.

## 2.2 Slantwise moist convection, moist PV and SCAPE

As outlined above, latent heating in clouds is an important diabatic process modifying PV and consequently both the structure and evolution of extratropical cyclones. Condensation and subsequent precipitation usually occur through the ascent, and so cooling, of an air parcel, leading to the characteristic cloud and precipitation structures along the frontal zones in cyclones. This ascent can be forced ascent, commonly from ageostrophic frontal circulations or orography, or occur through the release of mesoscale instabilities (usually following initial forced ascent). Condensation also occurs through ascent driven by large-scale processes, defined in the mid-latitudes by Doswell (1987) as those that are quasi-geostrophic such that the ascent can be diagnosed using the omega equation. However, as noted by Doswell (1987), this ascent is likely too slow to initiate the release of instabilities leading to deep moist convection. The importance of convective instability release, particularly the release of CSI (conditional symmetric instability) leading to slantwise convection, for explosive cyclone development (see Sect. 4.1.2.) and for the formation of "sting jets" (see Sect. 5.1.2) means that a brief review of the definition and methods of diagnosis of CSI, and related mesoscale and convective instabilities (including the more well-known conditional instability (CI), the form of moist gravitational instability that leads to deep upright convection on release) is merited here.

The definitions of the different types of dry and moist gravitational and symmetric instabilities, as well as inertial instability, are summarised in Table 1 of Schultz and Schumacher (1999). Dry and inertial instabilities are released in response to infinitesimal amplitude perturbations, conditional instabilities are similarly released when an air parcel reaches saturation (through finite-amplitude lifting), and potential instabilities are released when an atmospheric layer is lifted a finite distance to saturation. While the three types of gravitational instability (dry, conditional and potential) can be defined solely by negative vertical gradients of different forms of a hydrostatically balanced mean-state potential temperature $\overline{\theta}$ ($\overline{\theta}$ for dry, and $\overline{\theta_e}$ for potential where $\theta_e$ is the equivalent $\theta$, and $\overline{\theta_e^*}$ for conditional where $\theta_e^*$ is the saturated equivalent $\theta$ ), the equivalent definition for symmetric instabilities also considers geostrophic absolute momentum defined as $M_g = v_g + fx$ where $v_g$ is the geostrophic wind component in the along-front direction, and $x$ is the cross-front direction (pointing towards the warmer air). The three forms of symmetric instability can be diagnosed as their gravitational equivalents along surfaces of constant $M_g$ (e.g., CSI exists where the vertical gradient of $\overline{\theta_e^*}$ is negative along an $M_g$ surface). Equivalently, these symmetric instabilities can be diagnosed as a negative horizontal gradient of $M_g$ along an iso-surface of the appropriate form of $\overline{\theta}$. This latter form reveals the link of these symmetric instabilities to inertial instability, defined as existing where there is a negative product of $f$ and the horizontal gradient of $M_g$ ($\partial M_g / \partial x$), which is equivalent to negative absolute vorticity. An air parcel can be unstable to slantwise displacements (in a direction between the surfaces of $M_g$ and the appropriate $\overline{\theta}$) while being stable to vertical displacements, due to the corresponding gravitational stability, and stable to horizontal displacements, due to inertial stability.

This situation occurs where the appropriate $\bar{\theta}$ surfaces are more vertical than the $M_g$ surfaces, a situation that often arises in a portion of a cold front (see Fig. 3 of Clark and Gray, 2018). The close link between inertial, gravitational and symmetric instabilities often leads to more than one type of instability being diagnosed within frontal zones and the release of one type of instability generating another type of instability (see Sect. 5a of Schultz and Schumacher, 1999). As described in Sect. 4.1.2, many of the early papers considering the potential role of CSI release in explosive cyclogenesis inferred the presence of CSI by plotting vertical cross-sections across the frontal zone of $M_g$ (or the full instead of the geostrophic absolute momentum) and $\theta_e$ (noting that $\theta_e$ equals $\theta_e^*$ in cloud). As implied by the equation for $M_g$, the theory of symmetric instability is strictly two-dimensional and applies to a frontal zone in thermal wind balance. However, as discussed by Schultz and Schumacher (1999), many frontal regions are not two-dimensional leading to potential errors in the diagnosis of moist symmetric instability (MSI, a term that includes both potential and conditional symmetric instability).

In addition to relative slopes of surfaces of $M_g$ and $\theta_e^*$, two other diagnostics are commonly used to diagnose the presence of CSI: a form of moist PV and slantwise conditional available potential energy (SCAPE). As summarised in Schultz and Schumacher (1999), the definitions of dry, conditional and potential symmetric instabilities in terms of the relative slopes of $M_g$ and the appropriate form of $\theta$ can be expressed, when extended from two to three dimensions, as the conditions that the associated form of PV is negative (provided inertial instability and the corresponding gravitational instability are absent). Thus, dry symmetric instability exists where the PV is negative (PV is usually calculated using the geostrophic absolute vorticity vector and denoted $PV_g$, though see comments below); potential instability exists where the moist PV, MPV, is negative (MPV is PV calculated according to Eq. 1 but using $\theta_e$ instead of $\theta$); and CSI exists where the saturated equivalent PV, MPV*, is negative (MPV* is PV calculated using $\theta_e^*$). As explained above, the diagnosed presence of CSI does not guarantee that it will be released and so produce slantwise convection. Dixon et al. (2002) defined a diagnostic for the number of model levels (for each model grid column) with "realisable" CSI by combining the MPV criteria for CSI with the ingredients-based methodology for forecasting convection described above: they defined the vertically integrated extent of realisable symmetric instability as the number of levels that have negative MPV and ascent while also being inertially and moist-statically stable and near saturated.

SCAPE is the slantwise equivalent of the more familiar convective available potential energy (CAPE, for a definition see Sect. 6.3 of Emanuel, 1994) and can be calculated as CAPE along an $M_g$ surface, as first defined by Emanuel (1983) (see also Sect. 12.2 of Emanuel, 1994), thus enabling SCAPE to be calculated from vertical cross-sections across fronts. A practical method of extending this calculation to generate maps of SCAPE from numerical model output was proposed by Shutts (1990a). Rather than determining the direction of the frontal cross-section, the absolute momentum surfaces along which CAPE is calculated are diagnosed as the intersection lines of the two components of absolute momentum (starting with the values at the initial position of the air parcel) because parcels ascending along these intersection lines follow a succession of minimum energy states. Note that Shutts (1990a) used absolute momentum calculated using the full rather than geostrophic winds, a simplification that can be justified both on practical grounds (derived geostrophic winds are typically noisier than the full winds) and also on theoretical grounds—see discussion in Gray et al. (2011, Sect. 2.2.5). This SCAPE calculation implicitly assumes that the timescale over which CSI is released is fast relative to the timescale for the evolution of the meteorological

fields (e.g., the evolving frontal system). While the equivalent assumption can be considered reasonable for the release of CI, it is less appropriate for CSI with respective timescales typically considered to be about 4 h, 0.5 h and a day for slantwise convection, upright convection and baroclinic growth. Gray and Thorpe (2001) investigated the impact of this assumption by developing an appropriate extension of the parcel theory for CSI to three dimensions and then comparing SCAPE values derived using two and three dimensional approaches for a cyclone observed during the FASTEX field campaign, finding qualitative differences in the patterns of SCAPE relative to the cyclone features. A method for estimating SCAPE from a single (upright) thermodynamic sounding (such as measurements from a radiosonde) is also given in Emanuel (1983). This method has also been applied more recently to model output to simplify the calculation of SCAPE as required for the computational efficiency, e.g., for the calculation of a CSI climatology by Chen et al. (2018) and near-real-time diagnosis of the possible presence of sting jets in cyclones by Gray et al. (2021b). The diagnostic of Dixon et al. (2002) described above has advantages over SCAPE in that instability at all model levels is considered (in contrast to SCAPE for which air parcels are typically lifted from a single level or a maximum value is calculated by lifting from a small number of levels in the boundary layer) and that the diagnosed CSI is likely being released due to the criteria for ascent and saturation. The disadvantage of this diagnostic though is that the units (a number of model levels) cannot be directly related to the strength of the slantwise ascent in the same way as SCAPE, for which the value (in $J\,kg^{-1}$) can be equated to the theoretical maximum kinetic energy of air parcels when the SCAPE is released.

Finally, we note that an environment does not have to become unstable to CSI for its symmetric stability to modify both the structure of frontal zones and cyclone growth. A series of papers by Emanuel (1985), Thorpe and Emanuel (1985) and Emanuel et al. (1987) successively investigated the effect of small moist symmetric stability on frontal circulations and baroclinicity through finding analytic solutions to, and running time-dependent models based on, semigeostrophic equations. Considering frontal circulations, Emanuel (1985) and Thorpe and Emanuel (1985) found that the updraught collapses to a very small scale and the rate of surface frontogenesis is "somewhat enhanced" in environments that are nearly moist symmetrically neutral. Considering baroclinicity, Emanuel et al. (1987) found that the fastest growing baroclinic wave in the limit of zero moist PV (implying moist symmetric neutrality as moist symmetric instability exists if moist PV is negative) has a finite growth rate that is about 2.5 times the dry value and an associated horizontal scale reduction to about 0.6 of the dry value. This result was obtained from solving a two-dimensional two-layer semigeostrophic model analytically for the fastest growing baroclinic wave (see also Sect. 4.3.1). This behaviour reflects that the growth of baroclinic waves varies with the inverse square root of PV in the semigeostrophic system (implying that PV can be considered as having the analogous role to static stability in the quasi-geostrophic system).

For the interested reader, an extensive textbook description of convective instabilities can be found in Emanuel (1994). Also, a critical review of the diagnosis and interpretation of CSI is presented by Schultz and Schumacher (1999) and an extended historical review and discussion of whether conditional instabilities are truly instabilities can be found in Schultz et al. (2000), which was written in response to Sherwood (2000). In particular, Schultz and Schumacher (1999) emphasised that the release of conditional instabilities (both CI and CSI) can only occur where there is also moisture and a mechanism to lift air parcels to saturation (the so-called ingredients-based methodology for forecasting upright or slantwise convection). The authors also

argued for the inseparability of MSI and frontogenesis due to modulation of frontogenetically forced circulations by symmetric stability, possible existence of environments close to slantwise convective equilibrium (in which MSI is released before substantial build up of available potential energy) and similarity of timescales for frontogenetic circulations and slantwise ascent through the release of MSI. Finally, a more recent (and shorter) review of gravitational, symmetric and inertial instability definitions and processes can be found in Clark and Gray (2018, their Sect. 2.3).

## 3   Historic overview until 1980

This historic overview section starts in the first half of the 19[th] century and ends in the late 1970s. In these almost 150 years, the view on the importance of diabatic processes for explaining the dynamics of extratropical weather systems varied strongly. Whereas latent heating was seen as the essential ingredient for cyclone intensification in much of the 19[th] century, its role was considered as marginal after the breakthrough discovery of quasi-geostrophic baroclinic instability in the first half of the 20[th] century (Charney, 1947; Eady, 1949), which led to the "golden age of dry dynamics". The concise summary of more than a century of research in the following subsections focuses on four selected aspects: latent heating as an energy source of cyclones (Sect. 3.1); moist ascending airstreams, i.e., Lagrangian flow structures characterised by strong latent heating (Sect. 3.2); early considerations of diabatic PV modification (Sect. 3.3); and first studies emphasising that moist processes might play a key role in rapid cyclone intensification (Sect. 3.4).

### 3.1   The thermal theory of extratropical cyclones in the 19[th] century

The question about the source of the kinetic energy associated with the cyclonic circulation in extratropical cyclones was already at the heart of dynamical meteorology in the 19[th] century. The essential ingredients of the conceptual understanding at that time were the ascent of warm air in the centre of the cyclone, the condensation of water vapour in clouds, and the release of latent heat. The book by Kutzbach (1979) provides a fascinating historical overview of this so-called "thermal theory of cyclones" and its main proponents, including Espy and Loomis. In the 1830s, notably prior to the formulation of the first law of thermodynamics in 1850 by Clausius, Espy performed laboratory experiments and correctly inferred the expansion and cooling of ascending currents of air. These ideas about thermal convection go back to Halley (1687) who studied trade winds and monsoons. Espy (1841) went beyond a description of convection in cyclones and hypothesised that latent heating by condensation in this thermal convection is the "motive power" of cyclones. Loomis (1841) supported this viewpoint by detailed meteorological analyses of a winter storm in the U.S. (most likely one of the earliest documented cyclone case studies), based on observations of pressure, temperature, precipitation and windspeed from more than 100 surface stations, operated by the military and the academies in New York, respectively. Loomis concluded ". . . that the southerly current must have disappeared by being elevated in the upper regions of the air. We find, then, a warm current suddenly cooled, and its moisture must, of course, be in part precipitated", which sets the basis for the idea of precipitation formation in cyclones by ascending air currents or later airstreams (see Sect. 3.2). At that time, extratropical cyclones were seen as circularly symmetric with low-

level convergence into the cyclone, ascent in the centre and divergent outflow at the level of upper-tropospheric cirrus clouds (Kutzbach, 1979, their Fig. 1).

These early North American concepts were supported by theoretical considerations mainly in Europe a few decades later. Using concepts of theoretical thermodynamics, meteorology transformed into atmospheric physics in the late 19[th] century (McDonald, 1963). Among the leaders in this undertaking were Thomson (Lord Kelvin), Reye, and Peslin, who developed the concept of moist saturated ascent of an air parcel. Reye (1864, 1865) was the first to apply the first law of thermodynamics to meteorology (see review by Davies and Wernli, 2016). Reye introduced the "parcel method" and, by considering the expansion of air and the buoyancy changes related to ascent and cloud formation, examined stability criteria for dry and moist ascent. Buchan (1868) then combined detailed meteorological observations and theoretical thermodynamic principles and concluded that "the chief disturbing influences at work in the atmosphere are the forces called into play by its aqueous vapor", an early and strong statement about the importance of diabatic processes for atmospheric dynamics. The statement also conveys a clear understanding of the intimate relationship between atmospheric dynamics and the atmospheric water cycle.

## 3.2 Moist airstreams

Ascending air currents in the centre of extratropical cyclones, and the associated latent heat release, were essential elements in the thermal theory of cyclones. Bezold (1888) introduced the term potential temperature and noted that within the ascending air current, $\theta$ continuously increases in proportion to the amount of condensed water vapour. The 19[th] century air current concepts were refined near the turn of the century by Bigelow, Lempfert and Shaw. Based upon more detailed observations of the structure of cyclones, they abandoned the axisymmetric model of extratropical cyclones. Instead, they emphasised important features of the cyclones' three-dimensional structure, e.g., the ascending motion in the tongue of warm air ahead of a moving cyclone and the role of alternating tropical and polar air currents in accomplishing horizontal and vertical heat exchange (Bigelow, 1902; Ficker, 1911, Fig. 4a). In harmony with the ideas of Margules (1905), Bigelow (1906) wrote "instead of vertical convection being the primary cause of storms it is rather horizontal convection" and identified this quasi-horizontal interpenetration of currents of different temperatures as the true energy source of the storms (see also the critical discussion by Brunt, 1930). Bjerknes (1919) referred to the two currents as cold and warm, respectively, and highlighted that "the cold current is screwed underneath the warm one, and the warm current screwed up above the cold one[4] ... joining the general western drift in the higher strata". The introduction of the Lagrangian concept of an air parcel trajectory by Shaw (1903) (see also Shaw and Lempfert, 1906) made it possible to follow the actual path of an isolated volume of air (at that time only along the surface due to the lack of free-tropospheric wind data). Representing the trajectories relative to the translating cyclone, they deduced regions with pronounced surface convergence, and therefore ascent, in certain parts of the cyclone (Fig. 4c).

A few decades later, after establishing networks of upper-air observations with radiosondes, the method of isentropic analysis was pioneered by Rossby (e.g., Rossby and Collaborators, 1937) and Namias. This method can be regarded as an alternative Lagrangian tool for the analysis of three-dimensional airflows [see Eliassen (1986) for a summary of the historical development

---

[4]Interestingly, Bjerknes also used the formulation that "warm air is conveyed to previously cold air regions", which foreshadowed the term WCB introduced half a century later.

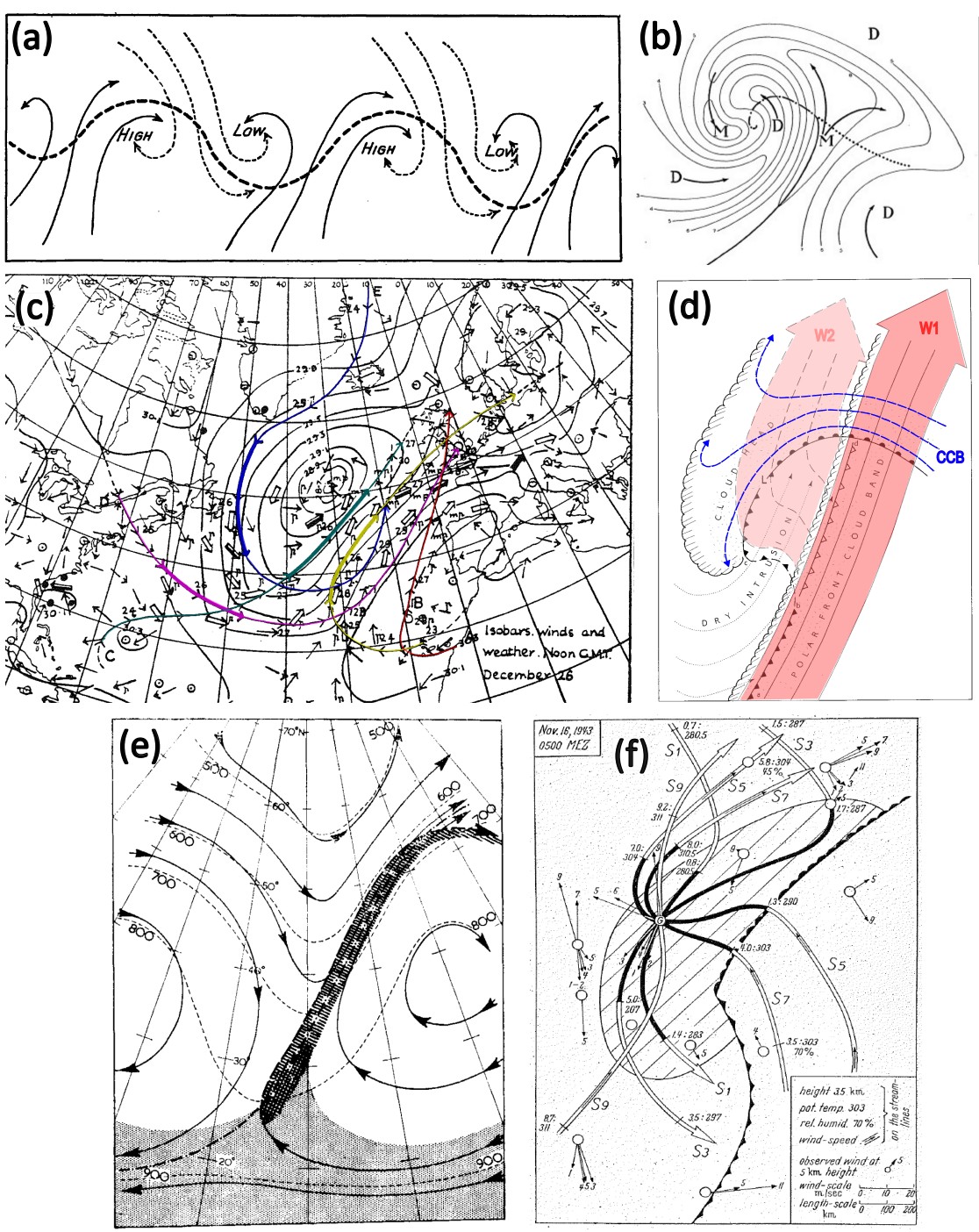

**Figure 4** *(previous page).* Historical illustrations of trajectories and moist airstreams. **(a)** from Bigelow (1902, their Fig. 9), curling of northward and southward airstreams relative to the position of low and high pressure systems; **(b)** from Namias (1939, their Fig. 1), schematic flow pattern (bold arrows) around an occluded cyclone indicated by isolines of moisture on an isentropic surface (M and D refer to moist and dry air, respectively); **(c)** from Shaw and Lempfert (1906, taken from Volkert (1999), their Fig. 4), 7-day surface trajectories (bold arrows) in a North Atlantic cyclone in late December 1882; **(d)** from Browning (1997, their Fig. 4), schematic of the WCB (drawn relative to the motion of the system) in a developing extratropical cyclone with cold, warm and bent-back fronts (W1 and W2 refer to two branches of the WCB, CCB refers to the cold conveyor belt, and precipitation is indicated by stippling); **(e)** from Green et al. (1966, their Fig. 2), schematic of the moist poleward ascending airflow associated with an elongated cloud band (dark hatching) in a trough-ridge system on an isentropic surface (numbers indicate the height of the isentrope in hPa); and **(f)** from Eliassen and Kleinschmidt (1957, their Fig. 35), five trajectories ascending through the "cloud head" of a developing cyclone (calculated as streamlines in a system relative flow). Colour has been added to (c,d) to enhance clarity. All figures used with permission: (a,c) from the American Meteorological Society, (f) from Springer Nature, and (d,e) from Wiley.

and Carlson (1991) for a detailed outline of the method]. Its basic idea is that air masses can be identified and classified either by the display of a quantity that is materially conserved for adiabatic flows (e.g., specific humidity or later PV) on surfaces of constant (equivalent) $\theta$. Namias (1939) emphasised the value of isentropic charts for "tracing air particles from day to day, and in this manner to define the principal flow patterns of the free atmosphere" (Fig. 4b). Later, Eliassen and Kleinschmidt (1957)

and Green et al. (1966) considered streamlines of the flow relative to a mature synoptic system (typically, an extratropical cyclone) on isentropic surfaces. Assuming that for such a system the isentropic flow configuration is in a steady state—Carlson (1991) referred to this as the "frozen-wave approximation"—observations made at different times can be included in the same analysis and streamlines can be interpreted as parcel trajectories (Fig. 4f). Green et al. (1966) identified a narrow and elongated ascending airflow on the downward side of the trough with air parcels lifted through the whole troposphere during a single

event of cyclogenesis (Fig. 4e). This study was clearly very influential for the emergence of the concept of the WCB (Browning, 1971; Harrold, 1973). If compared with the original air currents or "moist tongues"[5], the novelty of the conveyor belt concept (as extended by, e.g., Carlson, 1980; Browning and Mason, 1981; Browning, 1990) was the emphasis on the narrowness of the strongly ascending airflow and its direct association with cloud and precipitation patterns as observed from satellite imagery (Fig. 4d). It is worth noting that the analysis of airstreams in this period was strongly influenced by the newly available radar

and satellite imagery[6].

_______________________

[5]The recent review by Mo (2022) emphasised the concept of "moist tongues" identified by Rossby and Namias in the 1930s and their similarity with the later established concepts of WCBs and atmospheric rivers.

[6]The period since 1979 is regarded as the early satellite era, with a first Geostationary Operational Environmental Satellite (GOES) launched in 1975. However, a series of polar orbiting satellite, specifically dedicated to meteorology (TIROS I), was launched already in 1960 and in the following years, providing fascinating pictures of different cloud formations, including wave formation on a frontal cloud band and spiral cloud structures of a mature extratropical cyclone (e.g., Fig. 49 and 62 in Anderson et al., 1966). This document was later updated as a very comprehensive atlas for the interpretation of satellite imagery (Anderson et al., 1973). Early papers about using satellite images specifically for studying intense extratropical cyclones also appeared at this time (Böttger et al., 1975; Burtt and Junker, 1976)

The alternative approach to identify key atmospheric flow patterns was to explicitly calculate air parcel trajectories. After the pioneering studies of Shaw and Lempfert, this approach was resumed by Rossby (1945), assuming that absolute vorticity is conserved for flows in the free troposphere. With this approach, Rossby was the first to identify the out-fanning nature of equatorward descending currents in the cold air behind extratropical cyclones, later referred to as "dry intrusions". Similar flow patterns were identified, e.g., by Palmen (1953), Reiter and Mahlman (1965), and Danielsen (1968), based on the approximate calculation of isentropic trajectories.

## 3.3 Diabatic effects on PV in cyclones

In parallel to the studies mentioned above, a second more theoretical stream of research originated that used the concept of PV (Ertel, 1942) to investigate the role of diabatic processes. Interestingly, the pioneering studies in this field looked at the temporal change of PV along ascending and descending air parcel trajectories—a very modern approach, which emerged again several decades later as a key element of the PV dynamics of WCBs (Sect. 4.1.5) and of investigations about exchange processes at the tropopause (Sect. 4.1.6). Kleinschmidt (1950a, b) (see also Thorpe, 1993) estimated the diabatic change of PV along trajectories ascending near the cyclone centre. Trajectories were approximated by streamlines assuming that the flow is stationary. Using manual calculations, Kleinschmidt estimated a material increase of PV due to latent heating by more than $1\,\mathrm{pvu}\,(= 1 \cdot 10^{-6}\,\mathrm{K\,kg^{-1}\,m^2\,s^{-1}})$ during the ascent along the sloping front of a moist air parcel from the surface to a height of about 3 km, and a subsequent decrease of PV as the parcel ascends to the upper troposphere (see also Figs. 35 and 36 in Eliassen and Kleinschmidt, 1957). Kleinschmidt mentioned the importance of this diabatic PV production for cyclone development, in qualitative agreement with today's basic understanding of diabatic effects on PV (see Sect. 2.1). Eliassen and Kleinschmidt (1957) noted that "the air with increased PV gathers in a continually growing body, which is the producing mass of the new cyclone". Manabe (1956) quantified the diabatic modification of PV due to heat released by condensation in two extratropical cyclones over the U.S. They also emphasised the material increase of PV below, and decrease above, the level of maximum latent heating (Fig. 2a) and concluded that this diabatic PV production can affect the large-scale flow.

So far, this review focused mainly on the diabatic effects on the structure and intensification of extratropical cyclones. In terms of PV, the focus so far was on the low-level diabatic PV production, and much less so, on the simultaneously occurring diabatic effects on upper-level PV (discussed conceptually in Sect. 2.1). An explanation for this early "research bias" on diabatic low-level PV production is the occurrence of high-impact storms, that were poorly predicted and poorly understood based on dry theory, and which required intensified and coordinated research in this area. An early exception are remarks by Eliassen and Kleinschmidt (1957) that "if the jet stream is not too strong, the air masses leaving the cloud [i.e. the region with diabatic heating] with the upper stream form a high-level anticyclone ahead of the cyclone [and] if the jet stream is strong, the upper masses are carried forward and may continue their path in the form of a Rossby wave". These ideas will be discussed in more detail in Sect. 4.1.6 and 5.4. However, research on stratosphere-troposphere exchange (STE), which had a first peak period already in the 1960s after atmospheric nuclear weapon tests, also touched upon this theme. When defining the

tropopause as an iso-surface of PV (the so-called dynamical tropopause[7]), the transport of stratospheric air into the troposphere requires diabatic PV destruction to occur, either by turbulence, radiation, or cloud processes. Therefore, a part of STE research addressed questions related to diabatic effects near the tropopause.

An exemplary early study about STE and diabatic PV modification was performed by Staley (1960). They derived a variant of Eq. 3 and then estimated orders of magnitude for PV changes due to large-scale heating and frictional forces, respectively, based on plausible assumptions about the magnitude of the involved terms (mentioning that their estimates might be in error by an order of magnitude or more). They also identified actual PV changes along isentropic trajectories and thereby events of stratosphere-to-troposphere transport—notably all based on isentropic analysis with radiosonde data over the U.S. Concerning the relative role of PV modifying processes, Staley (1960) estimated that PV changes were mainly attributable to the vertical gradient of diabatic heating rather than to frictional forces. Danielsen (1968) presented aircraft *in-situ* observations near tropopause folds of $^{90}$Sr (used as a stratospheric tracer from bomb tests in the stratosphere) and $^{89}$Sr (used as a tropospheric tracer from tests in the Sahara). STE was identified by diagnosing strong PV changes along trajectories. Radiative cooling and mixing were mentioned as important processes for these PV changes, but no specific process identification was possible. The study by Shapiro (1976) investigated the mesoscale substructure of upper-level jets based on aircraft observations. It revealed localised isentropic PV maxima on the cyclonic shear side of the jet at the level of maximum wind speed. The hypothesis was made that these maxima were diabatically produced by diabatic temperature changes that arise from vertical, shear-induced, turbulent mixing in the vicinity of the jet. Shapiro's estimate of these clear air turbulence-induced diabatic heating rates is one order of magnitude larger than the estimate of radiative effects by Staley (1960). It is interesting to note that the initial research on STE near upper-level fronts and tropopause folds mainly discussed how radiation and turbulence can modify PV, but overlooked the potential effects of latent heat release in clouds. However, this focus changed almost 20 years later with the systematic analysis of STE in mesoscale modelling case studies (see Sect. 4.1.6).

### 3.4 Diabatic processes are required to explain rapid cyclone development

In the mid-1950s, the analysis of the mean sea level pressure (MSLP) evolution of extratropical cyclones led to the important observation that some cyclones intensify more rapidly than others. The question why this is the case was an essential driver of research on the dynamical role of diabatic processes. Petterssen (1955) suggested that a non-adiabatic contribution might be "not negligible" but difficult to evaluate. They primarily thought about heating by surface fluxes based on climatological evidence that most winter cyclones develop over open waters. Winston (1955) also considered heat fluxes from the ocean and latent heating in the lower troposphere to be important for rapid cyclogenesis in the Gulf of Alaska. They wrote "... it is desirable to learn more about how heat sources influence circulation changes and just how important these effects are".

Two quantitative studies on this subject were then performed by Aubert (1957) and Danard (1964). The former contrasted NWP experiments with and without latent heating due to cloud condensation, and the latter indicated, likely for the first time, that the contribution of diabatic processes and dry dynamics to cyclone intensification can be of the same order of

---

[7]There is no universally used PV threshold for the dynamical tropopause, but the most common choice is the 2 pvu surface—see the detailed discussion in Kunz et al. (2011).

magnitude. Both these pioneering studies used quasi-geostrophic dynamics to investigate cyclone case studies over the U.S. Aubert (1957) concluded that latent heating reduces near-surface geopotential height, and Danard (1964) that latent heating contributed to increased low-level kinetic energy and vorticity. A decade later, Tracton (1973) investigated the performance of numerical forecasts of continental U.S. cyclones (from models that had a highly simplified treatment of convection) and found a systematic underprediction of their initial intensification, which, according to observations, was accompanied by an outbreak of convection. In line with the earlier studies, Tracton hypothesised that latent heating in the vicinity of the cyclone centre was crucial for cyclone development (and not properly represented in the forecast models). In parallel, analytical, and highly idealised numerical studies of moist baroclinic instability indicated the importance of latent heat release for intense, small-scale cyclones. Nitta and Ogura (1972), including moisture in an idealised numerical model setting similar to the one used by Hinkelmann (1959) to study dry baroclinic instability, identified the growth of a shallow "intermediate-scale cyclone" with a wavelength of about 1500 km, which was not maintained in the simulation without moisture. Cyclones of this scale were then also observed during the AMTEX field experiment near Japan in 1975 (Saito, 1977), and it was noted that they do not appear to be related to an upper-level trough but associated with moist convection. Idealised studies with and without latent heating revealed an increase of the linear growth rate in the moist experiments (e.g., Gall, 1976), as discussed later in Sect. 4.3.

## 3.5 Two major U.S. storms as catalysts for research on diabatic processes

The brief historic overview in the previous subsections presents clear evidence that until 1980, several pioneering studies investigated many of the essential aspects of how diabatic processes influence extratropical cyclones, based upon observations, theoretical concepts, and the use of early (and still comparatively simple) numerical models. However, these studies have not yet been numerous, and it appears that some of the, in retrospect, main discoveries have been overlooked for decades by the scientific community (e.g., the work by Kleinschmidt as discussed by Thorpe, 1993). The rather marginal role of this research field changed—we think, dramatically—in the 1980s and 1990s. In these decades, research on diabatic processes in extratropical cyclones was strongly intensified, in particular in North America, and it involved many early career scientists who later became outstanding leaders in dynamical meteorology and NWP. We claim this paradigm shift was triggered by the succession of two major cyclonic storms in the U.S. in early and late winter 1978/79. Both cyclones were rapidly intensifying, produced extreme local weather conditions, and were very poorly predicted by operational forecast models.

The first storm on 10–11 September 1978 was termed the "Queen Elizabeth II or just QE II storm", according to the name of the liner that was battered by the storm's hurricane-force winds (Gyakum, 1983a). The core pressure of the cyclone deepened by 60 hPa in 24 h as it moved offshore from New Jersey. Numerical forecasts at the time missed this intensification completely (Gyakum, 1983a). The second storm on 18–19 February 1979, termed the "President's Day snowstorm", produced record 24-h snowfalls along parts of the U.S. east coast. The cyclone deepened explosively along a very shallow Carolina coastal front at the arrival of a short-wave trough from the continent. Bosart (1981, their Fig. 9b) emphasised the key role of the shallow front with a low-level ascent maximum at about 900 hPa and an associated intense vorticity maximum at 950 hPa along the coastal front, which was critical for the rapid spin-up of the cyclone upon arrival of the trough. The study also mentioned intense convection in the core region of the cyclone as a common feature of the two storms during intensification. It is interesting to

note that although the development of this blizzard was missed by the operational weather prediction model at the time, about a year before the same model provided useful guidance for a blizzard affecting Boston (Bosart, 1981).

The societal relevance of these and other explosively deepening cyclones is also reflected by the introduction of the terminology of "bombs" in meteorology (Sanders and Gyakum, 1980). A cyclone is called a "bomb" if its central pressure decrease in 24 h exceeds 24 hPa after normalisation[8]. Based on a statistical analysis of cyclone deepening rates, Roebber (1984) hypothesised that mechanisms of explosive cyclone deepening differ fundamentally from ordinary baroclinic instability. Both U.S. cyclones mentioned above classify as "bombs" and their evolution and the involved dynamical processes have been studied in profound detail (see Sect. 4.1). Even more importantly, they motivated comprehensive North American field campaigns to study diabatic processes in explosively deepening cyclones. As discussed later, also in Europe, two poorly predicted, devastating extratropical cyclones served as catalysts of dedicated research activities, field campaigns, and scientific progress in the field: the "Great October storm" in the U.K. in 1987 and the winter storm "Lothar" in Central Europe in 1999. Table 1 provides an overview on relevant field experiments.

Given the relevance of the four catalyst cyclones for research on diabatic processes, the Supplement provides an overview on their large-scale evolution, using a combination of synoptic charts, PV charts, and vertical cross sections, based on modern reanalysis data (ERA5, Hersbach et al., 2020). This material is prepared in a way that it can be easily used in lecture courses to discuss the evolution and PV dynamics of these four "(in)famous storms".

## 3.6 Summary

Early research on diabatic processes already considered moist airstreams, the diabatic modification of PV in the lower and upper troposphere, and the role of latent heating for the rapid intensification of extratropical cyclones—aspects that remained at the heart of this research field until today. It is impressive to see how relevant conceptual ideas about the role of latent heating emerged prior to the ready availability of satellite observations, numerical models, and gridded analysis products. While early research on diabatic processes in extratropical weather systems can be considered as marginal for the overall development of dynamical meteorology until about 1980, two poorly predicted U.S. cyclones with severe impacts led to a dramatic intensification of research in this field. This research boost was assisted by technological progress and included studies based on theory, numerical modelling, and observations, in particular from dedicated field experiments.

## 4 Diabatic effects on extratropical cyclones—studies from 1980 to 2000

This section attempts to summarise the key research ideas, concepts, and findings about diabatic effects on extratropical cyclones during the years 1980–2000. It contains essential results from the analysis of major international field experiments, in particular ERICA in 1989 and FASTEX in 1997. The three subsections focus on different research approaches: diagnostic studies (Sect. 4.1), real case numerical sensitivity experiments (Sect. 4.2), and idealised numerical modelling studies (Sect. 4.3).

---

[8]The pressure decrease is normalised by multiplying with $\sin(60°)/\sin(\phi)$, where $\phi$ denotes the averaged latitude of the cyclone center during this 24-h interval, such that the absolute pressure decrease threshold increases with latitude.

**Table 1.** Overview of field experiments that contributed essentially to the understanding of moist dynamics of extratropical weather systems. Two references are given where the overview publication is not easily accessible. Acronyms are defined in Appendix B.

| acronym | date | region | overview publication |
|---|---|---|---|
| CYCLES | winters 1973–1978 | eastern North Pacific | Hobbs et al. (1980) |
| AMTEX | Jan–Feb 1975 | western North Pacific | Saito (1977) |
| ALPEX | Mar–Apr 1982 | western Mediterranean | Smith (1986) |
| CASP | 1986/1992 | Canadian east coast | Stewart et al. (1987); Stewart (1991) |
| GALE | Jan–Mar 1986 | U.S. east coast | Dirks et al. (1988) |
| FRONTS-87 | Oct 1987–Jan 1988 | eastern North Atlantic | Clough and Testud (1988); Thorpe and Clough (1991) |
| ERICA | Jan–Feb 1989 | western North Atlantic | Hadlock and Kreitzberg (1988) |
| FRONTS-92 | Mar–Apr 1992 | eastern North Atlantic | Hewson (1993); Browning (1995) |
| FASTEX | Jan–Feb 1997 | North Atlantic | Joly et al. (1999) |
| T-PARC | Aug 2008–Mar 2009 | North Pacific | Parsons et al. (2017) |
| DIAMET | Sep 2011–Aug 2012 | eastern North Atlantic | Vaughan et al. (2015) |
| NAWDEX | Sep–Oct 2016 | eastern North Atlantic | Schäfler et al. (2018) |

### 4.1 Diagnostic studies based on observations, analysis fields and model simulations

As mentioned above, several meteorological field experiments were organised in the two decades before the turn of the century (Table 1), yielding a vast amount of novel data thanks to new instrumentation, e.g., dropsondes and airborne Doppler radar (Wakimoto et al., 1992; Protat et al., 1997). Some of these observations were analysed manually or using optimum interpolation to investigate spatial structures and their temporal evolution. In parallel, NWP centres developed sophisticated data assimilation techniques to obtain high-quality initial conditions for their forecasts. Although strongly influenced by the underlying model's

first guess field, analyses can be regarded as physically consistent observation-based datasets. They are available at least every six hours from weather centre archives and, together with short-term global and regional model forecasts, they have served as invaluable gridded datasets for diagnostic investigations since the early 1980's (see also Sect. 5.1). The studies discussed in this subsection were based on (a combination of) these observational and model-based datasets.

### 4.1.1 "Catalyst cyclone" cases

The first studies about the President's Day and QE II storms were mainly using observational data (conventional surface and upper-air weather charts, surface precipitation station data, ship reports, radiosondes, radar data, and various satellite products[9]) and analysed the structure of the cyclones and their evolution. The analysis of low-level thickness revealed the warm core character of the President's Day cyclone. Clear indications of deep convection in a region mainly north and east of the cyclone centre were obtained from radar and visible satellite imagery, and from convectively unstable layers between 850 and 500 hPa

in radiosonde profiles (Bosart, 1981). The same study also emphasised the important role of strong surface fluxes occurring in

---

[9]It is notable that the timing of these storms was at the start of what is now considered the satellite era (since 1979).

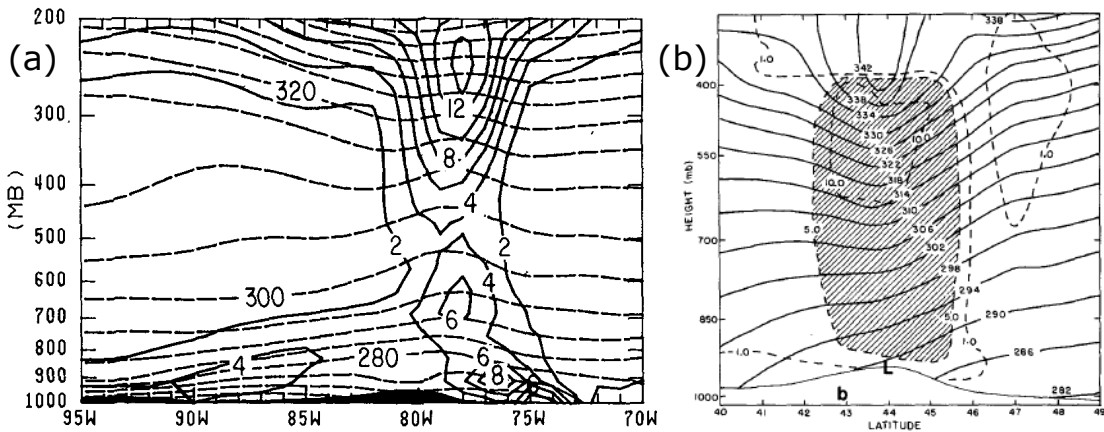

**Figure 5.** Vertical sections of $\theta$ and PV across the catalyst cyclones. **(a)** from Bosart and Lin (1984, their Fig. 11f), President's Day cyclone at 12 UTC on 19 February 1979 (solid contours show semigeostrophic PV in units of $10^{-4}\,\mathrm{s}^{-2}$ and dashed contours show $\theta$ in K); and **(b)** from Gyakum (1983b, their Fig. 7b), QE II storm at 12 UTC on 10 September 1978 (shading indicates PV larger than $5\cdot10^{-5}\,\mathrm{K\,hPa}^{-1}\,\mathrm{s}^{-1}$). Figures used with permission from the American Meteorological Society.

advance of the cyclone centre to increase baroclinicity along the coast. An observation-based analysis of the PV structure by Bosart and Lin (1984) revealed low-level PV production in the intensifying cyclone. In addition, the authors emphasised the precursor role of the downward extension of stratospheric PV for rapid cyclogenesis and the vertical alignment of the upper and low-level positive PV anomalies (Fig. 5a)—an early example of a so-called "PV tower", a term introduced by Hoskins
(1990) for a vertical, troposphere-spanning column of high PV (see below). Uccellini et al. (1985) investigated radiosondes over the U.S. and confirmed the important role of the deep tropopause folding to below 700 hPa, which occurred upstream and 12 h prior to rapid cyclogenesis. They hypothesised that the strong descent of stratospheric air together with the ascent to the north of the cyclone centre, likely enhanced by latent heat release, could have combined to increase the lower-tropospheric cyclonic circulation near the storm centre. The occurrence of rapid deepening at the time when the stratospheric PV anomaly
approached and overlay the low-tropospheric PV maximum over the east coast was confirmed by the results of a mesoscale model simulation with 60 km grid spacing by Whitaker et al. (1988) .

In their study about the QE II storm, Gyakum (1991) showed that prior to rapid intensification, low-level and upper-level precursor vortices developed independently, the former as a shallow frontal wave in a region of strong coastal frontogenesis. The potential role of diabatic processes for the formation of this low-level vortex was not discussed in this study. In the first
of two important earlier studies about the QE II storm, Gyakum (1983a) emphasised the occurrence of deep convection at the time of rapid intensification, again asymmetrically distributed around the storm centre as reported by Bosart (1981) for the President's Day storm. To identify how convection contributed to the intensification of the QE II cyclone, Gyakum (1983b) (in the second of these studies) first estimated temperature and humidity profiles in vertical sections across the cyclone between near-surface ship and upper-level aircraft observations, then calculated geostrophically balanced wind fields and finally the

635 cyclone's PV structure. The results indicate enhanced PV values in the lower troposphere in the centre of the cyclone—most likely the first visualisation of a diabatically produced positive PV anomaly diagnosed from observations (Fig. 5b). Gyakum attributed the formation of this PV anomaly to the latent heating due to convection in the cyclone centre. Also, they regarded the baroclinic forcing of the cyclone as weak and proposed that processes important in tropical cyclones were essential for the intensification of the QE II cyclone. In contrast, Uccellini (1986) emphasised the role of classical baroclinic instability

for the rapid intensification, involving an intense and deep frontal zone interacting with a prominent short-wave trough with a tropopause fold reaching down to below 700 hPa. However, they did not exclude the importance of diabatic processes and concluded that "It seems that explosively developing cyclones like the President's Day and QE II storms are related to the interaction of dynamical and diabatic processes over the entire extent of the troposphere". This controversy illustrates the difficulty in quantifying the relative role of different processes from early diagnostic studies, which was an important aim of

later diagnostic studies and model sensitivity experiments discussed in Sect. 4.2.

Manobianco (1989b) also studied the deepening of the President's Day cyclone (and of two other explosive cyclones that formed off the east coasts of the U.S. and Japan, respectively), using the surface pressure tendency equation (Sect. 5.3.2) to quantify the diabatic contribution to the deepening and data from global model simulations. They assessed the diabatic contribution to the deepening as, at most, 40% for the President's Day cyclone and substantially less for the two other cases

and concluded that "the bomb is fundamentally a baroclinic phenomenon".

### 4.1.2 Further explosive cyclone cases

The diagnostic analysis of the structure and dynamics of explosive cyclones was soon extended beyond the two "catalyst cyclones". Most studies considered cyclones that occurred over the U.S. and the western North Atlantic, where ERICA, a major field experiment in winter 1988/89, triggered a series of innovative and highly influential analyses. Other studies considered

cyclones over the Mediterranean and western Europe. In the following, key studies are presented in approximate date order and collated into themes. In several of the studies, the authors developed innovative diagnostic approaches that are introduced here and expanded upon in Sect. 5.3.

Boyle and Bosart (1986) used gridded fields based on optimum interpolation of observations to study the PV structure of an explosively deepening cyclone over the eastern U.S. in March 1971. They identified a very prominent upper-level stratospheric

PV anomaly down to 600 hPa just upstream of the deepening surface cyclone. The advection of this anomaly is regarded as the main reason for the rapid intensification, confirming the hypothesis by Kleinschmidt (1950a) and in line with the schematic of top-down cyclogenesis induced downstream of an approaching upper-level PV anomaly by Hoskins et al. (1985). However, they also noted the production of high ("stratospheric") values of low-level PV close to the cyclone centre beneath the level of maximum diabatic heating. For another explosively deepening continental U.S. cyclone in March 1984, Gyakum and Barker

(1988) found similar ingredients as for the exceptional QE II cyclone, i.e., deep convection and diabatic production of low-level PV in the cyclone centre. Rogers and Bosart (1991), using analysis fields, compared two intense cyclones along the east coast of North America and diagnosed deep convection near the cyclone centre in one case (developing over warm waters) but not for the second case (over cold waters).

**Early studies on the PV structure of cyclones in Europe.** The ALPEX field experiment in spring 1982 led to intensified research about Alpine lee cyclogenesis. These lee cyclones are typically less rapidly deepening than explosive North Atlantic cyclones. Bleck and Mattocks (1984) used isentropic PV fields calculated from radiosonde profiles and diagnosed upper-level PV advection as a necessary ingredient of lee cyclogenesis. We find it remarkable that they, as well as other authors investigating this type of extratropical cyclones, did not mention potential diabatic effects to cyclone intensification—at a time when these processes were heavily discussed, e.g., in the context of the U.S. catalyst cyclones. Later, Tafferner (1990) simulated a few of the ALPEX lee cyclones with a dry model and concluded that the role of moisture during lee cyclogenesis is not clear since dry simulations were able to describe the process satisfactorily. As an aside, we mention that Bleck and Mattocks (1984) were the first to use the term "PV streamer" to describe the shape of the meridionally elongated narrow PV filaments in the upper troposphere, which typically accompany lee cyclogenesis—a term that later became established to identify Rossby wave breaking on isentropic PV charts (Appenzeller and Davies, 1992; Appenzeller et al., 1996; Massacand et al., 1998). The similar sounding term "PV banner" was introduced by Smith and Smith (1995) and Aebischer and Schär (1998) to denote orographically produced low-tropospheric PV filaments, which are typically smaller and more ephemeral than stratospheric PV streamers.

The study by Hoskins and Berrisford (1988) of the U.K. Great October storm in 1987, based on more extensive diagnostics by Berrisford (1988), was the first case study of an explosively deepening cyclone entirely based on ECMWF analyses. Like the U.S. "catalyst cyclones", the October storm was poorly forecast; it had a devastating effect in parts of the U.K. and France and instigated a period of intensified research on cyclones in the eastern North Atlantic and plans for the international field experiment FASTEX. The U.K. Met Office's National Severe Weather Warning Service was also set up in the wake of this storm in recognition of the need to improve warnings. Hoskins and Berrisford (1988) showed maps of PV in the upper and lower troposphere and vertical sections across the cyclone centre, which revealed PV values exceeding 1 pvu in the lower troposphere, produced by latent heating ahead of the broad upper-level trough. Berrisford (1988) noted that advection could not account for this feature, indicating the relevance of diabatic PV production. This cyclonic low-level PV anomaly was regarded as crucial for the cyclone's intensification and the devastating intensity of the low-level winds. At the time of minimum MSLP, the vertical alignment of the diabatically-produced low-level PV with the upper-level trough resulted in a vertically coherent PV tower with PV values exceeding 1 pvu from the surface to the tropopause (Fig. 6a). The strong contribution of latent heating to the evolution of this storm was later confirmed by the rapid growth rate of moist singular vectors in the vicinity of the cyclone centre (Hoskins and Coutinho, 2005). It is noteworthy that, in their PhD thesis, Berrisford (1988) diagnosed the three-dimensional field of latent heating by condensation from operational analyses of temperature, humidity, and vertical motion, and from there, using Eq. 4, the field of diabatic PV modification due to cloud condensational processes. The same approach to diagnose PV changes from analysis fields was used later in the studies by Wernli and Davies (1997) and Rossa et al. (2000). Also noteworthy is the coincidence that the FRONTS-87 field experiment in the eastern North Atlantic started just two days after the passage of the October storm. Thorpe and Clough (1991) documented diabatically produced low-level PV maxima along cold fronts exceeding 2 pvu during two intense observation periods (IOPs) of FRONTS 87 (Fig. 6b), diagnosed

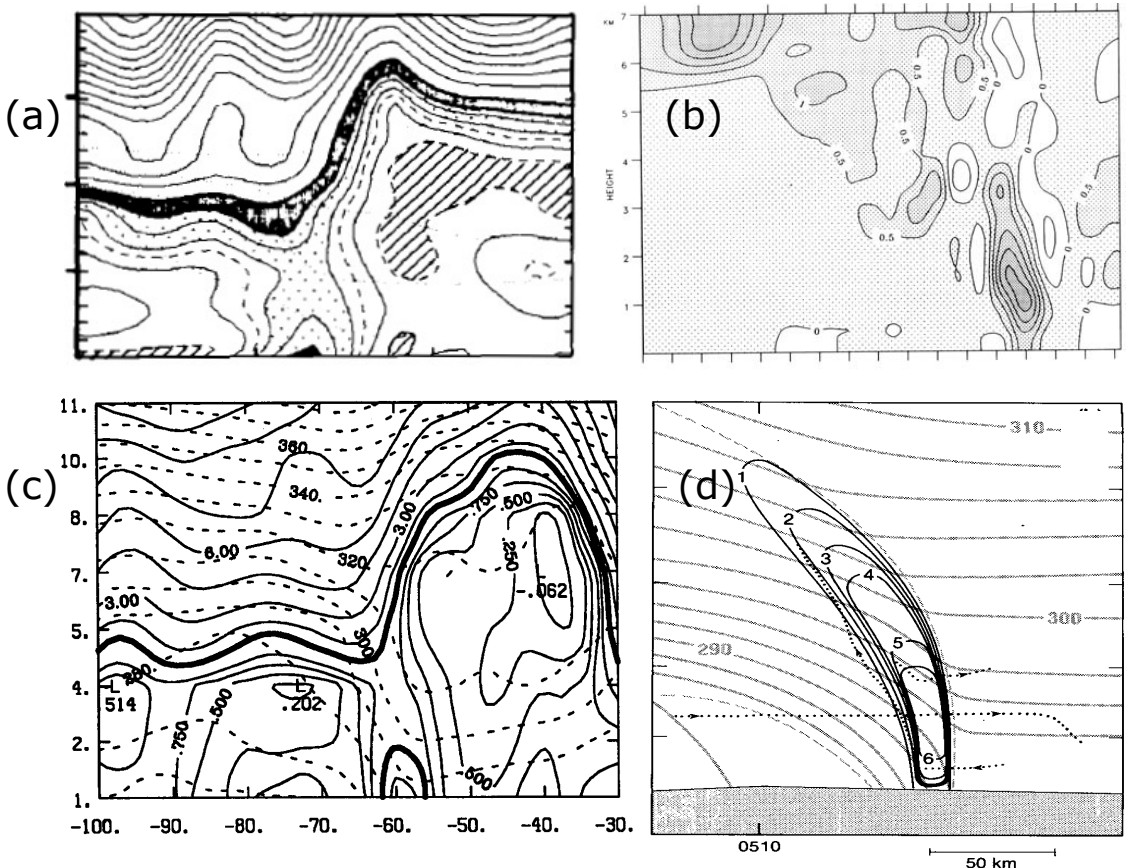

**Figure 6.** Vertical cross sections of PV. **(a)** from Hoskins and Berrisford (1988, their Fig. 4b), across the centre of the U.K. Great October storm (PV values of less than 0.25 pvu are hatched, 1-2 pvu are stippled, and 2-3 pvu are blacked; **(b)** from Thorpe and Clough (1991, their Fig. 13c), across the cold front of FRONTS-87 IOP8 (contour interval 0.5 pvu); **(c)** from Davis and Emanuel (1991, their Fig. 5d), across the centre of a North Atlantic cyclone (isentropes are dashed; PV contours at irregular intervals, 1.5 pvu is bold); and **(d)** from Neiman et al. (1993, their Fig. 12b), across the warm front of the ERICA IOP4 cyclone (isentropes in gray; PV contour interval 1 pvu). All figures used with permission: (a,b) from Wiley, and (c,d) from the American Meteorological Society.

from dropsonde data, and they emphasised that "the PV distribution is specifically valuable as an indicator of the effects of diabatic processes".

**PV inversion, trajectories and model simulations.** Davis and Emanuel (1991) used analysis fields from the U.S. National Meteorological Centre with 2.5° horizontal grid spacing to investigate the PV dynamics of an explosively deepening North American cyclone in February 1988. This cyclone was also characterised by a PV tower at the time of maximum intensity (Fig. 6c), very similar to the results by Hoskins and Berrisford (1988) for the U.K. Great October storm (Fig. 6a). As a particularly novel aspect, Davis and Emanuel (1991) performed a piecewise PV inversion, enabling a quantitative analysis of

the near-surface flow induced by individual PV anomalies. They concluded that "The low-level PV anomaly seems to result

from the condensation of water vapor rather than from advection. This feature grows rapidly, eventually contributing about 40% of the (low-level) cyclonic circulation in the mature storm".

A new level of detailed analysis became possible with the use of data from mesoscale model simulations, pioneered by Whitaker et al. (1988) who studied the President's Day cyclone (because of its focus on PV production along airstreams, this study will be discussed in Sect. 4.1.5). Reed et al. (1992) used the mesoscale model MM4 at 40 km grid spacing and with 15 vertical levels to simulate a U.S. east coast cyclone in February 1987. They emphasised the importance of three interacting anomalies of surface $\theta$, low-level PV, and upper-level PV, each associated with cyclonic flow, whose vertical alignment occurred at the time of maximum storm intensity and led to the formation of a PV tower. The low-level PV was produced by condensational heating in clouds and, as for the two catalyst cyclones, the upper-level PV was advected from upstream. As a specific innovation, they used air parcel trajectories to analyze the origin of the three anomalies, such that this study and Whitaker et al. (1988) are early examples of combining the PV framework with Lagrangian analyses (see Sect. 4.1.5). Reed et al. (1992) included preliminary results from comparing frictional and diabatic PV modifications and concluded that the latter are most likely larger.

The field experiment ERICA provided additional surface observations from drifting buoys in the western North Atlantic, and airborne in-situ observations at low and upper levels. Several explosively deepening cyclones were observed during ERICA, the three most intense ones during IOPs 4, 2, and 5 with deepening rates of 60, 45, and 33 hPa in 24 h, respectively (Reed et al., 1993b). One of these cyclones (IOP5) was not well predicted by the operational models even at short lead times. Reed et al. (1993b) investigated this cyclone with the aid of observations, including dropsonde data, satellite imagery, and precipitation estimates from a microwave sounder, and a mesoscale model hindcast simulation, which well reproduced the 24-h cyclone deepening. The analysis revealed a two-stage development with the formation of initially weak disturbances along the Gulf Stream front in the form of "large cloud masses with embedded convection", followed by the coupling of one of them with a narrow upper-level trough arriving from the continent. Reed et al. (1993a) showed that this low-level disturbance was associated with an initially small but intense, diabatically produced, positive PV anomaly at 850 hPa. The authors noted the transient nature of some of the low-level PV anomalies, which makes it difficult to assess their overall effect on the cyclone's structure and evolution. The strong upper-level PV anomaly, associated with the narrow trough, was considered the main reason for the rapid deepening of the ERICA IOP5 cyclone.

**Observational evidence for PV anomalies along bent-back front.** Neiman and Shapiro (1993) and Neiman et al. (1993) used an impressive set of airborne observations (*in-situ*, dropsondes, and Doppler radar) to present an overview of the synoptic and mesoscale evolution of the ERICA IOP4 cyclone, which showed an exceptional deepening of 60 hPa in 24 h over the warm Gulf Stream. In total, they evaluated observations from nine flights into this cyclone with four aircraft, and portrayed the exceptional depth of the upstream tropopause fold. Figure 6 in Neiman and Shapiro (1993) showed a fold reaching down to 700 hPa beneath an upper-level jet exceeding $80 \, \mathrm{m \, s^{-1}}$, which was instrumental in the rapid deepening of ERICA IOP4. Neiman and Shapiro (1993) then focused mainly on low-level processes and emphasised the unusually intense surface heat and moisture fluxes and their importance for the formation of a prominent warm-core seclusion in the lower troposphere. These fluxes were largest near the bent-back front, which was characterised by wind speeds up to $50 \, \mathrm{m \, s^{-1}}$ at 900 hPa induced by

very intense diabatically produced PV anomalies exceeding 5 pvu along the warm and bent-back front (Neiman et al., 1993, see Fig. 6d). Grønås (1995) also found a strong low-level PV anomaly along the bent-back front of a very intense cyclone affecting Norway in January 1992, contributing to hurricane-force winds of about $60\,\mathrm{m\,s^{-1}}$ at 850 hPa. Neiman et al. (1993) diagnosed the material change of PV due to diabatic heating by studying the intersection of contours of absolute momentum, $m = u - fy$ where $y$ is the cross-front direction, and diabatic heating, $\dot{\theta}$, since, in the two-dimensional approximation the material change of PV can be written as $DQ/Dt = \mathbf{i} \cdot g(\nabla m \times \nabla \dot{\theta})$ (their Eq. 1, where $\mathbf{i}$ is the along-front unit vector). Note that both estimates of latent heating and the resulting diabatic PV changes were entirely based on observations and diagnostics of the front-relative transverse circulation. This multi-faceted, observationally based study also suggested the so-called "escalator-elevator concept" of the WCB ascent at the warm front, based on Doppler radar observations of reflectivity and horizontal front-normal wind speed. This concept refers to alternating regions of slantwise ascent (escalator) and of embedded convective updrafts (elevator) associated with high radar reflectivity of more than 45 dBZ. At the time, this observed mesoscale variability of the type and intensity of diabatic processes could not be assessed with numerical model simulations. Only about 25 years later, the availability of convection-permitting numerical models enabled the re-discovery of the escalator-elevator concept (see Sect. 5.5.2). Neiman et al. (1993) foresaw this development when they wrote "We envision that the combination of spatially and temporally continuous datasets and sophisticated numerical models ... will be able to address the origins and importance of sub-synoptic structures and physical processes within a cyclogenetic environment."

**Diabatic PV tracers.** A next landmark study by Stoelinga (1996) applied a set of sophisticated diagnostics to mesoscale model simulations of an intense western North Atlantic cyclone in February 1987. Following an early implementation by Davis et al. (1993), the study introduced a so-called "partitioned PV integration", i.e., implemented an online model diagnostic that accumulates the PV modification by different diabatic processes during the model integration. This approach was inspirational for the later development and wide application of the PV tracer method at the University of Reading (see Sect. 5.3). In addition, Stoelinga (1996) was one of the first to quantify the role of surface friction for PV modification in extratropical cyclones (as discussed further in Sect. 5.2.1). Results of the PV integration showed that latent heating created a strong positive PV anomaly in the region of the surface warm and bent-back fronts at the level of maximum heating. Stoelinga realised that the levels of maximum latent heating and diabatically produced PV coincide because of a balance between PV modification and vertical advection (Fig. 2c,d):

"... all parcels that ascend through the frontal zone must first travel through a region of PV generation, achieving a maximum value of PV at the level of maximum ascent (and heating), and then spend an approximately equal amount of time losing the previously acquired PV in a similarly shaped region of PV depletion above the level of maximum ascent. In such a situation, a steady-state PV maximum generated by latent heating should exist at the level of maximum ascent (contrasting with the PV maximum generated by instantaneous heating which is located beneath the heating maximum); furthermore, parcels that complete their ascent should have roughly the same PV as they did in the boundary layer".

This link between the PV maximum and latent heating was independently derived by Persson (1995), using dropsonde data and a mesoscale model simulation from an eastern North Atlantic cold front observed during FRONTS-87, and by Wernli

and Davies (1997), based on trajectory calculations with ECMWF analysis wind fields. Inversion of the diagnosed low-level latent-heating-induced positive PV anomaly by Stoelinga (1996), using the method by Davis and Emanuel (1991), showed that it explained approximately 70% of the low-level circulation in the fully developed cyclone. Surface friction was found to also generate PV leading to mainly positive anomalies at low levels, particularly in the easterly warm frontal flow. Although inversion of these positive anomalies also yielded a weak low-level cyclonic circulation that would be expected to enhance 785 coupling, indirect effects yielded a stronger surface cyclone in a sensitivity experiment without surface friction.

**Moist symmetric instability.** The above studies of explosive cyclogenesis have in common low-level PV anomalies attributed to latent heating from condensational heating in clouds. While the diagnosis of convective-type instabilities are typically not a part of these studies, some studies have specifically discussed the possible importance of the release of MSI in such explosive systems. Here we distinguish these studies from other studies that have discussed the release of moist symmetric 790 instabilities as a possible explanation for frontal rainbands or snowbands (as first proposed by Bennetts and Hoskins, 1979) without linking this process to cyclone development. Studies attributing precipitation bands to moist instability release were critically reviewed by Schultz and Schumacher (1999, see Sect. 6 particularly) in terms of the criteria used for their diagnosis.

Reed and Albright (1986), in their analysis (using analyses and satellite imagery) of an explosively developing eastern Pacific storm from November 1981, noted that "symmetric instability can appear suddenly with the formation of the frontal cloud just 795 as in severe local storm situations convective instability appears suddenly when potentially unstable air is lifted and brought to saturation". They speculated that the rapidity with which this symmetric instability develops could be a crucial factor in the explosive deepening of cyclones, although in their case small effective static stability (defined as weak or neutral moist static stability in the cloud and with values less than normal dry static stability in the environment) together with strong deep baroclinic forcing was diagnosed as the reason for the explosive development with enhancement from strong latent heat release. 800 The presence of symmetric instability was noted as conducive to frontal uplift and an associated rapid generation of cyclonic vorticity near the cyclone centre, but the importance of its release in the explosive development was unclear because there was also satellite evidence of deep convection, consistent with convective instability (the lifted index became negative prior to rapid intensification) with destabilisation from surface fluxes. It should be noted that in this paper, as in many others discussed in this section, the authors considered the relative vertical slopes of isopleths of absolute momentum and $\theta_e$ when diagnosing 805 the presence of instability. The reason that they are able to diagnose CSI, rather than potential symmetric instability, is that they are considering the saturated frontal cloud region. As discussed further in Sect. 4.2, Kuo and Reed (1988) performed a set of numerical mesoscale model experiments (40 and 80 km grid spacing) on the same storm with one of the goals being to examine the effects of physical processes on the storm development. In their control simulation they found that the warm frontal cloud band was symmetrically neutral or slightly unstable with a narrow, sloping sheet of rapidly ascending air at the frontal 810 boundary (consistent with structures found in the theoretical analysis of Emanuel et al. (1987) described below). This ascent produced strong vertical stretching of the boundary layer, leading to spin-up of low-level vorticity. These authors concluded that "symmetric instability (or neutrality) may be an important factor in rapid cyclogenesis". As indicated in the quotation above, it is important to realise that an environment tending to moist symmetric neutrality can favour cyclone intensification even if the stability does not subsequently tip over into moist symmetric instability (or if the atmosphere is maintained in its

near neutral state by the constant release of the instability). This result was demonstrated by Emanuel et al. (1987) by solving a two-dimensional two-layer semigeostrophic model analytically for the fastest growing baroclinic wave in the limit of zero moist PV (see Sect. 2.2). These authors also noted that although latent heating from condensation can increase baroclinic growth rates, they did not believe that this process alone can lead to explosive cyclogenesis.

The first evaluation of CSI in an explosively developing storm through calculation of maps of the diagnostic SCAPE using
numerical model output was performed by Shutts (1990b) for the U.K. Great October storm, using the method described in Shutts (1990a). The air ahead of the main surface front was found to contain substantial SCAPE, and slantwise ascent followed by horizontal spreading of the outflow generated the broad cloud head in this storm. The role of CSI release in the formation of the cloud head is pertinent because it had been observed to be a precursor of intense storms (Böttger et al., 1975). Shutts concluded that the "small scale and vigour" of storms such as the Great October storm and the Pacific storm studied by both
Reed and Albright (1986) and Kuo and Reed (1988) may be associated with the organised release of SCAPE and that maps of SCAPE may indicate the likelihood of cyclone development.

Corroborating evidence for conditional symmetric neutrality (and sometimes even weak instability) in the vicinity of the warm front as a generic feature of explosive cyclogenesis came from several other modelling studies performed at this time: a case from the CASP by Mailhot and Chouinard (1989), the nine cases over the western Atlantic analysed by Kuo and Low-
Nam (1990), and a study of the QE II storm by Kuo et al. (1991b). As for the Great October storm, strong CSI was found for some cases including in one of the modelling studies of the ERICA IOP5 storm (Reed et al., 1993a), observational analysis of four other ERICA cyclones (IOPs 2, 4, 7 and 8; Reuter and Yau, 1993) and a modelling study of an explosive cyclone that hit Iceland in February 1991 named the Greenhouse Low[10] (Kristjánsson and Thorsteinsson, 1995). In the Reuter and Yau study, dropsonde observations revealed that the lower-tropospheric air on the warm side of the warm frontal zone was stable
or neutral to convective instability but had CSI in all four ERICA cyclones, including a deep layer of CSI in one cyclone that did not develop explosively (note that, in contrast, convective instability was found in the warm sector near the surface low for the two explosive cyclones during their rapid growth period). These authors also noted the importance of considering water loading in the assessment of CSI.

### 4.1.3   Frontal wave cyclones

Interest in secondary cyclones developing from frontal waves goes back to the Bergen school and the hypothesis that instabilities on pre-existing frontal zones can grow into extratropical cyclones that are typically smaller in scale (occurring on the mesoscale: 100–1000 km) and faster developing (within ∼2 days) than those explained by dry baroclinic instability theory (Eady, 1949). In the last two decades of the 20[th] century, these secondary frontal waves became a major focus of dynamical meteorology for three main reasons: (i) the need to develop a fundamental understanding of the phenomenon more than half
a century after the observation-based hypothesis by Bjerknes and Solberg (1922); (ii) the realisation that cyclones of this type can be extremely damaging, as shown by the U.K. Great October storm, but are often poorly forecast, with perhaps only about

---

[10]It was called the Greenhouse Low because of widespread damage to greenhouses during this event.

half of identified waves developing significantly (Parker, 1998); and (iii) the joint planning of the field experiment FASTEX by scientists from the U.K., France, and the U.S., to obtain detailed observations of frontal waves.

A breakthrough in the understanding of frontal waves came with the diagnostic study of the U.K. Great October storm by Hoskins and Berrisford (1988) and idealised numerical instability studies (Thorpe and Emanuel, 1985; Joly and Thorpe, 1990; Schär and Davies, 1990; Malardel et al., 1993). These idealised numerical instability studies will be discussed in more detail in Sect. 4.3.4 together with case studies from which thresholds for environmental strain inhibiting growth were diagnosed for comparison with values derived from idealised numerical experiments. These studies revealed the essential role of diabatically produced low-level bands of PV or surface $\theta$ as a prerequisite for frontal wave instability to occur; similarly, the earlier Presidents' Day storm study by Bosart (1981) noted the importance of the development of a low-level vorticity strip for subsequent rapid cyclogenesis. However, it was unclear whether latent heating is only important to provide the precursor low-level PV band, whose instability then can be regarded as an essentially dry dynamical problem, or whether it was also relevant for the instability process *per se*, as summarised in the review paper on the understanding of frontal wave dynamics by Parker (1998), written before results from the FASTEX field campaign were available.

In contrast to this low-level instability viewpoint, where diabatic processes are at least important for producing the precursor PV band, Thorncroft and Hoskins (1990) described an idealised modelling study of a finite-amplitude dry dynamical mechanism, where frontal waves emerge when an upper-level PV anomaly overrides a surface front. Ayrault et al. (1995) revealed the potential importance of both dry and moist processes through the first climatological analysis of observed frontal waves. Two types of frontal waves were distinguished. Type 1 waves are a baroclinic development (though with a barotropic contribution) with well-defined precursors forming along a well-defined cold front, in a weak background deformation field, on a smaller scale than for primary cyclogenesis; the smaller scale is attributed to either moist processes or a frontal instability mechanism. Type 2 waves develop spontaneously (without clear precursors) in a broadening baroclinic region with a frontolytic or diffluent deformation field and are dominated by their warm front. The dry dynamical mechanism was also identified in the diagnostic study of real case events by Appenzeller et al. (1996), based on ECMWF analysis and forecast fields. They found evidence of both the self-development of the low-level PV (e.g., break-up of the PV band into distinct elements) and mutual interactions of low-level and upper-level PV anomalies. This led to eight different hypotheses for frontal wave development (three of them are shown in Fig. 7a), with a variable role attributed to diabatic processes for both the initial wave perturbation and its subsequent growth. Regarding the numerical prediction of frontal waves, this study also emphasised the importance for models to accurately resolve the narrow width of the low-level PV band, since the instability is related to the PV gradient.

Fehlmann and Davies (1997) explicitly diagnosed the latent heating and associated vertical dipole of diabatic PV production and destruction for the low-level PV band associated with the early stage of a frontal-wave cyclone (Fig. 7b). Compared to strongly deepening larger-scale cyclones, the latent heating is comparatively shallow such that PV peaks at about 850 hPa, at a value of 2–3 pvu, and maximum positive and negative PV tendencies occur at about 950 and 700 hPa, respectively. A detailed synoptic analysis of the same case using model analyses and observational imagery determined that the cyclone developed at the northern end of the low-level PV band, which was collocated with the cyclone's WCB (Browning and Roberts, 1994). Similar vertical sections of PV, diabatic heating rates and the associated diabatic PV tendencies across the warm front in the

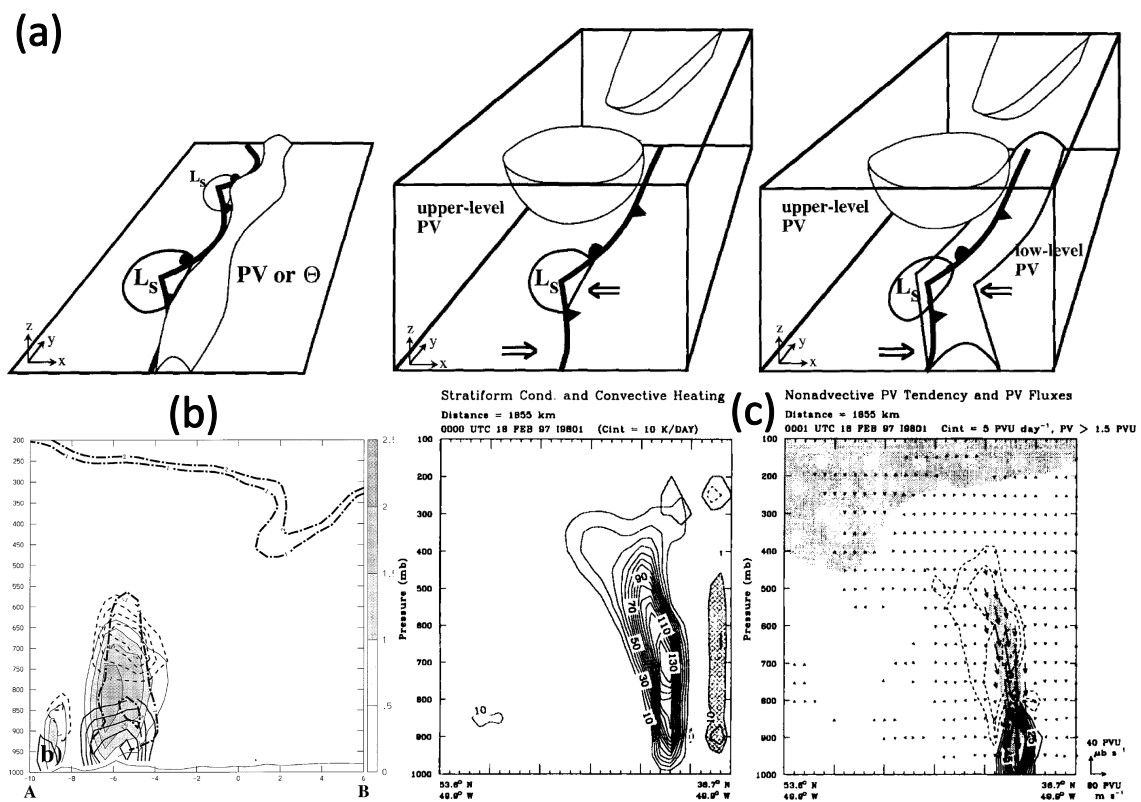

**Figure 7.** Low-level PV in frontal waves. **(a)** from Appenzeller and Davies (1996, their Fig. 9), schematic of three PV hypotheses of frontal wave instability, from left to right: low-level instability of a PV band associated with the surface front, an upper-level PV cutoff triggering a surface wave, and an upper-level PV anomaly interacting with the low-level PV band; **(b)** from Fehlmann and Davies (1997, their Fig. 6b), vertical section across incipient frontal wave in the eastern North Atlantic at 12 UTC on 13 January 1993 with diabatic heating rate (shading, in $K\,h^{-1}$), diabatic PV rate (solid and dashed lines every 0.1 pvu $h^{-1}$ for positive and negative values, respectively), and PV (dash-dotted lines for 1 and 2 pvu); and **(c)** from Mallet et al. (1999b, their Fig. 8a,d), vertical sections across cold front of FASTEX IOP17 at 00 UTC on 18 February 1997 of (left) diabatic heating (every 10 $K\,d^{-1}$) and (right) associated diabatic PV tendencies (every 5 pvu $d^{-1}$, dashed contours for negative values). All figures used with permission: (a) from Springer Nature, (b) from the American Meteorological Society, and (c) from Wiley.

early phase of the FASTEX IOP17 cyclone by Mallet et al. (1999b) also showed maximum PV values exceeding 2 pvu at about 800 hPa (Fig. 7c).

### 4.1.4   Statistical studies of explosive cyclones

To go beyond individual case studies and to identify typical and representative structural features of cyclones, composite studies of cyclones were first performed by Rogers and Bosart (1986), using rawinsonde data, and by Sinclair and Elsberry (1986), using ECMWF analyses. The latter study is about polar lows, which will be discussed in Sect. 5.3.4. Rogers and Bosart

(1986) composited 30 explosively deepening cyclones in the North Atlantic and North Pacific and concluded that "[they] are baroclinic phenomena whose development may be enhanced in some cases by the bulk effects of cumulus convection near the cyclone centre". For one of their cases, soundings launched during the rapid deepening revealed a conditionally unstable layer extending from the surface to 750 hPa ahead of the cyclone (their Fig. 9), indicating the possibility of convection if ascent occurs. Manobianco (1989a), using ECMWF analyses, composited 24 explosive U.S. east coast cyclones. The averaged fields revealed strong upper-level forcing during intensification, which was enhanced by strongly reduced static stability in the low troposphere, likely due to surface fluxes prior to cyclone development. Composite cross-sections of PV indicated the consistently low tropopause above explosive cyclones but did not show any diabatically produced PV signals in the lower troposphere, potentially because of the low resolution of the data (2.5° horizontal grid spacing, 7 vertical levels) and due to substantial case-to-case variability. Also using ECMWF analyses, Sinclair and Cong (1992) studied the composites of 30 cyclones in the Australasian region and emphasised the role of latent heating in small-scale cumuli during the early stages of the cyclone lifecycle, and of heating due to large-scale ascent later in the development. The composite of the most intense cyclones showed stronger surface heating than for weaker cases. McMurdie and Katsaros (1996)[11] analysed the deepening of 23 North Atlantic cyclones and the associated surface precipitation and vertically integrated water vapour as observed by the satellite based SSMI microwave imager. Their results indicate that, although in general more rapidly deepening cyclones have higher water vapour anomalies, rapid deepening can also occur without these and, occasionally, cyclones have high water vapour anomalies yet do not deepen rapidly. The relatively small number of events included in these early statistical analyses may reflect that, at the time, downloading and archiving large datasets from meteorological centres was not straightforward—which changed substantially with the advent of reanalysis datasets in the late 1990s (Sect. 5.1).

### 4.1.5 Lagrangian view and coherent airstreams

Early research on (moist) airstreams has been introduced in Sect. 3.2, together with the diagnostic methods of isentropic flow analysis and the explicit calculation of air parcel trajectories. With the advent of six-hourly global analysis data and high-temporal resolution output from numerical models, the latter approach to calculate trajectories became technically straightforward and increasingly popular to address Lagrangian aspects of flow dynamics. Trajectories enable, for instance, identifying the "origin" of air parcels that constitute a PV anomaly of interest, or assessing the coherency of a specific atmospheric flow. Trajectories are purely diagnostic; their computation requires four-dimensional wind fields with sufficient spatial and temporal resolution, and their accuracy depends much more on the quality of the underlying wind fields than on the details of the trajectory algorithm (Stohl et al., 2001). In particular, the temporal resolution of the wind fields—which has been typically 6 h and 1 h when using reanalysis data and mesoscale model output, respectively—is a limiting factor for trajectory accuracy, as already discussed by Reed et al. (1992). Also, important for the interpretation of trajectories calculated from an early to a later time is that their classical visualisation, e.g., in the form of lines on map, is not instantaneous, which must be kept in mind when

---

[11]This is the earliest paper with a female first author referenced in this review article. Lynn McMurdie did her PhD in 1989 at the University of Washington, U.S., where she has been a research associate professor since 2017.

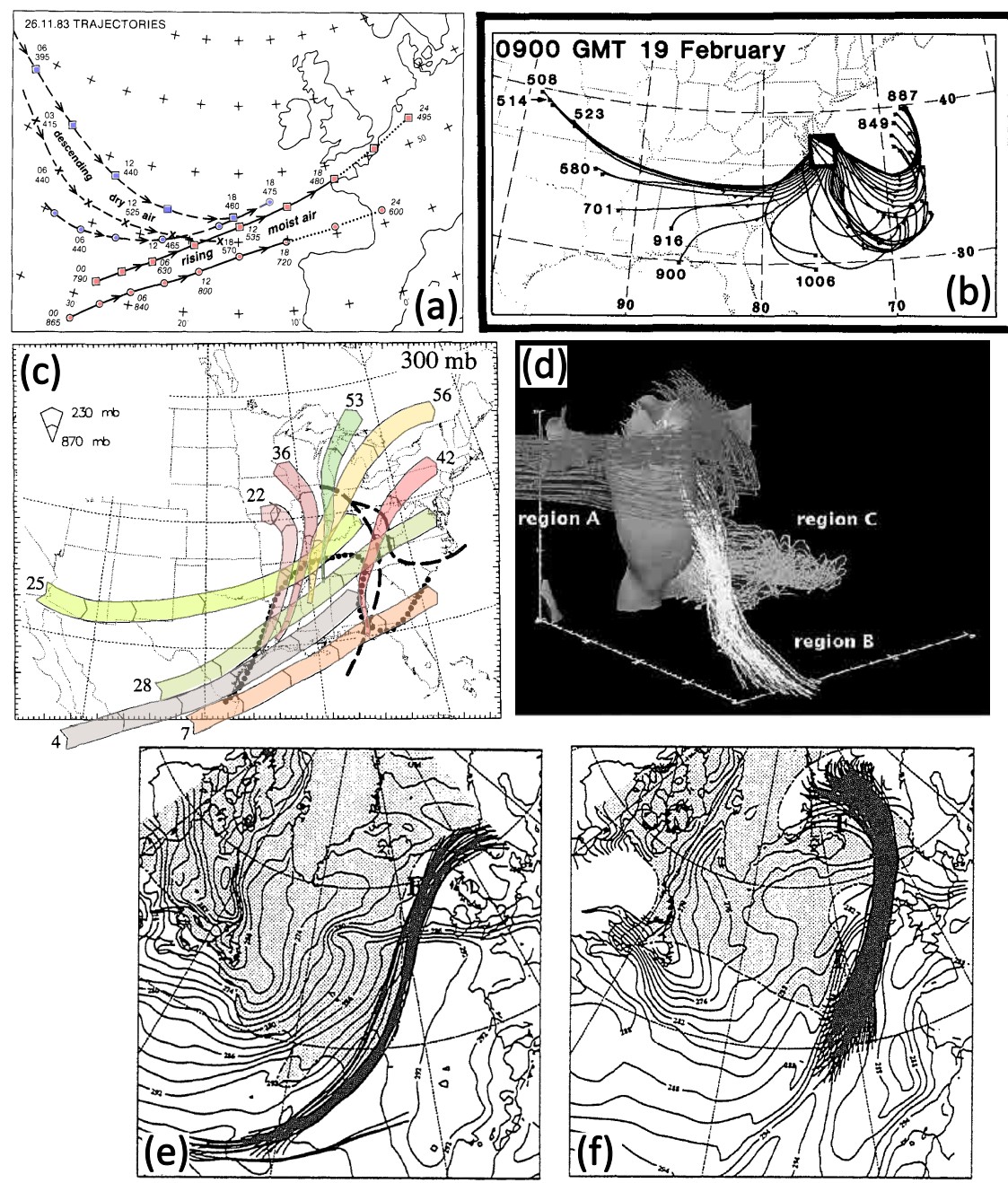

.

combining Eulerian and Lagrangian information in the same figure. This subsection summarises research from 1980–2000 on using trajectories to investigate (moist) airstreams in extratropical cyclones.

**Figure 8** *(previous page)*. Trajectories and airstreams. **(a)** from Young et al. (1987, their Fig. 8), manually calculated backward trajectories in a North Atlantic cyclone from 18 UTC on 26 November 1983, which (solid) ascended into the leaf cloud (forward extensions shown by dotted lines) and (dashed) descended to the rear of the leaf cloud (time and pressure, in hPa, are indicated every 6 h); **(b)** from Whitaker et al. (1988, their Fig. 16d), 24-h backward trajectories from centre of President's Day cyclone on 700 hPa at 12 UTC on 19 February 1979 (labels denote pressure in hPa at 12 UTC on 18 February); **(c)** from Mass and Schultz (1993, their Fig. 18d), 15-h backward trajectories from region around U.S. cyclone on 300 hPa (surface fronts at two times are shown by dashed and dotted lines; pressure is indicated by the width of the trajectories); **(d)** from Rossa et al. (2000, their Fig. 6), three-dimensional perspective of the PV tower (defined as the 1-pvu surface) from the south-west at 18 UTC on 23 November 1992; thin lines show a representative selection of 48-h backward trajectories from the PV tower; and **(e,f)** from Wernli (1997, their Fig.12b,e), WCB trajectories identified in the same cyclone as shown in (d) that start their ascent at 06 UTC on 21 November and 12 UTC on 22 November, respectively; dash-dotted lines show $\theta$ at the level of the WCB at the starting time and the shading shows regions with PV larger than 2 pvu at the level of the WCB outflow 42 h after the starting time. Colour has been added to (a,c) to enhance clarity. Note that (e,f) were taken from the first author's doctoral thesis as the reproductions there are better quality than in the paper. All figures used with permission: (a,e,f) from Wiley, (b,c) from the American Meteorological Society, and (d) from Springer Nature.

Isentropic trajectories were helpful to diagnose the descent of high-PV stratospheric air towards the centre of extratropical cyclones and their contribution to the upper part of the cyclone's PV tower. Early examples of manually constructed individual isentropic trajectories are from Boyle and Bosart (1986), who found fairly good PV conservation along the descent of stratospheric air to the 600 hPa level, and Young et al. (1987), who diagnosed the contrasting airflows of ascending air within a so-called "leaf cloud" in the early period of cyclogenesis and descending air behind it (Fig. 8a). One of the trajectories ascended by more than 300 hPa in 18 h, which was interpreted as moist slantwise ascent within the WCB. The authors also discussed the differences between the trajectory approach and the isentropic relative flow analysis, which assumes the system to be in steady state (Sect. 3.2). The descending trajectories were part of the dry intrusion and showed diffluence at lower levels, leading to a "hammer-head shape" at the forward part of the dry intrusion that marks the sharp western edge of the leaf cloud. Since only very few trajectories were calculated, no consideration was possible of the coherency of the identified flows.

A new level of trajectory analysis was achieved in the study by Whitaker et al. (1988) on the President's Day storm—see also the three-dimensional visualisations of these trajectories in Uccellini (1990). They used wind fields from a mesoscale model simulation and an automated trajectory algorithm to investigate the interaction of a tropopause fold with a low-level PV anomaly. Backward trajectories from the cyclone centre on the 700 hPa pressure level at different times of the lifecycle showed a transition from early ascending air from the oceanic boundary layer to later mainly descending air from the upper-level fold (Fig. 8b). The authors subjectively attributed their cyclone-centre trajectories to the dry airstream or to the cold or warm conveyor belt and discussed in detail the evolution of key variables along these trajectories, including $\theta$, PV, absolute vorticity, and vorticity stretching and tilting terms. The dry airstream trajectory showed a descent of almost 300 hPa in 24 h, again fairly good conservation of PV, and substantial vorticity stretching at low levels. The WCB trajectory rapidly ascended by more than 600 hPa in 12 h, and experienced strong latent heating by about 15 K and a decrease of specific humidity by more than 10 g kg$^{-1}$ due to condensation. PV values showed a rapid increase at low levels followed by a gradual decrease. This PV

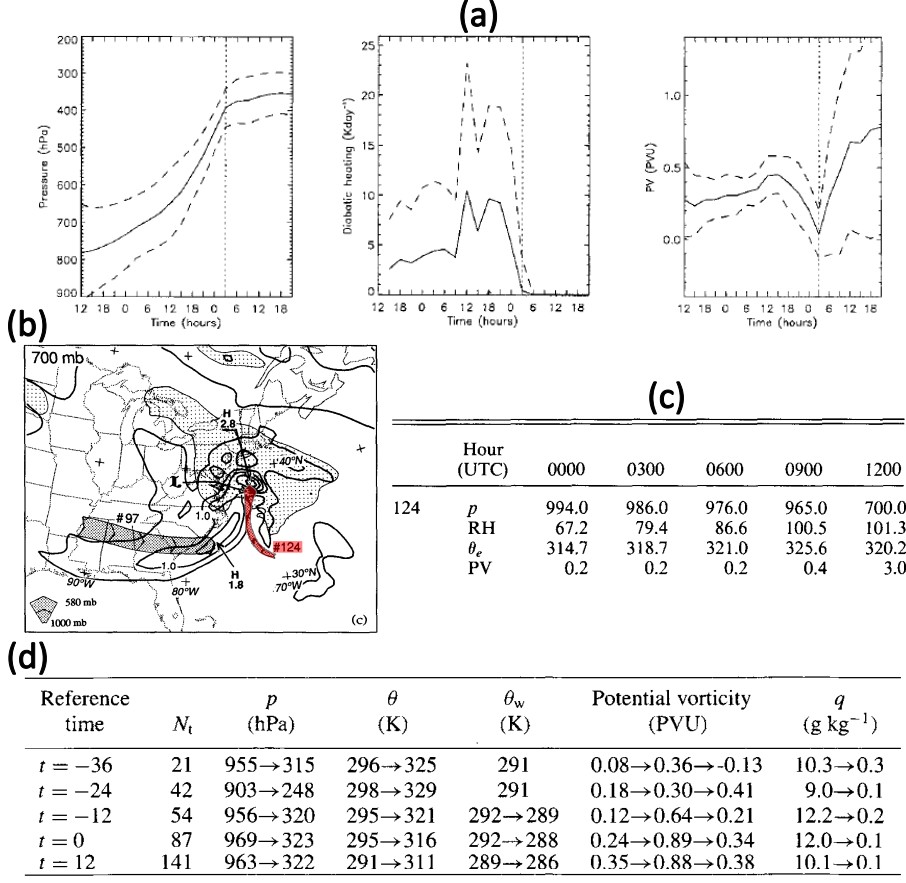

**Figure 9.** Evolution of atmospheric variables along trajectories and airstreams. **(a)** from Pomroy and Thorpe (2000, their Fig. 3a,d,e (left to right)), pressure, diabatic heating and PV of trajectories passing through a region of reduced upper-level PV caused by latent heating (forward and backward trajectories from the time indicated by the dotted line; solid and dashed lines show mean and one standard deviation); **(b)** from Reed et al. (1992, their Fig. 6c), fields of 700 hPa PV (solid contours, PVU), relative humidity (> 90% shaded) and overlaid by the two trajectories (including the most rapidly ascending one, #124; in contrast #97 descends by over 100 hPa over the same period) with the width of the trajectories proportional to pressure level (scale bottom left) and the surface low centre marked by "L"; **(c)** from Reed et al. (1992, their Table 4), characteristics of a specific trajectory (#124, path shown in panel (c)) at three-hourly intervals; **(d)** from Wernli (1997, their Table 5), Lagrangian characteristics of the ensembles of WCB trajectories shown in Fig. 8(e,f) for different starting times (where $t = 0$ is when the cyclone first appeared) showing the number of trajectories, $N_t$, and the mean values of the each characteristic at the initial and final times of the 48-hour period (with additionally the mean value in the middle of the time period for PV). Colour has been added to (b) and the rows of data have been thinned in (c,d) to enhance clarity. All figures used with permission: (a-c) from the American Meteorological Society, and (d) from Wiley.

increase in the WCB was interpreted as the process responsible for the formation of the low-level PV anomaly. Importantly, the fact that the WCB trajectories gain and lose PV while ascending through the low-level PV anomalies implies that these

anomalies are continuously reproduced by latent heating in newly ascending trajectories (see also Fig. 7a in the later study by
Wernli and Davies, 1997). Clearly, the pioneering trajectory study by Whitaker et al. (1988) illustrated the great potential of Lagrangian analyses and revealed several results that were later confirmed for other cyclones and with refined approaches.

     Studying an U.S. east coast cyclone in February 1987 based on output from a mesoscale model simulation, Reed et al. (1992) used trajectories to investigate the origin and evolution of the $\theta$ and PV anomalies in the cyclone centre. The surface warm anomaly resulted primarily from poleward advection with a minor contribution from surface fluxes. Similar to the study
mentioned in the previous paragraph, the low-level positive PV anomaly formed in moist and rapidly ascending air parcels. For selected trajectories, they found a rapid increase of PV from 0.1 to more than 4 pvu in 3 h (their Table 4; Fig. 9b,c here show the trajectory path and along-trajectory characteristics, respectively, of a closely related trajectory from this paper, selected as that experiencing the greatest ascent during the time period in the Table), and a subsequent decrease to typical tropospheric values. They noted that, in order for the cyclone to be intensified by this diabatic PV production, the trajectory must ascend close to
the cyclone centre. A systematic trajectory-based analysis of a PV tower in the mature phase of a North Atlantic cyclone was presented by Rossa et al. (2000), who considered the PV evolution along backward trajectories from 100 hPa layers of grid points near the cyclone centre with PV > 1 pvu from 900 to 300 hPa. They showed that the PV tower was built by the vertical superposition of three elements (Fig. 8d): (i) stratospheric air in the upper part of the PV tower that descended adiabatically within an upper-level trough; (ii) a mid-tropospheric layer mainly originating from the cyclone's cold front and associated with
diabatic PV production during its ascent from about 800 to 550 hPa; and (iii) a low-tropospheric layer of diabatically-produced PV that originated from the flow along the warm and bent-back front.

     At the same time, the WCB concept was further developed in the U.K., primarily for the interpretation of cloud structures in extratropical cyclones (Bader et al., 1995) and particularly so-called "cloud heads" near the centre of developing frontal waves (Browning and Roberts, 1994). The latter study referred to WCB branches that ascend close to the cyclone centre and
are responsible for the cloud head formation as type W2. (In contrast, W1 refers to WCB branches that ascend further east over the warm front and typically turn anticyclonically at upper levels.) They concluded that airstream W2 was related to high PV at 900 hPa, associated with convective clouds and diabatic production of PV, qualitatively in line with the trajectory interpretation by Reed et al. (1992). Calculating trajectories for a simulated summer cyclone in northern Germany associated with localised heavy precipitation, Grønås et al. (1994) found that rapidly ascending trajectories, referred to as the WCB, were
responsible for the most intense precipitation. However, not all studies investigating trajectories in numerical simulations of extratropical cyclones readily adopted the WCB concept. Mass and Schultz (1993) investigated backward trajectories from selected starting points at different levels in an intense cyclone over the eastern U.S., and these trajectories revealed complex flow patterns rather than coherent airstreams as featured by the conveyor belt model. In particular the authors emphasised that trajectories ascending from the warm sector fan out at upper levels with some turning cyclonically and others anticyclonically
(Fig. 8c). They concluded that the conveyor belt model provided a coarse description of the real flow in extratropical cyclones. In hindsight, this controversy suffered from the schematic depiction of conveyor belts in early studies (which does not do justice to cyclone variability), from drawing conclusions based on results from few cases only, and from a rather subjective selection of starting points in the trajectory-based studies.

Schär and Wernli (1993) and Wernli and Davies (1997) introduced an alternative approach to potentially identify coherent, conveyor-belt-like airstreams in extratropical cyclones. Their method was based on a two-step procedure with the first step being the calculation of a large amount of typically two-day forward trajectories from a dense regular grid in a region encompassing the lifecycle of the considered cyclone (e.g., the entire North Atlantic). This brute-force approach avoids a subjective pre-selection of relevant trajectories. In the second step, a geometric or physical selection criterion is applied and only those trajectories are retained that fulfil this criterion. For instance, inspired by the notion of WCBs as the maximum precipitation-producing airstreams in extratropical cyclones (Browning, 1990), Wernli and Davies (1997) showed that coherently ascending airstreams from the oceanic boundary layer to the upper troposphere can be identified using any of the three selection criteria of maximum ascent, maximum increase in $\theta$, or maximum decrease of specific humidity (along the two-day trajectories). These criteria therefore serve to objectively identify coherent airstreams in extratropical cyclones that are akin to WCBs. At the same time, with this method, it is possible to investigate the evolution of WCBs during different stages of the cyclone lifecycle. Wernli (1997) showed how the WCB, identified with the criterion of ascent exceeding 620 hPa in 48 h and referred to as a "moist ascending coherent ensemble of trajectories", moves poleward with the evolution of the cyclone and transitions from a purely anticyclonically curved outflow in the upper troposphere to a strong fanning-out during maximum cyclone intensification (cf. the results of Mass and Schultz, 1993), and eventually to a cyclonically curved outflow at later times (Fig. 8e,f, with associated along-trajectory characteristics in Fig. 9d showing the increase and subsequent decrease of PV as the air ascends and dries). These studies have shown that, using trajectories, WCB-like, strongly poleward-moving airstreams can be meaningfully identified in many case studies of extratropical cyclones and that their coherent phase is limited to the rapid ascent (with a much less coherent inflow at lower and outflow at upper levels). In terms of physical characteristics, they experience strong latent heating of about 20 K in two days mainly due to condensation, they produce clouds and intense precipitation during their ascent, and they have a characteristic PV evolution related to the intense heating. This PV evolution is such that PV first increases from typical low-tropospheric values to often more than 1 pvu (e.g., 1.5 pvu in one of the examples in Wernli, 1997) followed by a decrease to low values ($< 0.5$ pvu) in the outflow at about 300 hPa, which are similar to the values in the WCB inflow at about 950 hPa. However, since these low-PV outflows typically occur at high latitudes (e.g., 50–70°N in the case study mentioned above) in regions that are climatologically already in the stratosphere, these PV values constitute intense negative PV anomalies. The significance of this result was not yet fully appreciated in these studies. Wernli (1997) merely noted that the PV values in the WCB outflow were anomalously low and coincided with intensifying upper-level ridges. However, Pomroy and Thorpe (2000) picked up on this theme, noting that "... very little research has been undertaken on the existence and role of diabatically produced negative PV anomalies". They used a mesoscale model simulation to consider low-PV features at 300 hPa that were observed during FASTEX (with some values showing even absolute negative PV) and, based on backward trajectories from the anomalies, also concluded that the negative anomalies formed due to latent heating (Fig.9a shows the along-trajectory pressure, latent heating and PV of trajectories passing through such a PV region). Other studies that strongly emphasised the role of diabatically-induced negative upper-level PV anomalies for ridge amplification, Rossby wave breaking, and the formation of blocks are summarised in the next subsection and in Sect. 5.5.

### 4.1.6 Diabatic effects in the upper troposphere

The effects of latent heating on the upper-level flow and, in particular, on the structure of the tropopause, were considered, in early diagnostic and modelling studies of STE (as already mentioned in Sect. 3.3). The innovative study by Price and Vaughan (1993) used ECMWF analysis fields (available at the time with 2.5° grid spacing on six pressure levels) in combination with total ozone satellite observations and qualitatively identified different mechanisms leading to STE in upper-level cutoff lows. The mechanisms considered important were diabatic PV erosion by convective heating and turbulence, while radiative processes were estimated to be weaker. The detailed modelling study of STE in a cyclone over the south-eastern U.S. by Lamarque and Hess (1994) identified latent heating, and in particular the negative vertical gradient of diabatic heating below the tropopause, as essential for near-tropopause PV modification leading to cross-tropopause exchange (Fig. 10a). Like Price and Vaughan (1993), they referred to this process as convective erosion and mentioned that "the heating rate gradient at tropopause level is mainly due to latent heat release in the clouds [which slightly penetrated into the lower stratosphere above the warm sector of the cyclone] and infrared cooling at the cloud top". Investigating STE in a decaying upper-level PV cutoff over the western Mediterranean in a simulation with the ECMWF model, Wirth and Egger (1999) identified direct PV erosion at the tropopause by latent heating in deep convection as the most important diabatic process, in line with an earlier idealised study of a diabatically eroded PV cutoff (Wirth, 1995) and the real case study of Lamarque and Hess (1994). In contrast, Gouget et al. (2000), studying another PV cutoff that moved into the western Mediterranean, quantified that filamentation and eventually three-dimensional mixing were even more relevant for the decay of the system than latent heating due to convection. The filamentation could not be identified in the relatively coarse analysis fields but was diagnosed with the aid of airborne observations of ozone and humidity. Other idealised model studies by Zierl and Wirth (1997) and Forster and Wirth (2000) focused on radiative PV modification near the tropopause and the associated STE, considering upper-level anticyclones and stratospheric filaments, respectively. In all these studies, vertical gradients in diabatic heating, either due to cloud physics or radiation, can affect the dynamical tropopause if a substantially strong heating gradient extends across the 2-pvu iso-surface. Although these studies were primarily interested in STE, their results are also clear indications that different diabatic processes (latent heating in convection, radiation) can, in addition to filamentation and turbulence, affect tropopause dynamics.

Other studies addressing diabatic effects at upper levels did not focus specifically on STE. Chen et al. (1983) compared vertical sections across a cyclone in moist and dry simulations and noted that latent heating affected the orientation of the upper-level trough. Holt and Chang (1993) mentioned that latent heat release due to cumulus convection influenced the alignment of the trough with the surface cyclone by retarding its eastward propagation, a behaviour that was later used as an identifying characteristic of a new category of cyclone, so-called type-C cyclones (Deveson et al., 2002) (see Sect. 5.3.1). Studying the President's Day cyclone, Atlas (1987) was most likely one of the first to infer the important role of diabatic heating for the intensification of the upper-level ridge downstream of the cyclone. Hoskins and Berrisford (1988), in their study about the U.K. Great October storm, which developed beneath a very intense upper-level jet associated with a prominent meridional "tropopause jump" (Fig. 6a), mentioned the advection of the (diabatically influenced) low PV outflow of tropical cyclone Floyd towards the eastern North Atlantic and its importance for the formation of the strong isentropic PV gradient in the

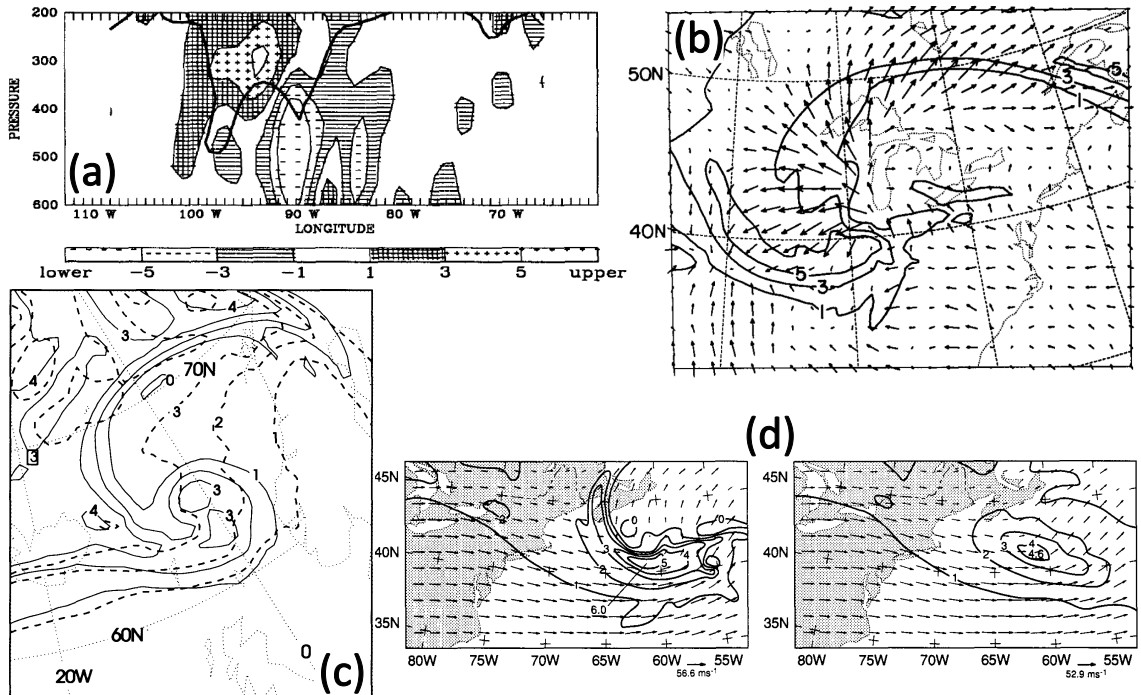

**Figure 10.** Diabatic effects near the tropopause. **(a)** from Lamarque and Hess (1994, their Fig. 6d), vertical section across upper-level PV anomaly and tropopause fold with 2-pvu tropopause (bold line) and diabatic PV tendencies (shading, in units of $0.1\,\mathrm{pvu\,d^{-1}}$; **(b)** from Davis et al. (1993, their Fig. 18a), irrotational horizontal wind vectors and PV contours (1, 3 and 5 pvu) in moist simulation; **(c)** from Grønås (1995, their Fig. 6), PV on 300 K in moist control simulation (solid lines) and dry sensitivity experiment (dashed contours); and **(d)** from Reed et al. (1993b, their Fig. 6e,f), PV at 400 hPa from (left) moist and (right) dry simulations of the ERICA IOP5 cyclone. All figures used with permission: (a,b,d) from the American Meteorological Society.

region of the storm that later severely affected the U.K. In the full-physics simulation of the ERICA IOP5 cyclone, but not in the dry run, Reed et al. (1993a) found anomalously low PV in the upper cloud shield. The upper-level PV pattern at the time of the mature cyclone differed strongly between the two simulations. Whereas, at 400 hPa, the moist simulation showed a prominent PV maximum of 6 pvu in a cyclonically wrapping PV streamer and a broad low-PV ridge further north, the PV distribution in the dry simulation indicated a broad trough with a less prominent PV maximum and a much weaker ridge (Fig. 10d). Similarly, Davis et al. (1993) found that latent heating intensified the downstream ridge aloft and led to an enhanced upper-level cyclonic wrapping of positive and negative PV anomalies. Studying the processes that led to the amplification of the upper-level ridge, they emphasised the upward and poleward advection of the tropopause by horizontal wind divergence above ascending motions that were intensified by latent heating (Fig. 10b). They referred to these effects of latent heating on PV advection by the divergent wind as "indirect", in contrast to the "direct" upper-level PV erosion by latent heating. In their case, however, the differences in the upper-level PV evolution between moist and dry simulations were less pronounced than in the study by Reed et al. (1993a). Larger differences in upper-level PV when turning off latent heating were again found by

Grønås (1995), with a cyclonic wrapping of stratospheric and tropospheric PV in the moist simulation that is absent in the dry run (Fig. 10c). Note that such large differences in upper-level PV and PV gradients directly translate in large differences of the upper-level jet amplitude. The role of upper-level divergence of the horizontal wind was also discussed by Stoelinga (1996). In their case, comparison of the moist and dry simulations showed that latent heating also enhanced upper-level divergence, which expanded the downstream ridge, retarded the upstream trough, and supported the vertical coupling of the PV anomaly of the trough with the low-level disturbance.

In an overview paper of the March 1993 "superstorm" over the eastern U.S. (also called "Storm of the Century"), Kocin et al. (1995) emphasised the strong increase of winds in the upper-level jet at the crest of an amplifying ridge downstream of the rapidly developing cyclone over the Gulf of Mexico. The wind speed increase by $30 \, \text{m s}^{-1}$ in 12 h coincided with intense convection and latent heat release within the cyclone. Examining operational numerical forecasts for this extreme event, Dickinson et al. (1997) showed that even short-range forecasts failed in simulating the intense deep convection over the Gulf of Mexico, the associated diabatic PV modification at lower and upper levels, and the lifting of the tropopause over and downstream of the developing storm. In a detailed model-based study of the early phase of the FASTEX IOP17 cyclone, Mallet et al. (1999a) also discussed the modification of the upper-tropospheric winds by latent heat release along the warm front, which, in this case, led to a splitting of the North Atlantic upper-level jet into two jet streaks during the cyclone intensification. Strongly divergent wind vectors at 300 hPa occurred above the latent heating maximum between the jet streaks. The model simulation also produced a vertically extended region with absolute negative PV from 800 to 400 hPa directly poleward of the prominent diabatically-produced PV anomaly that reached from the surface up to 500 hPa. Interestingly, in addition to direct PV erosion due to convective latent heating and strong divergent outflow above the heating, Morgenstern and Davies (1999) proposed a third process by which latent heating in a developing cyclone can influence the upper-level flow. Their model simulations of a cyclone in the Gulf of Genoa provided some evidence that sustained deep convection, triggered by southerly flow towards the Alps, produced a localised mid-tropospheric positive PV anomaly, whose induced wind field was strong enough to distort the upper-tropospheric PV streamer leading to its breakup and cutoff formation. This mechanism emphasised the far-field effect of diabatically produced PV anomalies. PV diagnostics, providing fundamental understanding of moist dynamical synoptic-scale weather systems, have also been applied to study the effect of latent heating on the dynamical evolution of mesoscale convective systems and their downstream impact (Davis and Weisman, 1994; Gray et al., 1998; Gray, 2001).

### 4.1.7 Summary

A multitude of diagnostic cases studies of typically rapidly intensifying extratropical cyclones provided clear evidence for the relevance of diabatic processes, in particular condensational latent heating, for the evolution of these systems. These research activities were based on observations, analyses, and numerical model simulations (mainly with limited-area models) and the development of novel diagnostics. These diagnostics combined in innovative ways the concepts of PV, MSI, and air-parcel trajectories. The growing use of Lagrangian methods was instrumental in elucidating the characteristics of coherent airstreams, in particular WCBs, and in developing an approach for their objective identification. A major challenge that remained was the

very large case-to-case variability and the need for more systematic assessments of the quantitative role of diabatic processes for the modification of cyclones and upper-level Rossby waves.

## 4.2 Real case numerical sensitivity experiments

It is challenging to identify cause-effect relations with only diagnostic studies due to the complicated feedback mechanisms involved (e.g., Yau and Jean, 1989); this is where numerical models can lead to further insight. The method pioneered by Aubert (1957) and Danard (1964) (Sect. 3.4) of using a limited-area weather prediction model to run sensitivity experiments with "full physics" (rendering a "realistic" dynamical evolution of the system) and with certain physical processes turned off became a widely used approach to quantify the effect of specific processes on the considered system (see Table 2). Simulations with all moist processes turned off are often referred to as "dry runs". (Different techniques have been used for setting up the dry runs, e.g., by initialising relative humidity in the entire model domain with a low value, or by turning off the parametrisations of convection, cloud microphysics and surface fluxes, or by setting the latent heat constants to zero.) The interpretation of these experiments appears to be straightforward as the difference between two simulations can be unambiguously attributed to the process turned off—and as summarised below, such experiments contributed enormously to improve the understanding of how cyclones are affected by diabatic processes. However, there are also at least five important caveats with this approach:

(i) The differences between sensitivity experiments show the effect of diabatic processes as they are represented in the model (typically by simplified parametrisations). But the effect of a specific process quantified in the model world may differ substantially from its "real effect", which of course is impossible to assess. It is likely that results would look different if early moist vs. dry comparison case studies were repeated with today's models;

(ii) The outcome of a moist vs. dry comparison for a particular case depends on the initialisation time of the simulations. This becomes particularly critical when considering the effect of surface fluxes: they might be essential for "preconditioning" the boundary layer and this might happen a few days prior to the onset of cyclogenesis. Therefore, a sensitivity experiment initialised after the preconditioning might erroneously conclude that surface fluxes were of minor importance. Also, the initialisation time is important because dry simulations can be influenced by artificially persistent PV anomalies that were diabatically produced prior to the start of the simulation and enter the simulation via the initial conditions, as critically discussed by Fehlmann and Davies (1999). Therefore, dry simulations might be "less dry" than anticipated;

(iii) The outcome also depends on the size of the model domain. In most cases, pairs of dry and moist experiments are relaxed to the same boundary conditions (e.g., from global analyses) and therefore, a too small model domain can lead to spuriously small differences between the experiments. It is therefore advisable to use a large model domain, often with repercussions for the model resolution;

(iv) Somewhat related to the three issues above, the "turning off" of a certain process affects the entire model domain and not only the specific cyclone (or another weather system) under consideration. Consequently, the impact of diabatic

processes in the larger environment of the cyclone cannot be distinguished from the impact of the same processes within the cyclone. A few studies therefore turned off diabatic processes only in a small subregion of the model domain, which was identified as particularly important for the evolution of the considered cyclone (e.g., Rossa et al., 2000); and

(v) The nonlinearity of the atmospheric flow implies that if the modified cyclone follows a modified track, then it may interact differently with an upper-level jet streak or pass over, e.g., different sea surface temperature (SST) anomalies and thus
undergo a markedly different evolution as a result. Related to this is a major caveat of these early sensitivity studies, which are (understandably) single deterministic simulations. Hence, the robustness of the findings to small changes in initial or boundary conditions is not known. A more robust approach would be to compare ensembles of full physics vs. dry simulations.

Different metrics have been used to quantify the difference between moist and dry simulations. A frequently used straight-
1135 forward measure is to consider the temporal evolution of the cyclone's core pressure and/or maximum near-surface vorticity (see examples in Fig. 11). Alternative diagnostics used were, e.g., the eddy energetics equations (Robertson and Smith, 1983) and later, in many studies, the cyclone's PV structure (e.g., Kuo and Reed, 1988). In the following subsections, we discuss a selection of studies with model sensitivity experiments, whose methods and/or findings we regard as particularly noteworthy and refer to Table 2 for a more complete overview of process-related sensitivity experiments and their results regarding
the quantification of diabatic processes for cyclone intensification. Also, Balasubramanian and Yau (1995) summarised in a table many studies on the sensitivity of rapid cyclogenesis on physical processes and distinguished between simulations that included slantwise convection vs. upright convection vs. non-convective heating. Section 4.2.2 summarises various moist vs. dry simulations of extratropical cyclone case studies, and Sect. 4.2.3 contains studies that applied the same approach to a set of simulations to identify the moist effects more systematically. Finally, Sect. 4.2.4 provides an overview on studies that used
the PV framework for the interpretation of the often striking differences between moist and dry simulations.

### 4.2.1 Moist vs. dry experiments: Cyclone deepening

In an ECMWF seminar contribution, Hoskins (1980) presented a forecast of the ECMWF model and compared it to a dry sensitivity simulation (setting the latent heat constant to zero). The most explosively deepening cyclone in this forecast, off the east coast of Asia, was dramatically underestimated in the dry run (core pressure of 987 instead of 940 hPa, and 24 h deepening of 9
vs. 44 hPa), whereas other cyclones were almost unaffected. Danard and Ellenton (1980)[12] investigated sensitivity experiments with surface fluxes turned off and, by comparing with a full-physics control run, found a negligible effect of surface fluxes during rapid cyclogenesis. A simulation of a continental U.S. cyclone without latent heating by Chang et al. (1982) failed in reproducing the intensity of the cyclone, in contrast to the fairly realistic moist control simulation.

---

[12]Gloria E. Ellenton is the earliest female co-author of the studies summarised in this review. She earned her PhD in Mechanical Engineering at the University of Waterloo in 1978 and became one of Canada's first women weather forecasters. Her professional career included weather forecasting, atmospheric modelling and running her own consulting company. Source: https://www.legacy.com/ca/obituaries/theglobeandmail/name/gloria-ellenton-obituary?pid=189971546.

**Table 2.** Overview on real case moist vs. dry numerical sensitivity experiments of extratropical cyclones. The studies are listed in chronological order and are considered until 2010. The columns list the name of the investigated cyclone(s) (if available), the region where and date when the cyclone(s) occurred, and (if available) an estimate of the contribution of diabatic effects to total cyclone deepening (diabatic deepening contribution, DDC). A DDC of 100% signifies that no cyclone formed or no deepening occurred in the dry simulation.

| Reference | Name/*number* of cyclone(s) | Region | Date | DDC (in %) |
|---|---|---|---|---|
| Danard (1964) | | U.S. | Jan 1959 | $\simeq 50$ |
| Robertson and Smith (1983) | *2 cyclones* | U.S. | Jan 1975 / May 1977 | $\simeq 50$ |
| Anthes et al. (1983) | QE II storm | N. Atl. | Sep 1978 | $> 90$ |
| Chen et al. (1983) | AMTEX | Taiwan | Feb 1975 | $> 50$ |
| Dell'Osso and Radinovic (1984) | ALPEX case | western Med. | Mar 1982 | $\sim 0$ |
| Liou and Elsberry (1987) | | western N. Pac. | Jan 1979 | 25 |
| Atlas (1987) | President's Day cyclone | U.S. east coast | Feb 1979 | 100 |
| Chen and Dell'Osso (1987) | | western N. Pac. | Nov 1982 | 44 |
| Kuo and Reed (1988) | | eastern N. Pac. | Nov 1981 | 50 |
| Mullen and Baumhefner (1988) | | N. Pac. | Jan GCM simulation | 50 |
| Reed et al. (1988) | *3 cyclones* | N. Atl. | Jan 1986 | 40–50 |
| Shutts (1990b) | U.K. Great October storm | eastern N. Atl. | Oct 1987 | 67 |
| Kuo and Low-Nam (1990) | *9 cyclones* | western N. Atl. | Dec–Apr 1981–1985 | 10–61 |
| Kuo et al. (1991b) | QE II storm | N. Atl. | Sep 1978 | 70 |
| Reed et al. (1993a) | ERICA IOP5 | western N. Atl. | Jan 1989 | 61 |
| Davis et al. (1993) | | central U.S. | Dec 1987 | 40 |
| Grønås et al. (1994) | | Germany | Aug 1989 | $\sim 50$ |
| Kristjánsson and Thorsteinsson (1995) | Greenhouse Low | N. Atl. | Feb 1991 | 34 |
| Grønås and Kvamstø (1995) | polar low | Norway | Mar 1992 | 40 |
| Grønås (1995) | New Year's Day storm | Norway | Jan 1992 | 50 |
| Stoelinga (1996) | | N. Atl. | Feb 1987 | 60 |
| Huo et al. (1996) | CASP II IOP14 | western N. Atl. | Feb 1992 | $< 30$ |
| Mailhot et al. (1996) | polar low | Labrador Sea | Jan 1989 | $> 80$ |
| Bresch et al. (1997) | polar low | Bering Sea | Mar 1977 | $> 50$ |
| Seluchi and Saulo (1998) | | S. America | Nov 1989 | $> 80$ |
| Carrera et al. (1999) | CASP IOP14 | western N. Atl. | Mar 1986 | 100 |
| Mallet et al. (1999b) | FASTEX IOP17 | N. Atl. | Feb 1997 | 40 |
| Rossa et al. (2000) | | N. Atl. | Nov 1992 | 40 |
| Wernli et al. (2002) | Lothar | eastern N. Atl. | Dec 1999 | 80 |
| Zhang et al. (2002) | | western N. Atl. | Mar 1992 | 35 |
| Davolio et al. (2009) | medicane | central Med. | Sep 2006 | 80 |
| Rivière et al. (2010) | Lothar | eastern N. Atl. | Dec 1999 | 100 |

Anthes et al. (1983) performed sensitivity experiments for the QE II storm and found that latent heating contributed strongly to cyclone intensification, in qualitative agreement with the observation-based studies mentioned in Sect. 4.1.1. Investigating numerical simulations of the same storm, Uccellini et al. (1987) could only reproduce the cyclone deepening with reasonable accuracy when including latent heating and surface fluxes. They emphasised that a synergistic interaction must exist among the upper-level jet streak and diabatic processes to account for the MSLP decrease. Similar conclusions were found by Kuo et al. (1991b)[13] who repeated the Anthes et al. simulations of the QE II storm with an updated version of the model and emphasised the importance of nonlinear interactions between (dry) baroclinic and diabatic processes. For the President's Day cyclone, Atlas (1987) identified very strong effects of surface fluxes on the evolution of the cyclone. They found that diabatic heating resulting from oceanic fluxes increased low-level baroclinicity, decreased static stability and significantly contributed both to the generation of low-level cyclonic vorticity and to the intensification of an upper-level ridge over the western North Atlantic—making this paper one of the first studies highlighting the simultaneous effects of latent heating on the flow evolution in the lower and upper troposphere (see Sect. 4.1.6).

An early model study of a cyclone outside of the North American region by Chen et al. (1983) considered a cyclone east of Taiwan that was observed during the AMTEX field experiment in 1975. They introduced a novel type of sensitivity experiment in which they strongly reduced the large-scale forcing by freezing the lateral boundary conditions, thereby disabling the upper-level trough (which instigated cyclogenesis in reality) from entering the model domain and consequently to a shallow and disorganised cyclone. This result about the importance of the large-scale forcing agrees with the findings by Liou and Elsberry (1987), who quantified that a dry run produced 75% of the observed explosive deepening of a North Pacific cyclone. Consequently, in their case, only 25% of the deepening was attributed to diabatic effects—a measure that shows high case to case variability (see Table 2). The simulations by Dell'Osso and Radinovic (1984) for an Alpine lee cyclone revealed an even slightly more intense cyclone in the dry compared to the full-physics simulation. Danard (1986) altered SSTs in simulations of cyclones in the eastern North Pacific. A uniform increase of SSTs by 5 K led to strongly increased surface precipitation and an almost doubled cyclone deepening, indicating a high sensitivity of cyclone deepening to SST and in turn surface fluxes. In contrast, the short-term surface fluxes were not relevant for the explosively deepening North Pacific cyclone simulated by Kuo and Reed (1988). However, latent heating was essential as dry simulations produced only about 50% of the deepening in the moist simulations (Fig. 11a). Vertical cross sections of PV revealed the strong PV production by condensational heating in the lower troposphere (their Fig. 9d). Also, Chen and Dell'Osso (1987), simulating a rapidly intensifying East Asian coastal cyclone, concluded that latent heating was essential for the deepening. Within two days, the deepening of minimum MSLP was 33 hPa in the full-physics simulation and only 14 hPa without latent heating.

Given the remarkable case-to-case variability in these early sensitivity experiments, which were performed with different models, parametrisations, and simulation setups, efforts were made to obtain more robust results by simulating a set of cyclones with the same model and simulation setup. In an early study of this kind, Mullen and Baumhefner (1988) analysed sensitivity experiments for 11 explosive North Pacific cyclones that occurred in a 150-day perpetual January simulation with the NCAR

---

[13]This paper was authored by Ying-Hwa (Bill) Kuo, Melvyn Shapiro, and Evelyn Donall (later Donall-Grell), and therefore is the 2nd contribution referenced in this review with a female co-author (more than a decade after Danard and Ellenton, 1980).

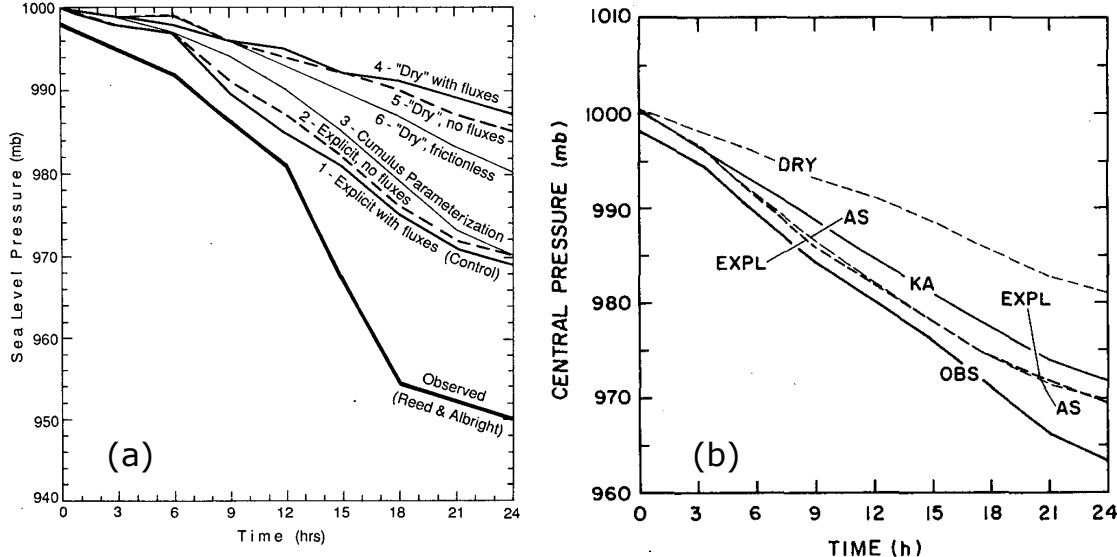

**Figure 11.** Cyclone intensification in moist vs. dry numerical experiments. **(a)** from Kuo and Reed (1988, their Fig. 18a), time evolution of minimum MSLP along eastern North Pacific cyclone from observations, moist and dry simulations; and **(b)** from Kuo and Low-Nam (1990, their Fig. 2), the same but averaged for nine explosively deepening cyclone in the western North Atlantic for observations (OBS), dry experiments (DRY), and simulations using different convection and cloud schemes (KA, AS and EXPL). Figures used with permission from the American Meteorological Society.

climate model. On average, total diabatic heating accounted for 50% of the deepening of the cyclones, with about equal shares from cloud latent heating and surface latent heat fluxes. Variability among the 11 cases was significant and the authors emphasised the potential danger in interpreting results from a single case of explosive cyclogenesis as representative. Reed et al. (1988) identified three explosive North Atlantic cyclones in operational ECMWF forecasts, and re-runs with a dry model version showed that 40–50% of the cyclone deepening was due to large-scale latent heating. Interestingly, they also considered in more detail an event for which the full physics forecast underestimated the storm intensification and attributed the forecast failure mainly to inaccuracies in the initial conditions. This can be regarded as an early example of a "forecast bust" process study, which became more prominent in the last decade (see Sect. 5.5.3). Based on sensitivity experiments (but no classical dry experiments) for three (explosive) winter storms during CASP, Mailhot and Chouinard (1989) emphasised the role of ocean evaporation and the transport of heat and moisture from the evaporative source region into the cyclone centre by the low-level jet along the cold front (which they regarded as the WCB).

An unprecedentedly systematic modelling study was performed by Kuo and Low-Nam (1990), who studied nine explosive cyclones in the western North Atlantic, each with 14 simulations at horizontal grid spacings of 80 and 40 km. The simulations include a dry run and simulations with different convective parametrisation schemes. It was found that the resolvable-scale precipitation parametrisation for slantwise ascent near the warm front was essential for rapid intensification. On average for

the nine cyclones, the 24-h deepening amounted to 35 hPa in the observations, 34 hPa in the 40 km moist simulations with 23 vertical levels, 31 hPa in the 80 km moist simulations with 15 vertical levels, and only 19 hPa in the 80 km dry runs, such that about 37% of the deepening was attributed to moist processes (Fig. 11b). For the individual cyclones, this number varied strongly between 10 and 61%. The dry runs also suffered from larger position errors of the simulated cyclones. In contrast to large-scale latent heating, deep convection and surface fluxes were of minor importance during the 24 h of strongest deepening. In a follow-up study, Kuo et al. (1991a) revealed that surface fluxes are however important for cyclone deepening in simulations initialised at least one day prior to rapid deepening, which they referred to as a "delayed effect of earlier fluxes".

For an event of secondary cyclogenesis in the western North Atlantic, Carrera et al. (1999) found that the model failed to produce the cyclone in the absence of latent heating. They therefore emphasised that their results differ from those generally found for large-scale extratropical cyclogenesis where upper-level induced baroclinic instability tends to dominate. Similarly, for an explosive cyclone in the la Plata river basin, Seluchi and Saulo (1998) identified latent heat release as the main mechanism acting to intensify the system (almost no cyclone formed in their dry sensitivity experiment). In contrast, a rare event of a North Atlantic cyclone, observed during CASP II in 1992 still underwent explosive deepening in the dry simulation (28 hPa in 24 h, which was 70% of the total deepening). Huo et al. (1996) attributed this particularly intense dry baroclinic intensification to the large-amplitude and deep upper-level PV anomaly, with the 2-pvu tropopause reaching down to 600 hPa.

Dry simulations have also been performed to investigate the sensitivity of other types of cyclones to diabatic effects. Bresch et al. (1997) addressed a polar low development in the Bering Sea with front-like features and found a substantially weakened cyclone in the dry run. The authors suggested that the development of most polar lows can be regarded as similar to that of midlatitude ocean cyclones, with essential interlevel interaction between upper-level PV anomalies and low-level thermal or PV anomalies produced by thermal advection and/or diabatic heating. An extended discussion of polar lows follows in Sect. 5.3.4.

### 4.2.2 Moist vs. dry experiments: PV perspective

A new level of dynamical insight was obtained by applying the PV framework to the comparison of moist vs. dry experiments (e.g., Davis et al., 1993; Reed et al., 1993a; Grønås, 1995). Davis et al. (1993) quantified what they called the "integrated effect of condensation" for a North American continental and two North Atlantic oceanic cyclones, applying PV inversion to PV differences between moist and dry MM4 simulations. They found that "... the primary effect of condensation at low levels was simply to superpose a positive PV anomaly onto the cyclonic circulation that would exist without latent heating". For the cyclones they investigated, the feedback of latent heating on the dry dynamical interaction of upper-level PV and surface $\theta$ anomalies was small. Reed et al. (1993a) performed sensitivity experiments of the ERICA IOP5 cyclone and found an unusually high sensitivity to latent heating—in the dry run the cyclone deepened in 24 h only by 13 hPa compared to 31 hPa in the moist simulation—and weaker sensitivity to surface fluxes, the Gulf Stream position, and horizontal model grid spacings varying between 90 and 10 km. They attributed the pronounced role of latent heating for cyclone intensification to the track of the cyclone parallel to the warm Gulf Stream waters and to the formation of a high-amplitude (> 10 pvu) low-level PV anomaly near the cyclone centre.

In Europe, model sensitivity experiments were mainly conducted for poorly-predicted high-impact storms that occurred in the 1980s and 1990s, e.g., a mesoscale storm in the Baltic Sea in summer 1985, the U.K. Great October storm in 1987, a northern Germany storm in August 1989, the Greenhouse Low in Iceland in February 1991, the New Year's day storm with Hurricane-force winds in western Norway in January 1992, and the infamous storm Lothar in December 1999. The Baltic Sea summer cyclone was unusual in the sense that it did not intensify (i.e., minimum MSLP did not decrease), however it

became more compact during the evolution, leading to very strong winds and heavy precipitation. A simulation without latent heating led to a 6 hPa higher central MSLP and much reduced near-surface winds (Kristjánsson, 1990). For the much more spectacular Great October Storm, in a simulation with 15 km grid spacing[14], Shutts (1990b) identified a diabatically produced low-level PV anomaly close to the cyclone centre, which was similarly intense as the one simulated for the ERICA IOP5 cyclone mentioned above. The dry sensitivity experiment without this anomaly only produced a 24-h cyclone deepening of

10 hPa compared to 32 hPa in the moist simulation. Kristjánsson and Thorsteinsson (1995) studied the Greenhouse Low with an MSLP deepening of 55 hPa in 24 h and a minimum MSLP below 945 hPa that hit western Iceland in early February 1991. The full-physics simulation produced an intense low-level diabatic PV anomaly. However, sensitivity experiments showed that the contribution of latent heating to the decrease of MSLP was only about one third, i.e., less than what was found in other studies of comparable cyclones, most likely due to the exceptionally strong baroclinicity in this case. Another strongly

diabatic and comparatively small-scale storm occurred in western Norway on the New Year's Day in 1992. Moist and dry simulations with 50 km grid spacing showed a 30-h MSLP decrease of about 52 hPa and 28 hPa, respectively (Grønås, 1995). Of particular interest is the analysis that the rapid intensification and generation of extreme near-surface winds occurred during a near-surface warm-air seclusion process at the western tip of the bent-back front and beneath the prominent upper-level PV anomaly. Intense precipitation along the bent-back front led to the diabatic formation of an intense low-level PV anomaly

(similar to the case described by Neiman and Shapiro, 1993, cf. Sect. 4.1.2). During the seclusion process, warm air and low-level PV secluded in tandem from the bent-back frontal band and formed what Grønås (1995) referred to as "the seclusion low", which induced the extremely intense low-level jet. In the dry simulation, no seclusion occurs, and surface winds remain much weaker. A special case is the late summer cyclone that developed over northern Germany in 1989 (Grønås et al., 1994). Compared to the many explosively deepening cyclones discussed in this section (most of them occurring in the cold season),

this cyclone was moderate with a minimum MSLP of 995 hPa. However, it was small-scale, slow-moving, and associated with strong winds and heavy precipitation—and the moist vs. dry simulations again revealed an essential contribution from latent heat release for the intensity and location of the storm, in agreement with the formation of a positive low-level PV anomaly. The authors interpreted the overall weak deepening of the cyclone as due to weak coupling of this low-level anomaly with upper-level PV. In addition to these high-impact cases, sensitivity experiments were also performed on two observed FASTEX

frontal cyclones. The effects of ice sublimation were studied in FASTEX IOP16 by comparing experiments with and without sublimational cooling (Clough et al., 2000). These experiments revealed that ice sublimation had an important impact on the

---

[14]Interestingly, while the operational simulation with the fine-mesh model of the U.K. Met Office failed in capturing the explosive development of this cyclone, the hindcast simulation with the same model by Shutts (1990b) accurately predicted the intensity and location of the storm. The successful simulation used a modified initial analysis, which included additional aircraft reports (AIREP) data.

mesoscale structure and circulation, with the cooling enhancing the presence of negative PV sheets along the sloping warm front. Moist PV was also negative implying the presence of CSI; the release of CSI can generate rain bands, as were observed in satellite imagery. FASTEX IOP17 was studied with moist vs. dry simulations by Mallet et al. (1999b). In the early phase of this cyclone, a weak upper-level trough triggered baroclinic instability and latent heat release led to the formation of a rapidly propagating low-level PV anomaly that did not couple with the upper-level trough. In contrast, in the dry simulation the cyclone evolved solely in response to the weak trough and therefore propagated more slowly and intensified less than in the moist simulation (MSLP decrease of 30 hPa compared to 43 hPa in 36 h). This is the first study that related the early phase of an observed cyclone to the concept of a diabatic Rossby wave, discussed further in Sect. 5.3.5.

A novel type of sensitivity experiments was introduced by Demirtas and Thorpe (1999) and Fehlmann and Davies (1999). Both studies considered North Atlantic cyclones and performed sensitivity experiments with modified initial conditions, where distinct finite-amplitude upper-level and/or low-level PV structures were artificially removed (and balanced initial conditions obtained via PV inversion). The differences to the control experiment (with the unmodified initial conditions) then yielded information about the role of the removed PV structure for the evolution of the cyclone. Demirtas and Thorpe (1999) mainly focused on the role of horizontal positioning errors of upper-level PV anomalies on cyclone forecasts; however, for one case they also briefly discussed the effect of removing a diabatically-produced low-level PV anomaly on the cyclone evolution, and they found a reduced deepening of MSLP in the simulation with the removed low-level PV anomaly. As discussed by Fehlmann and Davies (1999), who investigated a North Atlantic frontal wave cyclone, an interesting issue occurs when removing diabatically produced low-level PV anomalies in the initial conditions: since they are typically ephemeral and not materially conserved, the removed PV anomaly might rapidly regenerate in the full physics simulations and the low-level PV field and subsequent cyclone lifecycle then become similar to the evolution in the control simulation. However, Fehlmann and Davies (1999) also performed dry simulations with initially removed low-level PV anomalies and thereby could persistently eliminate their effects. From the set of 10 moist and dry simulations with various PV structures removed, the authors concluded that the diabatic effects and the initial low-level PV band played only a modulating role for this frontal wave cyclone, and that, however, upper-level PV features were essential. This indicates that this cyclone did not form as an instability of the low-level PV band, but rather via interlevel interactions of upper-level PV and the surface front, modified by diabatic processes. The same type of simulations with removed PV anomalies was performed by Zhang et al. (2002) for several frontal wave cyclones in the western North Atlantic. They also found that frontal waves still developed in dry simulations, and that the upper-level PV anomalies provided the necessary forcing for frontal cyclogenesis.

### 4.2.3 Summary

In agreement with the results from Sect. 4.1, this compilation of 20 years of numerical sensitivity experiments illustrates the high case-to-case variability of the contribution of diabatic processes to cyclone intensity (see Table 2) and, at the same time, provides convincing evidence for the key role of these processes. Particularly popular and insightful was the comparison of moist vs. dry simulations, despite the caveats of this methodology outlined at the beginning of this section. Two questions that emerge when realising this variability of the role of diabatic processes for the cyclone lifecycle are (i) whether certain

cyclone characteristics determine their susceptibility to diabatic processes, and (ii) whether NWP forecasts tend to have larger systematic errors and/or uncertainties for strongly compared to weakly diabatic systems, as hypothesised by Harr et al. (1992). Both are important themes of current research.

## 4.3 Idealised numerical simulations of cyclones

Complementary to real case model investigations, idealised numerical experiments of extratropical cyclones contributed to identifying the role of diabatic processes. In contrast to real case studies where control experiments are typically done with "full physics" and selected processes are then turned off, the tradition of idealised cyclone experiments is to perform dry simulations (e.g., Simmons and Hoskins, 1978; Hoskins and West, 1979; Davies and Schär, 1991) and the studies discussed here then added one or several diabatic processes to investigate their effects. The typical setup of idealised (dry or moist) baroclinic

instability experiments includes the stipulation of a zonally uniform basic state and an initial perturbation, which in most cases is chosen as either the most unstable normal mode or a localised finite-amplitude perturbation, e.g., in the form of an upper-level PV anomaly. Such simulations in tandem with novel analytical studies were essential for extending the understanding of the classical Eady model of dry baroclinic instability to moist flows. The results are summarised in four subsections, first for channel models using moist Eady-type basic states (i.e., with uniform baroclinicity and a rigid-lid tropopause, Sect. 4.3.1), then

for channel models using basic states with more realistic, meridionally confined upper-level jets (Sect. 4.3.2), then for global models using plausible climatological mean basic states (Sect. 4.3.3), and finally for investigating the physical basis of surface frontal instability (Sect. 4.3.4).

### 4.3.1 Moist baroclinic instability—Eady-type studies

An important strand of theoretical studies is based on analytical work about moist baroclinic instability, often assuming a

1320 so-called wave-CISK[15]-mechanism to represent latent heating. Most studies in this direction considered a quasi-geostrophic moist Eady model (Tokioka, 1973; Gambo, 1976; Mak, 1982, 1994; Bannon, 1986; Wang and Barcilon, 1986; Craig and Cho, 1988) and found that latent heat release, leading to reduced static stability, can enhance the growth rate and reduce the scale of the most unstable disturbance. In these highly idealised setups, the strength of the moist enhancement and scale reduction of maximum baroclinic growth depends nonlinearly on the amplitude of the heating (Mak, 1994, see Fig. 12a), as well as on the

1325 structure of the vertical heating profile. As an alternative to the wave-CISK parametrisation of cumulus convection, a refined approach would be to assume a saturated vertically extended cloud layer such that the heating in the cloud becomes proportional to the local vertical motion. Applied to the moist Eady problem, both approaches yield comparable results (Bannon, 1986). Interestingly, Bannon (1986) mentioned that the structure of the most unstable moist Eady mode with shallow convective

---

[15]Convective instability of the second kind, a concept that proved very helpful in understanding the dynamics of tropical cyclones (e.g., Charney and Eliassen, 1964), was supposed to be also useful to study cyclones in the extratropics and polar lows (Rasmussen, 1979). In essence, the wave-CISK approach to parameterise cumulus convection assumes that the amplitude of the heating in the cloud is determined by the vertical velocity at cloud base.

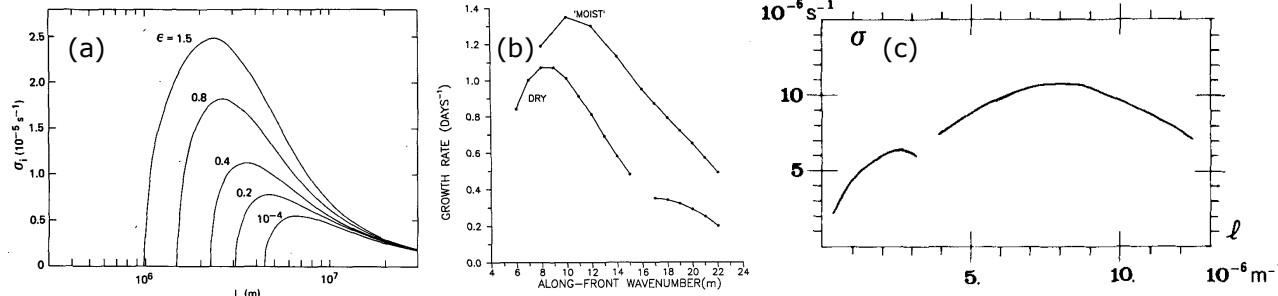

**Figure 12.** Idealised moist baroclinic instability studies. **(a)** from Mak (1982, their Fig. 1), growth rate as a function of wavelength for different heating intensities for an Eady-type basic state; **(b)** from Thorncroft and Hoskins (1990, their Fig. 9a), growth rate as a function of wave number in a dry and moist atmosphere for a frontal basic state; and **(c)** from Joly and Thorpe (1990, their Fig. 2), growth rate as a function of wave number for a frontal basic state with a low-level PV anomaly. All figures used with permission: (a,b) from the American Meteorological Society, and (c) from Wiley.

heating resembles observed intermediate-scale disturbances on the Baiu front[16]. Also, for establishing links between idealised
modelling approaches and real cyclones, Bonatti and Rao (1987) applied a multi-level wave-CISK model to real cases in the North Pacific and central South America and found a good agreement with the observed horizontal scale and vertical structure.

     Moist Eady-type studies with the wave-CISK approach also revealed that purely diabatically growing disturbances (a solution of the CISK-problem with zero basic-state baroclinicity, referred to as "pure CISK disturbances") are possible when the heating rate exceeds a threshold value (Craig and Cho, 1988, 1992). These authors also discussed the relevance of this
finding for explosive cyclogenesis, where the often observed warm core structure might be regarded as an indication for CISK. However, in their idealised study, Craig and Cho (1988) observed a warm core structure for heating below the CISK-threshold, and they concluded that "while a warm core indicates the presence of large amounts of heating in a baroclinic system, it may occur long before the disturbance can be characterised as CISK". Parker and Thorpe (1995) used the CISK approach to study convective frontogenesis. For strong heating they found a "solitary regime" where the growth rate of the moist unstable system
becomes independent of the wavelength of the imposed frontogenetic flow. They referred to this phenomenon as "diabatic Rossby waves" that propagate at the level of the low-level heating, a concept introduced by Snyder and Lindzen (1991) and discussed in more detail in Sect. 5.3.5. Parker and Thorpe (1995) showed that the PV tendencies due to diabatic heating act similarly to meridional PV advection in a Rossby wave, and therefore the diabatically-driven disturbances can be interpreted in terms of interacting Rossby wave pairs, as proposed by Hoskins et al. (1985) for dry Eady waves. An extreme case of the influ-
ence of diabatic heating on baroclinic instability was investigated by Snyder and Lindzen (1991), who considered wave-CISK with an Eady basic state in a vertically unbounded atmosphere, which is not baroclinically unstable, and in which the growth of waves is solely due to diabatic processes. They also briefly discussed the PV dynamics of their moist unstable modes and mentioned that heating may act as a dynamical surrogate for PV gradients, as confirmed later by Parker and Thorpe (1995).

---

[16]The Baiu front (Japanese) or Meiyu front (Chinese) is a persistent nearly stationary low-tropospheric baroclinic zone, extending from the east coast of China and Taiwan to the Pacific Ocean south of Japan in mid-spring to mid-summer.

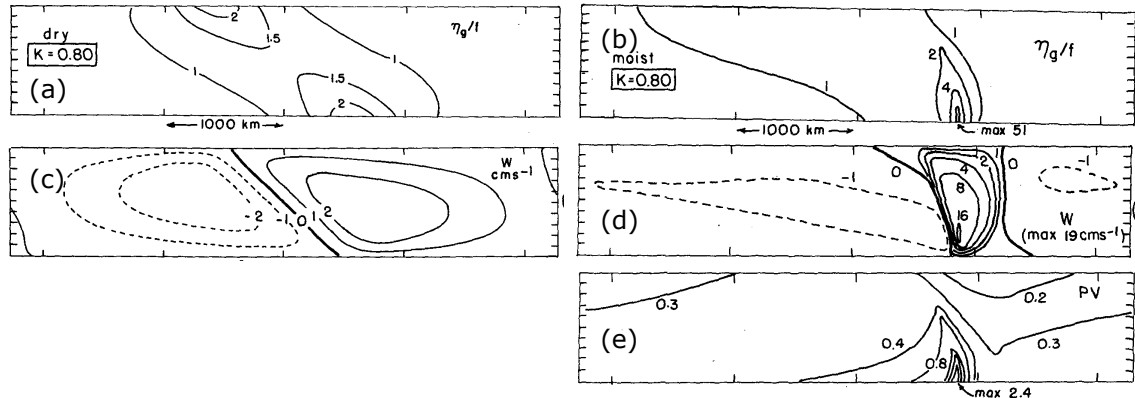

**Figure 13.** Moist Eady-type baroclinic instability. All panels from Emanuel et al. (1987, their Fig. 6c,e and 7b,d,e), showing the structure of the dry **(a,c)** and moist **(b,d,e)** waves with a wavelength of about 4500 km. Fields show **(a,b)** relative vorticity, **(c,d)** vertical velocity, and **(e)** PV. Figures used with permission from the American Meteorological Society.

Emanuel et al. (1987) used an alternative concept to parameterise latent heating and to study its effects on the development and structure of baroclinic waves. In essence, this concept regards near-zero moist PV (cf. Sect. 2.2) as the relevant constraint on the release of latent heat. Their results are based on two-level and multilevel semigeostrophic models with parameterised slantwise moist convection in an Eady-type basic state. As many previous studies using a wave-CISK parametrisation, they found an increase of the maximum growth rate (by a factor of 2.5 compared to the dry growth rate) and a shift of maximum growth to smaller scales. Particularly interesting are their results about the structural changes of the most unstable mode due to latent heating: with increasing heating, there is increasing horizontal asymmetry between ascent and descent (with much stronger and spatially concentrated ascent), and vertical asymmetry in the production of vorticity and PV (with high values near the surface front and very low values at upper levels), see Fig. 13. Using a similar approach to represent latent heating but using a primitive equation model with surface fluxes and varying air-sea temperature differences, Fantini (1990) revealed an interesting transition to an explosive phase of development, which goes along with a structural change from a moist baroclinic wave to a hurricane-like narrow maximum of $\theta_e$ associated with a local maximum of cyclonic PV and near-surface winds. It is important to emphasise the highly idealised quasi two-dimensional setup of these simulations, given the Eady basic state with uniform baroclinicity.

Whitaker and Davis (1994) built upon the slantwise convection approach of Emanuel et al. (1987), with important modifications. They did not assume the vertical stability to saturated ascent to be constant and near-zero but computed it consistent with local temperature and pressure. This led to a vertically increasing moist static stability profile. Using this parametrisation in a quasi-geostrophic Eady model led to important differences in the linear baroclinic instability characteristics compared to earlier studies: the most unstable mode in the moist atmosphere grows only marginally faster than the most unstable dry mode and the short-wave cutoff is eliminated. With the vertically increasing moist static stability, the effects of latent heat release were only felt at low levels and could not influence in a major way the deep Eady modes. The unstable short waves in the

1370 moist atmosphere are shallow, and their dynamics were described as a vertical interaction between anomalies of surface $\theta$ and low-level diabatically produced PV (Whitaker and Davis, 1994).

### 4.3.2 Moist baroclinic instability in jet-like channel model simulations

Early idealised moist channel model simulations with a jet-like baroclinic basic state led to contrasting results. Gall (1976), using small-amplitude initial temperature perturbations, found increased growth rates in the moist simulation at all wavelengths,
and no change of the most unstable wavelength. In the simulations by Nuss and Anthes (1987), who triggered baroclinic instability with a weak upper-level trough, the inclusion of latent heat release only led to a 10% increase of the deepening rate and was not necessary to get explosive deepening. They emphasised the essential role of the pre-existing strong low-level temperature gradient and weak static stability for cyclone intensification. Concerning the impact of surface fluxes, they found that the phase and amplitude of the fluxes relative to the low-level baroclinicity was crucial: surface heating that enhanced
baroclinicity led to a 15% increase of the deepening rate, while surface heating that reduced baroclinicity dampened the instability. This is in qualitative agreement with results from an analytic quasi-geostrophic diagnostic model based on the omega equation to study the effects of surface fluxes on marine cyclones (Roebber, 1989), which indicated that indirect effects of surface fluxes leading to a moistening and destabilisation of the boundary layer and an amplification of low-level baroclinicity are most relevant.

In contrast to these simulations with small-amplitude initial perturbations, which were designed much in the tradition of linear instability concepts, Montgomery and Farrell (1992) suggested adopting an initial-value viewpoint and shifting focus from studying linear growth rates to the physical processes involved in the nonlinear evolution of the systems. This viewpoint was also motivated by dry dynamical considerations of modal vs. non-modal baroclinic waves (e.g., Farrell, 1984) and idealised dry simulations of cyclones triggered by finite-amplitude upper-level troughs (Takayabu, 1991; Schär and Wernli, 1993), which
were conceptually in line with the archetypal schematic of upper-level induced cyclogenesis (Hoskins et al., 1985, their Fig. 21). Montgomery and Farrell (1992) performed simulations using the Emanuel et al. (1987) approach to represent latent heating and with large-amplitude initial PV anomalies both in the upper and lower troposphere. Although they focused on idealised polar low cyclogenesis, their proposed two-stage model of development can also be used to discuss the evolution of extratropical cyclones. In stage one, the cyclone intensifies in response to ascent induced by the upper-level trough and enhanced by CI, and
in stage two, the continuously diabatically produced low-level PV maintains the cyclone. The authors also suggested that "In exceptional instances of negligible upper-level forcing, the [stage two process] may also describe the gradual intensification of small-scale cyclones in regions of sustained neutrality and surface baroclinicity." This idea will be discussed again in Sect. 5.3.5 in the context of diabatic Rossby waves.

Balasubramanian and Yau (1994a, b) examined the effects of convection on explosively developing cyclones using two
models: A comparatively simple two-layer channel model with parameterised slantwise convection and an extended, more complex multi-level model with explicit equations for water vapour and cloud liquid water, respectively. They found a 36-h MSLP deepening of 30 vs. 55 hPa in the dry and moist simulations, respectively, in the simulation with the two-layer model, and a 48-h deepening of 45 vs. 56 hPa in the multi-level simulation. With both models, the moist simulation produced an

intense bent-back front with very high relative vorticity and PV values along the warm and bent-back front, and in the cyclone centre. Maximum relative vorticity along the bent-back front at 950 hPa was enhanced by a factor of almost five (Fig. 3b in Balasubramanian and Yau, 1994b). Slantwise convection along the warm and bent-back fronts enhanced frontogenesis in the moist simulation, as well as the associated vorticity and PV production and their subsequent horizontal advection towards the cyclone centre. With the aid of a piecewise PV inversion done separately for the surface $\theta$ and upper- and lower-level PV anomalies—analogously to Davis and Emanuel (1991) who studied a real case cyclone simulation—Balasubramanian and Yau (1994b) showed that in the moist simulation the diabatically-produced low-level PV anomaly and the surface temperature anomaly contributed about 40% and 35%, respectively, to the geopotential height anomaly at 900 hPa. In contrast, in the dry simulation, about 75% of the geopotential height anomaly was induced by the surface temperature anomaly. The findings from these idealised simulations agreed with the results from observed cyclones (Sect. 4.1 and 4.2) and emphasised the key role of PV production in the warm and bent-back front for explosive cyclogenesis. An earlier study by Hedley and Yau (1991) used a non-hydrostatic model that included surface fluxes and a Kessler-type cloud scheme with grid spacings from 50 to 10 km. The moist simulation showed explosive deepening of 49 hPa in 36 h and cloud processes along the intense warm front. Grid resolution mattered mainly for the strength of the warm frontal gradient and associated low-level vorticity, and cyclone deepening was reduced by about 10% at the coarsest resolution. Surface fluxes were essential for destabilising the lower troposphere in the region of warm frontogenesis. Very similar nonlinear simulations of moist baroclinic waves in a jet-like basic state with slantwise convection by Whitaker and Davis (1994), with both normal-mode and finite-amplitude initial conditions, confirmed maximum low-level diabatic PV production along the warm front and showed a 20% increased cyclone deepening in the moist simulation compared to the dry one.

A very different approach to investigate key processes for the evolution of idealised moist baroclinic waves is to perform an adjoint sensitivity study (Langland et al., 1996). This is a computationally efficient method to approximately quantify the sensitivity of a certain aspect of the model simulation (e.g., the cyclone minimum MSLP) with respect to perturbations of the model variables and parameters (see Sect. 5.4.3 for more explanation). A particular interesting property of adjoint-sensitivity information is that it quantifies how perturbations influence the future state of a forecast aspect. Using this approach in a moist nonlinear channel model simulation with a finite amplitude initial perturbation at upper levels, the adjoint sensitivities suggested that "diabatic heating near the warm front in the lower troposphere is optimal in the sense that it amplifies the dry processes of baroclinic instability in the location that potentially has the most impact on cyclone intensification ... If the Laplacian of diabatic heating is large near the warm front, this can reinforce a vorticity tendency in the direction of storm motion (to the north-east) ... ". This finding agrees nicely with the results discussed above that diabatic low-level PV production along the warm front is essential for rapid cyclone deepening.

In comparison to idealised studies about the effects of latent heating on baroclinic waves, few studies investigated the effects of surface friction. Valdes and Hoskins (1988) investigated baroclinic instability of the observed zonal-mean atmosphere with and without a boundary layer friction parametrisation in the form of Ekman pumping (using an idealised global circulation model based on Hoskins and Simmons, 1975). With friction the growth rate of the most unstable mode was reduced but still positive. In contrast, shorter wavelength modes were stabilised due to their shallow vertical scale. Hines and Mechoso

(1993) included surface drag and horizontal diffusion in a dry primitive equation channel model, initialised with a meridionally symmetric jet and the most unstable normal mode, and studied the effects on the surface frontal structure of the growing baroclinic wave. While their "no drag" and "ocean drag" simulations produced a well-defined frontal fracture and bent-back front, the warm and bent-back fronts had a much-reduced intensity and extent with "land drag". In contrast, surface drag hardly affected the intensity of the cold front. Qualitatively similar results about the effects of boundary layer turbulence to weaken surface warm and bent-back fronts were found by Thompson (1995), who simulated a baroclinic wave in a channel model with a second-order closure turbulence scheme and surface fluxes.

### 4.3.3   Moist baroclinic instability simulated with GCMs

In parallel to the studies discussed in the previous sections, which used highly idealised basic states in channel models typically on an $f$-plane, first baroclinic wave experiments were also made with global models and using basic states inspired by the climatological mean flow. A very early example is the study by Hayashi and Golder (1981), which compared the growth of small random perturbations added to a winter-mean baroclinic zone in dry and moist simulations with a spectral aquaplanet general circulation model (GCM). They found an increase of eddy kinetic energy at all wavelengths in the simulation with latent heating. A decade later, also using a global spectral model, Gutowski et al. (1992) investigated moist baroclinic waves for basic states that resemble the winter and summer zonal mean. Their simulations were initially perturbed with a low-amplitude wavenumber 7 temperature wave. Compared to the dry simulation, the inclusion of moisture and latent heating led a faster growth of the baroclinic wave and an increase of peak eddy kinetic energy by 22%, in qualitative agreement with results from channel model simulations.

Branscome and Gutowski (1992) was the first study that used an aquaplanet GCM to investigate explicitly how climate change affects moist baroclinic waves. Moist baroclinic instability was compared for basic states that were taken from a full GCM control simulation and one with doubled $CO_2$. In the doubled $CO_2$ simulation, zonal mean temperature and specific humidity were increased by about $4\,\mathrm{K}$ and $0.4\,\mathrm{g\,kg^{-1}}$, respectively, and the meridional temperature gradient was substantially reduced[17]. The same spectral model was used as in Gutowski et al. (1992), which has limited spectral resolution (simulating only wavenumber 7 and its two first harmonics). The simulations showed a decrease of eddy kinetic energy and surface precipitation of moist baroclinic waves when doubling $CO_2$. This early result is inconsistent with their later study (Gutowski et al., 1995), where they derived the initial conditions for the doubled $CO_2$ experiment from another GCM, which yielded a smaller meridional temperature reduction with increased $CO_2$, and in turn an increase of eddy kinetic energy of the simulated moist baroclinic waves. Also motivated by global warming and a concomitant increase of specific humidity, Pavan et al. (1999) used an aquaplanet GCM with T42 spectral resolution (equivalent to a grid spacing of about 300 km) and 11 vertical levels to investigate how increased moisture modifies the lifecycle of baroclinic waves, with a basic state flow taken from a

---

[17]From a technical point of view, these simulations could also be considered as just another contribution to the large set of sensitivity experiments that quantify the role of moist dynamics. However, what gives them a special flavour is the fact that they were motivated by climate change (and less so by understanding moist dynamics) and that they tried to modify humidity in their initial conditions in a way that is consistent with available climate change projections.

zonal 10-year winter average from a full-physics GCM simulation. Their simulations revealed a complex sensitivity pattern, with peak eddy kinetic energy of the most unstable wave increasing in simulations with moister subtropics but decreasing in simulations with moister extratropics. They also performed moist baroclinic instability analyses with basic states taken from control and doubled $CO_2$ GCM experiments and found the same result as Branscome and Gutowski (1992), i.e., that the reduction of baroclinicity dominates over the increase in moisture leading to a decrease of moist baroclinic instability. These early experiments with contrasting results clearly illustrate the challenge in assessing the relative magnitude of the opposing effects of climate change, which are to reduce baroclinicity and to increase moisture (and potentially latent heating). These aspects will be further discussed in Sect. 5.7.

### 4.3.4  Frontal wave instability

Idealised frontal wave instability studies are discussed here separately, because of the fundamentally different basic states and scale of instability compared to the moist baroclinic instability problem with an Eady-type, upper-level jet-like, or climatological mean basic state, discussed in Sect. 4.3.1–4.3.3. Also, a special aspect of most of the ground-breaking studies on this theme (Joly and Thorpe, 1990; Schär and Davies, 1990; Thorncroft and Hoskins, 1990; Malardel et al., 1993) is that they used dry models to investigate linear instability and the nonlinear evolution of the most unstable normal modes. However, diabatic processes are included in these studies in an indirect way. As discussed below, the basic states that enable frontal wave instability must include a low-level PV maximum or a surface warm anomaly. Both can be regarded as a manifestation of latent heating.

The study by Thorncroft and Hoskins (1990) established the link between large-scale baroclinic instability as discussed in the previous subsections and frontal wave instability, often also referred as secondary cyclogenesis. In their dry primitive equation simulation on the sphere, initialised with the most unstable normal mode of a symmetric upper-level jet, there is first classical baroclinic instability producing a large-scale and deep cyclone and intense surface fronts. After day 7 of their simulation, the primary cyclone decays, however, the (occluded) cold front remains intense. By day 9, in response to an advancing upper-level PV streamer, a wave forms on this cold front, which experiences moderate growth and dissipation within two days. This simulation indicates that, in principle, frontal waves can form in a dry atmosphere, given the suitable interaction of an upper-level PV disturbance and an intense surface front. However, the intensification of the simulated frontal wave was very weak and not comparable to observed cases like, e.g., the U.K. Great October storm. Thorncroft and Hoskins (1990) then also constructed a frontal basic state with a low-tropospheric cold frontal temperature gradient, an intense upper-level jet, and a prefrontal surface warm maximum to perform a linear instability analysis with a dry and moist model version with a very simple parametrisation of latent heating. Both model versions revealed instability for a broad range of scales. Maximum growth occurred for classical Eady modes, and growth rates at the horizontal scale of frontal waves (about 1000 km) were doubled when including effects of latent heating (Fig. 12b). The authors concluded that "... although the initiation of many frontal cyclone events in the atmosphere can be understood in terms of the dry dynamics, diabatic processes will play an important role ... The material generation and destruction of PV and the advection of such anomalies within the cyclone also requires further investigation".

In parallel, linear instability studies for basic states with intense surface fronts were performed by Joly and Thorpe (1990) and Schär and Davies (1990), the former with a prefrontal band of low-level PV in the basic state, the latter with a basic state of uniform PV but with a prefrontal surface warm anomaly. Both studies used a semi-geostrophic model and a semi-infinite domain, which excluded Eady-type large-scale baroclinic instability. Reconsidering the Charney-Stern instability criteria, Schär and Davies (1990, their Sect. 2b) derived, in the quasi-geostrophic framework, necessary conditions for instability. In addition to Eady and Charney-type baroclinic instabilities, they identified gradients of interior PV or surface $\theta$ as necessary prerequisites for frontal wave instability in an unbounded atmosphere on an $f$-plane. Both studies (Joly and Thorpe, 1990; Schär and Davies, 1990) revealed most unstable normal modes for perturbations with a horizontal wavelength of slightly below 1000 km, in stark contrast to Eady-type baroclinic instability and in agreement with the smaller scale of observed frontal wave cyclones (see Fig. 12c, where maximum growth occurs for a wavelength of about 800 km). The nonlinear evolution in a dry atmosphere of these most unstable normal modes, presented in Malardel et al. (1993) for the Joly and Thorpe basic state, showed several realistic features, including the shallow cyclone structure with low-level maxima of relative vorticity and PV. But, as in Thorncroft and Hoskins (1990), the intensification was only very moderate, which was attributed potentially to the missing diabatic effects and the absence of vertical coupling with upper-level PV anomalies. However, it is important to note that in the real atmosphere, many incipient frontal waves do not intensify significantly, as already noted by Bergeron and Swoboda (1924) and discussed by Parker (1998), indicating that real events often bear the conditions for frontal instability but not for strong intensification.

The importance of the large-scale environmental deformation field in frontal wave development was also the subject of a series of papers in the 1990s with initially idealised studies followed by comparisons with analyses of real case studies. The aforementioned study by Joly and Thorpe (1990) proposed a two-stage process for frontal instability: the development of a low-level PV band through diabatic heating along the front of a parent cyclone in a horizontal deformation field that strengthens the front and inhibits small-scale barotropic instability in the band, followed by the development of frontal waves when the deformation weakens in the mature cyclone. The horizontal stretching deformation threshold above which frontal instability is inhibited was quantified as $0.6-0.8 \cdot 10^{-5}\,\mathrm{s}^{-1}$, based on results from idealised modelling studies (Dritschel et al., 1991; Bishop and Thorpe, 1994a, b) and from real case studies (Renfrew et al., 1997; Rivals et al., 1998; Chaboureau and Thorpe, 1999; Baehr et al., 1999; Mallet et al., 1999a, the latter three papers considering cases from the FASTEX field campaign). In contrast, horizontal shearing deformation has a less discriminating role on frontal wave growth (Joly and Thorpe, 1991; Renfrew et al., 1997; Chaboureau and Thorpe, 1999).

### 4.3.5 Summary

Idealised studies provide an important theoretical backbone for the understanding of how diabatic processes influence extratropical weather systems. The approach of including moist processes to modify classical dry baroclinic instability is methodologically elegant, allows for a meaningful generalisation from dry to moist extratropical dynamics, and provided fairly consistent results about the main diabatic effects. Models of differing complexity showed that most unstable moist baroclinic waves have larger growth rates and smaller wavelength than their dry counterparts. Diabatic PV production along the warm and bent-back

fronts was found to be key for the increased intensification of idealised moist cyclones. An important milestone in the 1990s was idealised studies that could explain the small wavelength of frontal wave cyclones and that pointed to the key role of antecedent diabatic processes in producing the low-level PV band along the front.

## 5 Recent accomplishments (the last 20 years)

At the turn of the millennium, there was the clear understanding and appreciation that diabatic processes are important for the development of mainly intense and rapidly deepening extratropical cyclones (which were investigated mainly along the North Atlantic storm track) and it was realised that diabatic processes also affect the upper-level flow. This section provides an overview on key research accomplishments from the last roughly 20 years and does this in a more selective way compared to the detailed historical portrayal in the previous sections. We also mention that other recent review papers cover certain aspects

of the themes discussed in this Section, for instance the role of diabatic processes for Rossby wave generation and modification (Wirth et al., 2018, their Sect. 4) and for cyclones and fronts (Schultz et al., 2019).

While key research elements discussed in Sect. 4 (e.g., analysis of data from field campaigns, idealised numerical model studies, and model sensitivity experiments) have remained important and were used in many studies discussed in this section, very important progress has been made on the methodological and modelling approaches, and in the generation of new datasets.

Novel datasets and numerical models have became available, which have enormously benefited research on diabatic processes: reanalyses, ensemble forecasts, convection-permitting regional models, coupled ocean-atmosphere models, and climate model simulations. Given the key role of reanalyses for generalising earlier results from case studies and for establishing a climatological view on the theme, they are briefly introduced in Sect. 5.1. However, reviewing the evolution of numerical weather and climate models would go beyond the scope of this paper, and the interested reader is referred to recent overview articles (e.g.,

Bauer et al., 2015; Schär et al., 2020). More studies have addressed diabatic processes other than latent heating in cloud, and so radiative and surface flux related diabatic processes, as well as frictional processes, in extratropical cyclones are reviewed in Sect. 5.2. Examination of diabatic processes has extended from classical extratropical cyclones into other categories of cyclones and these are reviewed in Sect. 5.3. Novel diagnostics have been developed that are tailored for quantifying the role of diabatic processes (Sect. 5.4) and there has been an expansion of studies examining the role of diabatic outflows and WCBs

(Sect. 5.5). New field campaigns that have specifically targeted mesoscale cyclone features associated with diabatic processes motivated the inclusion of Sect. 5.6 and, finally, Sect. 5.7 is tightly focused on the changes in diabatic processes in extratropical weather systems under anthropogenic climate change.

### 5.1 Reanalyses and weather system climatologies

In contrast to archived weather analyses from operational forecasting systems, a reanalysis is produced with one specific version

of a data assimilation system and forecast model and is therefore not affected by methodological changes. Hence, they provide temporally consistent global fields[18] of the three-dimensional atmospheric circulation and are of great value for performing

---

[18]This simplified statement ignores the fact that inconsistencies still occur due to temporal changes in the underlying observing system.

climatological studies of atmospheric phenomena and processes. Early reanalyses include those produced by NASA for 1985–1989 (Schubert et al., 1993), NCEP for 1957–1996 (Kalnay et al., 1996), and ECMWF for 1979–1993 (ERA-15, Gibson et al., 1997) and 1957-2002 (ERA-40 Uppala et al., 2005). Two recent high-resolution ECMWF reanalyses are ERA-Interim for 1979–2019 (Dee et al., 2011) and ERA5 from 1940 to present (Hersbach et al., 2020).

The value of reanalyses for weather and climate dynamics cannot be overestimated. They enable assessment of the robustness and generality of findings from detailed case studies and idealised studies. With the use of suitable diagnostics, specific flow systems and physical processes can be objectively identified in multi-decadal datasets, in different seasons, and in different regions. This flexibility not only provides information about the average characteristics of these systems and processes, but also about their variability from case to case, and between seasons and regions. In this way, specific and often very detailed results from case studies can be compared with the climatological distribution, which helps with, on one hand, identifying exceptional events and, on the other hand, robustly quantifying the role of diabatic processes for the structure and evolution of weather systems. In addition, due to the consistency and long period covered, reanalyses allow the identification of potential trends in weather systems and physical processes. As an important aside, we mention that despite the great utility of reanalyses, they also have potential biases, especially related to moist processes. For instance Hawcroft et al. (2017) found, in a comparison of ERA-Interim with remote sensing observations, that the reanalysis has more clouds than observed at low levels in the WCB inflow region.

Early reanalysis-based, climatological studies of weather systems looked at extratropical cyclones, more specifically the geographical distribution of cyclone genesis, intensification, and tracks (e.g., Sinclair, 1995; Hoskins and Hodges, 2002) and of trends in cyclone frequency, size and intensity (e.g., Simmonds and Keay, 2000; McCabe et al., 2001). Other climatologies using reanalyses were produced for surface fronts (Berry et al., 2011; Simmonds et al., 2012), blocks (Pelly and Hoskins, 2003; Croci-Maspoli et al., 2007), and Rossby wave breakings (Abatzoglou and Magnusdottir, 2006; Wernli and Sprenger, 2007). While these examples are mentioned here to illustrate the broad range of phenomena that were climatologically investigated thanks to the advent of reanalyses, in the following paragraphs and sections only those climatological studies are mentioned that specifically investigated the role of diabatic processes.

Several studies, using very different methodologies, investigated statistically the importance of diabatic processes for the deepening of extratropical cyclones. Some of these studies focused on specific regions. For instance, Wang et al. (2002) composited about 20 explosive cyclones each in the western and eastern North Atlantic using operational ECMWF analyses from 1985–1996 and found an earlier and more pronounced formation of a near-surface PV anomaly during intensification for cyclones in the eastern North Atlantic. Similar regional differences were identified for more than 200 explosive cyclones in the western North Pacific by Yoshida and Asuma (2004), who investigated composites based on JMA analyses from 1994–1999. Cyclones that formed south of Japan had much more intense latent heating during their intensification compared to cyclones with genesis further north, and they hypothesised that this heating contributes to their slightly stronger deepening rates. Other studies compiled specific climatologies of explosively deepening cyclones in the Southern Hemisphere (Lim and Simmonds, 2002) and in the western North Pacific (Yoshiike and Kawamura, 2009). The latter study mentioned that the East Asian winter

monsoon strongly modulates the geographical distribution of explosive cyclones, and that these often, via a robust pattern of downstream development, induce genesis of Kona lows in the vicinity of the Hawaiian Islands.

Other studies considered cyclones globally (or at least in one hemisphere), often using refined diagnostics. Roebber and Schumann (2011) reconsidered the physical processes that are responsible for the rapid deepening tail of oceanic cyclone intensification identified first by Sanders and Gyakum (1980). They confirmed earlier arguments that the rapid deepening tail is evidence of a fundamentally distinct pattern of cyclone behaviour, which results from interactions of baroclinic dynamics and latent heat release. Figure 14a is a table from Roebber and Schumann (2011) that demonstrates that the rapid deepening tail, i.e., the significant skewness in the deepening rate distribution, is eliminated in experiments in which the latent heating is reduced (even if this reduced latent heating is combined with enhanced baroclinicity). Strong baroclinic forcing in the absence of moisture availability produced substantial cyclone intensification but without the skewed rapid deepening tail behaviour. Pirret et al. (2017) used a pressure-tendency diagnostic (Fink et al., 2012, see Sect. 5.4.4) to quantify diabatic relative to baroclinic contributions to cyclone deepening. Their results showed that contributions from diabatic processes vary strongly with values of up to 60%. Also, the diabatic contribution correlates with the time a given cyclone track spends on the equatorward side of the jet, where there is greater potential for diabatic processes in the warm, moist air. Čampa and Wernli (2012) produced a climatology of vertical PV profiles in the centre of almost 10,000 mature cyclones in the Northern Hemisphere and found that cyclones with a lower MSLP minimum have stronger PV anomalies both in the upper and lower troposphere (Fig. 14b), the latter most likely produced by latent heating. They also found regional differences with the strongest positive lower-tropospheric PV anomalies in cyclones in the western parts of the North Atlantic and North Pacific, respectively. Dacre and Gray (2013) similarly found stronger low-level PV values developed in western, compared to eastern, North Atlantic cyclones. However, using ensemble sensitivity analysis (Sect. 5.4.3) they revealed that the intensity, two days after genesis, of only the eastern North Atlantic cyclones had a strong association with the presence of that PV at genesis time and so concluded that latent heating is more important for cyclone development in this region.

The tendency that deeper cyclones have larger-magnitude low-level PV anomalies produced by cloud diabatic heating is qualitatively confirmed by several studies that found that surface precipitation correlates with cyclone intensity (Field and Wood, 2007; Rudeva and Gulev, 2011; Pfahl and Sprenger, 2016). Seiler (2019) was the first to climatologically quantify the contributions of the anomalies of surface $\theta$, low-level and upper-level PV, based on piecewise PV inversion, to the intensity of the about 3000 most intense extratropical cyclones, according to relative vorticity at 850 hPa, identified in ERA-Interim. For slightly more than half of all cases, the largest contributions during maximum cyclone intensity are associated with the diabatically produced low-level PV anomaly. Cyclones whose intensities are dominated by low-level PV are more frequent during the warm compared to the cold season, and more frequent in the North Pacific than in the North Atlantic. Chen et al. (2018) investigated the potential role of CSI for cyclone intensification and used reanalyses to quantify two metrics for CSI along cyclone tracks (SCAPE and the vertically integrated extent of realisable symmetric instability, see Sect. 2.2 for descriptions of these metrics). They found evidence that the release of CSI might contribute to the intensification of cyclones as both metrics had higher values before, and a larger drop after, the onset of rapid intensification in explosively deepening compared to non-explosively deepening cyclones.

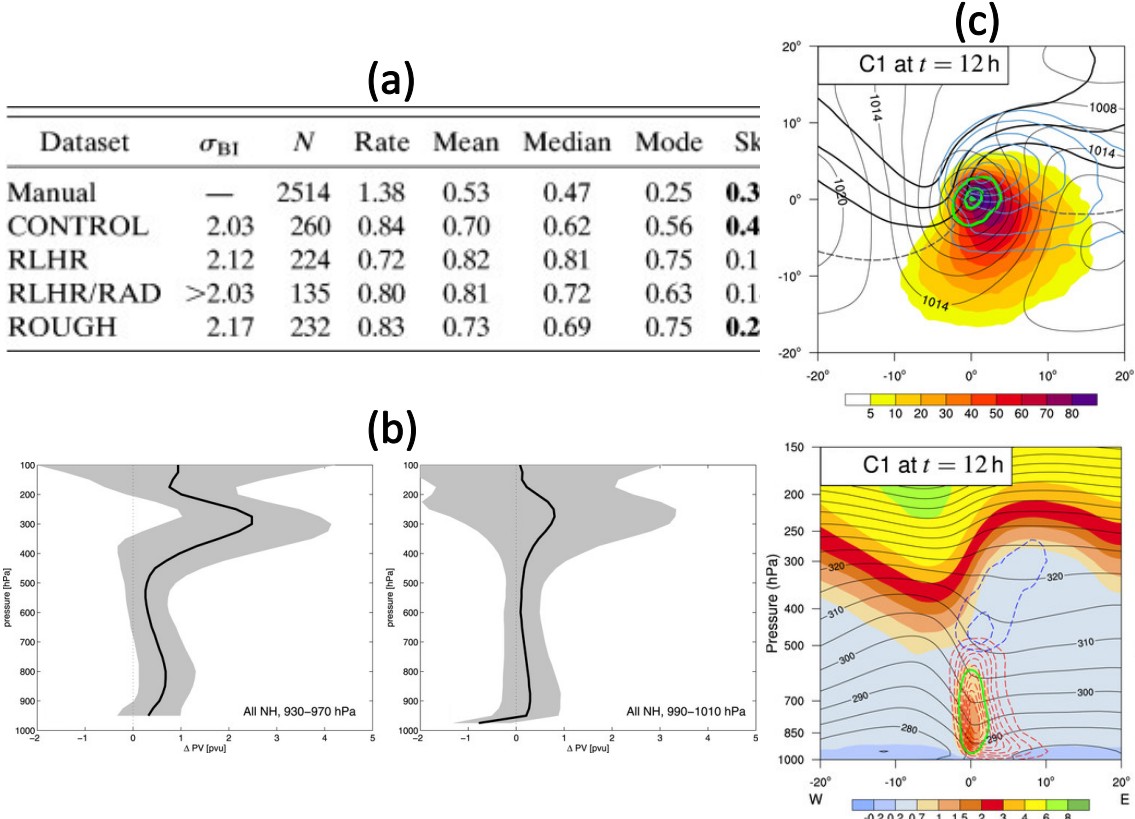

**Figure 14.** Linking diabatically generated PV, WCBs and either rapid deepening or intensity. **(a)** from Roebber and Schumann (2011, their Table 2), the Eady growth rate $\sigma_{BI}$ (day$^{-1}$), number $N$, occurrence rate (day$^{-1}$), the mean median and mode of the deepening rate (Bergerons), and the moment-based skewness ($Sk$ where bold indicates $>$ two standard errors of skew beyond zero) for Pacific maritime cyclones in a manual analysis and from four model experiments: CONTROL, enhanced roughness (ROUGH), reduced latent heating (RLHR), and both reduced latent heating and enhanced baroclinicity (RLHR/RAD); **(b)** from Čampa and Wernli (2012, their Fig. 3), composite Northern Hemisphere winter PV anomaly profiles with minimum MSLP in the ranges given and black line and shaded areas indicating mean and spread; and **(c)** from Binder et al. (2016, their Fig. 7), composite structure of explosive cyclones with strong WCBs in the middle of the strongest deepening period showing (top, low-level plan view) frequency of low-level WCB trajectories (shading), high-value PV low-level WCB trajectories (green contours) and upper-level WCB trajectories (blue contours ), together with MSLP (thin black contours) and PV on 315 K (thick black contours), and (bottom, west-east cross section view) PV (shading), $\theta$ (black contours) and relative WCB trajectory density (coloured contours for low-level (red), upper-level (blue) and high-value PV low-level WCB (green) trajectories). Figures used with permission from the American Meteorological Society.

Combining feature-based climatologies of cyclones and WCBs, Eckhardt et al. (2004) revealed that, considering both hemispheres and the summer and winter seasons, only cyclones in the Northern Hemisphere winter are usually accompanied by a strong WCB, identified with a Lagrangian criterion of large vertical ascent. In contrast, cyclones in the Southern Ocean are

not often associated with WCBs and therefore less influenced by intense cloud condensational heating. Dacre et al. (2012) described a North Atlantic extratropical cyclone "atlas" produced as a teaching tool by compositing the 200 most intense cyclones in 20 winters of ERA-Interim data. The associated webpage (http://www.met.rdg.ac.uk/~storms/) now also includes a Southern Hemisphere atlas. While the atlas is not designed to reveal the importance of diabatic processes in cyclone development, it is a useful tool to visualise the average cyclone structure and the WCB identified from system-relative flow analysis. A statistical evaluation of cyclone deepening and their associated WCB trajectories (Binder et al., 2016) considered in more detail how the latent heating in WCBs that ascend close to the cyclone centre contributes to low-level PV production in strongly deepening cyclones. They concluded that explosively intensifying cyclones typically have strong WCB-related PV production in the cyclone centre (Fig. 14c). This appears consistent with a climatological analysis of the linkage between so-called "atmospheric rivers" (Newell et al., 1992; Ralph et al., 2004; Dacre et al., 2015) and cyclone intensification (Zhang and Ralph, 2021), given that these moist filaments often lead into the inflow and ascent regions of WCBs (Sodemann et al., 2020)[19]. Zhang and Ralph (2021) found that cyclones that develop along a pre-existing atmospheric river receive nearly 80% more water vapour inflow on average, which enhances latent heating and cyclone intensification. However, the results by Binder et al. (2016) showed that variability is large, and a minority of explosive cyclones have no WCB according to the trajectory ascent threshold set (and most likely no atmospheric river) whereas some cyclones with intense WCBs do not experience strong deepening. Overall, these climatological studies confirmed results from earlier case studies and idealised simulations that diabatic PV production is essential for most explosive cyclones and also indicated a relatively large variability in terms of the link between cyclone deepening and diabatic processes.

### 5.1.1 Summary

The availability of reanalyses enabled fascinating new possibilities for investigating the role of diabatic processes. Together with the design of automated algorithms, e.g., for cyclone identification and tracking, systematic studies became possible to extend the scientific understanding beyond case studies and to obtain climatologically robust results. Statistical relationships could be investigated and have shown, for instance, that deeper cyclones have higher-amplitude low-level PV anomalies produced by cloud diabatic heating and that explosively deepening cyclones typically experience strong low-level PV production related to intense latent heating in the WCB. However, systematic investigations based on reanalyses also revealed a substantial variability (between regions, seasons, and types of cyclones), such that, for instance, not all cyclones with an intense WCB do deepen strongly. Reanalyses were also essential for developing concepts for objective cyclone classifications and the analysis of specific cyclone categories (see next section). As a caveat of this research progress, we mention that the community developed in the last decades a potentially confusingly diverse set of diagnostic tools (e.g., for the identification of cyclones, jets and Rossby waves), which often complicates the direct comparison of results from different studies. This has led to the design of specific model and method intercomparison projects.

---

[19]Note though the alternative paradigm for the relationship between WCBs and atmospheric rivers proposed by Dacre et al. (2019) from cyclone-relative airflow analysis, namely that atmospheric rivers can instead be a consequence of the export away from the cyclone centre of moisture that originates in a cyclone "feeder airstream" that enters the warm sector from ahead of the travelling cyclone.

## 5.2 Consideration of radiative and surface flux related diabatic processes

Prior to 2000 the majority of studies on diabatic processes in extratropical cyclones considered the effects of latent heating in cloud, with a few notable exceptions (e.g., baroclinic instability studies with surface friction as briefly mentioned in Sect. 4.3.2). However, in recent years an increasing number of studies have instead investigated other diabatic processes, particularly radiation, turbulence and frictional processes, and surface heat fluxes (including the effects of varying SSTs). Here we outline the main advances made while largely excluding those studies that address these processes utilising the novel diagnostics described in Sect, 5.4, which are instead reviewed there. Here we also focus on studies applicable to "standard" extratropical cyclones, whereas studies relating specifically to special categories of cyclones are discussed in the appropriate subsection of Sect. 5.3.

### 5.2.1 Frictional processes

As stated in Sect. 4.1.2, Davis et al. (1993) and Stoelinga (1996) were the first to quantify the role of surface friction for PV modification in case studies of an extratropical cyclone. Using PV tracers they found large positive PV generation along the warm front where there was a large wind component along the front, opposite to the direction of the thermal wind. Stoelinga (1996) found up to 0.8 pvu after 36 h and smaller magnitude negative PV generation along the cold front where there is a cyclonic circulation in a stably stratified boundary layer. As explained by these studies, these frictional sources and sinks of PV can be diagnosed from consideration of the frictional source term for PV. This term is the second source term on the r.h.s. of Eq. 3 in this review and can be expanded as

$$\frac{DQ}{Dt}\bigg|_{friction} = \frac{1}{\rho}\left[\left(\frac{\partial F_y}{\partial x} - \frac{\partial F_x}{\partial y}\right)\frac{\partial \theta}{\partial z} + \left(\frac{\partial F_x}{\partial z}\right)\left(\frac{\partial \theta}{\partial y}\right) - \left(\frac{\partial F_y}{\partial z}\right)\left(\frac{\partial \theta}{\partial x}\right)\right]. \tag{9}$$

The first term on the r.h.s. of this equation leads to the generation of negative PV for a cyclonic vortex with a stably stratified boundary layer (such that $\partial\theta/\partial z > 0$) with the approximation that the frictional force $\mathbf{F} \propto -\mathbf{v}$ where $\mathbf{v}$ is the horizontal velocity vector. In contrast, the sign of the second term depends on the relative directions of the surface wind and the thermal wind. For example if the surface wind is easterly, such that the frictional force is eastwards, then, assuming that the frictional force decreases with height, $\partial F_x/\partial z < 0$ and there will be PV generation if $\partial\theta/\partial y < 0$, which means that the thermal wind is westerly (as illustrated schematically in Fig. 15a).

These case study findings are consistent with the earlier work of Cooper et al. (1992) who showed, assuming bulk formulae for surface fluxes of heat and momentum and that turbulence fluxes decrease from maximum values at the surface to zero at the top of the boundary layer, that the Lagrangian rate of change of diabatic and frictional PV vertically averaged over a (dry) boundary layer can be decomposed into four terms, described by Plant and Belcher (2007) as barotropic and baroclinic terms for the frictional generation of PV, the direct effect of turbulent heat fluxes, and a term proportional to the gradient of these fluxes (which they showed is negligible). The barotropic term is the PV analogue to the Ekman pumping spin-down mechanism whereby low-level ascent is generated by convergence in the boundary layer above the low-pressure cyclone centre, which acts to spin-down the circulation through vortex squashing. Thus, the negative and positive PV generation regions found by Stoelinga (1996) are attributable to the barotropic and baroclinic terms (the first and second terms on the r.h.s. of

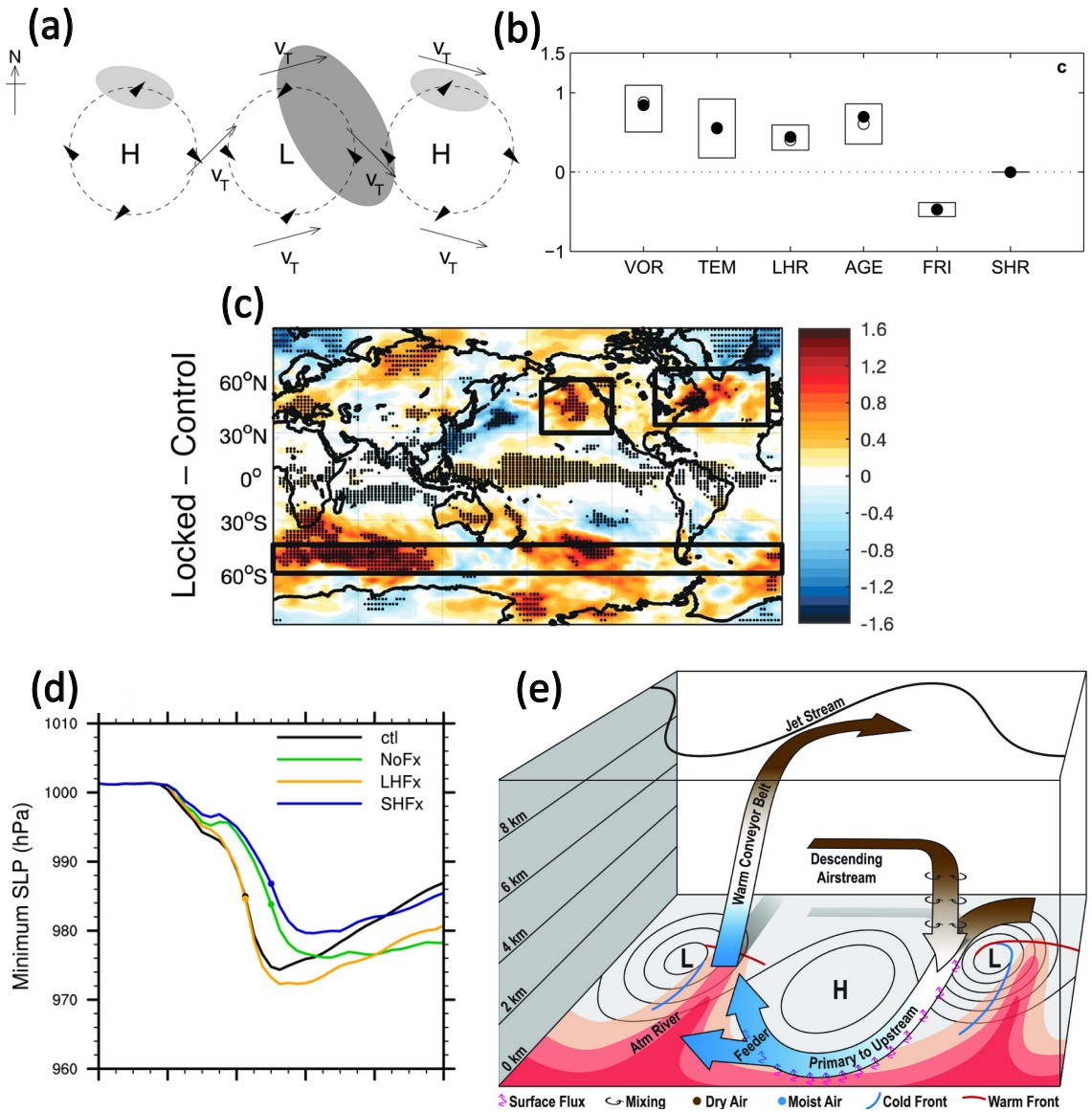

.

Eq. 9), respectively, using this terminology. The effect of latent heating was added as an additional term by Vannière et al. (2016). Using ERA-Interim reanalysis data, they argued for a PV-based indicator of the cold sector in cyclones based on their diagnosis of negative PV there from diabatic processes (surface sensible heat fluxes) and the barotropic frictional generation term in contrast to the positive PV generated by the baroclinic frictional term in the warm sector. A series of papers, mainly using idealised dry baroclinic life-cycle experiments, has explored the relative importance of Ekman pumping and the frictional baroclinic and barotropic terms in the PV budget in the spin down of cyclones. Adamson et al. (2006) and Plant and Belcher

**Figure 15** *(previous page).* Radiative and surface flux related and frictional effects on extratropical cyclones. **(a)** from Adamson et al. (2006, their Fig. 11c), schematic illustrating the baroclinic frictional mechanism for PV generation in the linear phase of a growing baroclinic wave where arrowheads denote near-surface winds ($\mathbf{v}_s$), long arrows denote the thermal wind ($\mathbf{v}_T$) and shading indicates regions where $\mathbf{v}_s \cdot \mathbf{v}_T$ is large with darker shading indicating negative values associated with baroclinic PV generation (and lighter shading indicating positive values associated with baroclinic PV destruction); **(b)** from Azad and Sorteberg (2014a, their Fig. 15c), total mean contributions (in units of $10^{-9}\,\mathrm{s}^{-2}$) to the 950-hPa geostrophic tendency at the cyclone centre for the entire development phase attributed to the vorticity advection (VOR), temperature advection (TEM), latent heat release (LHR), ageostrophic vorticity tendency (AGE), friction (FRI), and sensible heating (SHR) terms (boxes are the 25th—75th percentiles and white- and black-filled circles show median and mean values, respectively); **(c)** from Grise et al. (2019, panel from their Fig. 2), difference in annual-mean climatology of 850-hPa eddy kinetic energy for 2.5-to 6-day bandpass filtered eddies between runs with prescribed cloud properties (termed locked) and control runs (with standard interactive cloud-radiative effects), using a model coupled to oceans and sea ice, where stippling denotes statistically significant results; **(d)** from Bui and Spengler (2021, their Fig. 3a), time evolution of minimum MSLP (hPa) for a control idealised baroclinic wave simulation (with both sensible and latent heat fluxes) and simulations with no sensible or latent heat fluxes (NoFx), latent heat fluxes only (LHFx) and sensible heat fluxes only (SHFx); and **(e)** from Demirdjian et al. (2023, their Fig. 16), airstreams schematic with, in addition to the under-panel legend, orange-to-red coloring indicating small-to-large vertically integrated water vapour, "Atm River" the abbreviation for atmospheric river, black contours of MSLP at the surface, and a single contour of upper-level pressure. All figures used with permission: (a,c) from Wiley, and (b,d,e) from the American Meteorological Society.

(2007) showed that the baroclinic generation of PV dominated over the barotropic generation. The baroclinic term generates positive PV anomalies that are advected by the cyclone from the warm sector to above the surface low centre. There the PV anomaly damps the baroclinic growth due to the associated static stability anomaly (which is large because the PV anomaly is constrained to a thin layer) inhibiting baroclinic coupling between the upper and lower tropospheric layers. This damping occurs despite the inversion of the positive PV anomaly possibly giving rise to a (weak) low-level cyclonic circulation (as found by both Davis et al. (1993) and Stoelinga (1996)). In contrast, the negative PV anomaly produced by the barotropic term remains confined to the lower part of the boundary layer. Consistent with the damping effect of the baroclinic term, both Stoelinga (1996) and Davis et al. (1993) found that a more intense cyclone developed in simulations without surface friction (with an inviscid simulation in Davis et al. (1993) yielding the extremely low MSLP of 917 hPa). However, these studies attributed the intensity enhancement to other causes; e.g., Stoelinga (1996) described an enhancement of PV generation by temperature diffusion and a slight enhancement of the descent of the upper-tropospheric PV anomaly, an effect also found in the idealised GCM simulations performed with varying surface friction by Hines and Mechoso (1993).

Conversely to Adamson et al. (2006) and Plant and Belcher (2007), Beare (2007) found that the region of the cold conveyor belt (a rearward-directed low-level airstream on the cold side of the warm front), where there was greatest surface stress, dominated the frictional damping of cyclone growth (the Ekman pumping mechanism), which was associated with a well-mixed boundary layer and therefore had little PV signature. Boutle et al. (2007) then showed that these different conclusions were a consequence of the assumptions made regarding the SST distributions (which affected the surface temperature fluxes) and of

1725 the idealised cyclone initialisations in the two studies. More recently these two damping mechanisms were unified in the study by Boutle et al. (2015), which showed that Ekman pumping aids the ventilation of the baroclinically generated PV from the boundary layer and leads to its strong static stability signature. An alternative approach to diagnosing the importance of surface friction in cyclone spin down was used by Azad and Sorteberg (2014a, b). They combined an equation for diagnosing the near-surface geostrophic vorticity tendency, developed by Zwack and Okossi (1986), with the omega equation so as to partition

the adiabatic term in the former into different forcing mechanisms: vorticity and temperature advection, latent heat release, ageostrophic vorticity tendency, friction (calculated using a balance equation in the boundary layer) and sensible heating. From diagnosing these partitioned terms for the developing and decaying stages of 100 North Atlantic winter storms (in reanalysis data) they found a significant role of friction in opposing development (Azad and Sorteberg, 2014a, and Fig. 15b here) and demonstrated the dominant role of the ageostrophic vorticity tendency and friction terms in decay (Azad and Sorteberg, 2014b),

noting that adiabatic cooling associated with the Ekman pumping effect reinforces the damping effect of friction. Finally, another series of papers has diagnosed the role of turbulence processes in cyclones through the PV tracer and PV tendency diagnostics described in Sect. 5.4. An important distinction here is between the momentum and temperature contributions to the diagnosed turbulence term. These contributions are combined in some studies (and also sometimes referred to as a boundary layer term even though there will be contributions from above the boundary layer) making it difficult to disentangle the specific

role of surface friction. As these studies typically consider multiple diabatic processes as well as frictional processes, discussion of these studies is left to the Sect. 5.4.

### 5.2.2 Radiative processes

As described in Schäfer and Voigt (2018), the role of radiative processes in extratropical cyclone development has received little recognition in comparison to that of latent heating in cloud, likely due to its relatively slow impact, for example in terms of the

1745 generation of PV anomalies as shown in the PV tracer study of Chagnon et al. (2013). However, a small number of recent studies have isolated the role of radiative processes with apparent conflicting results. Here we consider only studies in which the impact of radiative processes on cyclones is addressed, excluding those in which cloud radiative forcing associated with cyclones is considered without consideration of its impacts on the cyclones themselves. The idealised baroclinic lifecycle experiment of Kunkel et al. (2016) addressed the effects of diabatic processes on the tropopause inversion layer (a quasi-permanent inversion

layer above the thermally defined tropopause) rather than on cyclone dynamics. However, their results showed that the addition of radiative processes to a dry dynamics simulation led to a reduction in deepening of the surface cyclone for both cyclonic and anticyclonic lifecycle experiments. Schäfer and Voigt (2018) instead focused specifically on the impact of radiation on cyclone dynamics. Also using idealised baroclinic lifecycle experiments with and without atmospheric radiative heating (which is dominated by longwave heating), they found that radiation substantially weakened the cyclone leading to a reduction of peak

eddy kinetic energy by 50% and an increase in the storm minimum central pressure of 17 hPa. Cloud-radiative interactions were responsible for between one third and one half of the total radiative impact. Schäfer and Voigt (2018) hypothesised that the weakening resulted from radiation acting to reduce mid-tropospheric PV by causing long-wave cooling at cloud top and heating at cloud base. The later study of Grise et al. (2019) examined specifically the role of cloud-radiative effects on the extratropical

storm tracks using GCM experiments and found, despite significant differences in experimental design, conclusions similar to Schäfer and Voigt (2018), specifically a damping effect on extratropical cyclones with decreased intensity and shorter lifetimes when radiative effects are coupled to the dynamics (Fig. 15c which shows enhanced low-level eddy kinetic energy in the storm tracks (black boxes) when interactive cloud-radiative effects are prevented). Grise et al. (2019) noted that their results highlighted the importance of the longwave radiative heating at cloud base in suppressing eddy kinetic energy. However, from idealised baroclinic lifecycle experiments using the same model as used by Schäfer and Voigt (2018), but applying a new method to isolate the impact of cloud-radiative heating on the cyclone (avoiding the changes in the mean state due to clear-sky radiative cooling that were included in Schäfer and Voigt (2018)), Keshtgar et al. (2023) found that cloud-radiative heating increased cyclone kinetic energy. One reason for the apparent conflict between these studies in terms of the impact of radiative processes on cyclone intensification has recently been proposed by Voigt et al. (2023) by teasing apart the differences between the Schäfer and Voigt (2018) and Keshtgar et al. (2023) studies. Voigt et al. (2023) attributed the different findings to the abundance of low-level clouds. The more abundant clouds in the Schäfer and Voigt (2018) study led to stronger radiative cooling of the boundary layer, which acted to suppress the cyclone by weakening static stability. In contrast, as found by Keshtgar et al. (2023), radiative cooling at cloud top sharpens the tropopause and can also lead to a reduction in static stability in the upper troposphere enhancing cyclone growth. Hence, the overall impact of radiative processes on cyclone dynamics, even in terms of whether it acts to damp or enhance growth, appears to be sensitively dependent on the vertical distribution of clouds within cyclones.

Finally, some studies have addressed the impact of radiative processes specifically on the tropopause PV gradient (Saffin et al., 2017) or the related tropopause inversion layer (Kunkel et al., 2016) within cyclones. These studies agreed that longwave radiative cooling enhances the sharpness and strength of these features, respectively, although both studies explained how cloud presence and processes are intertwined with the radiative effects. The PV tracer study of Chagnon et al. (2013) also found that longwave radiation resulted in a strong positive PV anomaly above the tropopause as a consequence of the strong humidity gradient there as well as cloud top cooling. They argued that the enhancement of the upper-level PV wave amplitude resulting from the tropopause-sharpening cross-tropopause PV dipole generated in combination by radiative and other diabatic processes leads to an enhancement of the growth rate of the associated baroclinic wave (illustrated schematically in Fig. 17b).

### 5.2.3 Surface fluxes and the effects of SST anomalies

As mentioned in Sect. 3.4, some of the very early studies on diabatic processes in rapidly developing cyclones, in addition to latent heating in clouds also pondered about the role of surface heat fluxes. In the 1980s, the role of surface fluxes was discussed for the rapid deepening of the catalyst cyclones (Sect. 4.1.1), and sensitivity studies with surface fluxes turned off revealed a very large case-to-case variability of the effect on cyclone intensification (Sect. 4.2.1). In the last two decades, several studies re-investigated the role of surface sensible and latent heat fluxes on cyclone development and precipitation in cyclones in a more systematic way. Many of these studies focused on regions with oceanic western boundary currents (the Gulf Stream in the North Atlantic and the Kuroshio extension in the North Pacific), which feature strong SST gradients, the strongest midlatitude surface heat fluxes, and the highest frequency of rapidly deepening extratropical cyclones. Several of these studies

explicitly investigated the effect of SST anomalies on surface fluxes, on the pathway of evaporated moisture, and eventually on cyclone precipitation and cyclone intensification. Bui and Spengler (2021, their Sect. 1) provided a concise summary of how the SST distribution can modulate the development of extratropical cyclones through surface heat fluxes. Here, we highlight some of the recent studies, emphasising again the diversity of approaches, including diagnostics based on reanalyses, real case sensitivity experiments, and idealised baroclinic wave simulations.

For the western North Atlantic, Tsopouridis et al. (2021) classified reanalysis cyclone tracks according to whether they intensify on the northern or southern side of the Gulf Stream SST front (categories C1 and C2), respectively, or whether they cross the front from the warm to the cold side during intensification (C3). Cyclones in categories C1 and C3 on average intensified more than cyclones that remained on the warm side of the SST front (C2). C3 cyclones featured the highest fraction of explosive cyclones, however, low-level baroclinicity was lower around C3 cyclones than around C1 cyclones. The more rapid intensification of C3 cyclones was attributed to the increased latent heat release within the cyclones' WCB. The authors concluded that the role of the SST front is secondary for the low-level baroclinicity and that the intensification of cyclones near the Gulf Stream is not directly associated with the intensity of surface heat fluxes. These diagnostic results appear to differ from those based on model experiments of selected Gulf Stream cyclones. For example, for a rapidly deepening East Coast winter storm, Jacobs et al. (2008) found that a coarse SST representation was detrimental for simulating storm intensification, as it does not capture important thermal gradient features of the Gulf Stream. However, these results are not necessarily in disagreement: the former study shows that cyclones crossing the SST front are not systematically experiencing higher baroclinicity compared to cyclones north of the front, which does not exclude the possibility that for cyclones that cross the front, the details of the SST front matter.

To gain more insight into the role of the SST pattern and surface heat fluxes, a series of studies performed sensitivity experiments with modified SSTs. Modification patterns used were either a smoothing of the SST gradient (which is equivalent to putting a cold anomaly on the warm side of the front and a warm anomaly on the cold side), imposing local SST anomalies (which in turn modifies SST gradients), or uniformly changing the SST (with no effects on SST gradients). Booth et al. (2012) and Sheldon et al. (2017) performed cyclone case studies in the Gulf Stream region with smoothed and uniformly modified SSTs. In the first of these studies, cyclone intensity increased monotonically with the magnitude of the uniform SST perturbation (varying from $-3.6$ to $+3.6\,\mathrm{K}$), and this response was driven by the latent heat release in the WCB that ascends from the warm side of the SST front. This suggests that, when considering the role of the SST on the development of already existing cyclones, the absolute values of the SST along the cyclone track matter. Consistently, Sheldon et al. (2017) found that, compared to the control simulation, the intensity of the WCB was strongly reduced in simulations with smoothed or uniformly cooled SSTs. They referred to this mechanism as a "warm path" for Gulf Stream–troposphere interactions, indicating that SST variations on the warm side of the SST front can have a major influence on cyclones and the downstream upper-level flow, via modifying the intensity of the WCB. This link is further discussed in Sect. 5.5. In related work, Ludwig et al. (2014) assessed the importance of the anomalously high SSTs experienced by the damaging windstorm Xynthia in 2010 as it followed an unusual track originating in the subtropical North Atlantic. Surface fluxes were reduced through experiments with perturbed

SSTs and laminar boundary roughness for heat and shown to reduce and retard development (and weaken the cyclone's PV tower).

Slightly different results about the importance of the WCB were found for explosive cyclones crossing the SST gradient near the Kuroshio Extension (Hirata et al., 2015, 2016). In their simulations of two cyclones, strong diabatic heating occurred over the bent-back fronts. According to trajectory calculations, this heating was mainly due to condensation of moisture imported by the cold conveyor belt. The initially dry air in this airstream received large amounts of moisture due to intense latent heat fluxes when the cyclone was still on the warm side of the SST gradient and the airstream moved over the warm currents, in particular in sensitivity experiments with warm SST anomalies. These results indicate that the well-organised nature of the cold conveyor belt can play a vital role not only in stimulating surface evaporation from the warm currents but also in importing the evaporated vapour into the vicinity of the cyclone's centre where it converges and in turn contributes to the low-level PV production and cyclone intensification. However, also to reconcile the results with the findings of the studies that emphasised the key role of the WCB raising from the warm side of the SST front, it seems plausible to assume that the role of SST anomalies, surface fluxes, and different moisture transport pathways may be very sensitive to details of the cyclone evolution relative to the SST pattern (e.g., the cold conveyor belt most likely can only play an important role if it is already fully established before the cyclone moves to the cold side of the SST front).

Further insight was obtained from idealised baroclinic wave simulations with modified SST patterns. Boutle et al. (2010) studied the moisture budget in a growing baroclinic wave and found that latent heat fluxes are largest into the post-frontal and anticyclonic boundary layer. The evaporated moisture is then transported over large distances within the boundary layer until it converges in the WCB inflow region of the growing cyclone. For the theme of this section, the important finding from this idealised study is that moisture that evaporates in a given cyclone can be relevant for condensation, precipitation, and diabatic PV production in the subsequent system. Bui and Spengler (2021) performed idealised baroclinic wave simulations with surface latent and sensible heat fluxes turned on or off (Fig. 15d, which showed that the latent heat fluxes enhance deepening whereas the sensible heat fluxes slightly dampen it) and with different SST distributions, varying either the absolute SST, the SST gradient, or the meridional position of the SST front. In agreement with real case experiments mentioned above, the surface latent heat flux, which increases with absolute SST, plays a key role in enhancing the moist baroclinic development. The moisture provided by these fluxes originates from about 1000 km ahead of the cyclone a day prior to the time of the most rapid deepening, when it ascends within the WCB. Overall, in these simulations, cyclone intensification is more sensitive to absolute SST compared to SST gradients. Using a highly idealised Eady model setup, Haualand and Spengler (2020) quantified direct and indirect effects of surface fluxes on baroclinic instability. Direct effects are the diabatic generation of eddy available potential energy due to sensible surface fluxes, whereas indirect effects refer to modifications of the circulation and latent heating via both sensible and latent heat fluxes. They found that surface sensible heat fluxes have a minor detrimental impact, whereas latent heat fluxes are important in determining the moisture, and eventually latent heating, in the WCB. Channel model experiments by Demirdjian et al. (2022) with suppressed surface fluxes confirmed the crucial role of latent heat fluxes for precipitation and latent heating in cyclones. In their simulations, moisture sources for precipitation near the cyclone centre span a relatively wide area covering both WCB and cold conveyor belt regions. The authors therefore emphasise "that the

answer to where the moisture is sourced from does not need to be binary". In a follow-up study, Demirdjian et al. (2023) simulated upstream and downstream development in a long baroclinic channel and found that moisture evaporated from the postfrontal sector of the primary cyclone travels south of the upstream anticyclone feeding into the WCB of the upstream cyclone. The WCB moistens substantially as a result of surface latent heat fluxes in both the primary cyclone's postfrontal sector and along the southern flank of the anticyclone. These results agree qualitatively well with the interpretation of moisture source diagnostics in real cyclones (e.g., Pfahl et al., 2014; Dacre et al., 2019; Papritz et al., 2021). Useful schematics about how WCB inflow moisture is related to remote surface fluxes can be found in Bui and Spengler (2021, their Fig. 11) and Demirdjian et al. (2023, Fig. 15e here).

### 5.2.4 Summary

Recent work considering diabatic processes other than latent heating in cloud as well as frictional effects has revealed the complexity of their effects on extratropical cyclones. Surface friction can lead to either positive, negative or negligible PV anomalies dependent on the frictional PV generation mechanism and distribution of surface temperature fluxes, although there is consensus that surface friction dampens cyclone development. Radiative processes have been found to either damp or enhance cyclone growth dependent on vertical cloud distributions, which determine whether long-wave cooling in the boundary layer or at cloud top dominates growth through modification of the cyclone PV distribution and tropopause sharpness. Considering surface fluxes, the synthesis of results from idealized and real case studies shows that their effect on extratropical cyclones depends strongly on the track of the cyclone relative to the underlying SST and on the evolution and structure of the cyclones (which determines surface winds that essentially influence surface fluxes). Sensitivity experiments with modified surface fluxes reveal fairly consistently that surface latent heat fluxes, which increase with absolute SST, enhance moist baroclinic development. These fluxes tend to be largest in the cold sector of cyclones, and moisture evaporated from these regions is often transported into the WCB inflow of the subsequent upstream cyclone where it leads to latent heating and cyclone intensification.

### 5.3 Diabatic processes in (special categories of) extratropical cyclones

The following subsections review studies[20] that have analysed diabatic processes in cyclone types that differ from the classical extratropical cyclones found in the main storm tracks over the North Atlantic, North Pacific and Southern Ocean. Specifically, we first consider an objective categorisation of cyclones that includes a category in which diabatic processes have a significant impact on the cyclone structure and evolution (Sect. 5.3.1) and then consider the following special categories of cyclones: subtropical cyclones (Sect. 5.3.2), Mediterranean cyclones (Sect. 5.3.3), polar lows (Sect. 5.3.4), diabatic Rossby waves (Sect. 5.3.5), tropical cyclones undergoing extratropical transition (Sect. 5.3.6), and upper-level cutoffs and tropopause polar vortices (Sect. 5.3.7). The length of these subsections is not uniform, due to personal preferences of the authors and to the fact that excellent review articles were recently published about some of these categories, e.g., on extratropical transi-

---

[20]In these subsections we sometimes deviate from our concept of strictly structuring the review according to time periods, i.e., we also include here some pre-2000 material. The justification for this rule breaking is that, in these cases, we regarded the thematic connection more important than the chronology.

tions. As for classical extratropical cyclones, PV concepts have frequently been used to assess the role of diabatic processes in these special cyclones. A sample vertical cross section illustrating the PV structure of each of the six cyclone types reviewed, and additionally monsoon depressions (also mentioned in Sect. 5.3.5), is shown in Fig. 16. With the exception of Fig. 16g, a composite PV field for tropopause polar vortices, the cyclones have in common (with each other and with classical cyclones that have strong diabatic forcing) enhanced amplitude lower-tropospheric diabatically generated PV, typically in the form of a vertical tower.

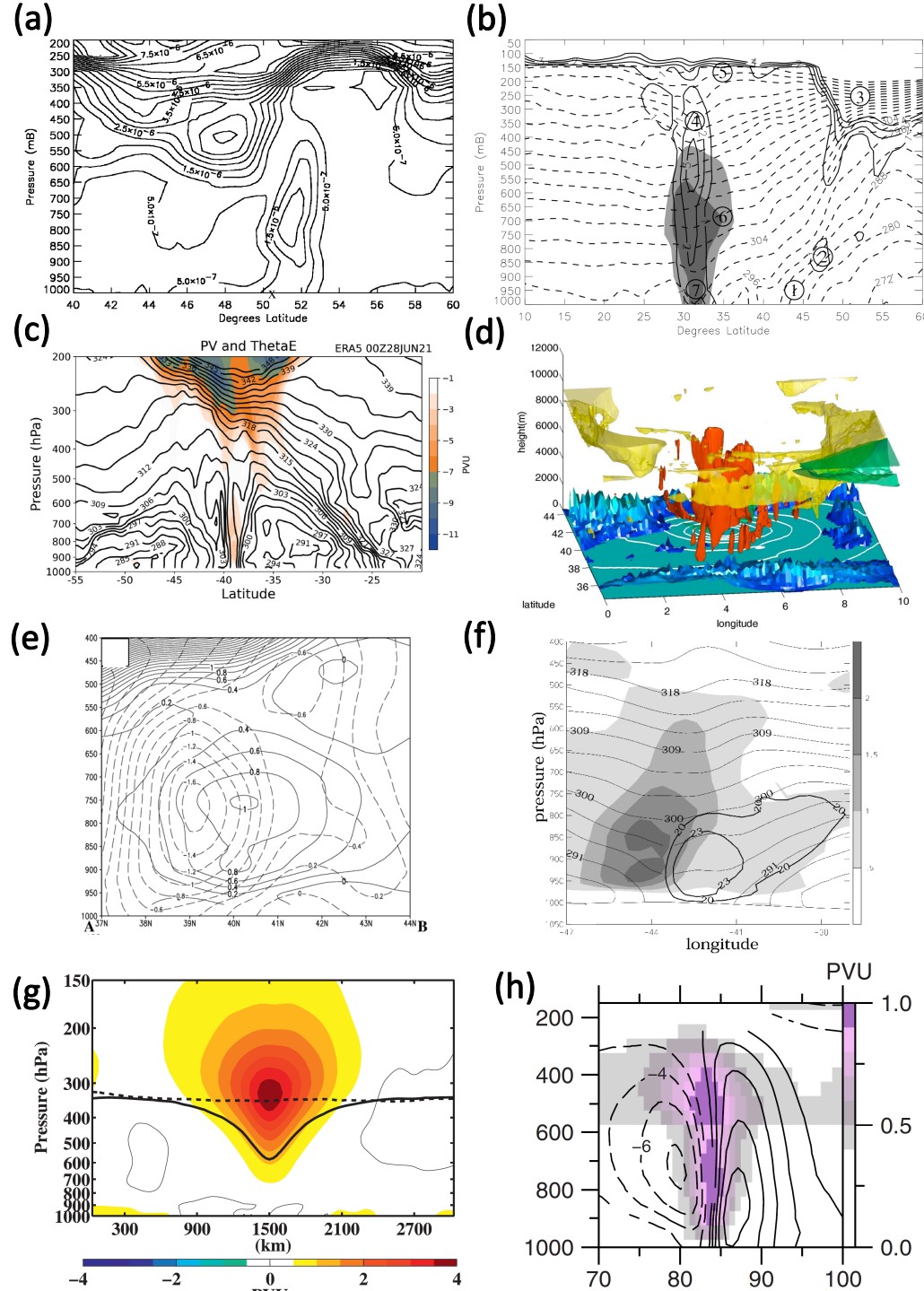

**Figure 16** *(previous page)*. Vertical cross-sections of PV in special cyclone categories. **(a)** from Plant et al. (2003, their Fig. 4a), north-south section through the centre of the cyclone observed during FASTEX IOP15 showing PV (pvu, contours) with the cross marking the location of minimum MSLP; **(b)** from Agustí-Panareda et al. (2004, their Fig. 3), north-south section through hurricane Irene as it undergoes extratropical transition showing PV (solid contours of 1, 2, 3, and 4 pvu) with $\theta$ (K, dashed contours), mixing ratio (shaded, $3 - 5 \, \mathrm{g \, kg^{-1}}$ light grey and $5 - 7 \, \mathrm{g \, kg^{-1}}$ dark grey) and labelling of the different PV anomalies; **(c)** from Reboita et al. (2022, their Fig. 7), section through Southern Hemisphere subtropical cyclone Raoni showing PV (only negative values shaded–recall that values are predominately negative for Southern Hemisphere systems) and $\theta_e$ (K, solid contours); **(d)** from Miglietta et al. (2017, their Fig. 2b), 3D view of a medicane showing 1-pvu isosurfaces of PV (yellow) and convectively produced diabatic PV (red) with the $30 \, \mathrm{m \, s^{-1}}$ isosurface of wind speed (green), MSLP isobars (white contours, 4 hPa interval) and orography (blue); **(e)** from Guo et al. (2007, their Fig. 5c), section through a polar low over the Japan Sea showing PV (pvu, solid contours) and omega ($\mathrm{Pa \, s^{-1}}$, dashed contours); **(f)** from Wernli et al. (2002, their Fig. 13a), east-west section of storm Lothar at the diabatic Rossby wave development stage showing PV (shaded) with $\theta$ (K, thin lines) and southerly wind ($\mathrm{m \, s^{-1}}$, thick contours); **(g)** from Cavallo and Hakim (2010, their Fig. 9c), east-west section of composite tropopause polar vortex showing anomalous PV (shading) with the composite and background tropopause in the thick solid and dashed lines, respectively; and **(h)** from Hurley and Boos (2015, their Fig. 9b), section through a regional monsoon depression composite for India showing PV (shading) with meridional wind ($\mathrm{m \, s^{-1}}$, contours). All figures used with permission: (a,b,c,d,f,h) from Wiley, (e) from Elsevier, and (g) from the American Meteorological Society.

### 5.3.1 Cyclone categorisation

The previous section summarised a set of studies that used reanalyses with the aim to diagnose the climatological and therefore, in comparison to earlier case studies, statistically more robust role of diabatic processes for the intensification of extratropical cyclones. The climatological cyclone datasets produced in these studies also offered the potential for objectively classifying cyclones—a theme that existed since the classical type A vs. B classification by Petterssen and Smebye (1971)—and for analysing the role of diabatic processes for the dynamics of specific types of extratropical cyclones. The Petterssen and Smebye

(1971) classification scheme classifies cyclones according to interactions between the upper-tropospheric trough and low-level frontal thermal gradient. The intensification of type A cyclones is mainly driven by low-level thermal advection whereas for type B cyclones the vorticity advection due to the approaching upper-tropospheric trough dominates. These two terms (or more precisely the horizontal Laplacian of thermal advection and vertical derivative of relative vorticity advection) are the two source terms in the omega equation (conventional formulation with $f$-plane approximation). The cyclone can be classified

by considering both the relative upper- and lower-tropospheric forcing strength and the evolution of the horizontal distance between the trough and surface cyclone (the tilt): the tilt stays constant until peak intensity is reached for type A cyclones, whereas it decreases during intensification for type B cyclones as the trough "catches up" with the surface cyclone [see Table 1 of Deveson et al. (2002) for a summary comparison between type A and B cyclones].

Deveson et al. (2002) and Plant et al. (2003) developed an objective method to extend the type A and B classification by

1915 including a third type C. This classification was based essentially on the ratio of the quasi-geostrophic vertical motion at 700 hPa forced by upper and lower-tropospheric levels, termed $U/L$. Although this diagnostic does not explicitly consider

latent heating, Plant et al. (2003) used in addition a PV inversion approach and found that the dynamics of type C cyclones, which have the highest values of $U/L$, are strongly influenced by mid-level latent heating that can generate important low-level anomalies of PV (Fig. 16a) and suppress the formation of strong near-surface thermal anomalies. The convective outflow and associated negative PV anomaly also acts to reduce the strength of the leading edge of the upper-tropospheric trough, so potentially maintaining the rearward tilt favourable for baroclinic intensification. The explosively deepening FASTEX IOP18 cyclone (Ahmadi-Givi et al., 2004) was considered as a typical type C cyclone. This was the first study that considered intense latent heating as a key characteristic of a particular category of extratropical cyclones. Later, Gray and Dacre (2006) produced a North Atlantic climatology distinguished into type A, B, and C cyclones and found similar proportions for each type. Dacre and Gray (2009) determined that when cyclones originating in the east and west North Atlantic were considered separately, then, while type B cyclones dominated in both regions, for the remaining cyclones a higher proportion of type C cyclones were found in the east Atlantic. Graf et al. (2017), using a very different classification approach based on a principal component analysis in a high-dimensional phase space of potential precursors of cyclogenesis (including variables related to diabatic processes like low-level PV and vertically integrated water vapour), found that the second principal component projects strongly on diabatic processes. This indicates that diabatic processes serve to meaningfully classify cyclones and that, in line with the type C category, they are particularly important for the development of a certain category of cyclones. In their comprehensive review on cyclone classification, Catto (2016) mentioned the importance of diabatic processes for type C cyclones, frontal wave cyclones, and for cyclones in specific geographical regions like the Mediterranean. Catto also emphasised the importance of applying cyclone classifications in climate simulations and of quantifying diabatic heating in cyclone development to address the impact of future changes associated with increased moisture availability (see Sect. 5.7).

Other categorisations did not emerge from statistical analyses, but either from geographical considerations (e.g., subtropical cyclones, Mediterranean cyclones, Tibetan plateau vortices) or specific processes involved (e.g., tropical cyclones transforming into extratropical cyclones or very rapidly propagating and shallow extratropical cyclones). Important studies that investigated the role of diabatic processes in these systems are summarised in the following.

## 5.3.2 Subtropical cyclones

Subtropical cyclones were first mentioned by Evans and Guishard (2009) and Guishard et al. (2009) as a category of cyclones that form at relatively low latitudes and should be distinguished both from classical tropical and extratropical cyclones. Subtropical cyclones develop in baroclinic zones, but baroclinicity is typically weaker and shallower than for extratropical cyclones, and deep convection is mentioned as a typical characteristic. Davis (2010) investigated their dynamics in idealised moist baroclinic channel simulations, which showed the formation of relatively small-scale, intense, and essentially diabatically driven cyclones. In terms of PV, their dynamics is characterised by the interaction of an upper-level disturbance on the subtropical jet, which thermodynamically destabilises the troposphere, with a low-to-mid-level PV anomaly related to organised deep convection, and a negligible surface $\theta$ anomaly (similar to type C cyclones discussed in Sect. 5.3.1). Davis (2010) concluded that "subtropical cyclones ... belong to a general class of previously identified cyclonic disturbances in which the principal agent of development is condensation heating, but where there is a guiding, baroclinic structure that organises the

heating". Building on ideas from the idealised study by Davis (2010), Bentley et al. (2016) used reanalyses to study about 200 subtropical cyclones in the western North Atlantic, which formed in the presence of an upper-tropospheric disturbance and transformed into tropical cyclones. They used strong diabatic PV production at low-levels as a criterion for the identification of subtropical cyclones. A rare event of a cold season subtropical cyclone over the South Atlantic was described by Reboita et al. (2022), which evolved from a mature extratropical cyclone with a prominent and moist warm seclusion. Important for the transition into a subtropical cyclone was an upper-level PV cutoff, which provided an environment with weak vertical wind shear favouring the organisation of deep convection and the diabatic production of a low-level PV anomaly in the centre of the subtropical cyclone (Fig. 16c). Numerical sensitivity experiments where turbulent heat fluxes were turned off produced a much weaker cyclone and revealed the importance of air-sea interaction for these intense cyclones, in agreement with Quitián-Hernández et al. (2020), who studied a subtropical cyclone in the eastern North Atlantic. Together, these studies clearly show the importance of diabatic processes (in particular, surface fluxes and low-level PV production, and upper-level PV erosion due to latent heating in organised deep convection) for the dynamics of this type of cyclones.

A special property of subtropical cyclones is that some of them transform into tropical cyclones, a process referred to as tropical transition (Davis and Bosart, 2004). The most likely first case study of this process was presented by Bosart and Bartlo (1991), who discussed the initial development of tropical storm Diana (1984) in a baroclinic environment. Diabatic PV production at low levels in an initially baroclinic system and PV erosion at upper levels were identified as important factors for the transition. Based on a numerical simulation of the genesis of Diana (1984), Davis and Bosart (2001) concluded that the low-level circulation was strengthened through the axisymmetrisation of PV anomalies that were generated by condensational heating and then advected toward the centre of the incipient tropical storm (their Fig. 9). Investigating other simulations of tropical transition events, the follow-up study by Davis and Bosart (2003) again emphasised the role of diabatic processes for the dramatic vertical shear reduction that accompanies tropical transitions. Diabatic outflow near the tropopause in the initially baroclinic system (discussed in detail in Sect. 5.5) can laterally displace the gradients of PV away from the surface circulation. The tropical transition concept was also used to study events of Mediterranean cyclones that developed into so-called medicanes (e.g., McTaggart-Cowan et al., 2010b), as discussed in the next section.

### 5.3.3 Mediterranean cyclones

An important part of cyclone research in the last 20 years considered cyclones in the Mediterranean. These cyclones are interesting for different reasons (Flaounas et al., 2022): (i) compared to extratropical cyclones in the main storm track regions, they have a different geographical setting with the comparatively small ocean basin of the Mediterranean enclosed by several high mountain ranges and partly arid continental areas, (ii) Mediterranean cyclones affect highly populated coastal regions, (iii) and despite their limited size and less-deep minimum MSLP value, they can produce high impact weather with extreme precipitation and near-surface winds. The frequent occurrence of Mediterranean cyclones was realised already in the 1950s (see the detailed review by Flaounas et al., 2022). Those in the western Mediterranean in the lee of the Alps, also referred to as "Alpine lee cyclones", were of specific interest during the field experiment ALPEX in 1982 (mainly focusing on dry dynamical aspects, see Sect. 4.1.2). Thirty years later, the special observation period of HyMeX in 2012 (Ducrocq et al., 2014) considered

the interplay of Mediterranean cyclones with the hydrological cycle, and thereby also the role of diabatic processes for their dynamics. This field campaign, which was embedded in the multidisciplinary HyMeX program from 2010–2020 (Drobinski et al., 2014), together with the earlier cyclone-specific initiative MEDEX from 2000–2010 (Jansa et al., 2014), triggered international collaboration and intensified research activities about the specific dynamical characteristics of Mediterranean cyclones.

In 2007, the term "medicane" appeared in the scientific literature (Fita et al., 2007) to describe a peculiar and rare sub-category of intense Mediterranean cyclones with tropical-like characteristics (e.g., a cloud-free "eye" observable in satellite imagery), as occasionally described in earlier studies (e.g., Ernst and Matson, 1983; Pytharoulis et al., 2000; Emanuel, 2005). These systems have received a lot of attention in recent years, but a consensus definition of medicanes is currently missing (Flaounas et al., 2022). Similar to subtropical cyclones, medicanes appear as hybrids (Fita and Flaounas, 2018) with baro-clinic dynamics triggered by an upper-level PV anomaly and intense deep convection and latent heating in their warm core. Numerical sensitivity experiments with selected physical processes turned off at different times of the evolution of a medicane that occurred in September 2006 (Davolio et al., 2009) indicated that surface fluxes were important mainly in the initial phase of the cyclone deepening, whereas convective heating was essential during the transition to a tropical-like cyclone. However, case-to-case variability of the relative role of different processes is large, as revealed by the analysis of a set of medicanes by Miglietta and Rotunno (2019) and Dafis et al. (2020). Miglietta et al. (2017) investigated the vertical PV structure of five medicanes and showed that, in the mature stage, the upper-level PV anomaly evolved differently for each case. An intense PV tower, typical for intense extratropical cyclones (Sect. 4.1.2), formed only in one of the medicanes and was disabled by intense upper-level diabatic PV erosion in the other cases (see the example in Fig. 16d).

For the general ensemble of Mediterranean cyclones, several studies investigated in detail their PV structure, combining methodologies developed previously for cyclones in the North Atlantic and North Pacific storm tracks (Sect. 4.1) with novel approaches like the factor separation method (Stein and Alpert, 1993), which is a systematic approach using model sensi-tivity experiments with a set of "factors" being turned on or off. These factors can be physical parametrisations or, as for the Mediterranean cyclone study by Romero (2008), a set of PV anomalies in the upper and lower troposphere, and surface thermal anomalies, respectively. For an intense cyclone in November 2001, which led to devastating winds and flooding near Algiers (Algeria) and in the Balearic Islands, this approach revealed the important role of diabatically produced PV during the intensification of this comparatively small-scale cyclone with a minimum MSLP of about 995 hPa. A similar set of simulations for another intense cyclone in the western Mediterranean in September 1996, also including simulations with removed Atlas mountains and with surface fluxes turned off, identified the joint action of the upper-level anomaly, as a spin-up agent, and the latent-heat flux, for maintaining convection, as the primary factor for the genesis and evolution of the cyclone.

Flaounas et al. (2015) investigated the composite vertical PV structure of the 200 most intense Mediterranean cyclones simulated in a 20-year regional climate simulation and found that, in terms of interaction between prominent PV anomalies at upper and lower levels (the latter produced diabatically), Mediterranean cyclones are similar to extratropical cyclones over the main storm track regions. However, they rarely fulfil the criterion for "explosive deepening" with on average only about 6 events per year in the entire basin (Kouroutzoglou et al., 2011). Such an explosive cyclone occurred in January 2004 with a core

pressure of 972 hPa and associated with one metre of snowfall in eastern Greece (Lagouvardos et al., 2007). Model sensitivity experiments, with either surface fluxes turned off or with removing the upper-level PV anomaly in the initial conditions via PV inversion, indicated that the upper-level anomaly was essential for the explosive deepening, while surface fluxes contributed to its further deepening during the cyclone's mature phase. Flaounas et al. (2021) quantified the baroclinic and diabatic forcing for 100 intense Mediterranean cyclones by applying piecewise PV inversion to the diabatically produced PV anomalies (determined using PV budget diagnostics, see Sect. 5.4.1) and found that, while both forcings were important for most cyclones, there was an inverse relationship between their strength (see their Fig. 10); interestingly, the 10 cyclones that have been diagnosed as medicanes in previous case studies did not appear as a distinct subset with the strongest diabatic forcing.

Flaounas et al. (2021) also considered frictional PV production in Mediterranean cyclones. They found a positive contribution from momentum mixing and frictional forces to the low-level PV averaged over all 100 cyclones, and a particularly strong contribution (comparable to the PV produced due to latent heat release) for a few cyclones that developed near the Gulf of Genoa. This result is qualitatively consistent with the model sensitivity experiment by McTaggart-Cowan et al. (2010a), who found that the formation of a medicane was suppressed in an experiment where the Alps were removed. Scherrmann et al. (2023) systematically investigated the origin of low-level PV in almost 3000 Mediterranean cyclones in ERA5 and found that in almost 20% of the cyclones, PV production near the mountains surrounding the Mediterranean Basin played a significant role in forming the low-tropospheric PV anomaly and therefore in determining the intensity of these cyclones. Although their method could not separately quantify PV production due to friction or latent heating, it is very likely that frictional PV production, e.g. in the form of PV banners (Sect. 4.1.2), played an important role for the intensification of their category of "orographic cyclones", which occur mainly South of the Alps, north of the Atlas mountains, and over Turkey and the Black Sea (their Fig. 8).

Some of the recent studies about the role of surface latent heat fluxes on extratropical cyclones in general, summarised in Sect. 5.2.3, also investigated moisture sources, i.e., the pathway of the evaporated water from regions with intense surface fluxes to regions in the cyclone where condensation, latent heating, and PV production occur. Similar analyses have also been performed for Mediterranean cyclones, for individual case studies (Raveh-Rubin and Wernli, 2016, their Fig. 8) and for a climatology of 100 intense cyclones in a regional climate simulation (Flaounas et al., 2019, their Fig. 12). The results revealed large variability with, in addition to important local moisture sources, substantial contributions from long-range transport from the entire North Atlantic from 20–60°N, but also from the Black Sea, the Red Sea, and the Persian Gulf in some cases. For the explosively deepening Mediterranean cyclone Vaia in October 2018, Davolio et al. (2020) identified long-range moisture transport even from the tropical Atlantic across Northern Africa. Together these studies indicate that, most likely due to the special geographical setting of the Mediterranean basin, moisture transport into Mediterranean cyclones can differ quite strongly from moisture transport into cyclones over the main oceanic storm tracks illustrated in, e.g., Fig. 15e. It appears that some Mediterranean cyclones (and related extreme precipitation events) are able to import moisture from remote ocean basins.

### 5.3.4 Polar lows

Similarly to Mediterranean cyclones, polar mesocyclones have a broad spectrum of dynamical types with one common feature being their genesis location. Similarly to medicanes, polar lows are are a subset of polar mesocyclones, partly distinguished by their strong near-surface winds and by mature systems sometimes resembling hurricanes with a symmetric cloud structure surrounding a clear eye (e.g., the "most beautiful polar low" in Nordeng and Rasmussen, 1992); indeed, Emanuel and Rotunno (1989) proposed that some polar lows are "Arctic hurricanes". In additional to their location and intensity, polar lows are distinguished by their scale: they are mesoscale features and thus distinct from the synoptic-scale Arctic cyclones that also occur in the Northern Hemisphere Arctic. A commonly used definition comes from the book on polar lows by Rasmussen and Turner (2003): a polar low is "a small, but fairly intense maritime cyclone that forms poleward of the main baroclinic zone (the polar front or other major baroclinic zone)..." with the horizontal scale and surface windspeed thresholds described as 200–1000 km and at or above gale force, respectively. A global climatology based on ERA-Interim data by Stoll et al. (2018) found that although polar lows occur in all high latitude ocean basins, they are most common near ice edges and in coastal zones; they are also less intense in the southern than the Northern Hemisphere. As mesoscale systems, polar lows are strictly beyond the synoptic-scale scope of this review. However, we argue that their similarities to medicanes and the use in many analyses of these systems of diagnostic tools commonly used to assess the importance of diabatic processes in extratropical cyclones (including PV thinking) justifies their inclusion.

As described in the recent review by Moreno-Ibáñez et al. (2021) [with updates in Moreno-Ibáñez (2024)], polar lows develop in a range of environments and there has been scientific debate about the relative importance of baroclinic and convective processes consistent with a paradigm of a spectrum of polar low development. A summary of the early scientific debate on development mechanisms, starting from their attribution to thermal instability within cold air masses flowing over relatively warm seas [as described in "a course in elementary meteorology" written by what was then called the Meteorological Office in the U.K. (Sect. 10.3.2 in Meteorological Office, 1962)], can be found in the conference paper by Reed (1987). Events where cold air masses flow over warm seas are now commonly referred to as a marine cold air outbreaks and, according to Stoll et al. (2018), there is now general consensus that only systems occurring during such outbreaks can be classified as polar lows. As one example illustrating that both baroclinic and convective processes can be important, Reed and Duncan (1987) considered a train of four polar lows and found that the characteristics, including the mean observed wavelength of 570 km, were in partial agreement with those obtained from a dry, linear quasi-geostrophic model. However, they argued that baroclinic instability alone was insufficient to explain the rapid development of the polar lows and that "some other mechanism, presumably latent heat release in organised deep convection, also played an important part in the growth". Similarly, from their case analysis of the "most beautiful polar low", Nordeng and Rasmussen (1992) concluded that, while it was triggered by an upper-level PV anomaly in a baroclinic zone, it was partly driven by latent heat release. In the same year, but by using a moist nonlinear idealised model (a geostrophic momentum model) to study interacting upper- and lower-level PV structures, Montgomery and Farrell (1992) proposed a two-stage conceptual polar low development model with initiation by an upper-level trough in a conditionally neutral environment (termed "induced self-development") followed by diabatic destabilisation producing low-level

PV. In addition to baroclinic and convective theories of polar low development (the latter, particularly focusing on the CISK mechanism in early studies[21]), Emanuel and Rotunno (1989) proposed air-sea interaction instability (now commonly termed WISHE: wind-induced surface heat exchange) as a possible mechanism following on from the proposal of the same mechanism for tropical cyclone development by Emanuel (1986). However, from their self-organising maps based classification, Stoll et al. (2021) found no evidence of hurricane-like polar low intensification in a low vertical wind shear environment. Surface latent heat fluxes were instead argued to enhance baroclinic intensification through reducing the atmospheric static stability.

The first demonstrations of the importance of diabatic processes to polar low development through model sensitivity studies were by Nordeng and Rasmussen (1992), Grønås and Kvamstø (1995) and Mailhot et al. (1996). Nordeng and Rasmussen (1992) and Grønås and Kvamstø (1995) both noted that weaker simulations of polar lows were formed in simulations without latent heating. Comparison of mesoscale model full physics and dry (with surface fluxes but no condensation processes) simulations of a Labrador Sea polar low by Mailhot et al. (1996) showed that condensation processes were the major cause of its rapid deepening in its mature phase; another simulation in which only evaporation from the ocean surface was omitted produced very similar results to the dry simulation implying that evaporation was a prerequisite for the condensation. A larger set of mesoscale model sensitivity experiments by Bresch et al. (1997) similarly demonstrated the importance of both surface fluxes and condensation processes for a polar low forming over the Bering Sea. Later, Føre and Nordeng (2012) found condensation essential to set up the baroclinic environment favourable for a polar low over the Norwegian Sea (but it then had a modest role), while Føre et al. (2012) found that condensation was not essential for a polar low over the Barents Sea and instead sensible heat fluxes fuelled the polar low in its mature stage. In the Southern Hemisphere, the sensitivity study of Papritz and Pfahl (2016) demonstrated the importance of a train of polar mesocyclones to the decay of a cold air outbreak. The erosion of the cold air mass through latent heating in the warm sectors of the mesocyclones was reduced, and the surface pressure raised, in simulations without sensible and/or latent surface fluxes. Idealised model simulations have similarly yielded varying conclusions regarding the relative importance of baroclinic and diabatic processes in polar low development and maintenance (Emanuel and Rotunno, 1989; Yanase and Niino, 2007; Adakudlu, 2012). Of particular note is the idealised high-latitude moist baroclinic channel simulation study of Terpstra et al. (2015) in which they showed that a weak disturbance with a PV structure consistent with a shallow diabatic Rossby wave (Sect. 5.3.5) is able to develop in the absence of an upper-level initial perturbation, surface fluxes, friction and radiation; however, they noted that additional forcing is likely required to produce more realistic polar low intensities.

While several earlier studies had discussed the potential importance of diabatically generated low-level PV anomalies in polar low development (e.g., Moore et al., 1996; Guo et al., 2007, see Fig. 16e), Bracegirdle and Gray (2009) were the first to quantify, through piecewise PV inversion, the relative importance of the different PV anomalies in polar low development. They applied this methodology to the system named Le Cygne (the swan) in Claud et al. (2004) and identified three stages to the development of this system with the first two, an initial baroclinic stage followed by a baroclinic stage with strong lower-tropospheric latent heat release, consistent with Type C cyclogenesis ( Sect. 5.3.1). In the final stage intensification occurred

---

[21]When considering polar lows and other warm-core vortices, the CISK mechanism describes development through a cooperative interaction between organised moist convection and large-scale moisture convergence due to the cyclonic vortex which releases CAPE.

through the non-baroclinic WISHE mechanism. A similar piecewise PV analysis of another polar low, this time over the Sea of Japan, also identified the first two of these stages but found the development to be dominated by forcing from the upper-tropospheric PV anomaly with that due to latent heating playing a lesser role (Wu et al., 2011). The PV inversion approach was used together with sensitivity experiments initialised with modified PV structures to determine the relative roles of the different PV anomalies involved in the development of polar low case studies by both Nordeng and Røsting (2011) and Føre et al. (2011); Nordeng and Røsting (2011) also evaluated PV along backward trajectories to determine that diabatic heating rather than friction was the source of the lower-tropospheric PV anomaly. Both studies revealed the dominance of the upper-tropospheric PV anomaly, although in Nordeng and Røsting (2011) the effect of the lower-tropospheric PV anomaly created by latent heating became large after cyclogenesis and the polar low was described as consistent with the type C classification, whereas in Føre et al. (2011) the lower-tropospheric PV anomaly became less important over time.

Finally, while many climatologies of polar lows exist (see reviews by Moreno-Ibáñez et al., 2021; Moreno-Ibáñez, 2024), that by Bracegirdle and Gray (2008) specifically considered the dynamical forcing including diabatic effects by applying the type A, B, C classification scheme to polar lows in the Nordic Seas using the quasi-geostrophic omega equation approach of Deveson et al. (2002). Overall 31% of the objectively identified systems met the criteria for type C (similar to the 32% found by Gray and Dacre (2006) for North Atlantic cyclones in the 30–60° latitude band), but type C cyclones dominated in the central and southern Norwegian Sea. The differing prevalence of type C polar lows was consistent with the analysed environment, with type C systems forming where there is weak low-level baroclinicity and weak static stability.

### 5.3.5 Diabatic Rossby waves / vortices

A special category of cyclones, which rely essentially on latent heating, are so-called diabatic Rossby waves (DRW) or diabatic Rossby vortices [see Appendix A of Boettcher and Wernli (2013) for a discussion of the wave-vortex terminology issue]. As mentioned in Sect. 4.3.1, the concept and terminology of DRWs was introduced in idealised studies of convective heating in a baroclinic atmosphere by Snyder and Lindzen (1991) and Parker and Thorpe (1995). The basic idea of DRW dynamics is that an intense low-level PV anomaly in a moist baroclinic zone can propagate faster than the mean flow as it regenerates itself at its downstream edge due to diabatic PV production in ascending air induced by its own cyclonic wind field that impinges on the tilted isentropes (Fig. 16f). The first real case study of an extratropical cyclone that explicitly established a link with this idealised DRW concept was the one by Mallet et al. (1999b) about the early phase of the FASTEX IOP17 cyclone. A meridional section across the cyclone in its early phase revealed a prominent diabatically-produced PV anomaly exceeding 2 pvu at 800 hPa about 700 km south of the upper-level jet axis (their Fig. 8c,d). A simulation with cloud processes turned off produced a weaker cyclone (deepening of 30 hPa instead of 45 hPa in 36 h). The authors concluded that after an initial baroclinic growth "the subsequent cyclone growth evolves towards a purely diabatic regime, which is nearly self-contained, i.e. decoupled from the synoptic far-field influence" and they attributed this regime to the concept of a DRW. In hindsight, however, it appears that much earlier studies of disturbances along the western part of the Baiu front (e.g., Saito, 1977; Yoshizumi, 1977) already described certain aspects of what later became the DRW concept, in particular the shallow vertical extension of the disturbances, the rapid propagation (note the easterly relative flow at 800 hPa shown in Yoshizumi, 1977, their Fig. 8a), the

moist environment, and the strong upper-level jet. Saito (1977) mentioned the potential for strong intensification of Baiu frontal depressions and attributed it to the development into a synoptic-scale low due to coupling with upper levels rather than to self-development (DRW propagation).

A prominent extratropical cyclone that was explicitly considered as a DRW, was the devastating, small-scale, and rapidly propagating winter storm Lothar (Wernli et al., 2002). When submitting the first version of their study, the authors were not aware of this potential linkage to the DRW concept, but Doug Parker (one of the reviewers) suggested considering the rapid propagation phase of Lothar as a realisation of the idealised notion of a DRW. The final version of paper then discussed the DRW character of Lothar with illustrations of the key ingredients of the mechanism (their Fig. 13f). A particular aspect of this mechanism is that DRW propagation, which typically is not associated with strong cyclone intensification, occurs in a moist baroclinic zone without an upper-level wave forcing. The explosive intensification of Lothar then occurred in a second phase when the diabatically-produced low-level PV anomaly of the DRW crossed the upper-level jet axis and induced the rapid downward motion of stratospheric PV along the strongly tilted isentropes (referred to as "bottom-up intensification" by Wernli et al., 2002). A classical sensitivity study with a dry hindcast simulation produced only a very weak surface development and confirmed that cloud diabatic heating was essential for the intensification of this unusual cyclone. Rivière et al. (2010), using another numerical model, confirmed that no cyclone growth occurs in a simulation with suppressed latent heating. Additional sensitivity experiments with modified initial conditions revealed that the location of the explosive development of Lothar depended primarily on the position of the low-frequency jet exit, i.e., on the position of the jet exit calculated with low-pass filtered winds.

In the last 20 years, research about DRWs has involved a broad spectrum of approaches, including idealised numerical modelling, further real case simulations, climatological studies, and several highly idealised theoretical studies. A first theoretical study by Moore and Montgomery (2004), using a two-dimensional semigeostrophic model with a parametrisation of latent heat release, diagnosed the structure and propagation characteristics of short-scale, diabatic normal modes in a moist, baroclinic atmosphere with the Eady basic state. They found that with a thermodynamically consistent vertical profile of latent heating, the short-wave cutoff vanishes, and fairly uniform growth rates appear for wavelengths shorter than about 2000 km. These short-scale modes correspond to DRWs as they exist due to the continuous PV generation by moist processes associated with warm air advection, rising motion, and latent heat release. De Vries et al. (2010) used a general quasi-geostrophic PV framework, with both wave-CISK and large-scale precipitation approaches to represent latent heating, to analyse four types of baroclinic instability. Their classification, based on the concept of phase-locking of counterpropagating Rossby waves (Heifetz et al., 2004), included type-C cyclogenesis and DRWs, the latter depending on both near-surface baroclinicity and latent heating. Deriving analytical solutions for DRW-like disturbances in a three-layer quasi-geostrophic framework, Oda and Kanehisa (2011) found a solution with an initial small-scale low-level PV disturbance that cannot effectively interact with an induced upper-level PV disturbance, resulting in no growth. However, a larger-scale low-level disturbance led to vertical interaction and exponential growth, suggesting that the scale of a DRW might be an important factor for its potential rapid intensification. Kohl and O'Gorman (2022) also used the analytical quasi-geostrophic framework and focused on the transition from periodic moist baroclinic waves to a DRW-like vortex mode maintained through latent heating. They created a phase diagram for

when the most unstable solution is a periodic wave versus an isolated vortex (DRW), with the latter emerging when the moist static stability and meridional PV gradients are weak. Together, these theoretical studies undermine the special characteristics of DRWs in the wide spectrum of extratropical cyclones, with their comparatively small horizontal scale and diabatically produced low-level PV as an essential ingredient.

Interestingly, only one study considered DRWs in an idealised channel model. Based on a set of experiments with differing environmental conditions, Moore and Montgomery (2005) revealed that the resulting intensity of the DRW vortex is most sensitive to the magnitude of environmental baroclinicity and moisture content, while the vertical moisture profile mainly determines the characteristic depth of the DRW. These results served as an important inspiration for the analysis of DRW propagation in ECMWF forecasts (Boettcher and Wernli, 2011) and the development of an objective DRW identification algorithm, which enabled the first compilations of short (10-year) DRW climatologies in both hemispheres, based on operational ECMWF analyses (Boettcher and Wernli, 2013, 2015)[22]. In the Northern Hemisphere, DRWs were found to be more frequent over the North Pacific than over the North Atlantic with on average 81 and 43 systems per year, respectively. Less than 15% of these systems intensify explosively, the majority dissipate when moving in regions with insufficient moisture and/or baroclinicity. DRWs are most frequent in summer but most of the explosively intensifying DRWs occur in autumn and winter. DRWs are generated typically between 30 and 50°N near the North American and Asian east coasts. This study also investigated different scenarios that can generate the initial low-tropospheric PV anomaly of DRWs (Boettcher and Wernli, 2013, their Fig. 8). These scenarios correspond to 1) flow around the subtropical high against the midlatitude baroclinic zone, 2) flow induced by an upper-level cutoff or a (tropical) cyclone against the baroclinic zone, 3) upper-level trough-induced ascent at the baroclinic zone, and 4) PV remnants of a tropical cyclone or a mesoscale convective system that are advected into the baroclinic zone where they start propagating as a DRW.

In addition to these climatological investigations, detailed case studies further illustrated the common dynamical ingredients of DRWs and their diverse impacts. Moore et al. (2008) investigated a DRW with a clear two-phase process, with an initial propagation of a low-level PV anomaly that originated from a convective complex over the U.S. central plains and a subsequent explosive intensification near the U.S. east coast associated with heavy snowfall. A simulation without latent heat release produced less than half of the observed decrease in MSLP. The authors further mentioned that the two catalyst cyclones discussed in Sect. 3.5 and 4.1.1 showed a qualitatively similar two-phase evolution. Investigating the evolution of the so-called Perfect Storm in 1991 and a subsequent North Atlantic cyclone, Cordeira and Bosart (2011) found for the latter also a two-phase development process remarkably similar to the development of Lothar, and, in their case, the initial low-level PV anomaly was most likely induced by the interaction of the circulation of Hurricane Grace with the baroclinic zone in the western North Atlantic. Laurila et al. (2020) studied an extratropical transition event in 1982 and found that a low-level PV-remnant of ex-Hurricane Debby propagated as a DRW-like feature across the North Atlantic before it coupled with an upper-level trough near the U.K., which led to rapid intensification and a severe windstorm over northern Finland. Another DRW-like cyclone occurred during the field experiment NAWDEX. Flack et al. (2021) analyzed the so-called Stalactite cyclone (see also Sect. 5.6.2), which

---

[22]At the time, the resolution of available reanalyses was insufficient to well capture the small-scale low-level PV anomalies associated with DRWs, and therefore higher-resolution operational analyses were used.

deepened due to the interaction of a DRW with an upper-level cutoff cyclone. Simulations with different resolutions indicated that a grid spacing of 50 km is sufficient to simulate this DRW, whereas coarser resolution failed to reproduce the amplitude of the low-level PV and the DRW propagation speed. And last but not least, recent studies also portrayed additional examples of summertime DRWs along the western part of the Baiu front that led to extreme precipitation events in Japan (Tochimoto and Kawano, 2017; Shibuya et al., 2021), and a DRW-like summertime phenomenon of eastward propagating heavy rainfall events along the Yangtze River (Zhao et al., 2023).

As an aside, we mention that the structure of DRWs has also been compared to the one of monsoon depressions (Cohen and Boos, 2016) and so-called Borneo vortices that occur in boreal winter (Hardy et al., 2023). Cohen and Boos (2016) found that, during their amplification phase, monsoon depressions share only few similarities with DRWs but many with hurricanes (see their Fig. 8). The monsoon depressions' vertical structure is characterised by a relatively weak PV tower (Fig. 16h) that does not connect to the tropopause. The westward propagating Borneo vortex investigated by Hardy et al. (2023) showed clear DRW signatures, with strongest diabatically produced PV at 5 km altitude that coupled with a lower boundary thermal wave. Because both phenomena are both geographically and dynamically mainly tropical phenomena, they are not further discussed in this review.

### 5.3.6 Extratropical transitions

Extratropical transition refers to the process of tropical cyclones that move poleward and transform into extratropical cyclones when reaching a baroclinic zone at higher latitudes—a phenomenon that occurs most frequently in the western North Pacific, and less frequently in the western North Atlantic, Southwest Pacific and Indian Ocean (Jones et al., 2003). Many systems re-intensify after the transition and develop into deep extratropical cyclones like, e.g., ex-Hurricane Lili in 1996 (Browning et al., 1998) or ex-supertyphoon Nuri in 2014 (Keller et al., 2019). Diabatic processes including surface fluxes and latent heating are of course essential for the tropical phase of these systems, but also play a major role during the interaction with the extratropical flow, which typically leads to a strong poleward moisture transport during the transition. One of the most infamous examples of an extratropical transition was the North Atlantic Hurricane Sandy in 2012, which hit the northeastern U.S. coastline with hybrid tropical and extratropical characteristics and led to a devastating storm surge and winds, and large amounts of precipitation (Galarneau et al., 2013). From a PV perspective, the extratropical transition process can be synthesised as, first, the interaction of a troposphere-deep diabatically-produced PV anomaly (associated with the tropical cyclone, Fig. 16b) with an upstream upper-level PV anomaly (associated with the approaching trough), and, subsequently, with the production and intensification of a new PV tower (associated with the extratropical cyclone) extending from the near-surface to the tropopause at the southern edge of the baroclinic zone, while the tropical cyclone-related PV anomaly decays (Agustí-Panareda et al., 2004; Grams et al., 2013). And as already discussed in the review by Jones et al. (2003), the diabatic low-PV tropical cyclone outflow can lead to strong ridge amplification and anticyclonic Rossby wave breaking further downstream (Riemer and Jones, 2010; Grams and Archambault, 2016), a theme that will be discussed further in Sect. 5.5. The interested reader can find more information about extratropical transitions and their downstream impacts in a series of excellent review articles (Jones et al., 2003; Evans et al., 2017; Keller et al., 2019).

 ### 5.3.7 Upper-level cutoffs and tropopause polar vortices

As a last special category of extratropical cyclones we consider diabatic processes in so-called upper-level cutoffs or cutoff lows, which are typically identified either by closed geopotential height contours in the upper troposphere or by isolated vortices of stratospheric PV on isentropic surfaces. Such upper-level cutoffs can grow into vertically deep, quasi-barotropic vortices that are also associated with local minima in MSLP. Cutoffs result from Rossby wave breaking and the subsequent breakup of PV streamers into vortices, due to an instability of the streamer's elongated band of enhanced PV (Appenzeller and Davies, 1992; Browning, 1993). They are, therefore, mainly of dry dynamical origin, and, in contrast to most other cyclones discussed in this section, diabatic processes are primarily important for their maintenance and decay. In Sect. 4.1.6 it was briefly mentioned that early studies of PV cutoffs found that diabatic erosion by latent heating in deep convection and filamentation both contributed to the decay of PV cutoffs, and therefore to intense downward STE. This association with deep convection also indicates that cutoffs can play an important role in triggering heavy precipitation by favoring the release of CI via dynamical forcing (Romero et al., 2000). Here, we briefly summarise more recent studies that investigated the role of diabatic processes in the lifecycle of cutoffs.

Portmann et al. (2021) compiled a 40-year global climatology of PV cutoffs and, after tracking the cutoffs, quantified whether their lysis occurred due to PV erosion (diabatic decay) or reabsorption of the cutoff by the main stratospheric body (adiabatic process, e.g., Knippertz and Martin, 2007). They found that, in the global average, both scenarios occur with a similar frequency, with diabatic decay dominating at lower and reabsorption at higher latitudes. Considering only cutoffs in a particular region, then diabatic decay is more frequent for cutoffs on lower isentropes and reabsorption for cutoffs on higher isentropes. Relatively few studies investigated the detailed diabatic processes that occur in cutoffs. Bourqui (2006) studied a decaying PV cutoff over the Mediterranean and found that both latent heating in clouds and radiative cooling above clouds can lead to negative PV tendencies at the tropopause, depending on the cloud top height relative to the tropopause. Investigating the evolution of a PV cutoff (referred to as a northeast China cold vortex) in a convection-permitting simulation, Fan et al. (2023) found strong diabatic erosion related to intense precipitation in the bent-back frontal region of the associated surface cyclone. For two long-lived PV cutoffs over Europe, Portmann et al. (2018) identified a more complex time evolution of their vertical structure than the generally assumed gradual diabatic decay process. They found that due to the combination of convective latent heating, long-wave radiative cooling, and turbulent entrainment of overshooting clouds, PV cutoffs can simultaneously weaken on one isentrope and intensify on another one. Their result that PV is diabatically produced in the lowermost stratosphere due to maximum long-wave radiative cooling at the level of the tropopause in a region where the tropopause is anomalously low is consistent with the results of Chagnon et al. (2013).

Radiative effects were also found to be important for the formation and maintenance of so-called tropopause polar vortices (TPVs). This term refers to mesoscale positive PV anomalies in the lower stratosphere over polar regions (with amplitudes of several pvu, Fig. 16g), which can have long lifetimes beyond two weeks (Hakim and Canavan, 2005). TPVs can strongly affect Arctic weather conditions, as they play an important role for the intensification of about one third of all summer Arctic cyclones, in particular for those over the Canadian Arctic Archipelago (Gray et al., 2021a), and for the formation of intense

cold air outbreaks from Fram Strait in winter (Papritz et al., 2019). A numerical model case study by Cavallo and Hakim (2009) revealed that cloud-top radiative cooling is the primary mechanism that intensifies TPVs. This result was confirmed by a composite analysis of more than 500 simulated TPVs over the Canadian Arctic, which showed that intense longwave cooling beneath TPVs leads to diabatic PV production within the TPVs of up to $0.3\,\mathrm{pvu\,d^{-1}}$ (Cavallo and Hakim, 2010, their Fig. 12b).

### 5.3.8 Summary

The concept of extratropical cyclones clearly became much more "colorful" in the last 20 years. Although we emphasised already the large diversity among cyclones in the main storm track regions in Sect. 4, this diversity became substantially larger with the increased research focus on specific categories of extratropical cyclones. The motivation for investigating specific cyclones can be a geographical focus (e.g., the unique setting in terms of land-sea contrasts for Mediterranean cyclones or of sea ice-ocean contrasts for polar lows), or a unique element of their structure (cutoff cyclones) or lifecycle (extratropical transitions). Importantly for this review, diabatic processes play an important role for all the considered types of cyclones, though the relative importance of the different processes (heating in clouds, deep convection, surface fluxes, and radiation) characteristically differ between the cyclone types (e.g., radiation is more important for TPVs whereas convection is more relevant for intense Mediterranean cyclones).

### 5.4 Novel diagnostics of diabatic processes

A notable advance in the past 20 years has been the development of diagnostics that enable the attribution of changes in the PV structure of cyclones or near the tropopause to individual diabatic processes. These approaches are distinct from simple assessments of cyclone structure in terms of PV (or moist PV) and the attribution of different parts of the cyclone structure, such as upper-level PV or boundary layer $\theta$ anomalies, to the cyclone circulation through so-called PV surgery and inversion. However, some authors have instead performed PV inversion on diagnosed diabatically generated PV features. These diagnostics are also distinct from diagnosis of latent heating effects using sensitivity studies, e.g., through comparison of model simulations with and without latent heating (discussed in Sect. 4.2.2). There are three main reasons to attribute PV changes to individual diabatic processes: to improve our fundamental understanding of the development of cyclones, to investigate the growth of uncertainty in forecasts, and to diagnose model error. To address this first reason, an assumption is made that the partitioning of processes by the model's code is reflective of nature, e.g., the modification of PV by the convective parametrisation scheme relates to that which occurs in the cyclone due to convective processes. In this example this assumption obviously breaks down in km-scale models in which the convection is handled explicitly instead of being parameterised: the lack of PV modification by a convective parametrisation scheme does not mean that there is no PV-modifying convection occurring in the cyclone. Diabatic processes attribution has been performed mainly using the two approaches of passive tracers and tendency diagnostics and these are covered in the following two subsections. A third subsection reviews the, as yet limited, use of ensemble- and adjoint-based approaches to interrogate diabatic processes. The final subsection presents some other approaches that have been applied.

### 5.4.1 Passive tracers

The use of passive tracers developed from the pioneering work of Stoelinga (1996) that extended earlier work by Davis et al. (1993), introduced in Sect. 4.1.2, in which they integrated online the accumulation of PV due to various physical processes in the model's Eulerian framework. With this online approach, Stoelinga produced partitioned PV integration applied to a cyclone case study. They considered six contributions to the non-conservation of PV: grid-scale (from the cloud microphysics parametrisation) and convective (from the convection parametrisation) latent heating and cooling, surface heat and momentum fluxes, and parameterised temperature and momentum diffusion. These contributions comprise the diabatic and frictional terms that theoretically complete the PV budget, summing to the Lagrangian rate of change of PV (Eq. 3, see discussion in Sect. 2.1). Integration of the above equation by time shows that the PV field at any given time can be written as the sum of a conserved component and non-conserved components, both advected as passive tracers (leading to the terminology of PV tracers or diabatic tracers in later papers). The conserved component is initialised as PV field from an earlier time, typically the start of a model forecast. The non-conserved components comprise accumulated sums of the increments from each of the non-conservative model processes. The tracer code runs online, embedded within the model code. For example, the contribution of the convective parametrisation scheme to the PV field is calculated by diagnosing the change in PV after each call to this scheme and adding this increment to the associated passive tracer, which is then advected at each time step with the model's advection scheme. The non-conservative PV tracers are typically initialised to zero at the start of the model forecast and at lateral domain boundaries (in limited area models); the conserved PV tracer is typically set to the model's PV field on these boundaries. These lateral boundary assumptions limit the time over which non-conservative PV tracers can accumulate to the time that weather features traverse the domain. Stoelinga (1996) acknowledged a caveat with the PV tracers approach: the PV budget may not balance exactly because the model equations are not formulated so as to conserve PV. This residual leads to a growing difference between the PV field diagnosed from the instantaneous model fields and that calculated from the sum of the conserved and non-conserved PV components. A later study showed that the non-conservation of the PV by the dynamical core could be calculated as an additional tracer and that this tracer had magnitude comparable to that of the other non-conservative tracers and accounted for the majority of the residual term in the PV budget equation for the MetUM model (Saffin et al., 2016).

Despite the usefulness of the PV tracers approach for diagnosing diabatic processes in models demonstrated by Stoelinga (1996), the requirement to incorporate the PV tracers within the model code presents a barrier to easy use. It was ten years before a series of papers emerged using PV tracers in the MetUM beginning with a study of cross-tropopause transport (Gray, 2006) and followed by partitioning of the diabatic processes in cyclones (Chagnon et al., 2013; Chagnon and Gray, 2015). The latter two studies demonstrated that diabatic processes can generate a PV dipole straddling the tropopause in extratropical cyclones (Fig. 17a) with the positive stratospheric pole generated by longwave cooling due to the sharp reduction in humidity across the tropopause and the negative tropospheric pole generated both locally from the longwave radiation and non-locally by advection out of the top of heating associated with the large-scale cloud, convection and boundary layer schemes (with the negative pole used by Schäfer and Voigt (2018) to hypothesize how radiation weakened their idealised cyclone through reduced

mid-tropospheric PV). The position of the tropopause was generally not found to be directly modified by diabatic processes, but the diabatically enhanced tropopause PV gradient can influence the tropopause indirectly through the winds associated with a jet strengthening and subsequent tropopause advection. Figure 17b schematically shows the influence of near-tropopause level diabatically generated PV anomalies on the subsequent Rossby wave evolution: the positive anomalies associated with long-wave cooling in the troughs enhance the PV anomalies associated with equatorward advection of stratospheric air and the negative anomalies in the ridges enhance the PV anomalies associated with the poleward advection of the tropospheric air leading to an amplification of the Rossby wave. Figure 17a also shows other features including a tripole of diabatically generated PV extending down from the main tropopause fold to the cold front, which was attributed to a dipole in latent heating across the front (heating in the WCB and evaporative cooling behind the front), and negative PV behind the surface cold front (between approximately longitude indices 470–500) attributed to surface-flux-driven boundary layer heating. The PV tracers diagnostics in the MetUM have also been used for other applications including assessment of diabatic processes in African easterly waves (Tomassini et al., 2017) and in maintaining the sharpness of the tropopause (Saffin et al., 2017). The current use of PV tracers is no longer restricted to the MetUM. A recent implementation of PV tracers in the WRF model, together with PV inversion, has been applied to 100 intense Mediterranean cyclones to determine the relative diabatic and baroclinic contributions (Flaounas et al., 2021).

Analogous tracers for $\theta$ (with non-conservative components contributed by the parametrisations for the boundary-layer, convection, cloud microphysics and radiation, and a residual) were also developed within the MetUM. These $\theta$ tracers were used alongside offline trajectory calculations to determine the origin location of the diabatic contributions and to assess how the activeness of the convection parametrisation scheme affects evolution of an extratropical cyclone (Martínez-Alvarado and Plant, 2014). As the conserved component of the $\theta$ tracers represents the advected initial $\theta$ field, it can be used to track air mass evolution and the non-conservative tracers can be used to identify diabatic processes leading to cross-isentropic transport. When the convective parametrisation scheme was less active, Martínez-Alvarado and Plant (2014) found that the large-scale latent heating released convective instability, which would otherwise have been released by the convection parametrisation and that the consequent more rapid and stronger latent heating led to the WCB extending to higher altitudes. As another extension to the PV tracers approach, cylindrical volume integral diagnostics using both the PV and $\theta$ tracers were used in Martínez-Alvarado et al. (2016a) to assess the difference in diabatic processes and their contributions to cross-isentropic mass transport (heating) and absolute circulation, respectively, in two contrasting summer cyclones. The circulation around the cyclone was found to increase much more slowly than the amplitude of the PV tower generated by diabatic processes in the stronger of the two cyclones, which could be explained by the impermeability theorem of PV (Haynes and McIntyre, 1990).

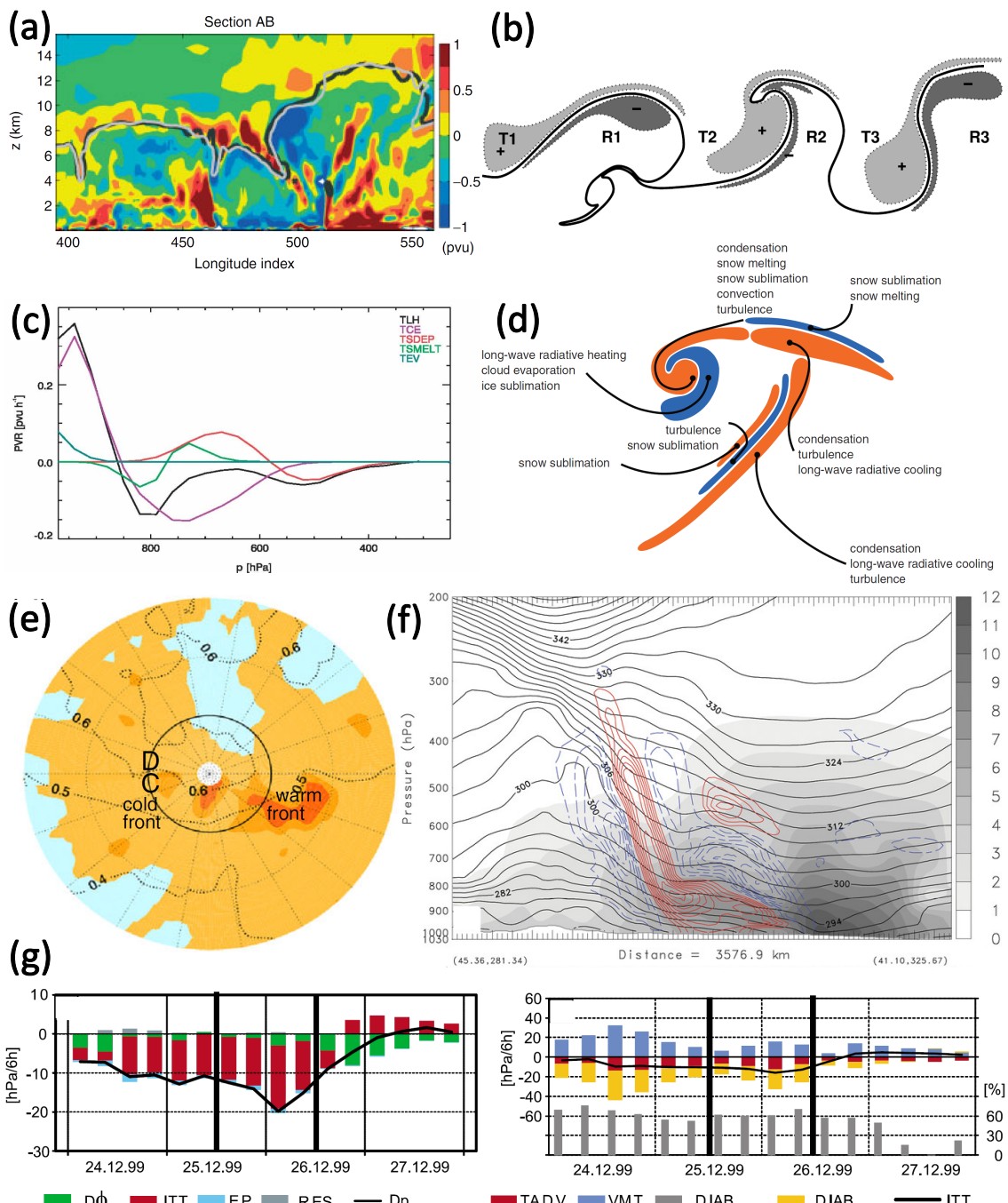

**Figure 17** *(previous page)*. Novel diagnostics. **(a)** from Chagnon et al. (2013, their Fig. 5c), vertical cross-cold-front section of diabatically generated PV (shading) with black and grey contours of 2 pvu from the full and advection-only PV, respectively; **(b)** also from Chagnon et al. (2013, their Fig. 10), schematic representation of the net diabatic PV (shaded: positive in troughs (T1–3) and negative in ridges (R1–3)) relative to large-amplitude waves on the tropopause (solid line); **(c)** from Joos and Wernli (2012, their Fig. 10c), mean rates of diabatically generated PV from different microphysical processes along WCB trajectories (TCE denotes condensation and evaporation, TSDEP depositional growth of snow, TSMELT melting of snow, and TEV evaporation of rain; in addition, TLH shows the total diabatic PV rate); **(d)** from Attinger et al. (2019, their Fig. 10), synthesised depiction of the low-level PV field for a North Pacific cyclone case study at the time of maximum intensity with regions of anomalous positive and negative PV in orange and blue, respectively, the origin of the air masses shown by lines and the most important processes responsible for the formation of the PV anomalies labelled; **(e)** from Dacre and Gray (2013, their Fig. 5d), sensitivity of cyclone intensity to 700-hPa precursor PV for east Atlantic cyclones (shaded) overlaid by composite-mean 700-hPa PV (contours, pvu); **(f)** from Doyle et al. (2019, their Fig. 4a), vertical cross-frontal-zone section of water vapour (shading, $\mathrm{g\,kg^{-1}}$) and water vapour sensitivity (positive red, and negative, blue contours, interval of $0.02\,\mathrm{m^2\,s^{-2}\,(g\,kg^{-1})^{-1}}$) with isentropes (black contours) for storm *Desmond* with a kinetic energy response function; and **(g)** from Fink et al. (2012, their Fig. 3a,f), pressure tendency equation analysis for storm *Lothar* with (left) Dp, the overall surface pressure tendency, and ITT, the vertically integrated virtual temperature tendency, and (right) a breakdown of ITT including the diabatic residual term, $\mathrm{DIAB_{RES}}$, and relative contribution of $\mathrm{DIAB_{RES}}$ to the total pressure tendency, $\mathrm{DIAB_{ptend}}$. All figures used with permission: (a-e,g) from Wiley, and (f) from the American Meteorological Society.

### 5.4.2 Tendency diagnostics

In parallel to the development and exploitation of PV tracers, tendency diagnostics, often combined with offline Lagrangian trajectories, have been used by other researchers to address closely related problems. In this approach tendencies from the different diabatic and/or frictional processes are determined. These tendencies and the associated PV changes are then evaluated along trajectories, which are usually calculated offline using the resolved wind components. Lamarque and Hess (1994) was most likely the first study to calculate PV online tendencies (in the MM4) and associated with offline calculated trajectories to identify the locations and importance of diabatic processes and diffusion in leading to STE across a tropopause fold (diabatic processes were found to dominate, see Sect. 4.1.6). Much later, a more detailed partitioning of processes was carried out by Joos and Wernli (2012) who used this approach to attribute PV changes in a WCB to different microphysical processes in the COSMO model (Fig. 17c). Rather than calculating PV tendencies online, as done by Lamarque and Hess (1994), they output heating tendencies from different parameterised processes at frequent intervals and calculated the corresponding diabatic PV rates, separately for each of these processes, using the first term on the r.h.s. of Eq. 3 (the modification of PV by frictional processes was not considered). This work revealed that the location of the diabatic heating relative to the cold front is vital for its contribution to PV changes due to the dependence of the PV changes on vorticity. Consequently, vapour condensation had a much larger impact on PV than snow depositional growth despite their similar diabatic heating rates. The relative importance of the vertical heating gradient and vorticity to PV changes in WCBs was assessed climatologically using ERA-Interim reanalyses

in Madonna et al. (2014b): high values of the vertical vorticity component were found to distinguish WCB trajectories with the highest values of low-level PV starting from a given region.

As a modification to the approach used in Joos and Wernli (2012), Crezee et al. (2017) integrated diabatic PV rates along trajectories running backwards in time from anomalous PV regions, using an idealised moist baroclinic wave simulation, to determine the microphysical processes causing the anomalies. Further developing this approach, Spreitzer et al. (2019) output

both temperature and momentum tendencies from simulations with the IFS model to enable calculation of a complete PV budget (including both diabatic and frictional source terms due to temperature tendencies in the large-scale cloud, convection and radiation schemes, and momentum tendencies in the convection and turbulence schemes); the diagnostic was then used to demonstrate the importance of both clear-air turbulence and cloud processes in modifying upper-tropospheric PV in a cyclone case study. The same approach was then used by Attinger et al. (2019) to attribute low-level PV anomalies in an extratropical

cyclone case study to diabatic and turbulent processes as summarised in Fig. 17d. In this figure turbulence leads to changes in PV through both temperature and momentum tendencies and so the relationship to the studies on frictional PV modification described in Sect. 5.2.1 cannot be discerned. However, the authors find that turbulent mixing of momentum in the baroclinic zone south of the warm front contributes to the broad positive PV anomaly along the warm front, consistent with the findings of Stoelinga (1996), Adamson et al. (2006), and Plant and Belcher (2007). In contrast, Attinger et al. (2019) found that the

enhanced low-level PV occurring in the centre of the cyclone is mainly produced by diabatic processes rather than turbulent mixing of momentum suggesting diabatic processes can dominate over the frictionally generated PV, which was advected into the cyclone centre in the idealised dry studies of Adamson et al. (2006) and Plant and Belcher (2007). An extension of the Attinger et al. (2019) work, considering 288 rapidly developing cyclones in 12 monthly IFS simulations (Attinger et al., 2021), found that the relative importance of the different processes varied between the cyclones and types of anomalies with, for

example, positive PV anomalies along the cold front in cold season cyclones mainly generated by condensation in half of the cases considered and by convection or longwave radiation in the other half of the cases. Both the passive tracers and tendency diagnostics approaches were applied to the attribution of diabatic heating rates to individual processes (in the MetUM and with COSMO output, respectively) in the two branches of a WCB in an exemplar cyclone case study by Martínez-Alvarado et al. (2010). The differences in diabatic PV changes found were attributed to differences in the models used rather than differences

in the diagnostic approach.

Except for Attinger et al. (2021), the above PV-based approaches have been applied to individual real or idealised cyclone case studies over relatively short periods (up to a few days). Motivated by the desire to determine the systematic role of latent heating in cyclones by considering many cyclones in reanalysis datasets or climate model output, Büeler and Pfahl (2017) proposed a simple diagnostic obtained by applying assumptions to the PV budget equation, which they argued are reasonable

when considering averages over a cyclone area (though not when considering individual grid points). By assuming a first order cancellation between the diabatic and advection terms in the PV equation (such that friction is neglected and PV is in steady state, i.e. the Eulerian rate of change is zero) and no horizontal PV advection across the boundaries of a cyclone's area, an expression for the vertical profile of diabatically generated PV is derived (where the diabatic processes are those due to latent heating only i.e. surface fluxes and radiation are excluded). This approach is related to that proposed in Berrisford

(1988) to calculate diabatic PV tendencies from the vertical advection of air, although here the calculation is performed using height rather than $\theta$ as a vertical coordinate and simplified by considering a cyclone volume in which the Eulerian rate of PV change is zero rather than using a Lagrangian approach. Büeler and Pfahl (2017) applied the expression to grid points at which the ascent exceeds a threshold value so as to restrict the calculation to those points where the balance between the diabatic and advection terms is most likely to apply, yielding the latent-heating-induced fraction of the positive lower-tropospheric

PV anomaly averaged over the cyclone area. Evaluation of the diagnostic using simulations of 12 cyclones with and without modified latent heating revealed a good correlation ($r = 0.71$) between the PV anomalies calculated using the simple diagnostic and directly from the sensitivity experiments.

### 5.4.3 Ensemble- and adjoint-based diagnostics

Experiments exploring the sensitivity of cyclones to diabatic processes by producing multiple simulations with modified initial

conditions, boundary conditions or model configurations have provided vital insight into the importance of diabatic processes (Sect. 4.2). However, a limitation of this approach is that a large number of bespoke simulations is potentially required to explore the full extent of possible relevant modifications. In recent years, several studies have exploited operationally generated ensemble weather forecasts for such sensitivity analysis. These ensembles are generated to provide realistic estimates of forecast uncertainty and so the perturbations made to the simulations are typically far smaller than the modifications used in

the sensitivity experiments described in Sect. 4.2, which include, e.g., contrasting simulations with dry and moist versions of a model. As an alternative to analysing an ensemble forecast of a specific weather event, some researchers have analysed ensembles generated by combining many forecasts, or (re-)analysis sequences, of similar weather events (typically by considering the weather systems independently of their precise geographical locations). Two related forms of ensemble analysis, ensemble sensitivity analysis and adjoint analysis, are discussed here.

A form of lagged linear regression commonly termed "ensemble sensitivity analysis" has become a popular statistical tool for determining relationships between a "response function" and model fields either at the same time or prior to the time when the response function is considered (with the latter sometimes referred to as a precursor field). The response function is typically simply a scalar forecast aspect of interest such as the mean-squared forecast error, cyclone intensity, or surface wind speed. Ancell and Coleman (2022) reviewed the usage of ensemble sensitivity analysis and argued that, as yet, the potential

of this technique has not been fully exploited. Ensemble sensitivity analysis yields a spatial field of sensitivities calculated at each grid point as the regression coefficients of the slopes of the linear regression between the response function and the model field (or field anomaly) across the ensemble members, where the regression coefficient is defined as the covariance between the response function and the model field divided by the variance of the model field. This sensitivity field thus yields the change in the response function for a unit change in the model field. To ease comparison between, or enable combination

of, sensitivity maps for different model fields and to remove the counter-intuitive dependence of a sensitivity field on the local variance, the "raw" sensitivity field can be multiplied by the standard deviation of the model field such that the sensitivity at each grid point can be interpreted as the change in the response function due to a one standard deviation increase in the model field at that grid point; this multiplication results in the units of sensitivity being the same as those of the response function. As

a concrete example, a sensitivity map of the minimum MSLP of a cyclone to a one standard deviation increase in the low-level humidity two days previously can be calculated using an operational ensemble forecast of the cyclone event. In some studies weighting functions have also been applied to filter out weak correlations, or significant sensitivities have been defined using the confidence interval on the regression coefficient. It is important to note that the term "sensitivity" is somewhat misleading as it implies a causal relationship between the model field and the response function (i.e., that the increase in low-level humidity two days previously causes a change in the cyclone MSLP). Instead only an association is found; to continue our example, it could be that the change in the cyclone MSLP is dynamically linked to changes in jet structure that happen also to be correlated with changes in low-level humidity rather than the humidity changes being the cause of the MSLP change. Hence, caution needs to be applied when inferring dynamical causation from ensemble sensitivity analysis although causation can be inferred if there exists a plausible physical link or mechanism between the precursor field and response function. Despite this caveat we use the term "sensitivity" below for consistency with the terminology used in the cited literature.

Since being introduced by Hakim and Torn (2008) and Ancell and Hakim (2007), ensemble sensitivity analysis has been used for a variety of purposes (as summarised by Ancell and Coleman, 2022) including investigation into the dynamical processes in, and predictability of, high-impact weather events. However, rather few studies as yet have used this technique to examine the dynamical importance of diabatic processes as reviewed here. As mentioned in Sect. 5.1, Dacre and Gray (2013) examined the relationship between cyclone intensity defined as 850-hPa smoothed relative vorticity (the response function) and the fields of $\theta_e$ and PV two-days earlier and found sensitivity to low-level PV, which was assumed to be diabatically generated, for an ensemble of cyclones in the eastern (but not western) North Atlantic; Fig. 17e shows this sensitivity to precursor PV for eastern Atlantic cyclones, with a peak in the region of the WCB. The sensitivity of the development of North Atlantic and Arctic cyclones to moisture in the WCB has also been investigated using case studies by Berman and Torn (2019) and Johnson and Wang (2021), respectively. For the North Atlantic case, an amplified ridge was found to be associated with stronger heat and moisture transport in the WCB of the upstream cyclone. For the Arctic case, the intensity (defined using MSLP), but not the track, was found to be sensitive to moisture at 500 hPa in the region of the WCB; the authors noted that most WCB transport occurs below 500 hPa but did not find a clear sensitivity to moisture at 850 hPa.

Another series of papers has used adjoint techniques as a computationally efficient way of determining the sensitivity of forecasts to the initial conditions. In this approach an adjoint model of a tangent linear version of the (non-linear) weather forecast model is used to determine the sensitivity of a chosen response function; the adjoint is the transposed operator of the tangent linear model. As explained by Homar and Stensrud (2004), the adjoint traces back the sensitivity of the chosen response function, computing the gradients of the response function with respect to the model state. When the adjoint is integrated back to the initial time, the gradient of the response function to the initial and boundary conditions is obtained; a fuller description of the adjoint approach can be found in Errico (1997). To quote from Homar and Stensrud (2004), "the traditional and the adjoint methodologies are essentially inverse strategies: whereas the traditional approach allows one to evaluate the effect of one perturbation to any number of response functions, the adjoint allows one to evaluate the effect of any perturbation to one particular response function". Ancell and Hakim (2007) showed that ensemble sensitivity analysis and adjoint analysis are mathematically related, but that the sensitivities diagnosed for a sample case study were very different in terms of location, scale

and magnitude, with the sensitivity field from the ensemble sensitivity analysis and adjoint analysis emphasising synoptic-scale and mesoscale features, respectively. This relationship was explained further in Ancell and Coleman (2022) where ensemble sensitivity is described as resulting from a mapping of the direct dynamical influence (given by the adjoint sensitivity) onto all other areas and variables with covariance relationships (see particularly their Fig. 2 where the ensemble sensitivity of future thunderstorms to the surface temperature field is spread along fronts as these are the areas of the atmosphere related to the much smaller region of adjoint sensitivity where temperature perturbations will directly affect the thunderstorms).

Several studies applying adjoint methods are of relevance to our review. For an idealised extratropical cyclone, Langland et al. (1996) (a study previously discussed in Sect. 4.3.2) demonstrated that the accuracy of the tangent linear and corresponding adjoint model was "much higher" when the adjoint model included parametrisations of moist processes, compared to a dry adjoint model. The enhanced cyclone deepening rates that occurred when moist processes were included were attributed to latent heat release in warm-frontal regions that were sensitive to temperature perturbations; notably though, the same regions were found to be sensitive to temperature perturbations in the both the dry and moist adjoint models, which the authors concluded suggested that moist processes do not yield a distinct deepening mechanism to dry baroclinic growth. Homar and Stensrud (2004) used a moist adjoint model to determine the most sensitive areas for the simulation of a Mediterranean cyclone, finding acceptable accuracy of the linear model despite the introduction of nonlinearities by the moist physics. The largest sensitivities were found for the temperature and specific humidity fields in the lower troposphere especially in association with sub-synoptic features along the cold front. Finally, forecasts of high-impact cyclones were found to be strongly sensitive to low to mid-tropospheric moisture in the initial state through the use of a moist adjoint modelling system (Doyle et al., 2014, 2019). The moisture sensitivity field has an upshear-tilted structure positioned along frontal zones (Fig. 17f) and perturbations introduced into both the linear tangent model and nonlinear model were found to expand vertically to interact with mid- and upper-level PV anomalies.

### 5.4.4 Other novel diagnostics

The surface pressure tendency equation as formulated by Knippertz and Fink (2008) and Knippertz et al. (2009) was extended, in response to Spengler and Egger (2009)[23], and used to quantify the contribution of diabatic processes to extratropical cyclone development by Fink et al. (2012); see also Zhang and Ma (2023). The surface pressure tendency is expressed as the sum of the change in geopotential at the upper boundary, the vertically integrated virtual temperature tendency (given by the sum of the effects of the horizontal temperature advection, vertical motions, diabatic processes and an integrated virtual temperature residual), the mass change due to surface precipitation and evaporation, and a pressure tendency equation residual (arising from discretisation errors). The diabatic contribution to the surface pressure tendency was calculated as a residual from the integrated virtual temperature tendency term (so includes a contribution from the integrated virtual temperature residual though this is stated to be small) using ERA-Interim data for five explosively deepening winter storms. Diabatic processes were found to

---

[23]This extension included the addition of a term for changes in the geopotential at the top of the considered atmospheric column (set to 100 hPa) to recognise that while heating can result in the adjustment of pressure and density profiles in an atmospheric column, reduction in hydrostatic surface pressure can only be caused by net mass removal; see also Spengler et al. (2011).

dominate over baroclinic processes for three of the five storms, including storm Lothar for which the diagnostic results are illustrated in Fig. 17g (the grey bars in the right hand panel show diabatic contributions to the negative pressure tendency of around 60% over a 2.5 d period). Pirret et al. (2017) extended this analysis to a set of 60 severe European windstorms with the surface pressure tendency equation integrated over the intensification period of the storm. Diabatic contributions varied from 15% to nearly 60% (for storm Xynthia in late February 2010) though the baroclinic term dominated in 48 of the 60 storms. The diabatic contribution was found to be larger for storms that spent more time to the equatorward side of the jet. More recently this surface pressure tendency diagnostic was applied to a medicane case study (Fita and Flaounas, 2018) where it was used, together with cyclone phase space diagrams and a water budget diagnostic, to formulate a three-stage conceptual lifecycle for these systems.

A novel diagnostic has also been developed to quantify diabatic effects on baroclinicity, motivated by a long-standing debate about the maintenance of baroclinicity in storm track regions. To this end, Papritz and Spengler (2015) derived an equation for the material derivative of the slope of isentropic surfaces, which corresponds to the modulus of the ratio between the horizontal gradient of $\theta$ and its vertical derivative. The equation shows that the material change of the slope is determined by isentropic advection, tilting of the isentropes due to differential vertical motion, and differential diabatic heating due to latent heating in clouds, radiation, and surface sensible heat fluxes. They studied the rapid intensification of winter storm Klaus in 2009 as an exemplar application to an individual cyclone and found that the decrease of slope due to tilting of the isentropes by baroclinic instability release is to a large degree compensated by an increase of slope due to latent heat release in the cyclone. This pattern was qualitatively confirmed by the climatological evaluation of the different terms of the slope equation over the North Atlantic, which revealed that surface sensible heat fluxes and latent heating in cold air outbreaks over the Gulf Stream region and in the Labrador Sea contribute essentially to restoring the slope in the lower troposphere. These results confirmed that diabatic processes are essential for the self-maintenance of baroclinicity along storm tracks (Hoskins and Valdes, 1990). The study by Weijenborg and Spengler (2020) used the slope diagnostic to show that, after the passage of the particularly intense North Atlantic cyclone Dagmar in December 2011, baroclinicity was increased due to strong diabatic heating, which was potentially important for the cyclone clustering that occurred during the following week.

A recent addition to the approaches available for diabatic processes attribution is through the use of the semi-geotriptic (SGT) balance tool created by Cullen (2018). This tool was used to partition the 3D ageostrophic flow into a "balanced-flow component" calculated from the geostrophic forcing of a generalisation to the omega equation, a "diabatic component" attributed to ageostrophic flow response to diabatic heating and a remainder termed the "unbalanced component". The contribution of diabatic processes to the ageostrophic advection of PV can then be diagnosed. Note that geotriptic balance is the same as the more familiar geostrophic balance but with the inclusion of Ekman friction in the boundary layer. Sánchez et al. (2020) used this tool to demonstrate that so-called "predictability barriers", defined as occurring when ensemble spread grows more quickly than usual but ensemble-mean forecast error grows even faster, are linked to events with strong diabatic influences on the advection of the tropopause—a topic that is discussed in detail in the next subsection. The SGT balance tool was also used by Hardy et al. (2023) in their investigation of Borneo vortices.

### 5.4.5 Summary

Sophisticated diagnostics have been developed since the turn of the century to diagnose diabatic processes in NWP simulations that lead to changes in PV and the influence of these processes on cyclone development. The related approaches of passive diabatic tracers and tendency diagnostics have enabled partitioning of the PV budget and determination of the causes of diabatic modification of the tropopause and cyclone structure. Both methods also emphasise the Lagrangian nature of PV modifications: Instantaneous PV tendencies must be integrated along the flow to explain the observed PV anomalies in extratropical weather systems. These approaches are built on the parametrisation schemes in the utilised model, which has the advantage that the behaviour of schemes can be compared between models. However, given the uncertainties of parametrisations, caution needs to be applied when interpreting the results in terms of real-world processes. Ensemble and adjoint-based diagnostics have been employed that provide more nuanced information about the sensitivity of cyclone evolution to diabatic-processes-related features, such as the initial humidity field, than the simple dry vs. moist sensitivity experiments described in Sect. 4.2. However, these diagnostics are yet to be widely applied to examine the dynamical importance of diabatic processes. All these methods are technically advanced, but the use of instantaneous diagnostics, like the surface pressure tendency equation, tendency of the slope of isentropic surfaces, and semi-geotriptic balance tool, can also yield valuable information about the role of diabatic processes. While surface pressure tendency equation diagnostics have a long history in the literature (albeit with relatively few studies considering diabatic processes in extratropical cyclones), there are currently very few studies on the application of the latter, much more recently proposed, two diagnostics; in particular, studies using the semi-geotriptic balance tool are currently restricted to those using the MetUM as the formulation was developed for consistency with the equations used in this model.

## 5.5 Diabatic outflows and Rossby waveguide dynamics

The novel diagnostics portrayed in the previous section help identify diabatic processes from the surface to the tropopause region. Research on diabatic effects near the tropopause, in particular for the dynamics of Rossby waves and blocks became very prominent in the last decades, continuing earlier research on this theme as summarised mainly in Sect. 4.1.6. A central theme in these studies is the evidence that so-called "diabatic outflows" substantially influence the shape and amplitude of Rossby waves and their downstream development. The term diabatic outflow refers to the cross-isentropic ascent due to latent heating in larger-scale cloud systems near the extratropical waveguide (normally in WCBs, WCB-like airstreams from tropical cyclones during extratropical transition, or mesoscale convective systems) and the subsequent horizontal divergence in the upper troposphere of the ascended air with typically low PV values (Sect. 4.1.5). These low-PV outflows often reach into pre-existing upper-level ridges, or they start forming such ridges, leading to the notions of "ridge amplification" and "ridge building" (e.g. Grams et al., 2013). This section has four subsections, addressing negative PV anomalies in diabatic outflows (Sect. 5.5.1), important findings from studies about extratropical transition events (Sect. 5.5.2), and the role of diabatic outflows for medium-range forecast errors (Sect. 5.5.3) and for the formation of blocks (Sect. 5.5.4).

 **5.5.1 Diabatic outflows and negative upper-level PV anomalies**

An early study that made a causal connection between a diabatic outflow from a North Atlantic WCB and the downstream Rossby wave breaking over western Europe was by Massacand et al. (2001). They used full physics and dry mesoscale model simulations to analyse the chain of events leading to a devastating flood in the southern Alps in September 1993. Their diagnostics and model experiments indicated that the anticyclonic circulation associated with a negative upper-level PV anomaly over the eastern North Atlantic, which was enhanced by the diabatic outflow of the WCB of a cyclone over Newfoundland, was important for transforming a farther-downstream pre-existing broad positive PV anomaly into an elongated PV streamer, which in turn instigated the heavy precipitation. Backward trajectories from the low-PV region in the ridge (mean PV value of 0.15 pvu) showed the characteristics of a WCB with more than 30 K of integrated heating. The dry simulation, initiated about 3 d prior to the fully established PV streamer, lacked the diabatic outflow and the formation of the PV streamer. Similar linkages between Rossby wave breaking and upstream diabatic outflows were found for heavy precipitation events over West Africa (Knippertz and Martin, 2007). The authors also emphasised the importance of "the strong divergent outflow near the tropopause that supports large negative isentropic PV advection", which became a specific research subject of the NAWDEX field campaign ("role of divergent outflow of WCBs for ridge amplification", Schäfler et al., 2018, Fig. 18d here). However, Meier and Knippertz (2009), who investigated a similar heavy precipitation event in the Cape Verde region found comparatively weak sensitivity to upstream latent heating, indicating that Rossby wave breaking is strongly influenced by upstream diabatic outflows in some but not in all cases. Madonna et al. (2014a) quantified the co-occurrence of PV streamers and WCB outflows statistically and found that about 60% of all WCB outflows co-occur with PV streamers in their vicinity, and in most cases the PV streamers form downstream (as in the studies mentioned above). While the case studies mentioned above investigated the potential causal link between diabatic outflows, Rossby wave breaking, and the occurrence of high-impact precipitation events, later studies showed a connection between diabatic outflows and other types of weather extremes downstream. Prominent examples of such studies are Grams and Blumer (2015) about severe convection in central Europe in September 2011 related to the outflow of the transitioning Hurricane Katia, Bosart et al. (2017) about the sequence of extreme events over North America in October 2007 (wildfires, cold surges, and heavy rainfall) that were related to a high-amplitude Rossby wave train influenced by WCB outflows from an intensifying North Pacific DRW and a western North Pacific tropical cyclone, and Oertel et al. (2023b) about the exceptional North American heatwave in June 2021 related to the outflow of two North Pacific WCBs during the week prior to the peak of the heatwave. These examples of weather extremes related to upstream latent heating provided additional support for the in-depth analysis of diabatic outflows in the last two decades.

For diabatic outflows to influence the downstream flow, it is essential that their low-PV air in the outflow forms an intense negative PV anomaly, which then induces an equatorward flow downstream (Fig. 18d). This is one of several mechanisms of how wave disturbances on a jet stream can be generated or enhanced (see the schematic by Davies and Didone, 2013, their Fig. 12). These authors explained this mechanism in the following way: "... the mere deposition of low-tropospheric air with low PV at tropopause levels would be dynamically significant because it could constitute a major negative PV anomaly relative to the ambient upper-level air". Therefore. the diabatic outflow, with typical PV values of 0.1–0.4 pvu (Madonna et al., 2014b,

their Fig. 6f—see also the conceptual considerations by Methven (2015) and Saffin et al. (2021)—must reach regions that are climatologically in the stratosphere, i.e., have much larger PV values. Indeed, Madonna et al. (2014b) quantified the PV anomalies in the outflow of WCBs (i.e., the deviations of the actual PV from the climatology) and found a broad distribution with frequent values between $-1$ and $-4$ pvu. Schemm et al. (2013), who compared idealised moist and dry baroclinic wave simulations, found upper-level PV reductions of about $-2.5$ pvu in the region of the WCB outflow in the moist compared to the dry simulation. Clearly, anomalies with such amplitudes can influence the upper-level flow dynamics if they occur close enough to the PV waveguide, i.e., the band with an intense isentropic PV gradient.

The study by Grams et al. (2011), which investigated an extratropical transition event over the North Atlantic and its impact on the downstream flow, was most likely the first to visualise the diabatic outflow as instantaneous isentropic-intersection points of WCB trajectories on isentropic PV charts. This type of visualisation reveals the region where the cross-isentropically ascending WCB amplifies the ridge, which typically agrees with a thick cirrus shield in the satellite imagery (Fig. 18a). Joos and Wernli (2012) studied a North Atlantic WCB that contributed to the formation of a prominent and large-scale ridge over Scandinavia, and in their case the diabatic outflow, again visualised as isentropic-intersection points, filled a large part of this ridge between 60 and 75°N (their Fig. 3a). In both these cases, the non-linear Rossby wave evolution was strongly affected by the diabatic outflows. It is briefly mentioned that in addition to contributing to ridge amplification and the formation of blocks (see Sect. 5.4.4), diabatic outflows can also lead to Rossby wave initiation, which is defined as events where an initially zonal and straight jet stream segment is perturbed, either from the stratospheric or tropospheric side, and starts to undulate for other reasons than downstream development. Röthlisberger et al. (2018) objectively identified a climatology of such events and found a frequency maximum in winter over the western North Pacific. Composites of the large-scale flow during these events showed that initially weak waves are often strongly amplified by WCB outflows (their Fig. 7).

### 5.5.2 Diabatic outflows during extratropical transition events

Very important contributions for understanding the dynamical relevance of extratropical diabatic outflows came from PV-based studies on extratropical transition events. Agustí-Panareda et al. (2004) investigated the explosive re-intensification of hurricane Irene after extratropical transition. Sensitivity experiments with and without the hurricane vortex and its humidity anomaly in the initial conditions prior to re-intensification revealed that extratropical cyclogenesis takes place regardless of the initial presence of the hurricane. However, with the hurricane, the track of the extratropical cyclone was more zonal and its deepening rate twice as fast as without the hurricane. As the main reason for these differences, the authors identified the negative PV anomaly and enhanced horizontal wind divergence at upper levels due to the diabatic outflow of the transforming hurricane. This diabatic outflow from the tropical cyclone and its effect on the upper-level jet can be enhanced by the diabatic outflow from so-called predecessor rain events, which occur poleward of the tropical cyclone along the zone of high baroclinicity and are fed by strong moisture transport induced by the cyclone (Moore et al., 2013). Riemer et al. (2008) performed idealised channel model simulations of an extratropical transition scenario and assessed the impact of the transition on the downstream flow using PV inversion and the partitioning of the flow into its rotational and divergent parts. They found that the upper-level diabatic outflow of the tropical cyclone with low PV contributed importantly to the formation and amplification of a jet streak and a

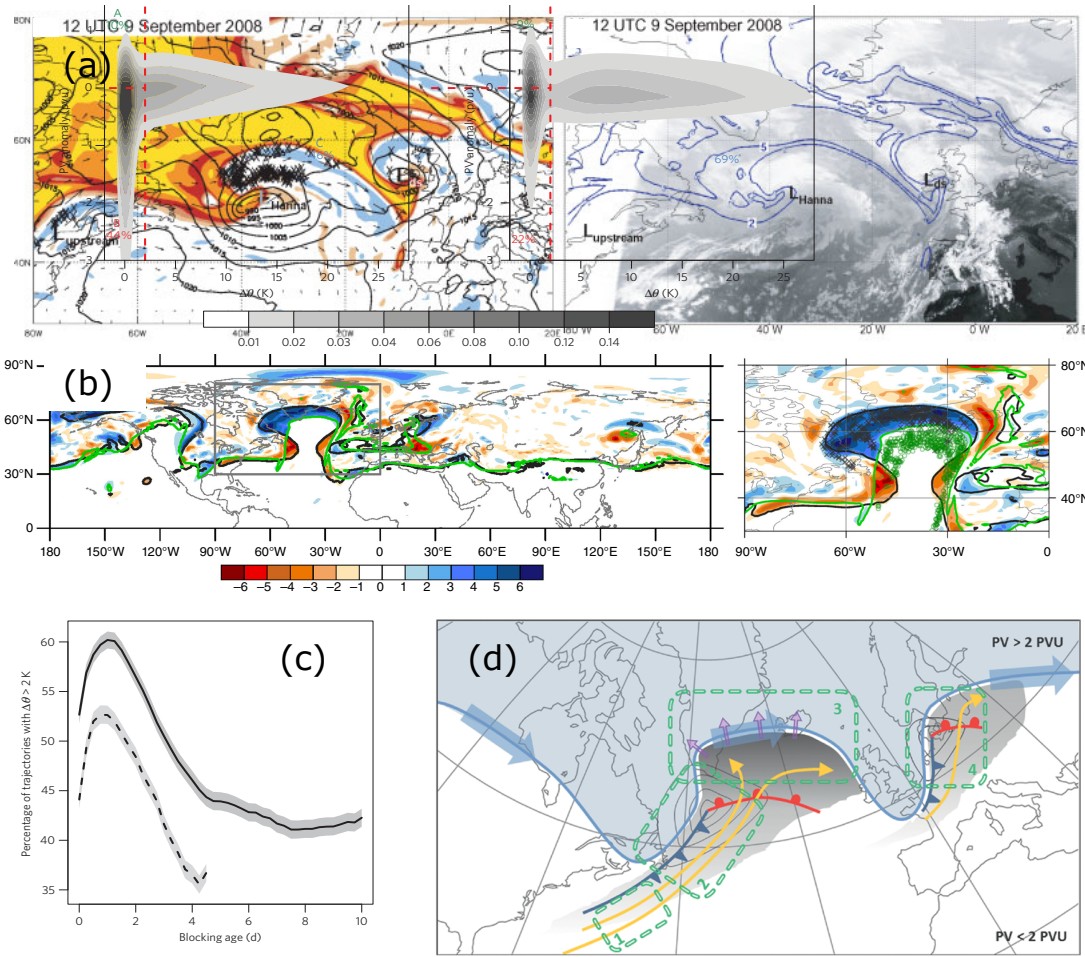

**Figure 18.** Diabatic outflows. **(a)** from Grams et al. (2011, their Fig. 4c,d), left: PV (colours) and wind vectors on 320 K, MSLP (black contours every 5 hPa), and isentropic-intersection points of WCB trajectories with the 320-K isentrope, and right: Meteosat IR imagery and 320-K PV contours (blue) at 12 UTC 11 September 2008; **(b)** from Martínez-Alvarado et al. (2016b, their Fig. 2c), error at day 5 of the forecast initialised at 12 UTC on 19 January 2011 in terms of PV on 320 K (positive values indicate an overestimation in the forecast); the right panel shows zoomed-in details inside the gray box marked in the left panel; black and green crosses show WCB intersection points with the 320-K isentrope in the analysis and the forecast, respectively; **(c)** from Pfahl et al. (2015a, their Fig. 2), percentage of three-dimensional backward trajectories affected by diabatic heating as a function of blocking age (solid line); the dashed line shows the percentage of trajectories affected by diabatic heating for anticyclonic anomalies that persist less than 5 d and are thus not classified as blocking; and **(d)** from Schäfler et al. (2018, their Fig. 1) schematic with Rossby waves, cyclones, and WCBs over the North Atlantic: the blue line indicates the waveguide separating stratospheric (blue background) from tropospheric air (white background); the jet stream (dark blue arrows) follows the waveguide; cyclones develop ahead of upper-level positive PV anomalies; grey lines indicate MSLP; standard notation is used for surface fronts; grey-shaded areas indicate clouds related to WCBs (yellow arrows); purple arrows mark divergent outflow at the tropopause; and green boxes outline (1) WCB inflow, (2) ascent, and (3) outflow, and (4) a region of expected downstream impact. All figures used with permission: (a,b) from Wiley, (c) from Springer Nature, and (d) from the American Meteorological Society.

ridge–trough couplet downstream of the transition. Torn (2010) investigated ensemble forecasts for the extratropical transition of two typhoons and found via multivariate regression calculations that the precipitation along the baroclinic zone northeast of the tropical cyclone and its associated latent heating have a large impact on the amplitude and area of the downstream ridge. Importantly, Torn (2010) pointed out that this precipitation is modulated by the moisture flux on the east side of the tropical cyclone, which is related to the cyclone's winds induced by the diabatic low-level PV production[24]. Several later studies about extratropical transitions confirmed the potentially strong modification of the downstream flow by the diabatic outflows of the tropical cyclones and discussed reasons for the large case-to-case variability in the amplitude of this modification, as summarised in the review by Keller et al. (2019). In particular, it was shown that in situations with a strong extratropical jet, the low-PV air associated with the outflow of the transitioning tropical cyclone is immediately advected downstream and ridge building is reduced (Riboldi et al., 2019). Conceptually, the diabatic outflow of transitioning tropical cyclones and of WCBs associated with extratropical cyclones act similarly on the upper-level waveguide via their negative PV advection by the divergent outflow. However, there are at least two important differences related to (1) the main PV features that eventually lead to the diabatic outflows and (2) the intensity and altitude of the outflow. Considering the PV features, for WCBs, the poleward ascent is part of the growing moist baroclinic wave and strongly influenced by the positive upper-level PV anomaly of the upstream trough; in contrast, the poleward ascent associated with an extratropical transition is strongly related to the positive low-level PV anomaly of the tropical cyclone. Considering the outflows, these are typically higher for transitioning tropical systems, suggesting that the diabatic outflow during extratropical transition can have a particularly strong impact on the downstream flow in cases where the outflow reaches far enough poleward.

### 5.5.3 Diabatic outflows and forecast errors

The finding that diabatic outflows can have an important downstream impact is consistent with a series of studies that reported an exceptionally large medium-range forecast uncertainty (ensemble spread) downstream of extratropical transitions (e.g., Anwender et al., 2008; Grams et al., 2015). Similarly, case studies of particularly poor forecasts highlighted the role of misrepresented WCB outflows, which led to strong error amplification and sometimes also downstream propagation (e.g. Lamberson et al., 2016; Martínez-Alvarado et al., 2016b; Grams et al., 2018). In the case study of Martínez-Alvarado et al. (2016b), the WCB intensity was overpredicted in the ECMWF high-resolution forecast, but the outflow was too far south, resulting in an underestimation of the magnitude of its negative PV anomaly by up to 5 pvu (Fig. 18b). Essentially the same forecast error also occurred in all members of the ECMWF and U.K. Met Office ensembles. Whereas in this case the forecast errors were largest in the direct environment of the WCB outflow, the poor forecast described by Lamberson et al. (2016) featured an inaccurate representation of a WCB over the North Pacific, which led to an erroneously explosive cyclone development over western Europe, indicating the possibility for rapid downstream error propagation along the PV waveguide. The forecast bust of the ECMWF ensemble described by Grams et al. (2018) was similar to the case described by Martínez-Alvarado et al. (2016b),

---

[24]This consideration is a nice reminder of the fact that diabatic PV modification occurs simultaneously and with opposite signs at lower and upper levels, and that both low-level cyclonic and upper-level anticyclonic PV anomalies influence the subsequent moisture transport, vertical motion, diabatic heating and PV modification.

featuring rapid error growth due to the misrepresentation of a North Atlantic WCB. The outflow of this WCB was too far south and therefore the ensemble strongly underestimated the amplitude of its negative PV anomaly and the onset of blocking over Europe.

The two systematic predictability studies by Rodwell et al. (2013) and Gray et al. (2014) did not explicitly consider diabatic processes, but both presented results that indicate a potential key role of diabatic outflows. Rodwell et al. (2013) identified a set of more than 500 forecast busts over Europe at day 6 among the forecasts made within the ERA-Interim project for the period 1989–2011. The composite of this set of busts showed in the initial conditions a trough over the Rockies and high values of CAPE over the eastern U.S. Together this indicates that many of these poor forecasts occurred in situations with mesoscale convective systems with intense diabatic outflows over the eastern U.S., which triggered a particular Rossby wave pattern across the North Atlantic that led to a meridional ridge-trough pattern over Europe. The authors concluded that "... [our] results do confirm that the physics [e.g., in diabatic outflows] is, in general, an active (non-negligible) component in the evolution of the bust forecasts". Gray et al. (2014) quantified systematic forecast errors from three global forecast centres of upper-level Rossby waves in winter and found that the wave amplitude reduced, i.e., both ridges and troughs became too weak, with lead time up to about five days. This forecast error behaviour is consistent with the assumption of too weak latent heating and diabatic outflows into upper-tropospheric ridges. However, the authors close with a note of caution that "... causality is not proven here. Further work needs to be done to attribute the systematic errors in Rossby wave structure shown here to model limitations in the representation of diabatic processes". An important study in this direction by Pickl et al. (2023) found that the forecast skill of the ECMWF ensemble is generally reduced when the WCB activity is high, and that WCB activity is particularly increased when the error growth is largest. The key role of diabatic processes for forecast error amplification is confirmed by the results from the highly idealised baroclinic instability study by Haualand and Spengler (2021), which revealed that the sensitivity of baroclinic instability to latent heating is significantly larger than to tropopause structure (tropopause altitude and sharpness).

Potential reasons for occasionally misrepresented WCB outflows are errors in the initial conditions (e.g., of the humidity in the WCB inflow) and limitations of the physical parametrisations (in particular cloud microphysics and deep convection). Addressing the sensitivity to the initial humidity field, Schäfler and Harnisch (2015) ran simulations of a cyclone in the western North Pacific with a WCB inflow north of tropical cyclone Sinlaku. Airborne water vapour lidar measurements indicated a positive moist bias of the ECMWF analyses in this inflow region, and a simulation with the corrected inflow moisture led to weaker integrated latent heating along the WCB, a lower WCB outflow, and a modified jet structure downstream, which agreed better with the verifying analysis than the uncorrected simulation with the too moist WCB inflow. Other sources of forecast errors are parametrisations. The detailed analysis of microphysical processes along a WCB revealed that condensational growth of cloud droplets and vapour deposition on frozen hydrometeors each contributed $\sim 10\,\mathrm{K}$ to the total latent heating (Joos and Wernli, 2012). The dominance of these two processes, identified in a simulation with 14 km grid spacing, a one-moment microphysics scheme, and offline trajectories, has recently been confirmed by Oertel et al. (2023a), who investigated microphysical processes along the WCB of the NAWDEX IOP2 cyclone Vladiana, in a convection-permitting ICON simulation with two-moment microphysics and online trajectories. Considering these two microphysical processes, the parametrisation of depositional growth is clearly more uncertain in weather and climate models (e.g., due to the treatment of supersaturation). This process is therefore

a source of uncertainty for the total latent heating in WCBs (and, in turn, for the isentropic outflow level) and for the timing of the ascent. Joos and Forbes (2016) compared two simulations of the same WCB with different microphysical parametrisations and found a substantial effect on the position of the WCB outflow due to differences in the timing and amplitude of the cloud-diabatic heating. This led to initially small shifts in the location of the tropopause, which however amplified downstream and led to distinct differences in the upper-level PV pattern between the two simulations.

Also important for the simulation of WCBs in current NWP models is the parametrisation of deep convection. Martínez-Alvarado et al. (2014b) compared simulations of a North Atlantic WCB with two mesoscale models with grid spacings of 12–14 km and found that the different convection schemes caused significantly different contributions to the total heating along the WCB trajectories, which differed by about 5 K. They also emphasised that the split of a WCB into cyclonic and anticyclonic branches can be sensitive to parametrisations, with consequences for the resulting PV anomalies in the outflow of the two branches. Rivière et al. (2021) and Wimmer et al. (2022) compared the characteristics of the WCB of the explosively deepening NAWDEX IOP6 cyclone in simulations with 15 km grid spacing in the region of interest. The simulations differed only in the treatment of deep convection: one treated convection explicitly and the other two used different parametrisations with different approaches for closure. These differences influenced the ascent behaviour of the WCB, which was more rapid and abrupt without parameterised deep convection and more moderate but sustained with parameterised deep convection. Averaged over all WCB trajectories, heating was found to be stronger for parameterised deep convection and one of the schemes had a more prominent upper-level PV destruction compared to the other two simulations, which led to substantial differences in the jet position and intensity. Taken together, these studies clearly revealed that initial condition errors and parametrisation shortcomings can be large enough to considerably affect the large-scale flow evolution. In situations where this occurs, the WCB takes the role of integrating these effects along the flow and creating upper-level PV errors near the waveguide, with potentially strongly deteriorated forecasts downstream.

The PV framework is also well suited to systematically quantify the role of diabatic processes for forecast error growth on synoptic scales[25]. An early study in this direction by Dirren et al. (2003) diagnosed "PV errors", i.e. differences of isentropic PV fields between forecasts and the analysis, and mentioned the role of deficient diabatic effects in extratropical cyclones for the occurrence of prominent PV errors. The mathematical framework by Davies and Didone (2013) disentangled adiabatic and diabatic contributions to the material derivative of PV errors. Assuming that diabatic tendencies at the level of the tropopause are small, they concluded that on isentropes in the vicinity of the tropopause, the direct cloud-diabatic contribution to the PV error tendency after a mis-forecast of a diabatic event will be small, whereas the indirect effect due to earlier diabatic generation of PV errors accumulated along the ascending flow can be substantial. (This conclusion resonates with the time-integrated PV modification diagnostics discussed in Sect. 5.3.). For a specific synoptic sequence, Davies and Didone (2013) found that the PV-error pattern on the 320 K isentrope indicated that the failure to capture a significant cloud-diabatic event associated with rapid cyclogenesis off the eastern seaboard of North America contributed to substantial errors in the downstream Rossby wave evolution, wave breaking, and block formation over Europe. Baumgart et al. (2018) combined the concepts developed

---

[25]A discussion of the complex processes involved in upscale error growth from the smallest convective scales is outside the scope of this paper. The interested reader can find excellent contributions to this theme in recent studies (e.g. Durran and Gingrich, 2014; Judt, 2018; Selz et al., 2022).

by Davies and Didone (2013) and Teubler and Riemer (2016), who studied the amplitude evolution of individual troughs and ridges with a partitioning of the PV tendency equation, to further develop the PV diagnostic of forecast errors. They considered the local tendency of error potential enstrophy (the squared PV error) integrated over an area of interest on an isentropic surface.

In this equation, diabatic processes appear directly (via local PV non-conservation) and indirectly (via upper-level divergence related to latent heat release below). In their winter case study, Baumgart et al. (2018) found significant contributions to the error growth from the upper-tropospheric divergent flow associated with latent heat release below. However, these contributions to the error growth were smaller than those from non-linear near-tropopause dynamics. Direct PV non-conservative processes, albeit comparatively small, were largest in upper-level ridges and PV cutoffs, where radiative tendencies contributed to error

amplification. A similar methodology was used in the context of ensemble prediction, based on a tendency equation for the ensemble variance of PV, by Baumgart and Riemer (2019). Investigating the extratropical transition of tropical storm Karl, they found that locally, where the cyclone interacted with the PV waveguide, variance amplification was dominated by the divergent outflow and moist baroclinic vertical interaction. Despite clear evidence from several studies that diabatic outflows from extratropical transitions and WCBs in extratropical cyclones can be associated with downstream error growth, forecast busts,

and increased ensemble spread, there is no linear relationship of these downstream effects on predictability with the intensity of the latent heating. Torn (2017) compared downstream ensemble spread in transition cases with stronger and weaker latent heating and did not find a systematic difference of ensemble spread downstream, suggesting that "while large PV advection by the irrotational wind [in diabatic outflows] may produce a larger downstream response, that response is not necessarily less predictable".

### 5.5.4 Diabatic outflows and the formation and maintenance of blocks

One definition of atmospheric blocks is that they are quasi-persistent, synoptic-scale, positive anomalies of geopotential height, or, equivalently, negative anomalies of PV in the upper troposphere. Such anomalies can be created, amplified, and maintained either via isentropic poleward advection, or via cross-isentropic transport of low PV within diabatic outflows. A pioneering analysis of the potential role of diabatic heating for block formation was performed by Lazear (2007) in their MSc thesis.

Based on moist vs. dry simulations of two blocking episodes over the central North Atlantic, complemented by an adjoint-sensitivity analysis, they concluded that "the low PV within blocking anticyclones is related to condensational heating within strengthening upstream synoptic-scale systems". Using backward trajectories from an amplifying ridge, which developed into a block over Greenland, Archambault et al. (2010) diagnosed latent heating and negative PV tendencies along the trajectories during the last 6 h prior to arrival in the ridge. They concluded that adiabatic and diabatic processes collectively contributed to

the onset of blocking. A first systematic quantification of latent heating along trajectories that end up in blocks was done by Pfahl et al. (2015b), based on 20 years of ERA-Interim data. They showed that 46% of the air contributing to a block is heated by more than 2 K in the 3 d before its arrival in the block. The median heating along these trajectories amounted to 7 K. Some trajectories experienced a heating of more than 20 K, indicative of upstream WCBs whose outflow directly contributed to a block. (The complement of 54% non-heated trajectories experienced weak radiative cooling.) The percentage of trajectories

with latent heating exceeding 2 K is enhanced during the first 3 d of the lifecycle of the blocks (Fig. 18c), indicating that

latent heating is slightly more important during block onset than maintenance. Interestingly, this figure also shows that the percentage of trajectories with latent heating is larger for long-lived negative PV anomalies that constitute a block compared to shorter-lived negative PV anomalies. This climatological analysis revealed that, in addition to quasi-horizontal advection of air with low PV, ascent from lower levels associated with latent heating in clouds is of first-order importance for the formation and maintenance of the negative PV anomalies in blocks. These findings were later confirmed by the global, almost 40-y climatological analysis of Steinfeld and Pfahl (2019). An important finding of their study was that block-centred composites revealed prominent negative upper-level PV advection by the divergent component of the horizontal wind along the western flank of the block. This effect of the diabatic outflow contributes to a westward amplification of the ridge against the eastward advection by the background flow, and hence to the stationarity of the block. In order to better understand the causal link between latent heating and blocks, Steinfeld et al. (2020) refined the approach used by Lazear (2007) and performed "locally dry" sensitivity experiments with the global ECMWF model for five blocking events that occurred in different regions and seasons. In these simulations, latent heating in clouds was set to zero in a confined region upstream of the block, which was previously identified as contributing to the diabatic outflow into the block. This regional elimination of latent heating had substantial effects on the upper-tropospheric circulation in all case studies, but with large case-to-case variability. Some blocks did not develop at all without upstream latent heating, while others still developed but with reduced amplitude, size, and lifetime. Consistent with the earlier purely diagnostic studies, the strong influence of latent heating on the blocks was due to the cross-isentropic injection of low-PV air into the upper troposphere and the interaction of the associated divergent outflow with the upper-level PV structure. These results agree with the findings of Hauser et al. (2023) who investigated a case study of an European block with a combination of three complementary diagnostics. They concluded that pulse-like amplifications of negative upper-level PV anomalies before and during the life cycle of the block were mainly due to latent heating and the associated PV advection by the divergent wind. Finally, the relevance of these processes for operational forecast skill was revealed by several recent studies. The link between skillful block onset forecasts and skillful forecasts of the location and intensity of upstream cyclones in the days prior to the block onset was demonstrated through applying ensemble sensitivity analysis (see Sect 5.4.3 for an explanation of this technique) to the ECMWF operational ensemble by Maddison et al. (2019). Maddison et al. (2020) then showed that operational model development changes affecting diabatic processes were sufficient to impact forecast skill during blocking events due to changes in the WCBs of cyclones upstream of the blocks. In qualitative agreement with these results, a systematic analysis of ECMWF reforecasts of the onset of European blocking indicated that the model struggles to predict block onsets at lead times exceeding 10 d in line with a misrepresentation of WCB activity (Wandel et al., 2024).

## 5.5.5 Summary

Large progress has been made in the last two decades in understanding how diabatic processes influence upper-level Rossby wave dynamics in the extratropics. A central finding is the importance of so-called diabatic outflows from WCBs, the extratropical transition of tropical cyclones, and from organised convective systems, for the amplification of ridges, the formation of blocks, and downstream Rossby wave breaking. These outflows are produced by the cross-isentropic transport of air with low

PV to the upper troposphere, leading to the formation of intense negative PV anomalies and strongly enhanced PV advection by the divergent wind at the outflow level. Research in this area profited from increased interest in extratropical transition events, the systematic use of Lagrangian approaches, and detailed analyses of medium-range forecast errors. These errors appear to be often related to a chain of events starting with the misrepresentation of diabatic outflows, leading to errors in the amplitude of the upper-level negative PV anomaly, with implications for the downstream wave propagation and genesis of new weather systems. Realising their relevance also for upper-level flow dynamics and predictability, diabatic outflows, and in particular WCBs, are nowadays considered as essential elements of extratropical dynamics, in addition to their original role in explaining the observed cloud and precipitation structures.

## 5.6 Field experiments and the investigation of mesoscale substructures

The early major field campaigns on extratropical cyclones have been highlighted in Sect. 4.1 (see Table 1). Since the turn of the Century, three further major experiments using research aircraft have taken place: the international T-PARC research initiative with a summer field phase in August to October 2008 and a winter field phase in February and March 2009; the U.K.-based DIAMET campaign, which was part of the DIAMET research project and took place during four campaign periods between September 2011 and August 2012, and the international NAWDEX campaign in autumn 2016. T-PARC addressed the dynamics and short-range prediction of weather systems, in particular tropical cyclones and their extratropical transitions, over the eastern Asian and the western North Pacific and the impact of these systems on the formation of high-impact weather events over the "downstream" regions such as the eastern North Pacific and North America. The DIAMET project was one of three strategically-funded projects on "Storm risk mitigation". Notably, the structure and consequences of PV and moisture anomalies in cyclonic storms were specifically included as questions to be addressed in the call for this funding. NAWDEX was conceived as one of a series of campaigns of the World Meteorological Organisation coordinated by THORPEX and thus originally termed T-NAWDEX (with the "T" referring to THORPEX). It was included in the THORPEX European plan with the ambition for it to take place in autumn 2012 or 2013. NAWDEX was preceded by two pilot field campaigns, in the U.K. in November 2009 and in Germany in 2012 (the latter named T-NAWDEX-Falcon due to the use of the Falcon aeroplane). The aim of NAWDEX was to obtain measurements to enable the impact of diabatic processes on disturbances of the jet stream (Sect. 5.5) and the subsequent influence on downstream high-impact weather to be evaluated. A factor in the rationale for both the DIAMET field campaign and NAWDEX was the exploitation of new remote sensing instrumentation and the latest convection-permitting model configurations enabling study of mesoscale substructures (e.g., sting jets, embedded convection and associated PV modification) and validation with observations. Here the goals of the campaigns and key findings relating to the roles of diabatic processes on the dynamics of cyclones are given. The types of new observations and model configurations used allowed higher spatial resolution analysis than was typically possible from earlier campaigns and so, amongst other outcomes, led to new understanding about the importance of diabatically generated mesoscale substructures in cyclones.

Summaries and highlights of the DIAMET, T-NAWDEX-Falcon and NAWDEX campaigns are described in their respective overview papers (Vaughan et al., 2015; Schäfler et al., 2014, 2018). The DIAMET campaigns used the U.K.'s meteorological research aircraft (the BAe 146 aircraft of the U.K. FAAM) together with ground-based measurements from radar, wind profilers,

automatic weather stations and radiosondes (including 55 non-routine sonde launches). The research aircraft was equipped with instruments to measure in-situ winds (including turbulence), thermodynamic parameters, microphysics, and chemical tracers; dropsondes were also launched from the aircraft to measure vertical profiles of temperature, humidity and pressure. The aircraft was based in Cranfield (North of London, England) for three of the campaign periods and in Exeter in the southwest of England for the stormy 2011 winter campaign. The earlier U.K.-based T-NAWDEX pilot campaign used the same aircraft with similar instrumentation; flights from that campaign were also analysed as part of the DIAMET project (as described in Vaughan et al., 2015). There were 19 and 3 flights, respectively, during the DIAMET and T-NAWDEX campaigns. These flights were associated with 16 IOPs and most, but not all, of these IOPs were linked to cyclones. The focus of the DIAMET campaign was to obtain measurements of dynamics, cloud physics, and air–sea fluxes for calculation of diabatic heating rates in cyclones and subsequent model evaluation. The link between diabatic processes and PV anomalies was at the core of this project with scientific questions relating the mesoscale structure of the PV distribution to latent heating and air-sea fluxes, and the effect of that mesoscale PV structure on the local cyclone winds and precipitation and downstream cyclone evolution (including forecast error).

The T-NAWDEX-Falcon campaign was based at Oberpfaffenhofen airport in Germany but made stops at several other European airports. As well as in-situ measurements (including three instruments to measure total water and water vapour and instruments for chemical tracers), dropsondes could be launched. The focus of the campaign was WCBs in extratropical cyclones and measurements from three WCBs (three IOPs) were obtained from nine flights. A challenging goal of this campaign was to sample the same air parcels at different times (and thus locations) to obtain a Lagrangian estimate of the integrated diabatic effects. The successful achievement of this goal was demonstrated by aircraft sampling of an inert tracer gas released in the WCB inflow region, confirming the model-derived trajectory of the WCB from the Mediterranean boundary layer to the upper troposphere near the Baltic sea several hours later (Boettcher et al., 2021).

The NAWDEX campaign involved coordinated flights from four research aircraft (the German HALO, the DLR Falcon, the French SAFIRE Falcon 20, and the British FAAM BAe 146). The first three of these aircraft were based in Keflavik (Iceland) for the campaign period and the British aircraft was again based at Cranfield. The flight planning exploited the different flight capabilities and instrumentation of the aircraft. The aim was to deploy the HALO aircraft, with its extended range and ability to fly at up to 15 km altitude (above commercial aircraft routes), to observe moisture transport and diabatic processes in weather systems upstream of Iceland that impact the midlatitude waveguide. The two Falcon aircraft, with their shorter range and slightly lower maximum flight altitude, observed the approaching cyclones and evolving jet streams close to Iceland. The FAAM aircraft was deployed to observe those parts of the cyclones that came within range. The instrumentation on the aircraft varied with a notable campaign strength being the range of remote sensing instrumentation, which included a water vapour lidar, an airborne radar system, a Doppler wind lidar instrument, microwave radiometers, and cloud and aerosol spectrometers. Dropsondes were also available on the HALO, SAFIRE Falcon and FAAM aircraft. A total of 47 flights took place spanning 13 IOPs.

In the remainder of this section we focus on outcomes from DIAMET and NAWDEX that led to new insights into the role of diabatic processes on the dynamics of cyclones and their substructures, ignoring the many other important insights that

were obtained. The development of PV structures within several cyclones observed during DIAMET and T-NAWDEX has already been described in Sect. 5.4.1 (Chagnon and Gray, 2015; Martínez-Alvarado et al., 2016a) because they were analysed using novel diagnostics. Evidence of predictability barriers linked to diabatic processes (Sánchez et al., 2020), which was evaluated during the NAWDEX period, was also mentioned there. Related to this study, Binder et al. (2021) showed that for the NAWDEX upper-level PV cutoff called Sanchez an underestimation of the moisture in the WCB inflow led to forecast errors in the low- and upper-level circulation that caused an error in the location of a downstream heavy precipitation event. The impact of the latent heating and cooling associated with ice microphysical processes on cyclone development and structure was explored by performing convection-permitting model sensitivity tests using three different microphysics parametrisation schemes for the two summer DIAMET cyclones (Dearden et al., 2016). The simulations were found to be most sensitive to ice depositional growth, which affected the depth and position of the cyclones as well as the frontal precipitation structure. Mazoyer et al. (2021) performed a related study contrasting the effect of two different microphysics schemes on the WCB and ridge building of one of the NAWDEX cyclones (also using a convection-permitting model configuration). The scheme leading to the greatest heating rates in the WCB was found to lead to generally more rapid growth of the upper-level ridge. The two campaigns also led to new insights into two specific diabatically generated mesoscale features in cyclones: sting jets and embedded convection. These are discussed in the following two subsections.

### 5.6.1 Sting jets

A particular highlight of the DIAMET campaign was observations made over two flights of the intense windstorm Friedhelm, which caused widespread disruption and damage as it crossed Scotland on 8 December 2011. Measurements of the banding and low-level wind structure of the cloud located near the tip of the bent-back front in Friedhelm showed that these features were linked with weaker horizontal wind speeds within the cloud bands than in the clear interleaved regions (Vaughan et al., 2015). Cloud banding in this region of intense cyclones with a structure resembling the Shapiro-Keyser conceptual model (with a T-bone frontal structure and a bent-back front) is a "smoking gun" for the presence of a sting jet, defined in the review paper by Clark and Gray (2018) "as a coherent air flow that descends from mid-levels inside the cloud head into the frontal-fracture region of a Shapiro–Keyser cyclone over a period of a few hours leading to a distinct region of near-surface stronger winds". The transient nature of the sting jet thus contrasts with the typically much longer-lived wind jets associated with the cold and warm conveyor belts [as illustrated in the windstorm conceptual schematic in Hewson and Neu (2015), their Fig. 1]. The name "sting jet" evolved from the feature being referred to as "the sting at the end of the tail" by Browning (2004) in their meso-analysis of the winds associated with the U.K. Great October storm of 1987; the term "sting jet" is first used in the final sentence of that paper. The terminology evolved from that used in Grønås (1995). The evolving three-dimensional structure of this storm and its associated sting jet was then investigated using a model hindcast together with Lagrangian trajectory analysis by Clark et al. (2005). This study showed that the sting jet is distinct from the jet associated with the cold conveyor belt, and developed the conceptual model for the evolution of sting jet cyclones and the associated low-level wind jets (Fig. 19a). Chemical tracer measurements from the DIAMET flights proved that the cold conveyor belt and sting jet are distinct airstreams even when the associated low-level wind maxima cannot be spatially distinguished (Martínez-Alvarado et al., 2014a), providing observational

support for the findings of Clark et al. (2005). Figure 19b reveals that airstreams O1 and O2 have distinct behaviours and are associated with distinct air masses: O1, circling around the cyclone with intermediate observed carbon monoxide mixing ratios, was linked to the cold conveyor belt whereas O2, originating from the southwest, experiencing a cusp and with much higher mixing ratios, was linked to the sting jet. Martínez-Alvarado et al. (2014a) also pointed out similarities in the structure of the strong wind region in cyclone Friedhelm with that in two much older aircraft-observed cyclones with the same Shapiro-Keyser structure: a storm from 16 March 1987 that occurred during the Alaskan storms programme (Shapiro and Keyser, 1990) and the ERICA IOP4 storm (Wakimoto et al., 1992; Neiman and Shapiro, 1993; Neiman et al., 1993, see also Sect. 4.1.2). In particular, radar reflectivity from the ERICA case (Neiman et al., 1993, their Fig. 19b) revealed cloud banding at the cloud head tip.

The observational and model evidence for the existence of sting jets in extratropical cyclones together with a synthesis of the knowledge on the mechanisms leading to their formation and maintenance, interaction with the boundary layer and climatology are reviewed in Clark and Gray (2018) and so only a very brief summary is given here. For a longer, but still short, introduction to sting jets see Schultz and Browning (2017). Sting jets have been identified in several of the most damaging northern European windstorms since they were first identified in Browning (2004). Although there is no known physical reason why sting jet cyclones should not exist in other midlatitude regions, at the time of writing only one published case occurring elsewhere of an analysed sting jet exists, for a cyclone originating in the Mediterranean (Brâncuş et al., 2019). Strong winds commonly occur in the cold sector of an intense cyclone and without detailed case analysis it is not possible to distinguish between the sting jet, cold conveyor belt jet or other causes (such as localised strong convective downdrafts). The sting jet initially lies above the cold conveyor belt jet as it descends, but, in some cases at least, descends to reach the top of boundary layer ahead of the cold conveyor belt where boundary layer processes can lead to air with large momentum reaching the surface (as demonstrated by the idealised modelling simulations of Rivière et al., 2020). Analysis of cyclone case studies (Gray et al., 2011; Volonté et al., 2018; Eisenstein et al., 2020) and idealised model simulations (Volonté et al., 2020) has led to the development of a conceptual model of sting jet evolution, revealing the role of mesoscale instabilities, including CSI, in their generation, descent and acceleration. However, to quote Clark and Gray (2018), "it seems likely that a continuum of behaviour occurs, from balanced descent partly associated with frontolysis in the frontal-fracture region, through horizontally smaller-scale and stronger frontolytic descent associated with weak stability to slantwise convective downdraughts, to multiple slantwise convective downdraughts associated with the release of CSI and even, possibly, SI" (where SI is symmetric instability, see also their Fig. 8 which illustrates the locations where these mechanisms act relative to the cyclone).

### 5.6.2 Embedded convection and negative PV bands

A key outcome of NAWDEX was evidence of the presence and importance of coherent and organised embedded convection within the WCBs of three cyclones (named Vladiana, Stalactite and Sanchez) from combined analysis of observations and convection-permitting model output (Oertel et al., 2019, 2020, 2021; Blanchard et al., 2020, 2021). An important tool in this analysis (and used in all five of these papers) is online Lagrangian trajectories. By being calculated online, i.e., evaluated within the model using wind fields at every model time step rather than afterwards from model output with lower temporal resolution (Miltenberger et al., 2013), these trajectories are able to represent the rapid and transient convective ascent simulated

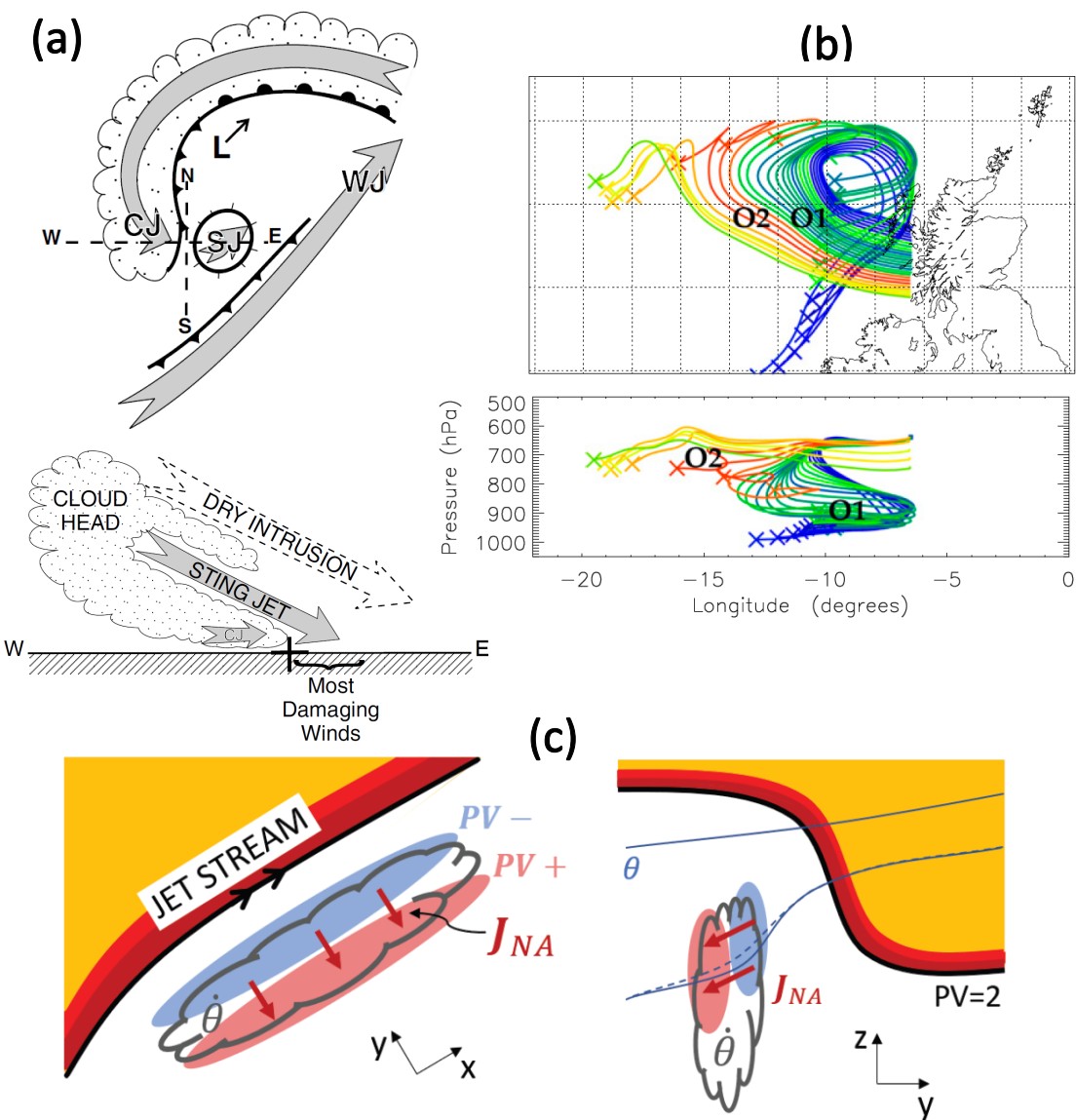

**Figure 19.** Mesoscale features. **(a)** from Clark et al. (2005, their Fig. 17b and 18a), conceptual model of near-surface flows during the frontal fracture stage of an extratropical cyclone containing a sting jet in (left) plan view and (right) west-east cross section; **(b)** from Martínez-Alvarado et al. (2014a, their Fig. 7b), back trajectories (1.125 days long), shown in two projections, from points along southwards heading aircraft track off the west coast of Scotland coloured by observed carbon monoxide at the start points (blue lowest values) with O1 and O2 identified as distinct air masses; and **(c)** from Harvey et al. (2020, their Fig. 5), schematic showing the orientation (red arrows) of the non-advective PV flux $\mathbf{J}_N$ (see Eq. 7 in Sect. 2.1, here denoted as $\mathbf{J}_{NA}$) in a typical WCB situation in (left) plan and (right) cross-section views. All figures used with permission: (a,c) from Wiley, and (b) from the American Meteorological Society.

by convection-permitting models. Rasp et al. (2016) was the first study to use online trajectories to distinguish between the slantwise ascent within the WCB and the more rapid ascent associated with embedded convection. They noted that embedded convection within a slowly upgliding WCB was conceptualised by Neiman et al. (1993) as the elevator-escalator model (Sect. 4.1.2). Despite the different character of the ascent, the net PV change between the inflow and outflow levels was found to be close to zero for both types of ascent in all three cyclone cases simulated.

Of particular relevance for this review article, analysis of the NAWDEX cyclone cases revealed that bands of absolute negative PV forming in the anticyclonic outflow region of the WCB in connection-permitting model simulations were generated by coherent and organised embedded convection, as reported for cyclones Vladiana (Oertel et al., 2020; Bukenberger et al., 2023), Stalactite (Blanchard et al., 2020, 2021), and Sanchez (Oertel et al., 2021). Analysis of Vladiana showed that at low-levels the convection led to enhanced positive PV anomalies and associated cyclonic circulation, whereas at upper-levels mesoscale horizontal PV dipoles (Fig. 2e,f) formed with negative PV values in one of the poles. Elongated persistent (lasting for several hours) PV dipole bands formed from the downstream amalgamation of these mesoscale dipoles. These bands were associated with large-scale circulation anomalies and capable of strengthening the cross-tropopause PV gradient and thus locally enhancing the upper-level jet, forming a jet streak, when advected towards the upper-level waveguide (see the schematic in Fig. 13 of Oertel et al., 2020). Bukenberger et al. (2023), calculating Lagrangian PV-gradient tendencies along trajectories ending in this jet streak of Vladiana, confirmed the importance of diabatic processes for the jet acceleration. In the Stalactite case, three types of organised banded convection were identified in the WCB region differing in the altitude of the base of the updraft and location relative to the low-level jet associated with the WCB (Blanchard et al., 2020). Of these only the convection originating at mid-levels is associated with coherent PV bands. Sensitivity experiments performed on this case by Blanchard et al. (2021) found that the jet stream is more intense and meandering in a reference simulation than in a sensitivity simulation in which heat exchanges in the cloud microphysical scheme are turned off, consistent with Oertel et al. (2020) and other previous studies that have demonstrated that diabatic processes can strengthen the cross-tropopause PV gradient and thus jet stream (Sect. 5.5). Finally in contrast, although a large influence of intense embedded convection on the cloud and precipitation structure was found for the Sanchez cyclone, comparatively weak upper-tropospheric PV dipoles were generated with the negative PV pole to the east of the convective ascent region, i.e., away from the upper-level jet core (Oertel et al., 2021). The weakness of the PV dipoles was attributed to the lack of a strong upper-level jet and associated vertical wind shear, unlike for the Vladiana and Stalactite cyclones.

Observational evidence, from a cross-jet-streak curtain of aircraft dropsonde measurements from the FAAM BAe146 aircraft, for the existence of negative PV on the anticyclonic (equatorward) side of the upper-tropospheric jet was shown for another NAWDEX case, post-tropical storm Karl, by Harvey et al. (2020). This study also provided the dynamical explanation for negative PV generation due to diabatic processes acting in regions of vertical wind shear. The quasi-horizontal PV dipoles are generated by a non-advective flux of PV, $\mathbf{J}_N$ (Eq. 7, see Sect. 2.1), directed "down the isentropic slope" due to thermal wind balance. This flux transports PV away from the jet core when caused by heating within a WCB, resulting in the observed more negative PV on the anticyclonic jet flank and consequent local enhancement of the jet stream (Fig. 19c). Harvey et al. (2020) demonstrated the evolution to a state with negative PV using an idealised semigeostrophic model and PV tracers in the MetUM

and showed that the negative PV bands are caused by a combination of the convection and large-scale rain scheme, consistent with heating in the WCB, in a model simulation of Karl (see Sect. 5.4.1 for an explanation of PV tracers). Evidence from both airborne in-situ measurements and model simulations was presented for multiple banding of the PV dipoles, which was hypothesised to arise as a consequence of the release of CSI: the instability release generates banded PV anomalies that then lead to negative PV bands due to the non-advective PV flux. As discussed by Harvey et al. (2020), the presence of negative PV bands can have consequences for downstream forecast evolution and aviation. Forecast errors can arise from model errors in jet streak formation and aviation can be impacted by both the influence of the PV bands on the jet stream, which affects flight routing, and clear air turbulence, which can occur in regions of negative absolute vorticity (the generated negative PV is associated with negative absolute vorticity rather than negative static stability).

### 5.6.3 Summary

Two major recent field campaigns have led to new knowledge of diabatically generated mesoscale structures within North Atlantic extratropical cyclones by confronting high resolution (particularly convection-permitting) models with observations from advanced instrumentation. Here we have focused on two specific diabatically generated features: the transient mesoscale descending sting jet and embedded convection within frontal zones. While the existence of frontal embedded convection is hardly a new discovery, the identification and conceptual model of the sting jet has been entirely developed within this century. For embedded convection, the field campaigns have led to new knowledge about the dynamically important interaction of embedded convection with the upper-tropospheric jet through the development of horizontal mesoscale PV dipoles. Lagrangian trajectory analysis and PV analysis have been fundamental to the advances made for both phenomena.

### 5.7 Linkage to climate change research

In 1990, the IPCC published their first scientific assessment report about climate change (Folland et al., 1990), and since then questions about how anthropogenic greenhouse gas emissions influence the Earth's climate, about forced trends vs. internal variability, about interactions and feedback processes in the coupled atmosphere-ocean-ice-land system, and eventually about consequences of climate change and effective mitigation measures have been at the heart of atmospheric sciences. The theme of this review article is affected by climate change in two essential ways: (1) global warming is not uniform, it is particularly strong over land, in the Arctic, and has a complex vertical structure, which leads to a reduced horizontal temperature gradient in the lower troposphere and modified static stability, and (2) according to the Clausius-Clapeyron relationship, every degree of warming leads to a 6–7% increase in precipitable water (assuming an approximately constant effective relative humidity[26], Schneider et al., 2010), which has the potential to further enhance the role of diabatic processes discussed in this article. Folland et al. (1990) already mentioned that there is evidence for an increase of global water vapour in the 20[th] century. In this section, we summarise the current understanding of how climate change influences latent heating in large-scale extratropical weather systems and, in turn, their dynamics. Thereby, we again focus on extratropical cyclones, WCBs, and Rossby wave

---

[26]Effective relative humidity denotes a vertical average of relative humidity weighted by the saturation vapour pressure, i.e., weighted toward the lower troposphere.

dynamics including the formation of blocks. Questions about how climate change and diabatic processes affect the extratropical circulation in general and specifically the location and intensity of the storm tracks extend beyond the scope of this review.

Instead, we summarise some general aspects in Sect. 5.7.1 and, for more details, refer the interested reader to overview articles on this broad and complex theme (e.g., Chang et al., 2002; Shaw et al., 2016; Chang, 2017; Chemke et al., 2022; Priestley and Catto, 2022). In the following, we first provide a selection of general considerations about climate change, water vapour, and weather system characteristics (Sect. 5.7.1), and then summarise relevant idealised (Sect. 5.7.2) and GCM (Sect. 5.7.3) modelling studies about climate change effects on extratropical weather systems.

**5.7.1  General considerations**

In their review about water vapour and its active role in shaping the global circulation of the atmosphere, Schneider et al. (2010) summarised the theoretical basis for understanding the role of water vapour, its phase changes, and the associated latent heating and cooling for the location and intensity of the extratropical storm tracks (and the tropical Hadley circulation). They emphasised that "the unclear role of water vapor in extratropical dynamics in the present climate and its changed importance

in colder or warmer climates are principal challenges in understanding extratropical circulations and their responses to climate changes". From arguments based on fundamental energetic constraints, building on earlier work by Boer (1993) and Held and Soden (2000), it follows that changes in near-surface relative humidity must be relatively small. Importantly, however, the energetics arguments cannot constrain free tropospheric relative humidity changes, which therefore can be larger than those near the surface. Considering the mean available potential energy, Lorenz (1979) estimated that this quantity is roughly

30% greater when including the potential release of latent heat in condensation of water vapour, and one might expect this to contribute to intensified storm tracks in warmer climates. However, idealised GCM simulations for a broad range of climates (O'Gorman and Schneider, 2008) revealed a nonlinear behaviour with maximum near-surface eddy kinetic energy—which serves here as a general measure of storm track intensity—in near-present-day climate conditions and smaller values towards much warmer and colder climates. The authors mentioned that "the 30% difference between mean dry and moist available

potential energies that Lorenz (1979) found for the present climate largely arises owing to water vapour in tropical low-level regions, which may not be important for midlatitude eddies".

Although based on idealised simulations, the above result is very important as it implies that the hypothesis that global warming, leading to increased tropospheric humidity and latent heating, generally increases the frequency and/or intensity of extratropical cyclones is too simplistic. It is essential to investigate how different types of weather systems in different

regions (Sect. 5.3) are affected by climate change. This is a formidable challenge as it includes studying climate change effects on several dimensions of weather system characteristics: their frequency, intensity, tracks, and structure. Eventually, it is the interplay of these aspects that determines how climate change effects on surface weather (e.g., precipitation) are modulated by weather system dynamics. For instance, cyclone-related precipitation (i.e., the total contribution of cyclones to the climatological precipitation) might increase due to global warming because cyclones become more frequent, because

they become more intense (i.e., produce more precipitation in a warmer environment, referred to as the "Clausius-Clapeyron effect"), because they become more stationary, and/or because they develop, e.g., more intense frontal structures. This challenge

was already mentioned in the 3rd IPCC report (Folland et al., 2001): "A more fundamental question is whether we would expect more or fewer extratropical cyclones with increased warming. As pointed out by Simmonds and Keay (2000), increased moisture should enhance extratropical cyclones, but Zhang and Wang (1997) suggested that cyclones transport energy more efficiently in a moister atmosphere, therefore requiring fewer extratropical cyclones". It is therefore important to understand the effects of climate change on single cyclones (i.e., their structure and intensity) as well as on their statistical ensemble (i.e., including their frequency). Consistent with the scope of this review, the following subsections focus on the first of these questions and thereby in particular on studies about how climate change affects the role of diabatic processes for the structure and intensity of cyclones (and other extratropical weather systems). For a more general overview about climate change effects on extratropical cyclones, the reader is referred to the review by Catto et al. (2019). We start with idealised numerical model studies where climate change effects were imposed by modified initial and boundary conditions, and then consider examples from the growing body of literature on diagnosing the characteristics of extratropical weather systems in GCM simulations, with a particular focus on studies also investigating their PV structure.

### 5.7.2 Idealised studies

Pioneering idealised numerical modelling studies about how climate change may affect baroclinic eddies were performed in the 1990s, as briefly summarised in Sect. 4.3.3. Recent studies with a similar motivation and objective, used either idealised channel model or GCM experiments or the semi-idealised pseudo-global warming approach[27], as summarised in the following. Booth et al. (2013) addressed the effect of increased moisture on the development and evolution of cyclones in baroclinic lifecycle experiments in an $f$-plane channel. In addition to the classical approach of using different initial relative humidity, $RH_0$ (varied between 0 and 95%), they also implemented a second approach to increase latent heating by keeping humidity constant but varying an artificial scaling factor inserted into the Clausius-Clapeyron equation. With both approaches they consistently found an increase in the strength of different storm intensity measures (central MSLP minimum, maximum surface winds, extreme and accumulated precipitation rates) with increasing heating. However, considering eddy kinetic energy shows a slightly different picture with a saturation of peak values for $RH_0 \geq 60\%$. This result is consistent with similar simulations by Boutle et al. (2011), who focused on quantifying vertical moisture transport in idealised cyclones.

Considering extratropical cyclones in idealised global aquaplanet simulations of changed climates, Pfahl et al. (2015a) showed that median cyclone intensity (minimum MSLP or maximum low-level relative vorticity), maximum deepening rates, and averaged eddy kinetic energy are largest in simulations with global mean temperature slightly warmer than present-day Earth. However, the relative vorticity of the 10% most intense cyclones continued to increase in simulations with substantially higher temperatures, due to intensified diabatic PV production in the cyclone centre, leading to a more pronounced and vertically more extended lower-tropospheric PV anomaly. The authors concluded that "moist processes may, therefore, lead to the further strengthening of intense cyclones in warmer climates even if cyclones weaken on average", which was further

---

[27]The pseudo-global warming approach denotes a method introduced by Schär et al. (1996), which imposes a thermodynamic climate change signal, quantified from global GCM simulations to the initial and lateral boundaries of a regional model. This approach allows investigating the thermodynamic effects of climate change to a specific meteorological event assuming that the large-scale flow conditions would occur unaltered in the future climate.

corroborated by Büeler and Pfahl (2019) using an area-averaged PV budget diagnostic defined in Büeler and Pfahl (2017) (and discussed in Sect. 5.4). Sinclair et al. (2020) performed aquaplanet simulations for a control and a warmer climate, with SSTs uniformly increased by 4 K. They also found that cyclones in the warmer climate were more diabatically driven, with increased precipitation by up to 50% in the most intense cyclones. Important structural changes of future cyclones included a poleward and downstream shift of the low-level PV anomaly, of mid-tropospheric ascent, and of the precipitation maximum associated with the warm front. In a follow-up study with similar aquaplanet simulations, Sinclair and Catto (2023) investigated potential changes due to 4 K uniform warming of the variability in precipitation structures associated with extratropical cyclones. They found that the major cyclone-related precipitation patterns, identified with a $k$-means clustering algorithm, remain fairly unchanged, and that the relationship between maximum cyclone intensity and precipitation depended strongly on the cyclone type.

In addition to quantifying the increasing skewness of vertical velocities in the extratropics with global warming, O'Gorman et al. (2018) identified the most unstable moist baroclinic waves for a wide range of global mean temperatures with an idealised GCM, including surface Rayleigh drag. A striking result of these simulations is the transition of the most unstable mode from a quasi-periodic wave to an isolated diabatic Rossby vortex at sufficiently high temperatures. A qualitatively similar transition from waves to a vortex-dominated regime with increased latent heating was found by Lapeyre and Held (2004), who used a nonlinear moist quasi-geostrophic two-layer model.

Kirshbaum et al. (2018) performed a systematic study with a classical $f$-plane channel model on the effects of increasing the mean basic state temperature on moist baroclinic instability while keeping relative humidity constant. They emphasised that the mean temperature $T_0$ does not affect dry baroclinic instability, as both the meridional baroclinicity and static stability do not depend strongly on $T_0$, but "midlatitude cyclones may still be regulated by $T_0$ through its indirect control over diabatic processes like latent heat release". For their chosen initial configuration, moist baroclinic waves developed larger eddy kinetic energy than corresponding dry waves, but the enhanced diabatic heating that occurs as $T_0$ is increased (at constant relative humidity) does not further increase eddy kinetic energy. Investigation of the low-level PV production in the different experiments showed that the diabatically produced warm-frontal cyclonic PV anomaly shifts eastward when increasing $T_0$, which feeds back negatively on the parent cyclone even though the diabatic PV production *per se* was intensified. A very important inference from this idealised study is that cyclone intensification eventually depends on the detailed mesoscale structure of diabatic PV production, in line with the study by Binder et al. (2016) whose results from analysing a large set of extratropical cyclones in reanalyses also indicated that the location of the diabatic low-level PV production relative to the cyclone centre is highly relevant for the resulting effect on cyclone intensity. This further implies that climate simulation studies of cyclones and storm tracks must have sufficient resolution to properly represent the formation of mesoscale PV anomalies and its variability between cyclones. As a caveat of the Kirshbaum et al. (2018) study it should be mentioned that they performed idealised simulations with only one specific basic state, which may not be representative for the climatology of cyclones as it is well known from idealised studies that frontal structures and associated latent heating and low-level PV production depend strongly on the basic state flow.

The study by Tierney et al. (2018) investigated the effect of increasing both $T_0$ (and the associated water vapour content) and baroclinicity, again using $f$-plane channel model simulations. Consistent with the studies mentioned above, they found that cyclone intensity and integrated eddy kinetic energy increase in higher baroclinicity environments, but both measures show a non-monotonic behaviour when increasing water vapour content above a threshold value. Investigation of the cyclone size, latent heating profile, and low-level diabatic PV production revealed that, with increased $T_0$, cyclones become smaller, latent heating extends to higher levels, precipitation becomes more convective, and, similar to Kirshbaum et al. (2018), the low-level PV is reduced near the cyclone centre.

We also discuss in this section a semi-idealised study that uses the pseudo-global warming approach to assess the impact of climate change on a previously identified moist dynamical feedback mechanism in a specific south-central U.S. flood event. This feedback mechanism involved a low-level jet along the cold front of an extratropical cyclone, poleward moisture flux and flux convergence, condensational heating, and diabatic low-level PV production, which in turn intensifies the low-level jet (Lackmann, 2002). Against the initial hypothesis that global warming and increased specific humidity would strengthen this feedback mechanism, the pseudo-global warming simulation did not indicate a strengthening of the low-level jet; however, precipitation still increased because of intensified convective updrafts (Lackmann, 2013).

### 5.7.3 GCM studies

Simulations with GCMs provide a new type of dataset to study diabatic processes in weather systems (recall that the studies discussed so far relied on observations, (re)analyses, idealised simulations, and real case simulations with weather prediction models). Because of the long duration of GCM simulations they are typically run at low resolution and it is not clear, a priori, whether diabatic processes, which are strongly influenced by mesoscale dynamics, are represented with high enough accuracy to quantify their effects on weather system dynamics. For instance, it is well known that low-resolution GCMs underestimate the intensity of tropical cyclones (e.g., Camargo and Wing, 2016) and it is therefore important to investigate whether they can realistically represent extratropical cyclones, moist airstreams, and the formation of blocks, and how strongly the accuracy of this representation depends on model resolution.

We first consider early compositing studies of extratropical cyclones and their cloud and precipitation structures that used data from different climate models with different grid spacings (Bauer and Del Genio, 2006; Bengtsson et al., 2009; Catto et al., 2010). Comparing composites for the 100 most intense North Atlantic cyclones in U.K. Met Office climate model simulations with 100 km grid spacing and ERA-40 reanalyses, Catto et al. (2010) concluded that the GCM's "ability to represent the key features of extratropical cyclone structure can give confidence in future predictions from this model". Similarly positive conclusions were drawn by Bengtsson et al. (2009), who investigated the structure of the 100 most intense cyclones in ECHAM5 simulations with a spectral resolution corresponding to 63 km grid spacing. As a caveat it is mentioned that the 100 most intense cyclones do not represent the entire spectrum of cyclones, and less strongly developing and smaller-scale cyclones might be more difficult to simulate with grid spacings on the order of 100 km. Despite the generally good agreement, Catto et al. (2010) identified weaker ascent and descent in the regions of the WCB and dry intrusion, respectively, in the GCM compared to reanalyses. Bauer and Del Genio (2006), investigating GCM simulations with a much larger grid spacing of 450 km, em-

phasised model deficiencies in capturing mesoscale frontal structures, with implications for the simulation of cyclones, which were generally weaker, slower-moving, and shallower compared to reanalyses.

Motivated by the hypothesis that GCMs with grid spacings of 100 km or more systematically underrepresent diabatic processes and with the intention of bridging the gap between climate modelling and synoptic dynamics, Willison et al. (2013) compared cyclones in regional simulations over the North Atlantic for 10 winters with grid spacings of 120 vs. 20 km and concluded that the intensity of cyclones is strongly sensitive to horizontal resolution. They found that the positive diabatic feedback on cyclone intensity is significantly enhanced when simulated with the finer grid spacing. This feedback included stronger vertical velocities along fronts leading to more condensation and stronger cyclonic PV in the lower troposphere, which in turn strengthened the low-level jet and moisture transport towards the cyclone centre. In a follow-up study, Willison et al. (2015) used the pseudo-global warming approach to quantify the effect of global warming on North Atlantic cyclones in their regional simulations and to test the sensitivity to resolution. Their most important finding was that the response to global warming (measured in terms of, e.g., precipitation, Eady growth rate, and eddy kinetic energy averaged in the region of the North Atlantic storm track) is amplified in the simulation with 20 km grid spacing. However, in this study no analysis was made of the structure and intensification of individual cyclones. Ma et al. (2017) conducted similar pairs of simulations with a regional climate model with 162 and 27 km grid spacing, respectively, for the entire North Pacific. They also found an important sensitivity to model resolution, which they attributed primarily to the fact that the coarser resolution lacks mesoscale SST structures in the western North Pacific, which, in the simulation with higher resolution, led to stronger cyclone intensification near the Kuroshio Extension and modified downstream development. They concluded that "it is only when the model has sufficient resolution to resolve small-scale diabatic heating that the full effect of mesoscale SST forcing on the storm track can be correctly simulated". Still higher resolution convection-permitting model simulations have been found to produce more increase in intense precipitation in the warm sectors of cyclones with climate change compared to that in their parent regional climate model (Berthou et al., 2022).

These results from a selection of studies about cyclone validation in climate models at varying resolutions can be summarised as follows: (i) simulations with grid spacings larger than 100 km most likely have serious deficits in representing frontal structures, which leads to an underestimation of vertical motion, diabatic processes, and cyclone intensity; (ii) with grid spacings in the range of 50–100 km, the composite structure of intense cyclones in climate simulations compares favourably with reanalyses at similar resolution; however (iii), results do not converge at that scale and simulations with grid spacings of about 20 km, and smaller, tend to show a further increase in the intensity of diabatic processes and their effects on synoptic-scale dynamics. It will therefore be extremely interesting to investigate cyclone properties and storm track dynamics in kilometre-scale simulations without parameterised convection in the near future.

The following paragraphs summarise findings from GCM and regional climate simulations about effects of global warming on the moist dynamics of extratropical weather systems. A consistent finding from studies with different GCMs is the intensification of precipitation associated with extratropical cyclones, as first quantified by Watterson (2006) in coarse simulations with grid spacings of about 400 and 200 km. Bengtsson et al. (2009) found an increase of cumulative precipitation along the tracks of cyclones by 11% from the end of the 20[th] to the end of the 21[st] century, which, in their simulation was about twice the increase

in global mean precipitation. Despite this strong thermodynamic effect, other aspects of the cyclone climatology changed only weakly. They found a small reduction in the number of intense cyclones, in qualitative agreement with the lower-resolution simulations by Watterson (2006), but no significant changes in the extremes of wind and vorticity in both hemispheres. Catto et al. (2011), who also found a slight decrease in the number of intense cyclones, argued that this result might imply that the effect of the reduced baroclinicity, which acts to reduce cyclone intensity, overcompensates the intensifying effect of the extra latent heat release in a warmer climate. The authors also indicated an important technical caveat: The identification of the most intense or the most strongly intensifying cyclones relies on a cyclone identification and tracking algorithm, which typically differs between studies. These algorithms all include subjective criteria (e.g., a minimum lifetime threshold or a maximum propagation speed), which influence the resulting cyclone climatologies, as discussed in detail in the cyclone tracking inter-comparison by Neu et al. (2013). However, this study also revealed that the most intense cyclones are typically those that are most consistently identified by the different algorithms. Using the pseudo-global warming approach and a regional model with 36 km grid spacing, Marciano et al. (2015) studied the thermodynamic effects of climate change on ten intense U.S. east coast cyclones. In their pseudo-global warming simulations, precipitation increased as well as cyclone intensity and 10-m winds, in response to strengthened diabatic PV production near the cyclone centre. These results support the hypothesis that enhanced latent heat release leads to an increase in future cyclone intensity; however, one must keep in mind that with this modelling approach, potential compensating effects by reduced baroclinicity are excluded by design. Applying a simplified pseudo-global warming approach (by adopting a uniform warming of 4 K) to 12 moderate to intense Northern Hemisphere cyclones, Büeler and Pfahl (2019) found enhanced latent heating in the warmer and moister simulations, which amplified low-level diabatic PV generation, leading to a stronger intensification of the cyclones. An important finding of their analyses was that observed, strongly diabatically driven cyclones have the potential to be substantially more devastating if occurring in a future warmer climate.

Related to potential future changes of cyclone intensity is the question how increased latent heating due to global warming might affect (extreme) surface winds in extratropical cyclones. While early simulations found no important changes (see above), Priestley and Catto (2022) reported a future strengthening of winds throughout the troposphere in winter cyclones (and more regionally variable results in summer), according to CMIP6 (coupled model intercomparison project, phase 6) simulations. The largest changes in wind speeds are projected to occur in the warm sector of cyclones. Intensified latent heating was mentioned as an important factor contributing to this increase. Dolores-Tesillos et al. (2022) found a similar future increase of surface winds in North Atlantic cyclones in CESM1 large ensemble simulations, in particular southeast of the centre of strong cyclones. Piecewise PV inversion revealed that both an amplified diabatically produced low-level PV anomaly and a dipole change in upper-level PV anomalies contribute to this wind intensification. These results indicate that a complex interaction of enhanced diabatic heating and altered non-linear upper-tropospheric wave dynamics shape future changes in near-surface winds in North Atlantic cyclones. Studies specifically considering the potential changes to sting jets in the future climate have found an increase in the likelihood that cyclones will contain sting jets and consequent increase in extreme surface winds (Martínez-Alvarado et al., 2018; Manning et al., 2021).

Other studies addressed the role of diabatic processes to understand the climate change effect on cyclone propagation (instead of cyclone intensity). Tamarin-Brodsky and Kaspi (2017) showed that, in CMIP5 models, the average poleward displacement of individual cyclone tracks increases under global warming. As a key process they diagnosed an enhancement of the upper-level poleward velocity in the region of the surface cyclones, induced by the zonal trough-ridge dipole, which contributes to their poleward propagation (in agreement with the quasi-geostrophic argumentation by Oruba et al., 2013). They regarded the role of latent heating as a separate, secondary factor that leads to increased vertical motion and vortex stretching, which contributes to the eastward and poleward propagation of the low-level cyclones. They found that also this contribution intensifies with global warming. These results from a comprehensive diagnostic analysis of cyclones in CMIP5 simulations qualitatively agree with findings from detailed process studies about how moist processes affect the track of individual extratropical cyclones. Tamarin-Brodsky and Kaspi (2017) showed that the poleward propagation of cyclones is larger for more intense cyclones, due to a combination of amplified latent heat release and more intense upper-level positive PV anomalies associated with intense cyclones (e.g., Čampa and Wernli, 2012). Coronel et al. (2015), however, emphasised that diabatic outflows amplify upper-level anticyclones (Sect. 5.5), which strengthens the upper-level zonally oriented PV dipole and induces poleward flow at low levels, leading to faster poleward propagation of the cyclone. These studies about diabatic effects on cyclone propagation are particularly relevant in light of the too zonal bias of the North Atlantic storm track since several generations of GCM simulations (Priestley et al., 2020). One hypothesis is that still substantially higher resolution is required to capture the full intensity of diabatic processes near the Gulf Stream SST front, which would lead to an increased poleward cyclone propagation. This idea is supported by recent simulations with an aquaplanet model (Schemm, 2023), which was set up with a nest down to 5 km grid spacing over an idealised Gulf Stream front. Compared to a simulation without the high-resolution nest, the results showed increased precipitation, more explosively deepening cyclones, and an overall stronger poleward deflection of the storm track.

Another long-standing bias in GCM simulations is the underestimated blocking frequency, particularly in winter over the North Atlantic (Woollings et al., 2018; Davini and D'Andrea, 2020). This bias hampers the interpretation of climate change projections of blocking, which according to the vast majority of CMIP6 models, show a future frequency decrease in most regions in winter and summer (Davini and D'Andrea, 2020). Given the recently established role of upstream latent heating for the formation and maintenance of blocks (Sect. 5.5), a better representation of diabatic processes might be required to reduce this GCM bias (Woollings et al., 2018). Steinfeld et al. (2022) performed a detailed Lagrangian analysis of the formation of negative PV anomalies in blocks in large-ensemble simulations with CESM1 in present-day and future climate conditions under the RCP8.5 scenario. Their results showed (i) an excellent agreement of the distribution of latent heating along the backward trajectories from blocks in the present-day simulation with ERA-Interim, and (ii) an increase of latent heating along such trajectories in the future climate, in particular of very intense accumulated latent heating exceeding 20 K, which is a useful threshold to identify WCBs. Also, blocks in the future climate become larger and have more negative PV anomalies. An important caveat is that such a process analysis can only be made when and where blocks form in the GCM, i.e., such a study cannot help with the identification of the reasons why blocking frequencies are too low. The hypothesis is still valid

that generally too weak diabatic processes might contribute to the underrepresentation of blocks in CESM1 and other climate models.

Last but not least, we briefly address the question how climate change affects WCBs, the airstreams in the extratropics characterised by very large integrated latent heating. Only recently, output became available from CESM1 large ensemble simulations with temporal and vertical resolution that allowed for the computation of air parcel trajectories and the identification of WCBs. Joos et al. (2023) showed that, in the present-day CESM1 simulations, WCBs are represented reasonably well in terms of location and occurrence frequency compared to ERA-Interim. In the future climate (RCP8.5 scenario), important changes occur in key characteristics of WCBs: inflow specific humidity increases by more than 20%, leading to (i) an increase in WCB-related precipitation by 7–28%, depending on region and season, (ii) a strong increase in diabatic heating, and (iii) an about 10 K higher isentropic outflow level, which might increase the interaction of WCB outflows with the PV waveguide. The companion study by Binder et al. (2023) investigated the effect of these changes in WCB characteristics on cyclone dynamics and found that WCB-related low-level PV production will be even more important for explosive cyclone intensification towards the end of the 21$^{st}$ century than in the present-day climate.

### 5.7.4 Summary

It is a formidable challenge to understand how anthropogenic climate change, leading to a future reduction of baroclinicity and increase of specific humidity, will affect the frequency, intensity, and location of extratropical weather systems. This was concisely formulated by Catto et al. (2019) as follows: "While precipitation intensity will most likely increase, along with associated increased latent heating, it is unclear to what extent and for which particular climate conditions this will feedback to increase the intensity of cyclones". In light of the discussion in Sect. 5.3, one could add "and how this feedback differs for different types of extratropical cyclones". As summarised in this section, a combination of theoretical considerations, idealised model studies, and process-based investigations of full-complexity climate model simulations is needed to obtain a robust picture of the projected changes and their implications for surface weather, weather extremes, and impacts. Most likely, the answer is strongly regionally dependent; for instance, weather systems might become less frequent but more intense in some parts of the world and more frequent but without major changes of intensity in others. We claim that systematically applying concepts and diagnostics established in weather research over the last decades, as documented in this review, to high-resolution climate simulation output will strongly increase our understanding of climate model capabilities (and shortcomings), and of how and why climate change will affect future weather.

## 6 Synthesis, implications, and outlook

### 6.1 Synthesis

This extensive historical review documents how research about the role of diabatic processes in extratropical weather systems evolved over more than a century; it highlights early ideas, pioneering studies around the middle of the last century, the

3305 "boom period" after 1980, and important innovations in the last two decades. Today, it is firmly established that diabatic processes are in general highly important for the structure and evolution of extratropical weather systems, with large case-to-case variability. The intensification of most high-impact extratropical cyclones cannot be understood without considering effects of latent heating, and diabatic effects are, nowadays, also considered as essential for understanding upper-level Rossby wave dynamics with consequent implications for downstream development and forecast predictability. This implies that the

3310 science of extratropical dynamics has reached a new level where the interplay of dry dynamics with effects of latent heating in clouds and other diabatic processes are considered as central to the field. This does by no means reduce the relevance of dry dynamics, which still constitutes the backbone for understanding large-scale atmospheric flows, but it adds complexity and highlights the need for increased interaction of atmospheric dynamics with neighbouring scientific fields like cloud physics, ocean-atmosphere interactions, radiation, and climate dynamics. As shown in this paper, the pathway to this new level of

3315 understanding was far from linear, and—what we regard as particularly fascinating—it emerged thanks to the integration of seemingly independent elements. In our view, the following factors were essential for the documented progress in this field:

– A few poorly-predicted high-impact storms between 1978 and 1999—we have referred to them as "catalyst cyclones" and portrayed them in some detail in the Supplementary—each time clearly evidenced a gap between the scientific understanding and the observed phenomenon. Very specifically, most NWP models at the time failed to predict these
3320 events, leaving people without warning when such alerting information would have been most needed. Most likely, the increased public attention and the articulated societal expectations after these events was helpful for funding dedicated research programs and costly field experiments to intensify research in this field and, eventually, improve predictions of explosively deepening extratropical cyclones and their associated weather. It would be too far stretched to claim that the catalyst cyclones directly triggered the planning of field experiments, but they clearly helped in engaging the
3325 research community to perform CASP and ERICA after the President's Day and QE II storms, FASTEX after the U.K. Great October storm (and DIAMET which followed the later discovery of a sting jet in this storm), and NAWDEX after Lothar.

– These field campaigns with many non-standard observations from aircraft and ships played an essential role for at least three reasons: (1) mainly thanks to sophisticated measurement devices developed for research aircraft, they provided
novel and detailed observational insight into (diabatic) processes that are essential for understanding the dynamics of extratropical weather systems; (2) given the advent of analysis data and numerical models with enhanced resolution, these observations were central for a critical validation of model products, which in general improved confidence in weather system simulations (e.g., observational analyses "confirmed" low-level PV anomalies produced by numerical models) and supported the use of numerical models as important research tools; and (3) the campaigns helped the
community to define and focus on a few specific research questions, and, after the campaign, to study a few particularly well observed weather events, the so-called "golden cases", in great detail. This happened sometimes in collaboration and sometimes in separate teams with complementary ideas and research methods. As shown for the catalyst cyclones and some golden cases from campaigns, series of studies about the same event, sometimes initially with contrasting and

conflicting results, and the subsequent scientific discussion can be extremely helpful for critically identifying important
gaps in scientific understanding. It is evident that the availability of satellite observations after the start of the satellite-era
in 1979, including time-lapse videos from geostationary satellites, which actually show how the WCB develops, played
an essential role for illustrating and investigating clouds and their role in weather systems, as well as for the design and
implementation of field campaigns.

– Numerical models contributed essentially to the investigation of diabatic processes, as they can provide four-dimensional
fields with high spatial and temporal resolution (e.g., of clouds, latent heating, and PV), which is essential for studying the
complex processes at play. In addition, they allow the performance of "experiments", e.g., by simulating a weather system
with a specific (diabatic) process turned off—a research approach that became popular with the advent of comparatively
high-resolution limited-area models in the 1980s. Clearly, such studies rely on the quality of numerical models, which
is not perfect. The representation of dry dynamics is affected by numerical shortcomings, and diabatic processes (cloud
microphysics, surface fluxes, radiation) rely essentially on so-called parametrisations, which are based on approximations
and simplifications of the observed complexity of the involved processes. Therefore, as emphasised throughout this
article, the confrontation of model results with observations has been extremely important. The enormous progress in
numerical modelling in the last 50 years, in particular the increases in resolution, went along with a similarly impressive
development of data assimilation, eventually enabling the production of high-quality reanalysis data sets. Whereas field
experiments and numerical model experiments were important for studying the dynamics of a few cyclones in great
detail, reanalyses led to climatological studies and the application of statistical approaches to identify more robust and
representative results, and to document the very high variability among events.

– Last but not least, theoretical concepts and idealised numerical studies contributed with fundamental insight, for instance,
about latent heating effects on baroclinic instability and the development of frontal waves, and helped to put results from
real case studies into perspective. Several papers have included well-designed schematic diagrams that have aided the
understanding and communication of atmospheric processes, airflows and weather features (e.g., those reproduced here
in Figs. 2, 3b, 4b,d,e, 7a, 15a,e, 17b,d, 18d, and 19a,c). We also highlight again the two main theoretical frameworks
emphasised in this review, the PV perspective and the concept of slantwise instability. Although early papers about the
usefulness of PV mainly emphasised the material conservation for adiabatic flows (and of course the invertibility prin-
ciple), it turned out that PV is also extremely helpful for identifying and quantifying the effects of diabatic processes on
the large-scale flow. To this end, sophisticated Eulerian and Lagrangian diagnostics have been developed, which enabled
a strongly improved understanding of how diabatic processes, integrated along the flow, can contribute to the formation
of PV anomalies. Slantwise instability was routinely evaluated in many of the early (pre-2000) cyclone case studies and
its release speculated to be an important factor in rapidly intensifying storms. Consideration of slantwise instability was
also a fundamental component of the design of early idealised cyclone simulations. More recently, slantwise instability
(and other mesoscale instabilities) has gained prominence due to its release being associated with the sting jet found to
lead to damaging surface winds in some intense cyclones.

When writing this historical review, we also considered the theme of gender contributions. In this particular research area, when did the transition occur towards more balanced contributions by female and male scientists? Within the text, we highlighted some of the pioneering contributions by female scientists. The earliest papers by female first authors, referenced in this review, were published just around the turn of the millennium (McMurdie and Katsaros, 1996; Demirtas and Thorpe, 1999; Mallet et al., 1999a; Pomroy and Thorpe, 2000; Gray and Thorpe, 2001; Massacand et al., 2001; Jones et al., 2003). A lot has been changing in the last 20 years, thanks to specific efforts to bring women into natural sciences and due to changes in academic culture. We claim that, in our field, the academic community is on a good pathway towards an equal share of the most innovative research contributions by female and male first authors, respectively. Considering the papers cited in this review and published in the last six years (in 2018 or later), about one third are first authored by women. We regard this as a promising sign that in the future, the full potential of scientific curiosity and capacity can participate in and contribute to the progress in this challenging research field.

## 6.2 Implications and outlook

At the end of this extensive review, what are the main implications for future research in this area? In our view, all the factors beneficial for scientific progress summarised in the previous section still hold, in particular the combination of scientific approaches and the development of sophisticated diagnostics. However, there are important general developments in atmospheric sciences, which offer fascinating possibilities for research in atmospheric dynamics. We would like to mention seven aspects, where the first four are mainly about how general developments in numerical modelling, field experiments and open science likely lead to further progress in understanding moist dynamics, whereas the last three aspects emphasise links to related research areas where moist dynamics has the potential to make important contributions.

1. **km-scale operational NWP.** Operational weather prediction models will reach horizontal resolutions of a few kilometres. Whereas several weather services have run limited area forecast models with grid spacings of 1–4 km for several years, the prospect is that only a few more years of optimised model development are required to reach such resolutions for global weather and climate models (Schneider et al., 2017; Schulthess et al., 2019; Schär et al., 2020). This improvement will allow for better representation of steep topography and the ability to turn off the parametrisation of deep moist convection. In the context of this review article, this implies that, e.g., fast-ascending motion in convective cloud systems and associated diabatic processes will be simulated with the same numerics and cloud microphysics as the more slowly ascending and larger-scale warm conveyor belts. As discussed in Sect. 5.6.2, this change will have direct implications for the diabatic modification of PV in the upper troposphere and, in turn, for the large-scale flow evolution. In parallel with this resolution enhancement are investigations into the sensitivity of diabatic processes to the complexity of the remaining parametrisation schemes (e.g., cloud microphysics, Mazoyer et al., 2023). Additionally, km-scale atmospheric models are now being run, at least in research mode, that are coupled to models for the ocean and waves (e.g., Ricchi et al., 2017; Lewis et al., 2019), providing an evolving and more consistent representation of surface fluxes of heat, moisture and momentum with potential (though as yet mainly un-quantified) benefits for the representation of

diabatic processes in, e.g., extratropical cyclones. We foresee great potential in studying the role of diabatic processes in such km-scale simulations, in particular if they are global (or have large enough domains to encompass the evolving synoptic-scale weather system) such that the potential large-scale effects of the explicit treatment of convection are not "damped" by the boundary conditions. Model intercomparisons for specific weather events will be highly valuable to assess the robustness of how simulated convective-scale effects are projected to larger scales in different models and flow situations.

2. **Ensemble weather forecasts.** A second major evolution in NWP, which was initiated more than 25 years ago (Palmer, 2019), is the ensemble approach. Research about the role of diabatic processes is beginning to profit from the availability of ensemble simulations (e.g., Dacre and Gray, 2013; Sánchez et al., 2020). Two challenges when using ensembles in the context of the theme of this review are that (i) the output from ensembles is huge and typically not archived with full vertical resolution—whereas diagnosing the effects of diabatic processes typically requires detailed knowledge about the vertical structure of clouds and latent heating, and (ii) most of the recently developed diagnostics dedicated to diabatic processes (see Sect. 5.4) are computationally intense and therefore cannot be straightforwardly applied to large ensembles. However, the potential to use ensembles for diagnostic studies is huge—they can be regarded as physically consistent sensitivity experiments (which are much more subtle than the classical dry vs. moist sensitivity experiments or the experiments where initial conditions were modified by PV surgery) that can serve to quantify the systematic effects of, e.g., moisture disturbances on cyclone dynamics. It is also potentially fruitful to consider the opposite direction of research, i.e., using moist dynamical understanding for improving ensemble predictions (see item 5).

3. **Field experiments.** Field experiments will most likely continue playing an important role for this research field, and plans exist, e.g., for a large international airborne campaign over the North Atlantic, with elements that build on NAWDEX but with a stronger emphasis on air-sea interactions in cold air outbreaks and the formation of surface wind gusts and heavy precipitation[28]. Future field experiments will also benefit from advances in observational capability such as airborne wind lidar and Doppler radar. An important challenge for future field experiments is the required transition towards net-zero also for research activities. Research aircraft and ships are large emitters of greenhouse gases, and the need for their effective compensation will most likely make field experiments even more expensive. One consequence is that the research community should, even more than in the past, carefully design and conduct such experiments and analyse the obtained data as comprehensively as possible.

4. **Open science.** The strong move towards open science will likely be very beneficial, as it should facilitate the open access and long-term archiving also of data from field experiments. Also, it will hopefully provide support for new model intercomparison projects, and for the sharing of data (e.g., from high-resolution simulations that can only be performed by a few institutions but analysed by the global community) and of diagnostic tools. Whereas there have been several very positive examples of data sharing and exchange of diagnostic and verification tools in the last decades, some dedicated efforts (e.g., in the framework of EU COST actions) and more standardised approaches between weather

---

[28]The field experiment NAWDIC is currently scheduled in January and February 2026, https://internal.wavestoweather.de/campaign/projects/nawdic/wiki

services and research groups could be highly beneficial. Excellent examples in this direction are the TIGGE archive of global ensemble forecasts (Swinbank et al., 2016), the German Hans-Ertel Centre to foster long-term collaboration between the weather service and universities (Simmer et al., 2016), and the OpenIFS initiative by the ECMWF (Szépszó et al., 2019).

5. **Atmospheric predictability.** As discussed mainly in Sect. 5.5, there are tight links between moist dynamics and atmospheric predictability. Diabatic processes are considered as important sources of model errors (due to limitations of physical parametrisations) and moist weather systems, in particular those associated with deep convection, is where small-scale forecast errors can grow to larger scales (as originally proposed by Zhang et al., 2003). In the last 20 years, several studies investigated the complex processes involved in upscale error growth (e.g. Durran and Gingrich, 2014; Selz and Craig, 2015; Judt, 2018; Leung et al., 2020), the role of mesoscale convective systems for the occasional occurrence of low-accuracy medium-range forecasts (Rodwell et al., 2013) and for ensemble forecast performance (Clarke et al., 2019a, b), and the relative importance of moist baroclinic processes for the amplification of ensemble spread (Baumgart and Riemer, 2019). Also, a recent flow-dependent reliability study indicated that ECMWF ensemble forecasts have too large spread in situations of intense moist baroclinic instability (Rodwell and Wernli, 2023). We see a great potential in further improving our understanding of ensemble forecast performance, and eventually in improving forecast systems, by continuing this line of research, which requires the combined expertise in weather system dynamics, numerical modelling, and theoretical predictability concepts.

6. **Extreme weather events.** A currently highly relevant theme in atmospheric research is the diversity of extreme events and their potentially huge socioeconomic consequences. These events include convective hailstorms, large-scale flood events and heat waves, as well as compound hazards such as precipitation and wind extremes (e.g., Zscheischler et al., 2020; Messmer and Simmonds, 2021; Owen et al., 2021). This review did not explicitly discuss the dynamics of different types of weather extremes, but it mentioned, where appropriate, that some of the strongly diabatically driven extratropical cyclones also led to extreme surface weather conditions and damage. It is obvious that diabatic processes play a role in the formation of most of these events, in particular of those associated with extreme precipitation. Currently many studies on extreme events use recently developed concepts and diagnostics to investigate the atmospheric water cycle (e.g., atmospheric rivers, moisture recycling, or so-called moisture source diagnostics). These concepts are very useful to understand, for instance, the anomalous water vapour transport leading to a large-scale flood event, but, in isolation, they cannot provide information about the interactions of the flow dynamics, the vapour transport, and the diabatic processes. Here, future research could make an important contribution in addressing these complex interactions in more detail.

7. **Climate change.** And last but not least, we get back to the relevance of studying diabatic processes in a changing climate. As outlined in some detail in Sect. 5.7, there are important open questions in how the currently ongoing global warming will affect diabatic processes, their effects on individual weather systems, and in turn the feedbacks on the climate system. We are convinced that the formidable challenges related to these research questions require integration of expertise from (i) weather and climate dynamics, (ii) detailed physical process understanding, and (iii) the integration

of the dynamical and physical understanding in high-resolution numerical models up to the complexity of Earth system models. Recent examples have shown that the availability of climate simulation output with high resolution in time and space (including vertical information) and the transfer of methodologies developed in weather system dynamics to climate simulations can yield fascinating and important insight into the future climate system. The increasing use of "storylines", defined by Shepherd et al. (2018) as "a physically self-consistent unfolding of past events, or of plausible future events or pathways", as an approach to representing uncertainties associated with climate change necessitates weather-scale physical processes understanding as highlighted by the rain-on-snow Alpine event example presented in that paper. We trust that the understanding about how diabatic processes influence the dynamics of synoptic-scale extratropical weather systems, as summarised in this review article, will be a central contribution for assessing these future changes.

Artificial intelligence approaches may also lead to progress in understanding moist dynamics. So far progress has been made in the development of atmospheric parametrisation schemes (e.g., O'Gorman and Dwyer, 2018; Gentine et al., 2018). Givon et al. (2023) is a rare example of the use of artificial intelligence approaches to infer diabatic processes. This study revealed the potential different contribution of diabatic effects to different types of Mediterranean cyclones by clustering their PV structures using self-organising map analysis, a type of artificial neural network. However, we have not included these artificial intelligence approaches in our list above because, while they may lead to improvements in models, it is difficult to anticipate how they could progress the understanding of moist dynamical processes. As a final remark, we would like to mention that we tried hard to include the most relevant literature from all parts of the world (published in English), but we most likely missed several important studies. We are therefore very grateful if future readers inform us about such oversights.

**Appendix A: Glossary of terms**

Table A1: Glossary of key terms. The early references included are those that specifically use this term; for some terms earlier papers have described the feature (e.g., PV tower), but not used this terminology. Note also that the online American Meteorology Society (2023) glossary includes the terms indicated by a *. The wording in the AMS glossary has influenced the definitions given of some of these terms. However, our wording reflects the needs for our review.

| Beginning of Table | | |
|---|---|---|
| Term | Definition | Important early references |
| Atmospheric river* | A ribbon-like, often transient, feature of strong horizontal water vapour transport; WCBs often ascend from regions that satisfy the criteria of atmospheric rivers, and atmospheric river criteria are also often met in orographic barrier jets such as the South American low-level jet and moist onshore airflows such as the monsoon winds. | Zhu and Newell (1994); Ralph et al. (2018) |
| Baroclinic instability* | A wave instability that occurs through the conversion of the potential energy that exists due to a meridional temperature gradient in the basic state flow (which is in thermal wind balance with a vertical wind shear) to kinetic energy; release of this instability is the primary mechanism for the intensification of extratropical cyclones | Hoskins et al. (1985); Charney (1947) |
| Cloud head* | A hooked-shaped cirrus cloud feature with a sharp convex poleward boundary, located on the poleward side of the main polar front cloud band | Böttger et al. (1975); Browning and Roberts (1994) |
| Conditional symmetric instability* | A form of moist convective instability that occurs when the atmosphere is unstable to air parcel displacements in a slantwise direction due to the combination of conditional and inertial instability, but stable to vertical and horizontal displacements due to conditional and inertial stability, respectively; the release of this instability leads to slantwise convection | Bennetts and Hoskins (1979); Schultz and Schumacher (1999) |
| Diabatic outflow | The cross-isentropic ascent due to latent heating in larger-scale cloud systems near the extratropical waveguide and the subsequent horizontal divergence in the upper troposphere of the ascended air with typically low PV values | Grams and Archambault (2016) |

| Term | Definition | Important early references |
|---|---|---|
| Diabatic Rossby wave | An intense low-level PV anomaly in a moist baroclinic zone that propagates faster than the mean flow due advection of moist air masses on the eastern flank of the low-level vortex that leads to upward motion, condensation, diabatic heating and PV production | Parker and Thorpe (1995) |
| Diabatic process* | A thermodynamic process occurring through the transfer of heat between the "system" and its environment. For weather systems the diabatic processes are latent and sensible surface heat fluxes, phase changes in clouds, and radiative processes | N/A |
| Elevator-escalator concept | WCB ascent consisting of interspersed regions of mesoscale/convective-scale ascent associated with deep convection (the elevator) and slantwise frontal ascent (the escalator) | Neiman and Shapiro (1993) |
| Frontal wave* | A horizontal wavelike deformation of a front in the lower levels that is commonly associated with a cyclonic circulation; it may lead to a wave cyclone and thus secondary cyclogenesis | Petterssen and Smebye (1971); Parker (1998) |
| Potential vorticity* (PV) | A materially conserved quantity in the absence of diabatic and frictional effects that can be defined as the scalar product of the vector absolute vorticity and potential temperature gradient all divided by density (Eq. 1) | Ertel (1942); Hoskins et al. (1985) |
| PV banners | Orographically produced low-tropospheric PV filaments | Smith and Smith (1995); Aebischer and Schär (1998) |
| PV streamers | Meridionally elongated narrow PV filaments in the upper troposphere, which typically accompany lee cyclogenesis | Bleck and Mattocks (1984); Appenzeller and Davies (1992) |
| PV tower | A troposphere-spanning vertically coherent column of anomalously high potential vorticity over a surface cyclone | Hoskins (1990); Rossa et al. (2000) |
| Reanalysis* | A consistent representation of the atmospheric (or any other system) state, typically at least several decades long, on a regular grid generated by blending past observations with short-range forecasts with a modern weather forecasting system | Bengtsson and Shukla (1988) |
| Sting jet | A coherent airflow that descends from mid-levels inside the cloud head into the frontal-fracture region of a Shapiro-Keyser type cyclone over a period of a few hours leading to a distinct region of near-surface stronger winds | Browning (2004); Clark and Gray (2018) |

| Term | Definition | Important early references |
|------|-----------|----------------------------|
| Warm conveyor belt* | A well-defined airstream in an extratropical cyclone between 100 km and 1,000 km wide and a few km deep, which initially flows parallel to and in advance of the surface cold front and which later ascends above the warm frontal zone; often the main rain-producing flow, transporting large amounts of heat and moisture | Harrold (1973); Browning and Roberts (1994) |

End of Table

## Appendix B: List of acronyms

Table B1: List of acronyms used.

| Beginning of Table | |
| --- | --- |
| Acronym | Meaning |
| AMTEX | Air Mass Transformation EXperiment |
| ALPEX | ALPine EXperiment |
| CAPE | Convective Available Potential Energy |
| CASP | Canadian Atlantic Storm Program |
| CESM1 | Community Earth System Model version 1 |
| CI | Conditional Instability |
| CMIP | Coupled Model Intercomparison Project |
| CSI | Conditional Symmetric Instability |
| CISK | Conditional Instability of the Second Kind |
| COSMO | COnsortium for Small-scale MOdelling |
| CYCLES | CYCLonic Extratropical Storms (experiment) |
| DIAMET | DIAbatic influence on Mesoscale structures in ExTratropical storms |
| DLR | Deutsches Zentrum für Luft- und Raumfahrt |
| DRW | Diabatic Rossby Wave |
| ECHAM5 | European Centre HAMburg climate model version 5 |
| ECMWF | European Centre for Medium-range Weather Forecasts |
| ERA | ECMWF ReAnalysis |
| ERICA | Experiment on Rapidly Intensifying Cyclones in the North Atlantic |
| FAAM | Facility for Airborne Atmospheric Measurements (in the U.K.) |
| FASTEX | Fronts and Atlantic Storm-Track EXperiment |
| GALE | Genesis of Atlantic Lows Experiment |
| GCM | General Circulation Model |
| HALO | High-Altitude and LOng-range aircraft (German research aircraft) |
| ICON | ICOsahedral Nonhydrostatic weather and climate Model |
| IFS | Integrated Forecasting System (of ECMWF) |
| IOP | Intense Observation Period |
| IPCC | Intergovernmental Panel on Climate Change |
| JMA | Japan Meteorological Agency |
| MetUM | (U.K.) Met Office Unified Model |

| Acronym | Meaning |
|---|---|
| | Continuation of Table |
| MM4 | PSU (Pennsylvania State University)-NCAR Mesoscale Model version 4 |
| MPV | Moist Potential Vorticity |
| MSI | Moist Symmetric Instability |
| MSLP | Mean Sea Level Pressure |
| NASA | National Aeronautics and Space Administration (in the U.S.) |
| (T-)NAWDEX | (THORPEX-)North Atlantic Waveguide and Downstream impact EXperiment |
| NAWDIC | North Atlantic Waveguide, Dry Intrusion, and downstream impact Campaign |
| NCAR | National Center for Atmospheric Research (in the U.S.) |
| NCEP | National Centers for Environmental Prediction (in the U.S.) |
| NWP | Numerical Weather Prediction |
| PV | Potential Vorticity |
| QE | Queen Elizabeth |
| SAFIRE | Service des Avions Français Instrumentés pour la Recherche en Environnement (French research aircraft) |
| SCAPE | Slantwise Conditional Available Potential Energy |
| SST | Sea Surface Temperature |
| STE | Stratosphere-Troposphere Exchange |
| THORPEX | THe Observing system Research and Predictability EXperiment) |
| TIGGE | The International Grand Global Ensemble (although the "T" stood for THORPEX prior to completion of the THORPEX programme) |
| T-PARC | THORPEX Pacific Regional Campaign |
| TPV | Tropopause Polar Vortex |
| WCB | Warm Conveyor Belt |
| WISHE | Wind-Induced Surface Heat Exchange |
| WRF | Weather Research & Forecasting (model) |
| | End of Table |

*Data availability.* The ERA5 data used in the supplement, Hersbach et al. (2023) can be downloaded from the Copernicus Climate Change Service (2023). The results contain modified Copernicus Climate Change Service information 2020. Neither the European Commission nor ECMWF is responsible for any use that may be made of the Copernicus information or data it contains.

*Author contributions.* Both authors designed this study and wrote the paper.

*Competing interests.* The authors declare that they have no competing interests. HW is one of the executive editors of *Weather and Climate Dynamics*.

*Acknowledgements.* We are most grateful to Sarah Jones for suggesting the writing of such a review paper years ago and for all her input and encouragement during the long conceptualization phase of this study. We also would like to thank Nicolai Krieger and Franziska Schnyder (both ETH Zurich) for their help with compiling the list of references, and Franziska Schnyder also for preparing the figures in the Supplement. Feedback from our colleagues at ETH Zurich and the University of Reading, Franziska Aemisegger, Hanin Binder, Helen Dacre, Michael Sprenger, and Alan Thorpe, on earlier versions of this article was highly valuable and much appreciated. Finally, we would like to thank Lance Bosart, Tyler Leicht, and Alexander Mitchell (all from SUNY Albany) and the five anonymous referees for reading such a long manuscript and for their very constructive and helpful comments.

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
