# Peer review of "The importance of diabatic processes for the dynamics of synoptic-scale extratropical weather systems—a review"

_EGUsphere, 2023_

## Referee Comment (RC1)

**The importance of diabatic processes for the dynamics of synoptic-scale extratropical weather systems—a review**

**Wernli and Gray**

This paper presents a comprehensive review of the role of diabatic processes in the development and structure of weather systems, as envisaged from the nineteenth century to the present day. This is a complex topic, as the physics is dominated by small to mesoscales and there is so much variability between cyclones that it took a long time for meteorologists to agree on the nature and importance of diabatic processes compared to the mathematically elegant theories of dry baroclinic instability. Much progress on this topic has been achieved over the past twenty years and this review is timely and welcome. It should be published with minor corrections – but because of its scope and length there are a lot of these.

This is not, and should not be, a comprehensive review of every paper ever published on diabatic processes in cyclones, as the authors make clear in the Introduction. It will be most valuable as a summary of these distinguished authors' own understanding of the topic, and of the papers that led them to that understanding, as this will give the review a coherence that those unfamiliar with the field will most appreciate. Of course, much of this already comes through strongly in the manuscript and my comments will mainly be aimed at improving the flow and readability of the paper.

I liked the structure of the review sections, with 'pauses for breath' or summaries at the end of each subsection. As my detailed comments will show, I think some of these could be developed further to provide a synthesis of the science that was revealed by the papers, rather than just reiterating what they did. Although I will concentrate here on sections 5 and 6 (following the Editor's request), these comments actually apply most strongly to the earlier sections where the evidence presented was sometimes contradictory, making it difficult to see what advances in the science actually occurred.

**Comments on section 5**

The advent of reanalyses has indeed been a game-changer for meteorological research generally, not just for this topic, and amply merits detailed discussion in 5.1. The section is informative and reads well, but the Summary is perfunctory and adds little. What is needed here is a summary of what the reanalysis papers found e.g. regarding the distribution among cyclones of the importance of diabatic heating, its possible added relevance to the more extreme cyclones, the link to 'atmospheric rivers' etc. How did this approach advance the science?

Section 5.2 discusses 'diabatic processes in (special categories of) extratropical cyclones', presenting seven loosely-connected subsections beginning with cyclone classification.

i. I recommend that the authors reconsider the order of their subsections, moving extratropical transitions to the penultimate slot with a more natural transition to tropopause level vortices. The transition from Type C cyclones to subtropical and then Mediterranean cyclones would then be smoother.

ii. Section 5.2.1 would make more sense to a reader unfamiliar with cyclone categorisation if the authors explained what the Peterssen and Smebye A/B scheme actually is before launching into a type C discussion.

iii. The Mediterranean section should be shortened, concentrating on distinctive properties of these cyclones, other than where they occur (e.g. paragraph 1736-1746 could be omitted, and I'm not sure what the last paragraph, on moisture sources, is adding to the science).

iv. It is appropriate to include polar lows in this review because of the contribution of convection to many of them, but the section could do with editing to make the key points clearer. I suggest that the text from 1815 to 1821 be removed as it lapses into jargon inconsistent with

the rest of the section and detracts from the theme of diabatic heating. Likewise, the paragraph 1843-1858 goes into a level of detail not required here, given the existence of reviews specifically of polar lows.

    v.    To my mind Diabatic Rossby Waves are of a different order of importance to the other subsections here, as this is a distinctive dynamical process in its own right. Could this become section 5.2, then section 5.3 would include the other subsections? I leave this to the authors' discretion but it would allow for mention of DRWs in the sections that currently precede it, and better overall coherence.

    vi.    The summary subsection 5.2.8 is appropriate to this section

Section 5.3 discusses novel diagnostics of diabatic PV modification. Again the subsections are appropriate but could benefit from some critical editing.

    i.    The first paragraph of 5.3.1 is too detailed – readers should consult the original papers for the detailed methodology – while the second paragraph could benefit from more examples of results obtained from PV tracer analysis.

    ii.    The final paragraph of 5.3.2 doesn't lead anywhere – did Büeler and Pfahl find anything useful from their study? If not, this paragraph could be deleted.

    iii.    On line 2175 we are cautioned that 'caution needs to be applied when inferring dynamical causation from ensemble sensitivity analysis' yet in the very next paragraph the word 'sensitivity' is used instead of 'association' three times! These paragraphs need to be consistent with each other.

    iv.    Has the adjoint technique led to any new insights into diabatic processes? The result that 'forecasts of high-impact cyclones were found to be strongly sensitive to low to mid-tropospheric moisture in the initial state' is hardly novel. Section 5.3.3 is one which could be considerably shortened.

    v.    The summary is again appropriate

Section 5.4 concentrates on the impact of diabatic processes on the dynamics at tropopause level, through the outflow of WCBs and tropical cyclones. I thought this was balanced and coherent, with an informative summary, and but for a couple of minor comments (see separate section below) I have no major problems with it.

Section 5.5 describes the two field campaigns DIAMET and NAWDEX which the authors consider to be the only two experiments of note since 2000 to study diabatic processes in cyclones.

    i.    Given that extratropical transitions fall into the domain of this review, mention should also be made of T-PARC (2008).

    ii.    The concept of the sting jet arose from analysis by Browning of the Great Storm that struck Southern England in 1987 and was developed by diagnosis of high-resolution models by Clark et al. This paper needs to explain more clearly what the DIAMET measurements contributed, for example by explaining what figure 16b is supposed to show. It also needs to acknowledge that the sting jet (defined as a descending airstream) is a transient phenomenon, especially when compared to the cold conveyor belt which dominates the low-level wind field in the southern quadrant of a mature cyclone.

    iii.    Although 5.5.2 is entitled 'Embedded convection', 'Embedded convection and negative PV bands' would better describe the content.

Section 5 concludes with a discussion of the relevance of diabatic heating in cyclones to climate change research. On the face of it, a warmer climate will mean more moisture in the atmosphere and more scope for diabatic heating. But, as 5.6.2 shows, the problem is far from linear and the location of diabatic heating relative to the cyclone centre is critical when considering its effect on the cyclone. It appears that in a warmer climate the PV source region will be further from the cyclone centre on

average – but the tail of the distribution, where the two line up, may result in a few very powerful windstorms. Very interesting! The long discussion in 5.6.3 of the effect of model resolution on the diabatic effects on cyclones in GCMs could be shortened considerably, as the details are covered in many other papers. This review could simply summarise the conclusion of these studies, i.e. a short introductory paragraph then pick up at line 2487. Although the summary of this section is informative, I recommend that it be expanded by a few sentences to cover the issues of propagation and blocking discussed in the text.

Section 6 provides a summary of the basic concepts presented in the review and looks to the future of the field. It reads well and I have no major comments on it.

Minor comments

l. 1496 millenium

2. l.1587 Either omit 'only' or restructure to '….revealed that only in the NH winter are cyclones usually accompanied by a strong WCB….', depending whether the sentence is meant to contrast winter and summer as well as the two hemispheres.

3. l. 1603 The idea of a cyclone with no WCB is most likely an artifact of the definition assumed for a WCB in Binder et al's study (or a problem with their method) than a reflection of the dynamics of the cyclone. Read literally it means there was no ascending airstream ahead of the cyclone, which is hard to square with explosive development. The idea is counter to the whole thesis of this review and requires more discussion if considered to be a real result.

4. l. 1635. Please explain what the Petterson and Smebye A and B cyclones actually are. The section is difficult to follow for someone not versed in cyclone classification because the story starts in the middle.

5. l.1851 lesser

6. l.2330-1 Isn't downstream development a consequence of Rossby wave propagation? The underlying dynamics are the same so this sentence needs to be re-phrased.

7. l.2420-2425 Which simulation corresponded best to reality in this case?

8. l. 2727 isn't the argument that, basically, the same cyclone in a warmer climate will produce more precipitation just because it is warmer (the 'Clausius-Clapeyron effect')?

9. l.2979. It is provocative to claim, without evidence, that the field campaigns were a direct result of a handful of high-priority storms. For example, FASTEX was not organised as a response to the Great October Storm and DIAMET (according to its description earlier) was not organised as a response to the discovery of sting jets.

10. l.2995 The authors have chosen not to say much about satellite measurements in this review, despite the crucial role they played in the thinking of key figures in the field, such as Keith Browning. That is their prerogative, though still an omission. But one of the key conceptual tools of the satellite era was time-lapse videos from geostationary satellites, which actually show how the WCB develops alongside the cyclone and the cold front, and that should be mentioned here. It is not speculation to point out the key role of satellite images in developing understanding, and that word should be removed. So also the 'real-time availability', so important for forecasting but not for research where the better-quality images available after the event (especially in the 1980s) were more useful.

---

## Referee Comment (RC5)

Review of egusphere-2023-2678

"The importance of diabatic processes for the dynamics of synoptic-scale extratropical weather systems—a review"
by
Heini Wernli and Suzanne L. Gray

**Recommendation: Major revisions**

**General Comments:**

The authors have put a lot of effort in summarizing and reviewing the evolution of our thinking of the effects of diabatic processes on synoptic-scale cyclone development. As usual, with such attempts, it is difficult to do the entire field of research justice and certain selections must be made by the authors in terms of focus. While the review is quite extensive, some recent literature and arguments have not been included thus far. Furthermore, potentially related to the background and work of the two authors, the manuscript focuses mainly on PV diagnostics based on Lagrangian, tracers, or other direct diagnostics of PV. Hence, the title might be a bit misleading and should potentially reflect this choice.

In addition, the manuscript reads more like an extensive summary of research conducted the last decades and one and a half centuries but does not necessarily do justice to the term "review". Review, to me, would imply that the authors go beyond summarizing the material and provide a more detailed discussion of the different methods, their pros and cons, as well as if the community along the way concluded on the adequacy, or lack thereof, of different concepts, diagnostics, and theories. Such a review of diagnostics, concepts, theories would be highly desirable. For example, most scientists would probably not anymore believe that CISC is really an appropriate concept for moist-baroclinic development.

The authors provide a general introduction to latent heating, which is maybe not needed given the audience. There is also a rather lengthy introduction of symmetric instability, which is potentially also beyond the focus of this review. In general, the part on slantwise convection seemed a bit out of place given the focus is mostly on synoptic scales.

Given the authors' background, their almost exclusive focus on PV is understandable, though there have also been other attempts to understand the effects of diabatic heating on the synoptic evolution that remain unmentioned. For example, the diagnostic of the tendency of baroclinicity (Papritz and Spengler, 2015), which has subsequently been used to postulate the hypothesis that cyclone clusters are formed diabatically (Weijenborg and Spengler, 2021) is not mentioned. This concept does neither rely on neglecting certain parts of the flow (PV only deals with the circulation on theta surfaces) nor on having to separate into a basic state and anomaly, while it clearly demonstrates the workings of diabatic forcing on the synoptic, and even larger-scale, evolution.

When the authors introduce the concept of PV in 2.1, the discussion lacks a discussion of the disadvantages of the usage of PV. For example, equation (1) highlights the fact that

only that part of the circulation that is perpendicular to the gradient of theta is represented by PV, i.e., only the circulation within a theta surface. All other parts of the circulation are neglected and can thus also not be addressed by the PV tendency in equation (3). Examples for such kind of circulations would be a sea-breeze, or frontal circulations. Of course, in a QG framework, these would be regarded as ageostrophic and thus potentially of lower relevance, but the implications and limitations of using a PV framework should be more clearly stated. The formulation in equations (5)-(7) could have been related to the circulation theorem, which yields the same result (implied impermeability), when only choosing that part of the circulation that projects on a theta surface.

With respect to the interpretation of the detected tendencies and changes in PV (e.g., section 5.3.2 and others), one should be aware that to understand the system and its evolution, it is not sufficient to merely quantify changes in PV. For example, while evaporation below an area of latent heating might yield a stronger PV anomy at the interface between these differently signed diabatic forcing, the actual effect on the evolution of the cyclone might still be detrimental (Haualand and Spengler, 2019). Thus, the discussion of diagnosed anomalies should be put in more context to the overall development when presenting the derived fields, trajectories, and tracers.

I am admittedly not too familiar with all the detailed historic developments in the field prior to the mid 20ies century and greatly appreciate the effort the authors put into providing a wider historic overview of the concepts of cyclone development. However, it feels strange that the work of the Bergen School of Meteorology is not mentioned when it comes to the concepts of the structure and airstreams in cyclones, as several seminal papers on the structure of cyclones came out of this school. Even though the naming of the airstreams and sectors of cyclones at the time was different, the cold and warm/moist airstreams were clearly depicted in their conceptual cyclone models. These contributions should be included in sections 3.1 and 3.2.

With respect to polar lows in section 5.2.5, Terpstra et al. (2015) stressed the role of diabatic processes, despite them often being argued to be small due to the low values of absolute available moisture. One of the main arguments of Terpstra et al. (2015) being that despite the diabatic heating being significantly smaller than in extratropical cyclones, the effect on the PV tendency is comparable to extratropical cyclones, because the vertical extent is also significantly reduced, thereby increasing the effect of the gradient of the heating on the PV tendency. Furthermore, Stoll et al. (2021) clearly classified polar lows as moist baroclinic cyclones in a recent climatological analysis, while clarifying that the genesis through hurricane-like processes is rather unlikely, due to the excessive amounts of baroclinicity and shear at the genesis time. In general, the notion of an upper-level PV anomaly must be applied with caution for polar lows, as the usual altitude chosen to detect these PV anomalies are well within the stratosphere and thereby not necessarily directly relatable to the development of the surface-based polar lows. Furthermore, these upper-level anomalies are often of much larger character than the developing polar low, also rendering a direct interaction questionable.

The surface pressure tendencies introduced at the beginning of 5.3.4 are significantly flawed (Spengler and Egger, 2009). A simple thought experiments directly reveals the false physical nature of the diagnostic. Assume a horizontally uniform atmosphere, equivalent to an atmospheric column with no horizontal advection, which is initially motionless and where a mid-tropospheric heating is applied. Note that all concepts are

hydrostatic. In such a hydrostatic setup, the heating cannot result in a direct change of surface pressure, which can only be caused by horizontal mass rearrangements (secondary circulation), or precipitation (Spengler et al., 2011). However, the diagnostic introduced by the various authors in section 5.3.4 directly implies a "diabatic" surface pressure tendency, which is not physical and points to a significant flaw in the diagnostic. Such an attribution could at best be achieved in a balanced framework, such as QG, though then the challenge is to define suitable boundary conditions for the inversion that is difficult to prescribe a priori (Spengler and Egger, 2012).

Related to predictability and the effects on the upper troposphere, a recent study highlighted the importance of diabatic heating compared to tropopause structure for initial error growth (Haualand and Spengler, 2021). Even more drastic changes, or implied initial errors, along the tropopause are easily dwarfed by the effects of misrepresenting latent heating.

Given that the authors state that they "touch on the effects of […] surface fluxes, in particular where studies have contrasted the effects of these processes with the effects of latent heating", it is surprising that recent studies highlighting the direct and indirect (enhanced latent heat release due to latent heat fluxes) effects of surface fluxes are not mentioned (Haualand and Spengler, 2020; Bui and Spengler, 2021). Previous studies in general showed that surface sensible heat fluxes have a detrimental effect, as they reduce baroclinic structure in the cyclone, while these more recent studies emphasise that the additional latent heat release available due to the surface latent heat fluxes can easily dominate the cyclone development in a favourable way.

Line 636-647: It is not clear what this discussion of lee cyclogenesis has to do with the main topic of the review article, i.e., the importance of diabatic processes. Consider removing or putting in context.

References:

Bui, H. and Spengler, T. (2021). On the Influence of Sea Surface Temperature Distributions on the Development of Extratropical Cyclones. J. Atmos. Sci., 78, 1173-1188, https://doi.org/10.1175/JAS-D-20-0137.1

Haualand, K. F., and Spengler, T. (2019). How Does Latent Cooling Affect Baroclinic Development in an Idealized Framework? J. Atmos. Sci., 76, 2701-2714, https://doi.org/10.1175/JAS-D-18-0372.1

Haualand, K. F., and Spengler, T. (2020). Direct and Indirect Effects of Surface Fluxes on Moist Baroclinic Development in an Idealized Framework. J. Atmos. Sci., 77, 3211-3225, https://doi.org/10.1175/JAS-D-19-0328.1

Haualand, K. F. and Spengler, T. (2021). Relative importance of tropopause structure and diabatic heating for baroclinic instability, Weather Clim. Dynam., 2, 695–712, https://doi.org/10.5194/wcd-2-695-2021

Papritz, L., and Spengler, T. (2015). Analysis of the slope of isentropic surfaces and its tendencies over the North Atlantic. Q. J. Roy. Met. Soc., 141, 3226-3238, https://doi.org/10.1002/qj.2605

Spengler, T., and Egger, J. (2009). Comments on "Dry-Season Precipitation in Tropical West Africa and Its Relation to Forcing from the Extratropics". 137, 3149-3150, https://doi.org/10.1175/2009MWR2942.1

Spengler, T., and Egger, J. (2011). How Does Rain Affect Surface Pressure in a One-Dimensional Framework? J. Atmos. Sci., 68, 347-360, https://doi.org/10.1175/2010JAS3582.1

Spengler, T., and Egger, J. (2012). Potential Vorticity Attribution and Causality. J. Atmos. Sci., 2600-2607, https://doi.org/10.1175/JAS-D-11-0313.1

Stoll, P. J., Spengler, T., Terpstra, A., and Graversen, R. G.: Polar lows – moist-baroclinic cyclones developing in four different vertical wind shear environments, Weather Clim. Dynam., 2, 19–36, https://doi.org/10.5194/wcd-2-19-2021

Terpstra, A., Spengler, T., and Moore, R. (2015). Idealised simulations of polar low development in an Arctic moist-baroclinic environment. Q. J. Roy, Met. Soc., 141, 1987-1996, https://doi.org/10.1002/qj.2507

Weijenborg, C., and Spengler, T. (2020). Diabatic heating as a pathway for cyclone clustering encompassing the extreme storm Dagmar. GRL, 47, e2019GL085777. https://doi.org/10.1029/2019GL085777

---

## Referee Comment (RC6)

Thursday 11 January 2024

A Review of Wernli and Gray (Ch. 2-4): The importance of diabatic processes for the dynamics of synoptic-scale extratropical weather systems—A review

Recommendation: Accept with minor revision (lots of little things)

A potpourri of possible additional references is also appended.

Overview:

Wernli and Gray have produced a very valuable, highly informative, and an extensively documented research review paper on the importance of diabatic processes that govern the dynamics of synoptic-scale extratropical weather systems. This extensive research review paper will be both a very valuable addition to the refereed literature and a "must have" document for advanced graduate students and early career scientists who have strong research interests in synoptic-dynamic meteorology. Noteworthy attributes of this synoptic-dynamic meteorology review paper include: 1) a broad-based historical perspective on the scientific ideas that have governed the field synoptic-dynamic meteorology going back to the 19th century, 2) an assessment of the original thinking that resulted in critical new breakthroughs in the development of innovative fundamental science ideas that have driven the growth of synoptic-dynamic meteorology, 3) an overview of critical past and present research in synoptic-dynamic meteorology that has driven the field forward, and 4) an outlook for future new research opportunities going forward that could further broaden and deepen the field. This extensive and detailed synoptic-dynamic meteorology review paper is suitable for both advanced graduate students and early career scientists who possess a strong interest in synoptic-dynamic meteorology.

Bottom line: "The importance of diabatic processes for the dynamics of synoptic-scale extratropical weather systems—a review" is thorough and offers a comprehensive overview of the history of extratropical cyclone dynamics using a PV framework. The authors demonstrate a strong foundation of synoptic-scale extratropical dynamics and present a clear and accessible synthesis of the historical literature. The inclusion of recent references and an overall balanced discussion add significant value for the reader.

Introduction:

Wernli and Gray write: "the main three aims of this review article are (i) to provide evidence that our understanding of how diabatic processes affect extratropical weather systems has grown considerably since the review article on PV by Hoskins et al. (1985) and the comprehensive book chapter on the rapid intensification of extratropical cyclones by Uccellini (1990), (ii) to portray in detail the historical evolution of a specific research field over several decades and thereby to exemplify how scientific progress results from the combination and integration of complementary research approaches,

and (iii) to promote the relevance of this research area in dynamical meteorology." This purpose statement is accompanied by a schematic overview figure that outlines the text pathways that follow.

Theoretical Background:

I didn't realize how much hard-core fundamental theoretical and dynamical meteorology I had forgotten (please forgive me, Joe Pedlosky) until I read through Ch. 2. That said, I think that this chapter as written in several places can come across as a bunch of research fluid dynamicists talking more to themselves than the intended audience. An example would be the discussion surrounding Fig. 2. Several of the panels in Fig. 2 (e.g., panels c and f) are quite obscure and need more context and explanation. In the case of panel c it would be helpful if the location of the snapshot relative to the larger scale flow on a weather map could be indicated for perspective purposes. Likewise, panel (f) supposedly references a convective updraft. Is this updraft located in the warm sector of an extratropical cyclone or elsewhere? Again, more context is needed.

Line 196: Check that "Q" was defined previously.

Omega in equation 3 needs to be defined for completeness.

The "M" lines on the Fig. 3c panel are undefined.

Define WCB on line 230 even though we all know what it means.

Lines 233–242:  Perhaps a relationship between the magnitude and spatial scale of a PV anomaly and the associated induced tropospheric wind and thermal fields can be referenced quantitatively here? In other words, consider including the Rossby penetration depth and its relation to the static stability in the context of moist processes to add clarity for sections 2.2, 4 and 5.2.1

Lines 259-262: What does the DSI (dynamical state index) tell us that we don't already know by other means?

Fig. 3b is not very readable. No point in showing a figure that requires the use of a magnifying glass to read it properly.

Likewise, Fig. 3c needs more explanation to be understood properly.

Define the two dipole axes in Fig. 3e. This figure panel needs more explanation to be understood.

Sections 3.3–3.4 are excellent.

Section 3.5 is also excellent.

You might want to emphasize that very important aspect of the Presidents' Day storm of Feb 1979 as discussed by Bosart (1981) was the comparatively low level (~900-hPa) of the ascent maximum associated with coastal front cyclogenesis (see his Fig. 9b and subsequent figures below) and the associated low-level frontogenesis maximum at ~950-hPa (Fig. 10c). Note also the derived kinematic ascent maximum below 800-hPa in Fig. 14. This low-level ascent maximum ensured that strong low-level cyclonic vorticity was being generated along the aforementioned coastal front. Subsequently, the arrival of the upper-level trough fostered impressive rapid surface cyclogenesis (see Fig. 18), given the presence of pre-existing cyclonic vorticity along the antecedent coastal front. Bottom line: The low-level d(omega)/dp profile was critical for the rapid spin-up of the Presidents' Day cyclone and should be mentioned.

Section 4.1.1 is excellent overall.

In section 4.1.1 consider showing and/or referencing a d(omega)/dp for the aforementioned Presidents' Day storm paper to help the reader better understand how the rapid growth of low-level vorticity along the coastal front occurred.

Section 4.1.2 is also excellent overall

PV Inversion, Trajectories, and Model Simulations (lines 670-677): Excellent discussion of Davis and Emanuel (1991) and Hoskins and Berrisford (1988). From my personal perspective, Hoskins and Berrisford (1988) was an eye-opening paper and a "must read" paper as well.

Line 704 and subsequent lines. Nice discussion of the warm-conveyor belt to include the Nieman and Shapiro papers. Other relevant Mel Shapiro papers that you cite include Shapiro (1976), and Keyser and Shapiro (1993). What may not be fully appreciated is that Mel Shapiro obtained all kinds of critical mesoscale data on upper-level fronts via his research flights on NCAR and NOAA aircraft. These aircraft-derived datasets permitted him (and others) to make the calculations on the evolution of PV in upper level fronts and associated PV anomalies that are discussed in this section.

Good discussion of the important escalator-elevator concept on lines 715-720.

Lines 754–760: Very important insight by Reed and Albright (1986) on how fast symmetric instability can develop in an explosively deepening bomb cyclone.

Lines 784–787: Thanks for reminding me of the importance of the Shutts (1990a, 1990b) papers with regard to SCAPE (an open and "shutt" case to make a bad pun).

Section 4.1.3: Frontal Wave Cyclones

Key point on lines 810-815: Breakthrough in understanding frontal wave cyclones inspired by the Hoskins and Berrisford (1988) diagnostic study on the infamous UK October storm and related papers by Thorpe and Emanuel, 1985; Joly and Thorpe,

1990; Schär and Davies, 1990; and Malardel et al.,1993). These papers (and others) collectively revealed how diabatically produced bands of low-level PV were a prerequisite for the occurrence of low-level frontal wave instability.

Lines 814-815 make an important point about the "essential role of diabatically produced low-level bands of PV or surface as a prerequisite for frontal wave instability to occur." Highlight this point a bit more?

Lines 847-850: Reference Fig. 9 from Rogers and Bosart (1986), given that this figure shows that the saturation equivalent potential temperature maximizes in excess of 315 K over the cyclone center at the time of lowest SLP? There is also an ~13 K increase in the saturation equivalent potential temperature over the cyclone center in the 24 h period ending 1200 TC 4 October 1965.

Lines 850-855: Discuss the assorted composites of East Coast cyclones from Manobianco (1989a) in a bit more detail?

Lines 895-900: This text "screams" for a supporting illustrative figure.

The excellent text on lines 900-920 would benefit from the addition of at least one new trajectory-related figure.

WCB discussion on lines 900-966 would benefit from the addition of a couple of new figures. Existing Figs. 7e,f are inadequate. Redo existing Fig. 7 into panels a-d and create a new figure consisting of the old e and f panels from Fig. 7? Check also whether Fig. 7f is adequately referenced in the text.

Section 4.1.5 (Lagrangian view and coherent airstreams) is very well done.

Cordeira and Bosart (2011; https://journals.ametsoc.org/view/journals/mwre/139/6/2010mwr3537.1.xml?tab_body=pdf) would be an appropriate additional reference for the evolution of PV structure in a tropical cyclone (Grace) that formed via a tropical transition (see Fig. 2, 4, 7, 11, and 12)

Figures:

The text-figure balance needs to be improved. There is a lot of text and comparatively few relevant illustrative figures. It is also unfortunate that few if any color figures are available from the older literature. I would like to suggest that the authors try to grab more relevant figures from the refereed literature to illustrate key points and bolster key arguments advanced in the text. It might also be appropriate for the authors to construct a few additional schematic figures to help to better illustrate/reinforce key points that they are trying to make in the text. Any new schematic figures should be constructed in color. Can existing black and white figures be digitized and then converted to color images using AI methodologies?

Multi-panel Figure 7 is in desperate need of improvement (or color!). Geography is mostly unreadable in panels b and c. Panel c is "fuzzy". Panels d-f demand improvement to be more readable because in present form they do not do justice to the text.

Lines 1152-1177: This discussion seems out of place here. I get that you have a separate section entitled: More Systematic Investigations. That said, the distinction seems a bit forced. Personally, I would have welcomed this discussion earlier in the text.

Section 4.2.3: PV perspective on moist vs. dry experiments: I can't decide whether this section works best as a stand-along section as it is now or whether the findings documented in this section should be integrated into earlier sections.

I am OK with section 4.3: Idealized numerical simulations of cyclones.

Lines 1305-1318: Discussion of Emanuel (1987) is well done.

Lines 1434-1457: This section, especially beginning with the discussion of Thorncroft and Hoskins (1990) on line 1441, seems especially relevant. That said, I could easily argue that this discussion is out of place and should appear earlier. Although I understand why the authors did what they did, it seems to me that an equally convincing argument can be made to embed this material into an earlier section. Personally, I find the climate change argument (footnote 15) unconvincing.

General comment after scrutinizing sections 2-4. I would like to reinforce what I said earlier. The ratio of text to figures is too high throughout. There are a number of places throughout the text where the inclusion of additional figures would make it easier to follow the discussion. That said, figure quality needs to be significantly improved in many places. I fully appreciate that this presents a problem for old non-digitized black and white figures. A possible compromise might be to separate multi-panel black and while figures that a marginally readable (at best) into individual single or double figures that are larger and possibly easier to read.

Additional Comments by Bosart Ph.D. students: Tyler Leicht and Alexander Mitchell

I've finished my review of the Wernli and Gray review paper. It is an exceptionally well-written paper, covering nearly 150 years of research in great detail. I think this will be a great research and educational resource for years to come. I only have a few minor comments that I think could improve the paper if considered.

In section 2.1, I would like to see a (brief) description of the QG height tendency and omega equations as they relate to diabatic heating. I know the paper mainly focuses on

a PV perspective, but in order to argue that PV thinking is best for understanding the impact of diabatic heating on midlatitude weather systems (stated on line 3019), one must first outline the QG perspective and compare the two frameworks. QG theory is alluded to in sections 2.2, 3.4, and 3.5, but the reader does not have an immediate reference in this text to the equations the way they do for the PV fundamentals. I would also rephrase the sentence ending in line 2965 stating that QG theory is synonymous with dry dynamics, since my view is that QG and PV can both be treated adiabatically and diabatically.

Otherwise, I think this was a very successful review paper. Let me know what you think of my suggestion, and I'll be curious to see what you and Alex think would improve this paper.

……………………….

Overview:

The manuscript, "The importance of diabatic processes for the dynamics of synoptic-scale extratropical weather systems—a review" is thorough and offers a comprehensive overview of the history of extratropical cyclone dynamics using a PV framework. The authors demonstrate a strong foundation of synoptic-scale extratropical dynamics and present a clear and accessible synthesis of the historical literature. The inclusion of recent references and an overall balanced discussion add significant value for the reader.

Lines 233–242: Perhaps a relationship between the magnitude and spatial scale of a PV anomaly and the associated induced tropospheric wind and thermal fields can be referenced quantitatively here? In other words, consider including the Rossby penetration depth and its relation to the static stability in the context of moist processes to add clarity for sections 2.2, 4 and 5.2.1

Minor Comments:

Section 1:

Line 98: Should quasi-geostrophic be abbreviated as (QG) throughout the article?
Line 110: Remove hyphen for "life-cycles"
Lines 110-114: Consider splitting these into 2 sentences
Line 116: Revise to "clouds can"
Line 117: Revise to "..Carbone, 2004), and…"
Line 121: Revise to "..up to several 1000 km"..
Line 138: Revise to "..and challenges to stimulate further.."

Section 2:

Line 206: Define $\dot{\theta}$?
Line 265: Revise to "..subsequent precipitation usually occur through"
Line 301: Revise to "..However, as discussed by Schultz and Schumacher (1999), many.."
Lines 349–355: Consider splitting this into two sentences.
Line 394: Remove the comma after "currents"

Section 3:

Lines 375–379: Insert "(a)", "(c)" and "(d)" as well since (b) is written in the sentence.
Line 402: Revise to "..and examined stability criteria.."
Lines 486–487: Revise to "..at the level of maximum wind speed. The hypothesis was made that these maxima were diabatically.."
Lines 489–492: Revise to "..It is interesting to note that the initial research on STE near upper-level fronts and tropopause folds mainly discussed how radiation and turbulence can modify PV, but overlooked the potential effects of latent heat release in clouds. However, this focus changed almost 20 years later.."
Lines 510–512: Omit "In"
Line 526: Revise to "We claim this paradigm shift.."

Section 4:

Line 568: Revise to "..novel data thanks to.."
Line 573: Revise to "..they have served as.."
Line 589: Revise to "..together with the ascent.."
Line 623: Add comma, "In several of the studies, the.."
Line 700: Revise to "..was considered the main reason.."
Line 703: Revise to "..to present an overview of the.."
Line 729: Revise to "..first to quantify.."
Lines 877–878: Revise to "This subsection summarizes research from 1980–2000 on using trajectories to investigate moist airstreams and extratropical cyclone dynamics."

Section 6:

Line 2974: Revise to "..time failed to predict these.."
Line 3012: Add a comma after "for instance"
Line 3024: Add a comma after "More recently"
Lines 3049–3054: Revise to "This improvement will allow for better representation of steep topography and the ability to turn off the parameterization of deep moist convection. In the context of this review article, this implies that, e.g., fast-ascending motion in convective cloud systems and associated diabatic processes will be simulated with the same numerics and cloud microphysics as the more slowly ascending and larger-scale warm conveyor belts. As discussed in Sect. 5.5.2, this change will have

direct implications for the diabatic modification of PV in the upper troposphere and, in turn, for the large-scale flow evolution."

………………

**Potential additional references (aka shameless plugs) to add to the Wernli and Gray Review Paper. Use anywhere from 0–48 of these suggested references as you see fit.**

1. **Archambault et al. (2010):** Relationships between Large-Scale Regime Transitions and Major Cool-Season Precipitation Events in the Northeastern United States
2. **Archambault et al. (2013):** A Climatological Analysis of the Extratropical Flow Response to Recurving Western North Pacific Tropical Cyclones
3. **Archambault et al. (2015):** A Composite Perspective of the Extratropical Flow Response to Recurving Western North Pacific Tropical Cyclones
4. **Bals-Elsholz et al. (2001):** The Wintertime Southern Hemisphere Split Jet: Structure, Variability, and Evolution
5. **Bentley et al. (2017):** Upper-Tropospheric Precursors to the Formation of Subtropical Cyclones that Undergo Tropical Transition in the North Atlantic Basin
6. **Bentley et al. (2019):** A Climatology of Extratropical Cyclones Leading to Extreme Weather Events over Central and Eastern North America
7. **Bell and Bosart (1993):** A Case Study Diagnosis of the Formation of an Upper Level Cutoff Cyclonic Circulation over the Eastern United States

8. **Biernat et al. (2023):** A Climatological Comparison of the Arctic Environment and Arctic Cyclones between Periods of Low and High Forecast Skill of the Synoptic-Scale Flow

9. **Bosart (1984):** The Texas Coastal Rainstorm of 17–21 September 1979: An Example of Synoptic Mesoscale Interaction

10. **Bosart and Bartlo (1991):** Tropical Storm Formation in a Baroclinic Environment
11. **Bosart and Dean (1991):** The Agnes Rainstorm of June 1972: Surface Feature Evolution Culminating in Inland Storm Redevelopment
12. **Bosart and Lackmann, 1995:** Postlandfall Tropical Cyclone Reintensification in a Weakly Baroclinic Environment: A Case Study of Hurricane David (September 1979)
13. **Bosart and Sanders (1986):** Mesoscale Structure in the Megalopolitan Snowstorm of 11–12 February 1983. Part III: A Large-Amplitude Gravity Wave
14. **Bosart and Sanders (1991):** An Early-Season Coastal Storm: Conceptual Success and Model Failure

15. **Bosart et al. (1996):** Large-Scale Antecedent Conditions Associated with the 12–14 March 1993 Cyclone ("Superstorm '93") over Eastern North America
16. **Bosart et al. (1998):** A Study of Cyclone Mesoscale Structure with Emphasis on a Large-Amplitude Inertia–Gravity Wave
17. **Bosart et al. (2000):** Environmental Influences on the Rapid Intensification of Hurricane Opal (1995) over the Gulf of Mexico
18. **Bosart et al. (2012):** An Analysis of Multiple Predecessor Rain Events ahead of Tropical Cyclones Ike and Lowell: 10–15 September 2008
19. **Bosart et al. (2017):** Interactions of North Pacific Tropical, Midlatitude, and Polar Disturbances Resulting in Linked Extreme Weather Events over North America in October 2007
20. **Cordeira and Bosart (2011):** Cyclone Interactions and Evolutions during the "Perfect Storms" of Late October and Early November 1991
21. **Davis and Bosart (2001):** Numerical Simulations of the Genesis of Hurricane Diana (1984). Part I: Control Simulation
22. **Davis and Bosart (2003):** Baroclinically Induced Tropical Cyclogenesis
23. **Davis and Bosart (2004):** The TT Problem: Forecasting the Tropical Transition of Cyclones
24. **DiMego and Bosart (1982):** The Transformation of Tropical Storm Agnes into an Extratropical Cyclone. Part I: The Observed Fields and Vertical Motion Computations
25. **DiMego and Bosart (1982):** The Transformation of Tropical Storm Agnes into an Extratropical Cyclone. Part II: Moisture, Vorticity and Kinetic Energy Budgets
26. **Galarneau et al. (2009):** Baroclinic Transition of a Long-Lived Mesoscale Convective Vortex
27. **Galarneau et al. (2015):** Development of North Atlantic Tropical Disturbances near Upper-Level Potential Vorticity Streamers
28. **Griffin and Bosart (2014):** The Extratropical Transition of Tropical Cyclone Edisoana (1990)
29. **Hakim et al. (1995):** The Ohio Valley Wave-Merger Cyclogenesis Event of 25–26 January 1978. Part I: Multiscale Case Study
30. **Hakim et al. (1996):** The Ohio Valley Wave-Merger Cyclogenesis Event of 25–26 January 1978. Part II: Diagnosis Using Quasigeostrophic Potential Vorticity Inversion
31. **Lackmann et al. (1996):** Planetary- and Synoptic-Scale Characteristics of Explosive Wintertime Cyclogenesis Over the Western North Atlantic Ocean
32. **Lackmann et al. (1997):** A Characteristic Life Cycle of Upper-Tropospheric Cyclogenetic Precursors during the Experiment on Rapidly Intensifying Cyclones over the Atlantic (ERICA)
33. **McTaggart-Cowan et al. (2006):** Analysis of Hurricane Catarina (2004)
34. **McTaggart-Cowan et al. (2007):** Hurricane Katrina (2005). Part II: Evolution and Hemispheric Impacts of a Diabatically Generated Warm Pool

35. **McTaggart-Cowan et al. (2010):** Development and Tropical Transition of an Alpine Lee Cyclone. Part I: Case Analysis and Evaluation of Numerical Guidance

36. **McTaggart-Cowan et al. (2013):** A Global Climatology of Baroclinically Influenced Tropical Cyclogenesis

37. **Moore et al. (2013):** Synoptic-Scale Environments of Predecessor Rain Events Occurring East of the Rocky Mountains in Association with Atlantic Basin Tropical Cyclones

38. **Moore et al. (2019):** Linkages between Extreme Precipitation Events in the Central and Eastern United States and Rossby Wave Breaking

39. **O'Handley and Bosart (1989):** Subsynoptic-Scale Structure in a Major Synoptic-Scale Cyclone

40. **Papin et al. (2020):** A Feature-Based Approach to Classifying Summertime Potential Vorticity Streamers Linked to Rossby Wave Breaking in the North Atlantic Basin

41. **Pyle et al. (2004):** A Diagnostic Study of Jet Streaks: Kinematic Signatures and Relationship to Coherent Tropopause Disturbances

42. **Röthlisberger et al. (2019):** Recurrent Synoptic-Scale Rossby Wave Patterns and Their Effect on the Persistence of Cold and Hot Spells

43. **Sanders and Bosart (1985):** Mesoscale Structure in the Megalopolitan Snowstorm of 11–12 February 1983. Part I: Frontogenetical Forcing and Symmetric Instability

44. **Schultz et al. (1997):** The 1993 Superstorm Cold Surge: Frontal Structure, Gap Flow, and Tropical Impact

45. **Schultz et al. (1998):** The Effect of Large-Scale Flow on Low-Level Frontal Structure and Evolution in Midlatitude Cyclones

46. **Winters et al. (2019):** The Development of the North Pacific Jet Phase Diagram as an Objective Tool to Monitor the State and Forecast Skill of the Upper-Tropospheric Flow Pattern

47. **Winters et al. (2020a):** Composite Vertical-Motion Patterns near North American Polar–Subtropical Jet Superposition Events

48. **Winters et al. (2020b):** Composite Synoptic-Scale Environments Conducive to North American Polar–Subtropical Jet Superposition Events

---

## Author Response (AR1)

*Paper egusphere-2023-2678*

**The importance of diabatic processes for the dynamics of synoptic-scale extratropical weather systems—a review**

by Heini Wernli and Suzanne L. Gray

***Replies to the reviewers' comments***

We are most grateful to the reviewers for reading this extensive review paper and for their thoughtful and constructive comments that help us to further improve the manuscript. Based on the reviewers' suggestions, we have implemented several changes in the manuscript. The main changes are that:

- We have written a new subsection of Sect. 5 on "Consideration of radiative and surface flux related diabatic processes" (Sect. 5.2) in which we have included studies on the effects on extratropical cyclones of radiative and surface flux processes including surface friction and the effects of sea surface temperatures. This subsection was added in response to comments from reviewers 2, 3, and 4, who all argued for the addition of extra material on these topics. We also made the appropriate adjustments to the schematic figure showing the article structure (Fig. 1) and associated text.

- We have revised some of the figures, particularly those from the older papers to try to increase their clarity (through increasing their size by rearranging the panels and adding some elements of colour where possible) as well as adding more description of these figures and their importance in the captions and corresponding text. We have included one additional figure illustrating how different variables evolve along warm conveyor belt trajectories (Fig. 9). In addition, there are two new figures in response to reviewers' suggestions, one visualising PV inversion (Fig. 3) and one with important figures accompanying the new subsection mentioned above (Fig. 15). Also, we have carefully considered the suggestion to include our own schematics in the paper (and indeed this is something that we had considered when writing the review). However, while we liked the idea, we struggled with its realisation. It is very difficult to do justice to the complexity of the theme with a few schematics. Instead, we have added the schematic Fig. 1 from Schäfler et al. (2016) in Fig. 18 as well as several other new schematics in the new figures and two added panels in what is now Fig. 17. Note that we emphasised the general value of schematics in the synthesis section (L3014 of the original manuscript). In this section we have now also cross-referenced to the 13 schematic figures now included in our paper.

- We greatly appreciate all of the suggested additional papers to include in our review. We have considered each of these and have included those that fall within the scope of our review (we have added comments on many of the suggestions below). When writing this review, we were very conscious to define the limits of the topics that fell within scope to avoid this review becoming a far larger review of extratropical cyclones in general (scope was a topic we discussed many times), and we have applied those same limits in considering these suggested additional papers.

Below we provide a one-to-one response to all points raised by the reviewers. The reviewers' comments are in black and our replies in blue.

Recommendation: minor revisions

This paper presents a comprehensive review of the role of diabatic processes in the development and structure of weather systems, as envisaged from the nineteenth century to the present day. This is a complex topic, as the physics is dominated by small to mesoscales and there is so much variability between cyclones that it took a long time for meteorologists to agree on the nature and importance of diabatic processes compared to the mathematically elegant theories of dry baroclinic instability. Much progress on this topic has been achieved over the past twenty years and this review is timely and welcome. It should be published with minor corrections – but because of its scope and length there are a lot of these.

Many thanks for your positive overall assessment of our review article.

This is not, and should not be, a comprehensive review of every paper ever published on diabatic processes in cyclones, as the authors make clear in the Introduction. It will be most valuable as a summary of these distinguished authors' own understanding of the topic, and of the papers that led them to that understanding, as this will give the review a coherence that those unfamiliar with the field will most appreciate. Of course, much of this already comes through strongly in the manuscript and my comments will mainly be aimed at improving the flow and readability of the paper.

Many thanks for your complimentary comments on our understanding and suggestions that help us to improve the readability of the paper.

I liked the structure of the review sections, with 'pauses for breath' or summaries at the end of each subsection. As my detailed comments will show, I think some of these could be developed further to provide a synthesis of the science that was revealed by the papers, rather than just reiterating what they did. Although I will concentrate here on sections 5 and 6 (following the Editor's request), these comments actually apply most strongly to the earlier sections where the evidence presented was sometimes contradictory, making it difficult to see what advances in the science actually occurred.

Thank you for your positive general remarks about the brief summary sections. We critically reviewed and where possible improved them such that they are most useful to the reader. In particular, we make sure that these summaries indeed provide "a synthesis of the science" and highlight the advances that actually occurred. However, we don't want to eliminate the fact the some results were contradictory – we think that it is an essential message of this review that complex research fields like the one reviewed here do not evolve linearly but rather in parallel strands, with sometimes contradicting results and based on completely different approaches such that it takes a lot of time and further research to establish links and elaborate the consensus between these different strands.

Comments on section 5

The advent of reanalyses has indeed been a game-changer for meteorological research generally, not just for this topic, and amply merits detailed discussion in 5.1. The section is informative and reads well, but the Summary is perfunctory and adds little. What is needed here is a summary of what the reanalysis papers found e.g. regarding the distribution among cyclones of the importance of diabatic heating, its possible added relevance to the more extreme cyclones, the link to 'atmospheric rivers' etc. How did this approach advance the science?

We slightly extended and improved this summary; however, it already contained two very important points, that reanalyses allowed going beyond case studies and finding statistically robust signals, and that they led to objective cyclone classifications. (The link to atmospheric rivers is, in our view, not primarily established because of reanalysis data, but again reanalyses helped to establish robust climatologies of atmospheric rivers and how they relate to cyclones etc.)

Section 5.2 discusses 'diabatic processes in (special categories of) extratropical cyclones', presenting seven loosely-connected subsections beginning with cyclone classification.

i. I recommend that the authors reconsider the order of their subsections, moving extratropical transitions to the penultimate slot with a more natural transition to tropopause level vortices. The transition from Type C cyclones to subtropical and then Mediterranean cyclones would then be smoother.

Thank you for this suggestion to improve the order of the subsections, which we have implemented.

ii. Section 5.2.1 would make more sense to a reader unfamiliar with cyclone categorisation if the authors explained what the Petterssen and Smebye A/B scheme actually is before launching into a type C discussion.

We have added a brief explanation about types A and B.

iii. The Mediterranean section should be shortened, concentrating on distinctive properties of these cyclones, other than where they occur (e.g. paragraph 1736-1746 could be omitted, and I'm not sure what the last paragraph, on moisture sources, is adding to the science).

We think that this section gained context and readability due to the inclusion of new section 5.2 about friction and surface heat fluxes. Several Mediterranean cyclone studies investigated these aspects. We also think that the paragraph on sensitivity studies is useful, as well as the paragraph about moisture sources, which relates to the discussion in the new Sect. 5.2.3. However, we rephrased and slightly shortened the last paragraph of this section to establish a better link with the moisture transport discussion in Sect. 5.2.3.

iv. It is appropriate to include polar lows in this review because of the contribution of convection to many of them, but the section could do with editing to make the key points

clearer. I suggest that the text from 1815 to 1821 be removed as it lapses into jargon inconsistent with the rest of the section and detracts from the theme of diabatic heating. Likewise, the paragraph 1843-1858 goes into a level of detail not required here, given the existence of reviews specifically of polar lows.

We consider these studies in lines 1815-1821 (in the original submission) discussing the relative importance of baroclinic and convective processes to polar low development are important to include. However, we have expanded on the terminology explaining that the geostrophic momentum model is an idealised moist nonlinear model and adding a footnote to explain the interpretation of CISK for warm-core vortices. Regarding the comment on reviews of polar lows, although the review by Moreno-Ibáñez et al. (2021) included some discussion of potential vorticity (the topic of paragraph L1843-1858) that review doesn't discuss diabatically generated PV, and we also felt it was important to re-evaluate the relevant cited papers to better link with our review. Note that we have now also included a reference to the "update" review of Moreno-Ibáñez et al. (2024).

v. To my mind Diabatic Rossby Waves are of a different order of importance to the other subsections here, as this is a distinctive dynamical process in its own right. Could this become section 5.2, then section 5.3 would include the other subsections? I leave this to the authors' discretion but it would allow for mention of DRWs in the sections that currently precede it, and better overall coherence.

We don't think that DRWs are of higher importance than the cyclone types described in the other subsections and we prefer to leave the subsections as they are. We understand the reviewer's remark that presenting DRWs earlier would make things easier in some other subsections, but every order has its particular pros and cons, and we think that the overall coherence is still fine.

vi. The summary subsection 5.2.8 is appropriate to this section.

Thanks.

Section 5.3 discusses novel diagnostics of diabatic PV modification. Again the subsections are appropriate but could benefit from some critical editing.

i. The first paragraph of 5.3.1 is too detailed – readers should consult the original papers for the detailed methodology – while the second paragraph could benefit from more examples of results obtained from PV tracer analysis.

We have shortened the detailed methodology slightly although we consider it important to get across the limitations of this tool (as well as its benefits). We have also added some more PV tracer and PV tendency results (including adding two new panels in the figure included in this section), in particular describing how these studies have helped to demonstrate the relative importance of different diabatic and frictional contributions to the PV budget.

ii. The final paragraph of 5.3.2 doesn't lead anywhere – did Büeler and Pfahl find anything useful from their study? If not, this paragraph could be deleted.

We have added a little more detail about what was found in this study. Our original idea of Sect. 5.3 (now Sect. 5.4) was to focus on the description of these different diagnostics rather than on the results of applying them, but we now embed more results throughout this section (as also indicated in our response to the comment above) and changed the title of Sect. 5.3. to "Novel diagnostics of diabatic processes".

iii. On line 2175 we are cautioned that 'caution needs to be applied when inferring dynamical causation from ensemble sensitivity analysis' yet in the very next paragraph the word 'sensitivity' is used instead of 'association' three times! These paragraphs need to be consistent with each other.

The reviewer makes a valid point here. We have used the term "sensitivity" following the terminology in the cited papers, which describe the diagnostics plotted as sensitivity diagnostics and discuss the implied sensitivity. It would confuse readers who go from our review article to the papers if we refer instead to association. Instead, we have clarified that despite the caution that must be applied when using the Ensemble Sensitivity Analysis technique as it merely diagnoses an association; causation can be inferred when there is a plausible physical link or mechanism between the precursor field and response function and we have stated that we have chosen to use the term "sensitivity" for consistency with the terminology used in the cited papers that follow.

iv. Has the adjoint technique led to any new insights into diabatic processes? The result that 'forecasts of high-impact cyclones were found to be strongly sensitive to low to mid-tropospheric moisture in the initial state' is hardly novel. Section 5.3.3 is one which could be considerably shortened.

We have reviewed this text and further highlighted the relationship between ensemble sensitivity analysis and the adjoint technique. We agree that this section is long but the ensemble sensitivity and adjoint techniques require a reasonable amount of text to explain, even simply. Studies using the adjoint technique are beginning to appear in the literature and so we consider it important to mention this technique even though we agree with the reviewer that there have been limited new insights related to our review as yet. However, in response to the reviewer's suggestion, we have removed one of the three paragraphs related to the adjoint technique, reducing the associated text by about a third.

v. The summary is again appropriate.

Thanks.

Section 5.4 concentrates on the impact of diabatic processes on the dynamics at tropopause level, through the outflow of WCBs and tropical cyclones. I thought this was balanced and

coherent, with an informative summary, and but for a couple of minor comments (see separate section below) I have no major problems with it.

Many thanks.

Section 5.5 describes the two field campaigns DIAMET and NAWDEX which the authors consider to be the only two experiments of note since 2000 to study diabatic processes in cyclones.

i. Given that extratropical transitions fall into the domain of this review, mention should also be made of T-PARC (2008).

Thank you for this suggestion. We included T-PARC, which was the basis of, e.g., the important study by Schäfler and Harnisch (2015), in Table 1 and briefly mention it in Sect. 5.5.

ii. The concept of the sting jet arose from analysis by Browning of the Great Storm that struck Southern England in 1987 and was developed by diagnosis of high-resolution models by Clark et al. This paper needs to explain more clearly what the DIAMET measurements contributed, for example by explaining what figure 16b is supposed to show. It also needs to acknowledge that the sting jet (defined as a descending airstream) is a transient phenomenon, especially when compared to the cold conveyor belt which dominates the low-level wind field in the southern quadrant of a mature cyclone.

We acknowledge that although we did say that the descent lasts only a few hours we didn't explicitly compare the timescales of the sting jet and cold conveyor belt jet. We have added "The transient nature of the sting jet thus contrasts with the typically much longer-lived wind jets associated with the cold and warm conveyor belts (as illustrated in the windstorm conceptual schematic in Hewson and Neu (2015), their Fig. 1)." and included more description of Fig. 16b, which is now Fig. 19b, (both in the caption and text).

iii. Although 5.5.2 is entitled 'Embedded convection', 'Embedded convection and negative PV bands' would better describe the content.

We thank the reviewer for the suggestion and have changed the heading accordingly (now Sect. 5.6.2).

Section 5 concludes with a discussion of the relevance of diabatic heating in cyclones to climate change research. On the face of it, a warmer climate will mean more moisture in the atmosphere and more scope for diabatic heating. But, as 5.6.2 shows, the problem is far from linear and the location of diabatic heating relative to the cyclone centre is critical when considering its effect on the cyclone. It appears that in a warmer climate the PV source region will be further from the cyclone centre on average – but the tail of the distribution, where the two line up, may result in a few very powerful windstorms. Very interesting!

Many thanks, writing this section was not easy; we are glad that you found it interesting.

The long discussion in 5.6.3 of the effect of model resolution on the diabatic effects on cyclones in GCMs could be shortened considerably, as the details are covered in many other papers. This review could simply summarise the conclusion of these studies, i.e. a short introductory paragraph then pick up at line 2847.

We considered shortening the text slightly. However, we think that a substantial discussion of model resolution is important for at least two reasons: 1) Resolution of model simulations and vertical and temporal resolution of simulation output are relevant prerequisites when diagnosing the structure and evolution of weather systems and the involved physical processes in numerical simulations. The fact that climate simulations have typically a coarser grid (compared to simulations used in studies on weather dynamics) and typically output a far more limited selection of fields than weather simulations has been a hindrance for applying sophisticated diagnostics about diabatic processes, as those discussed in Sect. 5.4 (formerly 5.3), to these simulations (as also mentioned in Sect. 6.2). 2) Because of this hindrance and the resulting scarcity of studies investigating weather system dynamics in climate simulations, there were limited possibilities to evaluate how well climate models capture these important processes. We therefore think that to address the important question of how global warming will affect weather system dynamics, the community must overcome the data issue, invest substantial efforts in applying clever diagnostics to climate simulations (see also point 7 in our outlook Sect. 6.2), and document the sensitivity of the results to model resolution. The first three paragraphs of Sect. 5.7.3 (formerly 5.6.3) shall contribute to this.

Although the summary of this section is informative, I recommend that it be expanded by a few sentences to cover the issues of propagation and blocking discussed in the text.

We prefer not to repeat this brief discussion in the summary, mainly because otherwise, a similar short expansion would be required for other specific types of weather systems like jet streams, upper-level cutoff cyclones, etc. Instead, the summary is written mainly in generic terms, and key points mentioned (overall challenge, need for combined approaches, regional dependency) in principle apply to any weather system under consideration.

Section 6 provides a summary of the basic concepts presented in the review and looks to the future of the field. It reads well and I have no major comments on it.

Many thanks.

Minor comments

l. 1496 millenium

Corrected.

2. l.1587 Either omit 'only' or restructure to '….revealed that only in the NH winter are cyclones usually accompanied by a strong WCB….', depending whether the sentence is meant to contrast winter and summer as well as the two hemispheres.

The sentence is actually correct as written. Northern Hemisphere cyclones are only associated with strong WCBs in the winter and, as stated in the following sentence in our review, Southern Hemisphere cyclones were found to not often be associated with WCBs. To improve clarity the sentence as been edited to read "Combining feature-based climatologies of cyclones and WCBs, Eckhardt et al. (2004) revealed that, considering both hemispheres and the summer and winter seasons, only cyclones in the Northern Hemisphere winter are usually accompanied by a strong WCB…".

3. l. 1603 The idea of a cyclone with no WCB is most likely an artifact of the definition assumed for a WCB in Binder et al.'s study (or a problem with their method) than a reflection of the dynamics of the cyclone. Read literally it means there was no ascending airstream ahead of the cyclone, which is hard to square with explosive development. The idea is counter to the whole thesis of this review and requires more discussion if considered to be a real result.

Thank you. Here "no WCB" means no WCB as defined in the many trajectory studies since Wernli and Davies (1997), i.e., with a threshold criterion for ascent. Therefore, "no WCB" does not mean "no ascent from the warm sector" but that the ascent is not strong enough to meet the criterion. We have changed the sentence to read "However, the results by Binder et al. (2016) showed that variability is large, and a minority of explosive cyclones have no WCB according to the trajectory ascent threshold set…".

4. l. 1635. Please explain what the Petterssen and Smebye A and B cyclones actually are. The section is difficult to follow for someone not versed in cyclone classification because the story starts in the middle.

See above, we have added a brief explanation of types A and B.

5. l.1851 lesser

Corrected.

6. l.2330-1 Isn't downstream development a consequence of Rossby wave propagation? The underlying dynamics are the same so this sentence needs to be re-phrased.

The study by Röthlisberger et al. (2018) identified so-called local Rossby wave initiation events that are not related to Rossby wave propagation from upstream. We agree that downstream development is related to Rossby wave propagation, but Rossby wave initiation can be independent from downstream propagation. No changes were made to the text.

7. l.2420-2425 Which simulation corresponded best to reality in this case?

There is no simple answer to this question. According to Rivière et al. (2021), one of the simulations with parameterised convection (B85) performs better in the representation of the double jet structure at 1 d lead time than the other two simulations, which was attributed to the more active WCB at upper levels. However, this effect is too strong and the simulation B85 becomes less realistic at lead times beyond 1.5 d. We decided to not discuss these details in the manuscript.

8. l. 2727 isn't the argument that, basically, the same cyclone in a warmer climate will produce more precipitation just because it is warmer (the 'Clausius-Clapeyron effect')?

This would be the argument for a single cyclone with a given intensity (e.g., in terms of minimum SLP) in a warmer climate. But here, when writing about "cyclone-related precipitation" we meant the contribution of cyclones to the climatological precipitation in a warmer climate. And when addressing this question, considering the CC effect is not sufficient, one must also consider how the frequency, intensity and propagation of cyclones changes due to warming. We clarified the formulation.

9. l.2979. It is provocative to claim, without evidence, that the field campaigns were a direct result of a handful of high-priority storms. For example, FASTEX was not organised as a response to the Great October Storm and DIAMET (according to its description earlier) was not organised as a response to the discovery of sting jets.

We agree that the field campaigns should not be considered a "direct result" of the catalyst cyclones and therefore we wrote that "It would be too far stretched to claim that the catalyst cyclones directly triggered the planning of field experiments" but we still think that they played a very important role in engaging the research community for these large campaigns. No changes were made to the text.

10. l.2995 The authors have chosen not to say much about satellite measurements in this review, despite the crucial role they played in the thinking of key figures in the field, such as Keith Browning. That is their prerogative, though still an omission. But one of the key conceptual tools of the satellite era was time-lapse videos from geostationary satellites, which actually show how the WCB develops alongside the cyclone and the cold front, and that should be mentioned here. It is not speculation to point out the key role of satellite images in developing understanding, and that word should be removed. So also the 'real-time availability', so important for forecasting but not for research where the better-quality images available after the event (especially in the 1980s) were more useful.

Thank you for this important remark. We have removed "speculate" in what is now L3302 (changing the text to read "It is evident that") as well as "real-time" and we mentioned the importance of time-lapse videos from satellite and radar for investigating cloud and precipitation processes in weather systems.

Recommendation: minor revisions

I enjoyed reading through this review. I really appreciate all of the work done by the authors to bring all of this information together and organise it in a coherent manner.

Many thanks for your very positive overall assessment of our review!

I was asked by Dr. Harnik to focus my review on Section 5. So that is what I have done.

All of my comments are minor comments. I have some additional papers that you might consider citing. I have listed them all at the end to make things more organised.

Line 1522: You mention an early reanalysis product produced by NASA for 1985– 1989, and then reference Schultz and Mass, 1993. Are you sure that is the reference you want there?

Many thanks for spotting this mistake. The correct reference here is Schubert et al. (1993): Schubert, S. D., R. Rood, and J. Pfaendtner, 1993. An assimilated dataset for Earth science applications. *Bulletin of the American Meteorological Society* **74**: 2331–2342.

Lines 1525 – 1533: I agree with you about the utility of reanalysis products, but they also have potential biases, especially in moisture processes. This seems like a good place to mention this. I believe that most of the results discussed in this review that utilise reanalysis are valid – or at least qualitatively correct at the synoptic scale. However, I also think the biases in precipitation rates and clouds in reanalyses give a reason to keep some open mind to the possibility that the reanalyses do not provide a complete and perfect picture of the physics.

Naud, C. M., J. Jeyaratnam, J. F. Booth, M. Zhao, and A. Gettelman, 2020: Evaluation of Modeled Precipitation in Oceanic Extratropical Cyclones Using IMERG. J. Climate, 33, 95–113, https://doi.org/10.1175/JCLI-D-19-0369.1

McErlich, C., McDonald, A., Renwick, J., & Schuddeboom, A. (2023). An assessment of Southern Hemisphere extratropical cyclones in ERA5 using WindSat. Journal of Geophysical Research: Atmospheres, 128, e2023JD038554. https://doi.org/10.1029/2023JD038554

We agree with your comment that reanalyses have biases (and we mention this now briefly in Sect. 5.1). However, we do not want to include a discussion of these biases and of comparisons with remote sensing data, as this would be a topic on its own for another review article.

Line 1575: I suggest adding a sentence or two regarding the fact that the relationship between the precipitation and cyclone intensity is also subject to details about the upper-level forcing and the latitude of the cyclone. Two papers that might be relevant here:

Booth, J. F., Naud, C. M., & J. Jeyaratnam, 2018: Extratropical cyclone precipitation life cycles: A satellite-based analysis. Geophysical Research Letters, 45, 8647-8654. https://doi.org/10.1029/2018GL078977

Sinclair, V. A., and J. L. Catto, 2023: The relationship between extra-tropical cyclone intensity and precipitation in idealised current and future climates, - Weather and Climate Dynamics.

Thank you; the 2$^{nd}$ paper is relevant for the theme of our review and has been included; however, it fitted even better to the discussion in Sect. 5.7.2 (formerly 5.6.2).

Line 1617: Section 5.2. Let me start by saying that I think this review article is already 100% sufficient. It is quite long as well. So, I am giving the following recommendation with caution. I will agree with the authors if they choose to ignore this suggestion. I wonder if you might consider adding a subsection in 5.2 on more recent work on energy fluxes from the surface and their role in the diabatic forcing of ETCs.

Many thanks for this suggestion, which resonates with comments from reviewers 3 and 4. We therefore decided to include an additional subsection (Sect. 5.2) about diabatic processes not related to latent heating in clouds. This subsection, entitled "Consideration of radiative and surface flux related diabatic processes", included discussions about the role of radiation, turbulence, and surface heat fluxes.

The most recent work that I know of on the topic is this one:

Demirdjian, R., J. D. Doyle, P. M. Finocchio, and C. A. Reynolds, 2023: Preconditioning and Intensification of Upstream Extratropical Cyclones through Surface Fluxes. J. Atmos. Sci., 80, 1499–1517, https://doi.org/10.1175/JAS-D-22-0251.1.

Within that work, the authors will see that there have been other relatively recent works focused on low-level diabatic heating and the associated surface fluxes. As I said though, I would be completely fine if the authors decided that this work is outside of the scope of their review.

Thank you, we included this and some other important recent studies on the subject in the new Sect. 5.2.

Line 2810 or elsewhere in Section 5.6.3: You should consider referring to Hawcroft et al. 2017.

Hawcroft, M. K., H. Dacre, R. Forbes, K. Hodges, L. Shaffrey, and T. Stein, 2017: Using satellite and reanalysis data to evaluate the representation of latent heating in extratropical cyclones in a climate model. Climate Dyn., 48, 2255–2278, https://doi.org/10.1007/s00382-016-3204-6.

Very good suggestion, thanks. However, we decided to include a reference to this paper in Sect. 5.1, where we briefly mention cloud biases in reanalysis data.

Recommendation: minor revisions

General comments:

This article reviews the historical progress and current understanding for the effect of diabatic heating on the extratropical weather systems. In the section 2, several kinds of theoretical framework to understand the effect of diabatic are described. Historical overview from 19th century for the studies for the impact of diabatic heating on the intensification of cyclones is described in the section 3. Then, the detailed results of studies for the diabatic effects on cyclones are reviewed in section 4. I really enjoyed reading the article, and I believe that this article is highly valuable for readers to know the historical overview and current understanding for the diabatic effects in extratropical weather systems. Overall, the article is well structured and is easy to read. Thus, I recommend that this article can be published, although I have just several recommendations to add references for the statistical study of cyclones in the Pacific.

Many thanks for your very positive overall assessment of our review! And many thanks for your suggestions to add more references about North Pacific cyclones.

Specific comments:

The descriptions for explosively developing cyclones in Pacific is relatively low compared with those in Atlantic. More descriptions and references may be added in the section 4.1.4. e.g.,
Yoshida, A., and Y. Asuma, 2004: Structures and environment of explosively developing extratropical cyclones in the northwestern Pacific region. Mon. Wea. Rev., 132, 1121–1142. https://doi.org/10.1175/1520-0493(2004)132<1121:SAEOED>2.0.CO;2
Zhang, S., G. Fu, C., Lu, and J., Liu, 2017: Characteristics of Explosive Cyclones over the Northern Pacific, J. Appl. Meteor. and Clim., 56(12), 3187-3210. https://doi.org/10.1175/JAMC-D-16-0330.1

A brief summary of the first of these papers by Yoshida and Asuma (2004) was already included in the original submission in Sect. 5.1, but due to a mistake on our side with the wrong reference (Yoshizaki et al. 2004 instead of Yoshida and Asuma 2004). This mistake was corrected. The paper by Zhang et al. (2017) does not explicitly discuss diabatic processes and is therefore out of scope for this review article.

The adding of descriptions for recent studies for the effects of SST front and associated surface fluxes on the structures and dynamics of extratropical cyclones would be useful for readers if it is possible. e.g.,
Bui, H., & Spengler, T. (2021). On the influence of sea surface temperature distributions on the development of extratropical cyclones. Journal of the Atmospheric Sciences, 78(4), 1173-1188.

Demirdjian, R., Doyle, J. D., Finocchio, P. M., & Reynolds, C. A. (2022). On the influence of surface latent heat fluxes on idealized extratropical cyclones. Journal of the Atmospheric Sciences, 79(9), 2229-2242.

Hirata, H., Kawamura, R., Kato, M., & Shinoda, T. (2015). Influential role of moisture supply from the Kuroshio/Kuroshio Extension in the rapid development of an extratropical cyclone. Monthly Weather Review, 143(10), 4126-4144.

Reeder, M. J., Spengler, T., & Spensberger, C. (2021). The effect of sea surface temperature fronts on atmospheric frontogenesis. Journal of the Atmospheric Sciences, 78(6), 1753-1771.

Tochimoto, E., & Niino, H. (2022). Comparing frontal structures of extratropical cyclones in the northwestern Pacific and northwestern Atlantic storm tracks. Monthly Weather Review, 150(2), 369-392.

Tsopouridis, L., Spensberger, C., Spengler, T., 2021a: Characteristics of cyclones following different pathways in the Gulf Stream region. Quart. J. Roy. Meteor. Soc., 147, 392–407. https://doi.org/10.1002/qj.3924

——, Spensberger, C., Spengler, T., 2021b: Cyclone intensification in the Kuroshio region and its relation to the sea surface temperature front and upper-level forcing. Quart. J. Roy. Meteor. Soc., 147, 485–500. https://doi.org/10.1002/qj.3929

Many thanks for this suggestion, which resonates with comments from reviewers 2 and 4. We therefore decided to include an additional subsection in Sect. 5 about diabatic processes not related to latent heating in clouds. This subsection includes discussions about the role of radiation, turbulence, and surface heat fluxes (and the role of SST). The new subsection, entitled "Consideration of radiative and surface flux related diabatic processes", references several of the studies listed by the reviewer.

Reviewer 4

Recommendation: Major revisions

General Comments:
The authors have put a lot of effort in summarising and reviewing the evolution of our thinking of the effects of diabatic processes on synoptic-scale cyclone development. As usual, with such attempts, it is difficult to do the entire field of research justice and certain selections must be made by the authors in terms of focus. While the review is quite extensive, some recent literature and arguments have not been included thus far. Furthermore, potentially related to the background and work of the two authors, the manuscript focuses mainly on PV diagnostics based on Lagrangian, tracers, or other direct diagnostics of PV. Hence, the title might be a bit misleading and should potentially reflect this choice.

We regard the title of the paper as appropriate, even if we agree that the paper has a focus on studies using PV diagnostics. Section 5.4 also discusses several new diagnostics that are not PV focused (we therefore changed the title of this section to "Novel diagnostics of diabatic processes"). Many other parts of the paper do not use PV concepts at all (e.g., large parts in Sect. 4.2, 4.3 and 5.7), or in other words, important parts of the paper would be out of scope if the PV focus was included in the title.

In addition, the manuscript reads more like an extensive summary of research conducted the last decades and one and a half centuries but does not necessarily do justice to the term "review". Review, to me, would imply that the authors go beyond summarising the material and provide a more detailed discussion of the different methods, their pros and cons, as well as if the community along the way concluded on the adequacy, or lack thereof, of different concepts, diagnostics, and theories. Such a review of diagnostics, concepts, theories would be highly desirable. For example, most scientists would probably not anymore believe that CISC is really an appropriate concept for moist-baroclinic development.

In our view there are several ways of writing "good", i.e., interesting and useful review papers and we fully acknowledge that with our more "descriptive and narrative style" we cannot do justice to those who expect a more "opinionated and judging style". We think that we included quite a bit about the pros and cons of the methods (e.g., in Sect. 4.2.1) and in the revised version of the paper, we considered adding some more pros and cons where appropriate. The point raised about whether "the community along the way concluded on the adequacy, or lack thereof, of different concepts, diagnostics, and theories" is an interesting one but difficult to address, mainly for two reasons: (i) it is risky to judge on this in hindsight (we read papers from 40 years ago with the knowledge obtained since then, which does not provide an objective view on what the community thought 40 years ago), and (ii) as we try to emphasise throughout the paper, this field profited from very complementary and diverse approaches, and it took decades for the community to see common elements of, for instance, highly idealised theoretical studies (e.g., Sect. 4.3.1) and observations from field campaigns (e.g., Sect. 5.6). Also, because of the broad support of the five other reviewers, we are confident that this style of a review paper also has its merits.

The authors provide a general introduction to latent heating, which is maybe not needed given the audience. There is also a rather lengthy introduction of symmetric instability, which is potentially also beyond the focus of this review. In general, the part on slantwise convection seemed a bit out of place given the focus is mostly on synoptic scales.

The section on slantwise moist convection, moist PV and SCAPE is about three pages long and the concept of conditional symmetric instability (CSI, the release of which often leads to slantwise convection) comes up in several places in the review: particularly in Sect. 4.1.2 on "Further explosive cyclone cases", Sect. 4.2.3 on the "PV perspective on moist vs. dry experiments", Sect. 5.1 on "Reanalyses and weather system climatologies", and Sect. 5.6.1 on "Sting jets". Hence, we argue that an introduction to this instability is necessary in our review. As this instability is somewhat complicated to explain, partly due to its link with other mesoscale and convective scale instabilities, the text needed to explain it at the level needed for a reader who is new to these instabilities is relatively substantial (even when linking, as we do, to reviews and textbook explanations).

Given the authors' background, their almost exclusive focus on PV is understandable, though there have also been other attempts to understand the effects of diabatic heating on the synoptic evolution that remain unmentioned. For example, the diagnostic of the tendency of baroclinicity (Papritz and Spengler, 2015), which has subsequently been used to postulate the hypothesis that cyclone clusters are formed diabatically (Weijenborg and Spengler, 2021) is not mentioned. This concept does neither rely on neglecting certain parts of the flow (PV only deals with the circulation on theta surfaces) nor on having to separate into a basic state and anomaly, while it clearly demonstrates the workings of diabatic forcing on the synoptic, and even larger-scale, evolution.

The study by Papritz and Spengler (2015), introducing a novel approach to diagnose diabatic effects on baroclinicity, is now referenced in Sect. 5.4.4 (formerly 5.3.4), together with the application study of Weijenborg and Spengler (2020).

When the authors introduce the concept of PV in 2.1, the discussion lacks a discussion of the disadvantages of the usage of PV. For example, equation (1) highlights the fact that only that part of the circulation that is perpendicular to the gradient of theta is represented by PV, i.e., only the circulation within a theta surface.

We are not sure why this is a "disadvantage". It reflects the fact that the balanced flow is determined by a quantity we call PV (and suitable boundary conditions). The omega equation then helps to obtain information about vertical motion.

All other parts of the circulation are neglected and can thus also not be addressed by the PV tendency in equation (3). Examples for such kind of circulations would be a sea-breeze, or frontal circulations. Of course, in a QG framework, these would be regarded as ageostrophic and thus potentially of lower relevance, but the implications and limitations of using a PV framework should be more clearly stated. The formulation in equations (5)-(7) could have been

related to the circulation theorem, which yields the same result (implied impermeability), when only choosing that part of the circulation that projects on a theta surface.

Thanks for sharing these views about the PV framework. When we mention the usefulness of the PV concept, then we don't regard PV in isolation. Rather, e.g., in a QG framework, we mean that PV determines the geostrophic flow, which in turn determines the ageostrophic flow. For the context of this review article, this conceptual framework still holds, with the additional (complicated) element that diabatic processes can create or destroy PV, and in this way modify the flow. We verified in our brief summary of the PV concept in Sect. 2.1 that it does not convey the wrong impression that PV directly represents the 3-dimensional flow, and we discuss more about PV inversion along the new Fig. 3 included in Sect. 2.1..

With respect to the interpretation of the detected tendencies and changes in PV (e.g., section 5.3.2 and others), one should be aware that to understand the system and its evolution, it is not sufficient to merely quantify changes in PV. For example, while evaporation below an area of latent heating might yield a stronger PV anomaly at the interface between these differently signed diabatic forcing, the actual effect on the evolution of the cyclone might still be detrimental (Haualand and Spengler, 2019). Thus, the discussion of diagnosed anomalies should be put in more context to the overall development when presenting the derived fields, trajectories, and tracers.

Thank you. We fully agree with the reviewer that understanding how diabatic processes affect individual PV anomalies is not enough to eventually understand their effect on weather systems and their evolution. For this, the mutual interaction of PV anomalies, which is strongly influenced by their location, size and aspect ratio, is key. But in order to study the interaction of PV anomalies, an essential step is to understand how they are formed and modified, which is discussed in detail in the paper. However, we also explicitly discuss examples that go against the first order assumption that stronger positive PV anomalies imply a stronger cyclone. For example, on L743 in the original submission we discuss that the indirect effects of a positive PV anomaly generated by surface friction led to a weaker cyclone (counter to the stronger cyclone expected from the inversion of the positive anomaly) in the Stoelinga (1996) paper, and in Sect. 5.4.2 (formerly 5.3.2) we discuss mechanisms for this from more recent studies. In the revised manuscript, we emphasise more broadly that the impact of PV anomalies on cyclones cannot be inferred simply from the sign of the PV anomaly, particularly in our new Sect. 5.2 on "Consideration of radiative and surface flux related diabatic processes" and also in Sect. 2.1, where we briefly discuss the mechanism of PV shielding (with the new Fig. 3e).

I am admittedly not too familiar with all the detailed historic developments in the field prior to the mid 20ies century and greatly appreciate the effort the authors put into providing a wider historic overview of the concepts of cyclone development. However, it feels strange that the work of the Bergen School of Meteorology is not mentioned when it comes to the concepts of the structure and airstreams in cyclones, as several seminal papers on the structure of cyclones came out of this school. Even though the naming of the airstreams and sectors of cyclones at the time was different, the cold and warm/moist airstreams were clearly depicted in their conceptual cyclone models. These contributions should be included in sections 3.1 and 3.2.

Many thanks, this is a very good point. We indeed overlooked the warm and cold air currents, e.g., in Fig. 6 of Bjerknes (1919) about "On the structure of moving cyclones". We are most happy to remedy this oversight and now discuss aspects of this paper in Sect. 3.2.

With respect to polar lows in section 5.2.5, Terpstra et al. (2015) stressed the role of diabatic processes, despite them often being argued to be small due to the low values of absolute available moisture. One of the main arguments of Terpstra et al. (2015) being that despite the diabatic heating being significantly smaller than in extratropical cyclones, the effect on the PV tendency is comparable to extratropical cyclones, because the vertical extent is also significantly reduced, thereby increasing the effect of the gradient of the heating on the PV tendency. Furthermore, Stoll et al. (2021) clearly classified polar lows as moist baroclinic cyclones in a recent climatological analysis, while clarifying that the genesis through hurricane-like processes is rather unlikely, due to the excessive amounts of baroclinicity and shear at the genesis time. In general, the notion of an upper-level PV anomaly must be applied with caution for polar lows, as the usual altitude chosen to detect these PV anomalies are well within the stratosphere and thereby not necessarily directly relatable to the development of the surface-based polar lows. Furthermore, these upper-level anomalies are often of much larger character than the developing polar low, also rendering a direct interaction questionable.

We already acknowledged the work by these authors in our review: "Of particular note is the idealised high-latitude moist baroclinic channel simulation study of Terpstra et al. (2015) in which they showed that a weak disturbance with a PV structure consistent with a shallow diabatic Rossby wave (former Sect. 5.2.6, now 5.3.4) is able to develop in the absence of an upper-level initial perturbation, surface fluxes, friction and radiation; however, they noted that additional forcing is likely required to produce more realistic polar low intensities". We had also included the earlier ERA-Interim based climatology of Stoll et al. (2018). We have now added the more recent paper by Stoll et al. (2021) as a recent addition to the debate over the relative importance of baroclinic and convective processes in polar low development as well as a recent review article by Moreno-Ibáñez (2024) (an update to her 2021 review) which includes recent studies such as Stoll et al. (2021) .

The surface pressure tendencies introduced at the beginning of 5.3.4 are significantly flawed (Spengler and Egger, 2009). A simple thought experiment directly reveals the false physical nature of the diagnostic. Assume a horizontally uniform atmosphere, equivalent to an atmospheric column with no horizontal advection, which is initially motionless and where a mid-tropospheric heating is applied. Note that all concepts are hydrostatic. In such a hydrostatic setup, the heating cannot result in a direct change of surface pressure, which can only be caused by horizontal mass rearrangements (secondary circulation), or precipitation (Spengler et al., 2011). However, the diagnostic introduced by the various authors in section 5.3.4 directly implies a "diabatic" surface pressure tendency, which is not physical and points to a significant flaw in the diagnostic. Such an attribution could at best be achieved in a balanced framework, such as QG, though then the challenge is to define suitable boundary conditions for the inversion that is difficult to prescribe a priori (Spengler and Egger, 2012).

We already had included reference to the reply by Knippertz et al. (2009) to the comment by Spengler and Egger (2009) in our submitted paper as we began what is now Section 5.4.4. (was Sect. 5.3.4) with "The surface pressure tendency equation as formulated by Knippertz and Fink (2008) and Knippertz et al. (2009) was extended to quantify the contribution of diabatic processes to extratropical cyclone development by Fink et al. (2012)...". In their reply to the comment, Knippertz and colleagues accepted that Spengler and Egger (2009) identified an issue that means that the magnitude and extent of the pressure fall diagnosed by the PTE is problematic in certain situations. They, however, argued that the overall interpretation of their original paper stands. In the later paper by this group of authors (Fink et al. 2012), the authors introduced an extended pressure tendency equation, in which changes of the geopotential at the top of the column, the effect of net evaporation minus precipitation on the mass in the column, and mass changes due to vertical changes in water vapour are considered. To increase the clarity we have slightly modified our introductory sentence to read "The surface pressure tendency equation as formulated by Knippertz and Fink (2008) and Knippertz et al. (2009) was extended, in response to Spengler and Egger (2009), by including changes at the top of the considered atmospheric column (at 100 hPa), and used to quantify the contribution of diabatic processes to extratropical cyclone development by Fink et al. (2012)..."

Related to predictability and the effects on the upper troposphere, a recent study highlighted the importance of diabatic heating compared to tropopause structure for initial error growth (Haualand and Spengler, 2021). Even more drastic changes, or implied initial errors, along the tropopause are easily dwarfed by the effects of misrepresenting latent heating.

Thanks for this suggestion. Indeed, the result about the importance of uncertainties in latent heating for baroclinic wave development, compared to uncertainties in the tropopause structure, quantified in an idealised simulation setup, are relevant for the theme of this review and have been included in Sect. 5.5.3 (formerly 5.4.3).

Given that the authors state that they "touch on the effects of […] surface fluxes, in particular where studies have contrasted the effects of these processes with the effects of latent heating", it is surprising that recent studies highlighting the direct and indirect (enhanced latent heat release due to latent heat fluxes) effects of surface fluxes are not mentioned (Haualand and Spengler, 2020; Bui and Spengler, 2021). Previous studies in general showed that surface sensible heat fluxes have a detrimental effect, as they reduce baroclinic structure in the cyclone, while these more recent studies emphasise that the additional latent heat release available due to the surface latent heat fluxes can easily dominate the cyclone development in a favourable way.

Thank you for emphasising the importance of surface heat fluxes. As mentioned above, we decided to include an additional subsection about diabatic processes not related to latent heating in clouds. This new subsection 5.2 includes discussions about the role of radiation, turbulence, and surface heat fluxes.

Line 636-647: It is not clear what this discussion of lee cyclogenesis has to do with the main topic of the review article, i.e., the importance of diabatic processes. Consider removing or putting in context.

We found it interesting that early studies about Alpine lee cyclones did not consider moist processes, in a period (1982-1998) when, e.g., North American cyclone research focused a lot on diabatic processes. We slightly rephrased this paragraph to better put it in context.

References:

Bui, H. and Spengler, T. (2021). On the Influence of Sea Surface Temperature Distributions on the Development of Extratropical Cyclones. J. Atmos. Sci., 78, 1173-1188, https://doi.org/10.1175/JAS-D-20-0137.1

Haualand, K. F., and Spengler, T. (2019). How Does Latent Cooling Affect Baroclinic Development in an Idealized Framework? J. Atmos. Sci., 76, 2701-2714, https://doi.org/10.1175/JAS-D-18-0372.1

Haualand, K. F., and Spengler, T. (2020). Direct and Indirect Effects of Surface Fluxes on Moist Baroclinic Development in an Idealized Framework. J. Atmos. Sci., 77, 3211-3225, https://doi.org/10.1175/JAS-D-19-0328.1

Haualand, K. F. and Spengler, T. (2021). Relative importance of tropopause structure and diabatic heating for baroclinic instability, Weather Clim. Dynam., 2, 695–712, https://doi.org/10.5194/wcd-2-695-2021

Papritz, L., and Spengler, T. (2015). Analysis of the slope of isentropic surfaces and its tendencies over the North Atlantic. Q. J. Roy. Met. Soc., 141, 3226-3238, https://doi.org/10.1002/qj.2605

Spengler, T., and Egger, J. (2009). Comments on "Dry-Season Precipitation in Tropical West Africa and Its Relation to Forcing from the Extratropics". 137, 3149-3150, https://doi.org/10.1175/2009MWR2942.1

Spengler, T., and Egger, J. (2011). How Does Rain Affect Surface Pressure in a One-Dimensional Framework? J. Atmos. Sci., 68, 347-360, https://doi.org/10.1175/2010JAS3582.1

Spengler, T., and Egger, J. (2012). Potential Vorticity Attribution and Causality. J. Atmos. Sci., 2600-2607, https://doi.org/10.1175/JAS-D-11-0313.1

Stoll, P. J., Spengler, T., Terpstra, A., and Graversen, R. G.: Polar lows – moist- baroclinic cyclones developing in four different vertical wind shear environments, Weather Clim. Dynam., 2, 19–36, https://doi.org/10.5194/wcd-2-19-2021

Terpstra, A., Spengler, T., and Moore, R. (2015). Idealised simulations of polar low development in an Arctic moist-baroclinic environment. Q. J. Roy, Met. Soc., 141, 1987-1996, https://doi.org/10.1002/qj.2507

Weijenborg, C., and Spengler, T. (2020). Diabatic heating as a pathway for cyclone clustering encompassing the extreme storm Dagmar. GRL, 47, e2019GL085777. https://doi.org/10.1029/2019GL085777

Recommendation: Accept with minor revision (lots of little things)
A potpourri of possible additional references is also appended.

Overview:
Wernli and Gray have produced a very valuable, highly informative, and an extensively documented research review paper on the importance of diabatic processes that govern the dynamics of synoptic-scale extratropical weather systems. This extensive research review paper will be both a very valuable addition to the refereed literature and a "must have" document for advanced graduate students and early career scientists who have strong research interests in synoptic-dynamic meteorology. Noteworthy attributes of this synoptic-dynamic meteorology review paper include: 1) a broad-based historical perspective on the scientific ideas that have governed the field synoptic-dynamic meteorology going back to the 19th century, 2) an assessment of the original thinking that resulted in critical new breakthroughs in the development of innovative fundamental science ideas that have driven the growth of synoptic-dynamic meteorology, 3) an overview of critical past and present research in synoptic-dynamic meteorology that has driven the field forward, and 4) an outlook for future new research opportunities going forward that could further broaden and deepen the field. This extensive and detailed synoptic-dynamic meteorology review paper is suitable for both advanced graduate students and early career scientists who possess a strong interest in synoptic-dynamic meteorology.

Bottom line: "The importance of diabatic processes for the dynamics of synoptic-scale extratropical weather systems—a review" is thorough and offers a comprehensive overview of the history of extratropical cyclone dynamics using a PV framework. The authors demonstrate a strong foundation of synoptic-scale extratropical dynamics and present a clear and accessible synthesis of the historical literature. The inclusion of recent references and an overall balanced discussion add significant value for the reader.

We are most grateful for this detailed and very positive overall assessment of our review and for the many constructive suggestions you and your PhD students made in the following comments!

Introduction:

Wernli and Gray write: "the main three aims of this review article are (i) to provide evidence that our understanding of how diabatic processes affect extratropical weather systems has grown considerably since the review article on PV by Hoskins et al. (1985) and the comprehensive book chapter on the rapid intensification of extratropical cyclones by Uccellini (1990), (ii) to portray in detail the historical evolution of a specific research field over several decades and thereby to exemplify how scientific progress results from the combination and integration of complementary research approaches, and (iii) to promote the relevance of this research area in dynamical meteorology." This purpose statement is accompanied by a schematic overview figure that outlines the text pathways that follow.

Theoretical Background:

I didn't realize how much hard-core fundamental theoretical and dynamical meteorology I had forgotten (please forgive me, Joe Pedlosky) until I read through Ch. 2. That said, I think that this chapter as written in several places can come across as a bunch of research fluid dynamicists talking more to themselves than the intended audience. An example would be the discussion surrounding Fig. 2. Several of the panels in Fig. 2 (e.g., panels c and f) are quite obscure and need more context and explanation. In the case of panel c it would be helpful if the location of the snapshot relative to the larger scale flow on a weather map could be indicated for perspective purposes. Likewise, panel (f) supposedly references a convective updraft. Is this updraft located in the warm sector of an extratropical cyclone or elsewhere? Again, more context is needed.

Thank you for telling us that some of the panels in Fig. 2 are not appropriately explained and discussed. We have improved the caption and made sure that each panel is briefly discussed in the text. However, since these schematics are often the synthesis of detailed papers, we cannot become extensive, and the very interested reader should consult the original references.
The schematic in (c) is an idealised version of a vertical cross-section through part of a warm-frontal zone - this information has been clarified in the caption and some additional information added. The convective updraft in (f) is embedded in a WCB - this information has been added to the caption.

Line 196: Check that "Q" was defined previously.
Omega in equation 3 needs to be defined for completeness.
The "M" lines on the Fig. 3c panel are undefined.
Define WCB on line 230 even though we all know what it means.

Q and absolute vorticity were already defined where they first appear (L104 and 108 in the originally submitted version). We think that the comment on the "M" lines refers to Fig. 2c, rather than Fig. 3c. The meaning of "M" (absolute momentum) was already defined in the caption. WCB was already defined where it first appeared (L221 in the originally submitted version).

Lines 233–242: Perhaps a relationship between the magnitude and spatial scale of a PV anomaly and the associated induced tropospheric wind and thermal fields can be referenced quantitatively here? In other words, consider including the Rossby penetration depth and its relation to the static stability in the context of moist processes to add clarity for sections 2.2, 4 and 5.2.1.

This is quite difficult as the quantitative relationship depends strongly on static stability. However, we agree that this is a very important aspect, and we added a new Fig. 3 on PV inversion including Alan Thorpe's figures (as reproduced in Hoskins et al. 1985) and also a schematic on inversion from the Hoskins and James textbook.

Lines 259-262: What does the DSI (dynamical state index) tell us that we don't already know by other means?

The DSI is mainly an alternative view on PV dynamics, which has not been applied a lot and it is therefore not yet fully clear how it adds to the general understanding of diabatic processes. Weber and Nevir (2008) investigated the DSI during the Lothar storm and concluded that "The DSI structures during the intensification phase of 'Lothar' correspond very well with the results of a high-resolution model simulation of the 2 PVU isosurface (Wernli et al., 2002) at this time". They found that intense cyclones are associated with intense dipoles of the DSI. They also compared the monthly DSI over the North Atlantic during a winter season with many rapidly propagating cyclones and another season with a long-lasting block. The DSI diagnostic distinguished well between the contrasting storm track seasons. In the revised version, we add a brief statement in Sect. 2.1: "Weber and Névir (2008) showed that strongly diabatic cyclones are associated with DSI dipoles in the lower troposphere, and that, evaluated on the seasonal timescale, this diagnostic can therefore serve as an alternative measure for storm track activity."

Fig. 3b is not very readable. No point in showing a figure that requires the use of a magnifying glass to read it properly.
Likewise, Fig. 3c needs more explanation to be understood properly.
Define the two dipole axes in Fig. 3e. This figure panel needs more explanation to be understood.

We improved all these panels. For the former Fig. 3b (note that the ordering of the panels has changed and a new figure has been added so that this is now Fig. 4c) we zoom in a bit more and add some colouring; and the former Figs. 3c and 3e (now Fig. 4f and 4e) are now better explained in the caption. We also increased the size of the panels (changing the aspect ratio of the figure from landscape to portrait). However with "Fig. 3e" the reviewer most likely meant Fig. 2e, which shows dipole axes. The two dipole axes in Fig. 2e are now explained in the caption.

Sections 3.3–3.4 are excellent.

Many thanks!

Section 3.5 is also excellent.

Many thanks!

You might want to emphasize that a very important aspect of the Presidents' Day storm of Feb 1979 as discussed by Bosart (1981) was the comparatively low level (~900-hPa) of the ascent maximum associated with coastal front cyclogenesis (see his Fig. 9b and subsequent figures below) and the associated low-level frontogenesis maximum at ~950-hPa (Fig. 10c). Note also the derived kinematic ascent maximum below 800-hPa in Fig. 14. This low-level ascent maximum ensured that strong low-level cyclonic vorticity was being generated along the aforementioned coastal front. Subsequently, the arrival of the upper-level trough fostered impressive rapid surface cyclogenesis (see Fig. 18), given the presence of pre-existing cyclonic

vorticity along the antecedent coastal front. Bottom line: The low-level d(omega)/dp profile was critical for the rapid spin-up of the Presidents' Day cyclone and should be mentioned.

Many thanks for this detailed discussion. We included these important aspects (and a reference to Fig. 9b of Bosart 1981) in the discussion of the President's Day storm in Sect. 3.5.

Section 4.1.1 is excellent overall.

Many thanks.

In section 4.1.1 consider showing and/or referencing a d(omega)/dp for the aforementioned Presidents' Day storm paper to help the reader better understand how the rapid growth of low-level vorticity along the coastal front occurred.

We added an additional sentence about the importance of the low-level maximum in vertical wind in Sect. 3.5 (see above).

Section 4.1.2 is also excellent overall.

Many thanks.

PV Inversion, Trajectories, and Model Simulations (lines 670-677): Excellent discussion of Davis and Emanuel (1991) and Hoskins and Berrisford (1988). From my personal perspective, Hoskins and Berrisford (1988) was an eye-opening paper and a "must read" paper as well.

Thank you, we fully agree about the relevance of the Hoskins and Berrisford paper!

Line 704 and subsequent lines. Nice discussion of the warm-conveyor belt to include the Nieman and Shapiro papers. Other relevant Mel Shapiro papers that you cite include Shapiro (1976), and Keyser and Shapiro (1993). What may not be fully appreciated is that Mel Shapiro obtained all kinds of critical mesoscale data on upper-level fronts via his research flights on NCAR and NOAA aircraft. These aircraft-derived datasets permitted him (and others) to make the calculations on the evolution of PV in upper level fronts and associated PV anomalies that are discussed in this section.

In Sect. 3.3, we already wrote "The study by Shapiro (1976) investigated the mesoscale substructure of upper-level jets based on aircraft observations" to emphasise the importance of these observations. In addition, we now mention in Sect. 4.1.2 that the aircraft data were also essential for investigating the structure of the upper-level fronts: "In total, they evaluated observations from nine flights into this cyclone with four aircraft, and portrayed the exceptional depth of the upstream tropopause fold. Figure 6 in Neiman and Shapiro (1993) showed a fold reaching down to 700 hPa beneath an upper-level jet exceeding 80 m/s, which was instrumental in the rapid deepening of ERICA IOP4."

Good discussion of the important escalator-elevator concept on lines 715-720.

Thanks.

Lines 754–760: Very important insight by Reed and Albright (1986) on how fast symmetric instability can develop in an explosively deepening bomb cyclone.

Thank you, we felt that no changes were needed.

Lines 784–787: Thanks for reminding me of the importance of the Shutts (1990a, 1990b) papers with regard to SCAPE (an open and "shutt" case to make a bad pun).

Thanks, due to your remark the first author learned about the expression "an open-and-shut case"!

Section 4.1.3: Frontal Wave Cyclones

Key point on lines 810-815: Breakthrough in understanding frontal wave cyclones inspired by the Hoskins and Berrisford (1988) diagnostic study on the infamous UK October storm and related papers by Thorpe and Emanuel, 1985; Joly and Thorpe, 1990; Schär and Davies, 1990; and Malardel et al.,1993). These papers (and others) collectively revealed how diabatically produced bands of low-level PV were a prerequisite for the occurrence of low-level frontal wave instability.

We agree; and we think that this point is well covered by our sentence "These studies revealed the essential role of diabatically produced low-level bands of PV or surface $\theta$ as a prerequisite for frontal wave instability to occur." Also, we have noted that the Bosart papers about the President's Day cyclone pointed out the importance of the development of the vorticity strip along the front.

Lines 814-815 make an important point about the "essential role of diabatically produced low-level bands of PV or surface as a prerequisite for frontal wave instability to occur." Highlight this point a bit more?

We changed the reference to Sect. 4.3 (which we had just above this line) to Sect. 4.3.4 (idealised frontal wave simulations) to be more specific.

Lines 847-850: Reference Fig. 9 from Rogers and Bosart (1986), given that this figure shows that the saturation equivalent potential temperature maximizes in excess of 315 K over the cyclone center at the time of lowest SLP? There is also an ~13 K increase in the saturation equivalent potential temperature over the cyclone center in the 24 h period ending 1200 UTC 4 October 1965.

Thanks for this suggestion, we added a reference to Fig. 9 from the mentioned paper with a brief explanation.

Lines 850-855: Discuss the assorted composites of East Coast cyclones from Manobianco (1989a) in a bit more detail?

We considered adding more details in the revised manuscript, but because these composites and their discussion did not specifically address the role of diabatic processes, we decided to not include more details.

Lines 895-900: This text "screams" for a supporting illustrative figure.

Please see our reply to the next comment.

The excellent text on lines 900-920 would benefit from the addition of at least one new trajectory-related figure. WCB discussion on lines 900-966 would benefit from the addition of a couple of new figures.

We included an additional figure (Fig. 9) with several panels supporting the text on lines 895-966 (line numbers from original submission). In particular, Fig. 9 illustrates how different variables evolve along WCB trajectories.

Existing Figs. 7e,f are inadequate. Redo existing Fig. 7 into panels a-d and create a new figure consisting of the old e and f panels from Fig. 7? Check also whether Fig. 7f is adequately referenced in the text.

This figure is now Fig. 9. Maybe it was not clear that the last two panels were referred to as Fig. 7e (there was no Fig. 7f). We have made this clearer by labelling the two panels separately. We also agree that these panels had low quality (side remark: something went wrong with this paper when formatted by the journal; the original figures looked much better). We therefore decided to now use the versions of these figures from the first author's doctoral thesis (which had nicer grey shading).

Section 4.1.5 (Lagrangian view and coherent airstreams) is very well done.

Many thanks.

Cordeira and Bosart (2011; https://journals.ametsoc.org/view/journals/mwre/139/6/2010mwr3537.1.xml?tab_body=pdf) would be an appropriate additional reference for the evolution of PV structure in a tropical cyclone (Grace) that formed via a tropical transition (see Fig. 2, 4, 7, 11, and 12)

Thanks, we included a reference to this paper in Sect. 5.3.5. (formerly Sect. 5.2.6).

Figures:
The text-figure balance needs to be improved. There is a lot of text and comparatively few relevant illustrative figures. It is also unfortunate that few if any color figures are available from the older literature. I would like to suggest that the authors try to grab more relevant figures

from the refereed literature to illustrate key points and bolster key arguments advanced in the text. It might also be appropriate for the authors to construct a few additional schematic figures to help to better illustrate/reinforce key points that they are trying to make in the text. Any new schematic figures should be constructed in color. Can existing black and white figures be digitized and then converted to color images using AI methodologies?

Many thanks for pointing out that the figures are useful, but that their quality needs to be improved and that even more figures might be appropriate. We implemented the following changes to the figures:
- In many figures, we arranged the panels differently such that they become larger.
- For certain black and white figures, we added subtle colouring of key features to enhance readability.
- We included additional figures about trajectories (Fig. 9), about PV inversion (Fig. 3), and for the new subsection on "Consideration of radiative and surface flux related diabatic processes" (Fig. 15).
- We also seriously considered the suggestion to add (newly produced) schematics, but while we liked the idea, we struggled with its realisation. It is very difficult to do justice to the complexity of the theme with a few schematics. However, we added several additional schematics from the literature (one in Fig. 3, two in Fig. 15, two in Fig. 17, and one in Fig. 18), and note that we already emphasised the general value of schematics in L3014 of the original manuscript.

Multi-panel Figure 7 is in desperate need of improvement (or color!). Geography is mostly unreadable in panels b and c. Panel c is "fuzzy". Panels d-f demand improvement to be more readable because in present form they do not do justice to the text.

See above.

Lines 1152-1177: This discussion seems out of place here. I get that you have a separate section entitled: More Systematic Investigations. That said, the distinction seems a bit forced. Personally, I would have welcomed this discussion earlier in the text.

We agree, and integrated Sect. 4.2.2 into 4.2.1. Thanks for this suggestion. We also slightly changed the titles of Sect. 4.2.1 and of the new Sect. 4.2.2 (formerly 4.2.3).

Section 4.2.3: PV perspective on moist vs. dry experiments: I can't decide whether this section works best as a stand-alone section as it is now or whether the findings documented in this section should be integrated into earlier sections.

We agree that your alternative suggestion would work as well. For pragmatic reasons, we leave the structure as is. Also, Sect. 4.2 is relatively long, such that the separation of the content in subsections might help the reader.

I am OK with section 4.3: Idealized numerical simulations of cyclones.

Thank you.

Lines 1305-1318: Discussion of Emanuel (1987) is well done.

Thanks!

Lines 1434-1457: This section, especially beginning with the discussion of Thorncroft and Hoskins (1990) on line 1441, seems especially relevant. That said, I could easily argue that this discussion is out of place and should appear earlier. Although I understand why the authors did what they did, it seems to me that an equally convincing argument can be made to embed this material into an earlier section.

We agree that here (and in many other places), there would be good arguments for rearranging some parts of the text. Here, our intention was to have enough emphasis on idealised frontal wave instability studies, which, in our view, are different from the classical baroclinic wave studies. One reason for this is that the pioneering studies by Eady and Charney set the basis for understanding large-scale baroclinic instability in the mid-20$^{th}$ century, but it took four more decades until similar instability concepts could satisfactorily explain smaller-scale frontal waves. We think that this is best highlighted with the separate section 4.3.4, which comes at the cost that the study by Thorncroft and Hoskins (and related material in this section) may appear a bit disconnected from the earlier parts of Sect. 4.3.

Personally, I find the climate change argument (footnote 15) unconvincing.

We were not aware of the study by Branscome and Gutowski (1992) before writing this review, and we found it fascinating that, technically, they did an idealised sensitivity experiment on the effects of enhanced humidity, but unlike the simulations summarised in Sect. 4.3.2, which were typically motivated by the need to understand the effects of latent heating, Branscome and Gutowski (1992) motivated their study with global warming. Since we found this duality of motivations remarkable, we added this footnote.

General comment after scrutinising sections 2-4. I would like to reinforce what I said earlier. The ratio of text to figures is too high throughout. There are a number of places throughout the text where the inclusion of additional figures would make it easier to follow the discussion. That said, figure quality needs to be significantly improved in many places. I fully appreciate that this presents a problem for old non-digitized black and white figures. A possible compromise might be to separate multi-panel black and white figures that a marginally readable (at best) into individual single or double figures that are larger and possibly easier to read.

See above, where we addressed the helpful comments about how to improve the figures.

Additional Comments by Bosart Ph.D. students: Tyler Leicht and Alexander Mitchell

I've finished my review of the Wernli and Gray review paper. It is an exceptionally well-written paper, covering nearly 150 years of research in great detail. I think this will be a great research

and educational resource for years to come. I only have a few minor comments that I think could improve the paper if considered.

Many thanks for your very positive overall assessment!

In section 2.1, I would like to see a (brief) description of the QG height tendency and omega equations as they relate to diabatic heating. I know the paper mainly focuses on a PV perspective, but in order to argue that PV thinking is best for understanding the impact of diabatic heating on midlatitude weather systems (stated on line 3019), one must first outline the QG perspective and compare the two frameworks. QG theory is alluded to in sections 2.2, 3.4, and 3.5, but the reader does not have an immediate reference in this text to the equations the way they do for the PV fundamentals.

Many thanks, we thought about including a (brief) discussion of QG theory, but we felt that this would make the scope of the paper too large. But we now reference the AMS encyclopaedia article by Davies and Wernli (205) about QG theory in Sect. 1. And when we emphasise the usefulness of the PV concept for understanding the impact of diabatic heating, then this was not meant as "in contrast to QG theory". The PV concept is a core element of QG theory, although for investigating the scales at which diabatic processes are most effective, QG theory is not always of high enough accuracy.

I would also rephrase the sentence ending in line 2965 stating that QG theory is synonymous with dry dynamics, since my view is that QG and PV can both be treated adiabatically and diabatically.

Thanks for pointing this out, we agree. We rephrased this sentence to avoid the misconception that QG theory is synonymous with dry dynamics. The best was to simply omit the reference to QG theory in this sentence.

Otherwise, I think this was a very successful review paper. Let me know what you think of my suggestion, and I'll be curious to see what you and Alex think would improve this paper.

We assume that this is a comment from Tyler to Lance and Alex.

Overview:

The manuscript, "The importance of diabatic processes for the dynamics of synoptic-scale extratropical weather systems—a review" is thorough and offers a comprehensive overview of the history of extratropical cyclone dynamics using a PV framework. The authors demonstrate a strong foundation of synoptic-scale extratropical dynamics and present a clear and accessible synthesis of the historical literature. The inclusion of recent references and an overall balanced discussion add significant value for the reader.

Many thanks!

Lines 233–242: Perhaps a relationship between the magnitude and spatial scale of a PV anomaly and the associated induced tropospheric wind and thermal fields can be referenced quantitatively here? In other words, consider including the Rossby penetration depth and its relation to the static stability in the context of moist processes to add clarity for sections 2.2, 4 and 5.2.1.

See above (this comment appears twice).

Minor Comments:

Many thanks for the careful reading and the many useful suggestions to improve our writing. Much appreciated! We implemented most of these suggestions listed on the following page for Sect. 1-6 (and we therefore don't respond to each of them separately).

Section 1:
Line 98: Should quasi-geostrophic be abbreviated as (QG) throughout the article?
Line 110: Remove hyphen for "life-cycles"
Lines 110-114: Consider splitting these into 2 sentences
Line 116: Revise to "clouds can"
Line 117: Revise to "..Carbone, 2004), and…"
Line 121: Revise to "..up to several 1000 km"..
Line 138: Revise to "..and challenges to stimulate further.."

Section 2:
Line 206: Define $\theta$?
Line 265: Revise to "..subsequent precipitation usually occur through"
Line 301: Revise to "..However, as discussed by Schultz and Schumacher (1999), many.."
Lines 349–355: Consider splitting this into two sentences.
Line 394: Remove the comma after "currents"

Section 3:
Lines 375–379: Insert "(a)", "(c)" and "(d)" as well since (b) is written in the sentence.
Line 402: Revise to "... and examined stability criteria.."
Lines 486–487: Revise to "..at the level of maximum wind speed. The hypothesis was made that these maxima were diabatically."
Lines 489–492: Revise to "..It is interesting to note that the initial research on STE near upper-level fronts and tropopause folds mainly discussed how radiation and turbulence can modify PV, but overlooked the potential effects of latent heat release in clouds. However, this focus changed almost 20 years later.."
Lines 510–512: Omit "In"
Line 526: Revise to "We claim this paradigm shift.."

Section 4:
Line 568: Revise to "..novel data thanks to.."
Line 573: Revise to "..they have served as.."

Line 589: Revise to "..together with the ascent.."
Line 623: Add comma, "In several of the studies, the.."
Line 700: Revise to "..was considered the main reason.."
Line 703: Revise to "..to present an overview of the.."
Line 729: Revise to "..first to quantify.."
Lines 877–878: Revise to "This subsection summarizes research from 1980–2000 on using trajectories to investigate moist airstreams and extratropical cyclone dynamics."

Section 6:
Line 2974: Revise to "..time failed to predict these.."
Line 3012: Add a comma after "for instance"
Line 3024: Add a comma after "More recently"

Lines 3049–3054: Revise to "This improvement will allow for better representation of steep topography and the ability to turn off the parameterization of deep moist convection. In the context of this review article, this implies that, e.g., fast-ascending motion in convective cloud systems and associated diabatic processes will be simulated with the same numerics and cloud microphysics as the more slowly ascending and larger-scale warm conveyor belts. As discussed in Sect. 5.5.2, this change will have direct implications for the diabatic modification of PV in the upper troposphere and, in turn, for the large-scale flow evolution."

Thanks, we revised these sentences as suggested.

Potential additional references (aka shameless plugs) to add to the Wernli and Gray Review Paper. Use anywhere from 0–48 of these suggested references as you see fit.

Many thanks for adding this impressive list of very interesting papers. We looked at all of them and included additional references to a selection of these papers that specifically address diabatic processes (and that we missed including in the first place). These are the studies by Archambault et al. (2010), Bosart et al. (2017), Cordeira and Bosart (2011), Davis and Bosart (2004), and Moore et al. (2013). Several papers in the list are about extratropical transition ( ET) – a topic for which we decided to be comparatively brief because of the recently published excellent two-part review paper about ET written by the ET experts (which we are not). Therefore, we prefer not to add more papers about ET. Thanks for your understanding.

1. Archambault et al. (2010): Relationships between Large-Scale Regime Transitions and Major Cool-Season Precipitation Events in the Northeastern United States
2. Archambault et al. (2013): A Climatological Analysis of the Extratropical Flow Response to Recurving Western North Pacific Tropical Cyclones
3. Archambault et al. (2015): A Composite Perspective of the Extratropical Flow Response to Recurving Western North Pacific Tropical Cyclones
4. Bals-Elsholz et al. (2001): The Wintertime Southern Hemisphere Split Jet: Structure, Variability, and Evolution
5. Bentley et al. (2017): Upper-Tropospheric Precursors to the Formation of Subtropical Cyclones that Undergo Tropical Transition in the North Atlantic Basin

6.  Bentley et al. (2019): A Climatology of Extratropical Cyclones Leading to Extreme Weather Events over Central and Eastern North America

7.  Bell and Bosart (1993): A Case Study Diagnosis of the Formation of an Upper Level Cutoff Cyclonic Circulation over the Eastern United States

8.  Biernat et al. (2023): A Climatological Comparison of the Arctic Environment and Arctic Cyclones between Periods of Low and High Forecast Skill of the Synoptic-Scale Flow

9.  Bosart (1984): The Texas Coastal Rainstorm of 17–21 September 1979: An Example of Synoptic Mesoscale Interaction

10. Bosart and Bartlo (1991): Tropical Storm Formation in a Baroclinic Environment

11. Bosart and Dean (1991): The Agnes Rainstorm of June 1972: Surface Feature Evolution Culminating in Inland Storm Redevelopment

12. Bosart and Lackmann, 1995: Postlandfall Tropical Cyclone Reintensification in a Weakly Baroclinic Environment: A Case Study of Hurricane David (September 1979)

13. Bosart and Sanders (1986): Mesoscale Structure in the Megalopolitan Snowstorm of 11–12 February 1983. Part III: A Large-Amplitude Gravity Wave

14. Bosart and Sanders (1991): An Early-Season Coastal Storm: Conceptual Success and Model Failure

15. Bosart et al. (1996): Large-Scale Antecedent Conditions Associated with the 12–14 March 1993 Cyclone ("Superstorm '93") over Eastern North America

16. Bosart et al. (1998): A Study of Cyclone Mesoscale Structure with Emphasis on a Large-Amplitude Inertia–Gravity Wave

17. Bosart et al. (2000): Environmental Influences on the Rapid Intensification of Hurricane Opal (1995) over the Gulf of Mexico

18. Bosart et al. (2012): An Analysis of Multiple Predecessor Rain Events ahead of Tropical Cyclones Ike and Lowell: 10–15 September 2008

19. Bosart et al. (2017): Interactions of North Pacific Tropical, Midlatitude, and Polar Disturbances Resulting in Linked Extreme Weather Events over North America in October 2007

20. Cordeira and Bosart (2011): Cyclone Interactions and Evolutions during the "Perfect Storms" of Late October and Early November 1991

21. Davis and Bosart (2001): Numerical Simulations of the Genesis of Hurricane Diana (1984). Part I: Control Simulation

22. Davis and Bosart (2003): Baroclinically Induced Tropical Cyclogenesis

23. Davis and Bosart (2004): The TT Problem: Forecasting the Tropical Transition of Cyclones

24. DiMego and Bosart (1982): The Transformation of Tropical Storm Agnes into an Extratropical Cyclone. Part I: The Observed Fields and Vertical Motion Computations

25. DiMego and Bosart (1982): The Transformation of Tropical Storm Agnes into an Extratropical Cyclone. Part II: Moisture, Vorticity and Kinetic Energy Budgets

26. Galarneau et al. (2009): Baroclinic Transition of a Long-Lived Mesoscale Convective Vortex

27. Galarneau et al. (2015): Development of North Atlantic Tropical Disturbances near Upper-Level Potential Vorticity Streamers

28. Griffin and Bosart (2014): The Extratropical Transition of Tropical Cyclone Edisoana (1990)

29. Hakim et al. (1995): The Ohio Valley Wave-Merger Cyclogenesis Event of 25–26 January 1978. Part I: Multiscale Case Study

30. Hakim et al. (1996): The Ohio Valley Wave-Merger Cyclogenesis Event of 25–26 January 1978. Part II: Diagnosis Using Quasigeostrophic Potential Vorticity Inversion

31. Lackmann et al. (1996): Planetary- and Synoptic-Scale Characteristics of Explosive Wintertime Cyclogenesis Over the Western North Atlantic Ocean

32. Lackmann et al. (1997): A Characteristic Life Cycle of Upper-Tropospheric Cyclogenetic Precursors during the Experiment on Rapidly Intensifying Cyclones over the Atlantic (ERICA)

33. McTaggart-Cowan et al. (2006): Analysis of Hurricane Catarina (2004)

34. McTaggart-Cowan et al. (2007): Hurricane Katrina (2005). Part II: Evolution and Hemispheric Impacts of a Diabatically Generated Warm Pool

35. McTaggart-Cowan et al. (2010): Development and Tropical Transition of an Alpine Lee Cyclone. Part I: Case Analysis and Evaluation of Numerical Guidance

36. McTaggart-Cowan et al. (2013): A Global Climatology of Baroclinically Influenced Tropical Cyclogenesis

37. Moore et al. (2013): Synoptic-Scale Environments of Predecessor Rain Events Occurring East of the Rocky Mountains in Association with Atlantic Basin Tropical Cyclones

38. Moore et al. (2019): Linkages between Extreme Precipitation Events in the Central and Eastern United States and Rossby Wave Breaking

39. O'Handley and Bosart (1989): Subsynoptic-Scale Structure in a Major Synoptic-Scale Cyclone

40. Papin et al. (2020): A Feature-Based Approach to Classifying Summertime Potential Vorticity Streamers Linked to Rossby Wave Breaking in the North Atlantic Basin

41. Pyle et al. (2004): A Diagnostic Study of Jet Streaks: Kinematic Signatures and Relationship to Coherent Tropopause Disturbances

42. Röthlisberger et al. (2019): Recurrent Synoptic-Scale Rossby Wave Patterns and Their Effect on the Persistence of Cold and Hot Spells

43. Sanders and Bosart (1985): Mesoscale Structure in the Megalopolitan Snowstorm of 11–12 February 1983. Part I: Frontogenetical Forcing and Symmetric Instability

44. Schultz et al. (1997): The 1993 Superstorm Cold Surge: Frontal Structure, Gap Flow, and Tropical Impact

45. Schultz et al. (1998): The Effect of Large-Scale Flow on Low-Level Frontal Structure and Evolution in Midlatitude Cyclones

46. Winters et al. (2019): The Development of the North Pacific Jet Phase Diagram as an Objective Tool to Monitor the State and Forecast Skill of the Upper-Tropospheric Flow Pattern

47. Winters et al. (2020a): Composite Vertical-Motion Patterns near North American Polar–Subtropical Jet Superposition Events

48. Winters et al. (2020b): Composite Synoptic-Scale Environments Conducive to North American Polar–Subtropical Jet Superposition Events

**Reviewer 6**

Recommendation: minor revisions

This very comprehensive review on the topic of diabatic processes in synoptic extratropical weather systems will become the definitive guide for researchers in this field.

Many thanks for your positive overall assessment of our review!

The review covers the history of the development of the topic over the past century. I was tasked with reviewing the introduction and the second half of the manuscript, which is concerned mostly with developments over the past 20 or so years.

I particularly like the schematic diagram showing the structure of the manuscript and where different weather systems and phenomena can be found. This will be very helpful for readers who wish to dip in and out of the manuscript to navigate the huge wealth of information.

Thanks, we are glad that you explicitly mentioned the usefulness of this schematic.

The structure of the second half of the paper (section 5) is logical and easy to follow, covering different categories of extratropical cyclones, diagnostics and methods of analysing diabatic processes, features ranging from waveguides down to the mesoscale, and finally field experiments and climate change. The figures included are very nicely chosen to illustrate the key concepts explored.

Many thanks.

Another aspect of the paper that I particularly like is the inclusion of details on the contributions from women that have increased in the past 20 or so years. It is heartening to see both the improvement in gender equality in this research area, and also the acknowledgement that there was an improvement to be made in the first place.

We both fully agree ☺

Given the comprehensive nature of the review, and the present quality of the writing, it is difficult to find much to add. So, I only have a few minor comments and suggestions.

Line 1496: "Millennia" -> "millennium".

Changed.

There are a number of uses of American English (where I think the majority of the text is written in British English). I will point out a few of these.

Line 1508: "generalizing".

Line 1631: "categorization".

Line 2236: "discretization".

Line 2322: "visualize".

Line 2506: "realizing".

Page 96: "summarize".

Line 1592: "southern hemisphere" -> "Southern Hemisphere".

Line 1792: "northern hemisphere" -> "Northern Hemisphere".

Line 1796: as points 3 and 4.

Thanks a lot, we implemented all these suggestions.

Line 1899: I don't think it is obvious what "low frequency jet" means.

We added a short explanation: "... position of the low-frequency jet exit, i.e., on the position of the jet exit calculated with low-pass filtered winds."

Line 1983: Perhaps the word "intensifies" is ambiguous here.

Thanks, we change the wording to "PV is diabatically produced ...".

Line 2079: Perhaps the reference for the impermeability theorem could be added here also.

We added a reference to Haynes and McIntyre (1990).

Line 2229: "which is defined as events", I think should be "which are defined as events".

We think that "is" is correct in L2329, but if not, then the copy editing will correct it.

Line 2414: Here I noticed an inconsistency in the spelling of parameterisation/parameterization.

Thanks for spotting this, we now consistently use the British spelling "parametrisation".

Line 2661: It would be helpful to reference the section here with the equation.

Indeed, we add "(Eq. 7, see Sect. 2.1)".

Line 3112: Suggest also citing Martius et al 2016 and Owen et al 2021.

References:

Martius O., Pfahl S., Chevalier C. (2016) A global quantification of compound precipitation and wind extremes, Geophys. Res. Lett., 43 (14), pp. 7709-7717.

Owen LE, Catto JL, Stephenson DB, Dunstone NJ. (2021) Compound precipitation and wind extremes over Europe and their relationship to extratropical cyclones, Weather and Climate Extremes, volume 33, DOI:10.1016/j.wace.2021.100342.

Thanks for suggesting these references. There is a lot of literature about compound extremes and we don't intend to include this topic in a comprehensive way in our paper because most of these studies are of statistical nature and not directly about processes. However, we added a reference to the Owen et al. paper, as it specifically links compound extremes to extratropical cyclones, which appear prominently in our review.

---

## Referee Report (RR1)

Review of egusphere-2023-2678-1

"The importance of diabatic processes for the dynamics of synoptic-scale extratropical weather systems—a review"
by
Heini Wernli and Suzanne L. Gray

**Recommendation: Minor revisions**

**General Comments:**
The authors state that their interpretation "reflects the fact that the balanced flow is determined by a quantity we call PV (and suitable boundary conditions). The omega equation then helps to obtain information about vertical motion." This is ok. My point was that all non-rotating components as well as all rotating components that do not project onto a theta surface are neglected when only using PV. Given that the authors also veer into mesoscale arguments with significant circulations perpendicular to theta surfaces, the caveats and limitations of only using a PV framework could have been elaborated on.

Regearing the authors' response: "When we mention the usefulness of the PV concept, then we don't regard PV in isolation. Rather, e.g., in a QG framework, we mean that PV determines the geostrophic flow, which in turn determines the ageostrophic flow. For the context of this review article, this conceptual framework still holds, with the additional (complicated) element that diabatic processes can create or destroy PV, and in this way modify the flow."

It is not clear what the authors mean by that the "conceptual framework still holds". In general, PV thinking is based on postulating the invertibility principle, i.e., as the authors state, that one makes balance assumptions to revert PV into a purely non-divergent balanced and hydrostatic flow. The original PV, however, was calculated using the full flow field; so depending on the actual state of the atmosphere, the flow field obtained by PV inversion can have significant deviations from the actual flow field. When moving more and more to meso and smaller scales, one should hence be allowed to wonder how justifiable a purely balanced flow assumption is. And regarding diabatic heating; it can neither destroy nor create PV, except if occurring at the boundary. It merely rearranges PV. When "creating" or "destroying" PV in the interior by diabatic heating, these creations and destructions cancel each other; so it can be seen as misleading to refer to as creation and destruction.

Regarding the authors' response to the flaws in the discussed surface pressure tendency. The "extensions" introduced by the authors who originally introduced the diagnostic did certainly not remedy the physical flaw in the diagnostic. A hydrostatic surface pressure cannot be changed directly by diabatic heating, as implied by their diagnostic (see Bannon 1996 and Spengler et al. 2011). It is therefore unfortunate that the authors decided to maintain their reference to this flawed diagnostic.

References:
Bannon, P. (1996): Hydrostatic Adjustment: Lamb's Problem. J. Atmos. Sci., 52, 1743-1752, https://doi.org/10.1175/1520-0469(1995)052<1743:HALP>2.0.CO;2

Spengler, T., J. Egger, and S. T. Garner (2011). How Does Rain Affect Surface Pressure in a One-Dimensional Framework? J. Atmos. Sci., 68, 347-360, https://doi.org/10.1175/2010JAS3582.1

---

## Author Response (AR2)

*Paper egusphere-2023-2678*

**The importance of diabatic processes for the dynamics of synoptic-scale extratropical weather systems—a review**

by Heini Wernli and Suzanne L. Gray

***Replies to the reviewers' comments (2nd round)***

We thank the three reviewers who looked again at our paper and the revisions. Below we address the minor comments from reviewers 4 and 5. The reviewers' comments are in black and our replies in blue.

**Reviewer 4**

Recommendation: Minor revisions

General Comments:
The authors state that their interpretation "reflects the fact that the balanced flow is determined by a quantity we call PV (and suitable boundary conditions). The omega equation then helps to obtain information about vertical motion." This is ok. My point was that all non-rotating components as well as all rotating components that do not project onto a theta surface are neglected when only using PV. Given that the authors also veer into mesoscale arguments with significant circulations perpendicular to theta surfaces, the caveats and limitations of only using a PV framework could have been elaborated on.

We agree that the validity of the balanced flow assumption required for agreement between the flow field obtained from PV inversion and the actual flow field becomes more questionable at the meso- and smaller scales. However, there is evidence to support that the PV concept is still useful at such scales. Davis and Weisman (1994) showed that mesoscale convective vortices evolving from mesoscale convective systems (with horizontal scales of 100-200 km) are nearly balanced, although their formation depends on unbalanced motions. Weijenborg et al. (2017) concluded that the statistically significant flow anomalies associated with PV anomalies resulting from cells of summertime deep moist convection imply that the PV dipoles might be invertible in a statistical way and discussed possible routes to inverting PV at the convective-weather scale. This study, and those of Weijenborg et al. (2015) and Chagnon and Gray (2009), also found that PV dipoles can have longer lifetimes than the convective updraught that initiated them, increasing the likelihood that balanced circulations exist. Finally, individual mesoscale PV anomalies can aggregate to form larger anomalies that are associated with coherent larger-scale horizontal circulation anomalies, implying the qualitative validity of PV inversion at this scale. For example, Oertel et al. (2020) showed how the upper-tropospheric PV anomalies associated with embedded convection in a WCB

aggregate to form elongated PV dipole bands. To better discuss this important aspect about the validity of PV inversion at the mesoscale, we added a slightly shortened version of this reply to Sect. 2 (new L252-263).

Regarding the authors' response: "When we mention the usefulness of the PV concept, then we don't regard PV in isolation. Rather, e.g., in a QG framework, we mean that PV determines the geostrophic flow, which in turn determines the ageostrophic flow. For the context of this review article, this conceptual framework still holds, with the additional (complicated) element that diabatic processes can create or destroy PV, and in this way modify the flow." It is not clear what the authors mean by that the "conceptual framework still holds". In general, PV thinking is based on postulating the invertibility principle, i.e., as the authors state, that one makes balance assumptions to revert PV into a purely non-divergent balanced and hydrostatic flow. The original PV, however, was calculated using the full flow field; so, depending on the actual state of the atmosphere, the flow field obtained by PV inversion can have significant deviations from the actual flow field. When moving more and more to meso and smaller scales, one should hence be allowed to wonder how justifiable a purely balanced flow assumption is. …

We think that this comment addresses the same issue as the first comment, namely that balance assumptions inherent in the conceptual PV framework are admittedly more questionable at the mesoscale than at synoptic scales. See our response to the first comment.

… And regarding diabatic heating; it can neither destroy nor create PV, except if occurring at the boundary. It merely rearranges PV. When "creating" or "destroying" PV in the interior by diabatic heating, these creations and destructions cancel each other; so, it can be seen as misleading to refer to as creation and destruction.

Here we disagree with the reviewer, or, more precisely, we prefer using a different terminology. As discussed in the Haynes and McIntyre papers, positive and negative PV tendencies cancel each other if integrated over the entire atmosphere (disregarding processes at the boundaries). However, the PV of air parcels can change, which we refer to as material PV production or destruction. As discussed at length in our paper, the fact that material PV production happens typically at the bottom of a region of intense diabatic heating, and material PV destruction at the top, has a profound influence on the evolution of cyclones and upper-level Rossby waves. The fact that these tendencies cancel each other if integrated over the entire domain does not imply that diabatic heating has no effect on the atmospheric flow and/or that this effect cannot be investigated with the aid of PV.

Regarding the authors' response to the flaws in the discussed surface pressure tendency. The "extensions" introduced by the authors who originally introduced the diagnostic did certainly not remedy the physical flaw in the diagnostic. A hydrostatic surface pressure cannot be changed directly by diabatic heating, as implied by their diagnostic (see Bannon 1996 and Spengler et al. 2011). It is therefore unfortunate that the authors decided to maintain their reference to this flawed diagnostic.

References:

Bannon, P. (1996): Hydrostatic Adjustment: Lamb's Problem. J. Atmos. Sci., 52, 1743-1752,

Spengler, T., J. Egger, and S. T. Garner (2011). How Does Rain Affect Surface Pressure in a One-Dimensional Framework? J. Atmos. Sci., 68, 347-360, https://doi.org/10.1175/2010JAS3582.1

We recognise the continued controversy regarding the use of this surface pressure tendency diagnostic. However, as this is a review article it is important to document the use of this diagnostic. We also note that the reviewer has not explained why the extension introduced in Fink et al., (2012) fails to address the physical flaw in the diagnostic described in Spengler and Egger (2009). Specifically, following the comment on their earlier work by Spengler and Egger (2009), in Fink et al. (2012) the authors introduced an extended form of the surface pressure tendency that includes a term for the changes in the geopotential at the upper boundary of the column, $\rho_{sfc}\frac{\partial \phi_{p_2}}{\partial t}$, (in addition to other changes). Sensitivity tests yielded a value of 100 hPa for the upper integration boundary, $p_2$, as integrals were found to remain nearly constant for upper integration boundaries beyond the tropopause. Figure S1, and the associated text, in their appendix showed that the authors recognise that diabatic heating can only modify the hydrostatic surface pressure if mass is removed from the column – as explained in Spengler et al. (2011) – and that this new term has been included as a consequence. When applied to the five explosively deepening extratropical storms considered in the study by Fink et al. (2012), this new term was found to be significant for a small number of time steps.

As a consequence of the reviewer's continued concern about the surface pressure tendency diagnostic, L2503 (in the last version; now L2526) has been edited to provide more information through the addition of a footnote:

"The surface pressure tendency equation as formulated by Knippertz and Fink (2008) and Knippertz et al. (2009) was extended, in response to Spengler and Egger (2009) [Footnote], and used to quantify the contribution of diabatic processes to extratropical cyclone development by Fink et al. (2012) …

Footnote: This extension included the addition of a term for changes in the geopotential at the top of the considered atmospheric column (set to 100 hPa) to recognise that while heating can result in the adjustment of pressure and density profiles in an atmospheric column, reduction in hydrostatic surface pressure can only be caused by net mass removal; see also Spengler et al. (2011)."

Recommendation: Minor revision

Outstanding paper on a decadal time scale. The scope of the paper prompted me to revisit some of the earlier re3search that i did with Chris Davis (and others) on tropical transition. Appended are comments and additional references on tropical transition for consideration by the authors ...

I reread sections 1-4 of the Wernli and Gray manuscript (and I skimmed the remaining chapter) as you requested. Appended is my second review (I had trouble navigating some of the chapters due to user error at my end which made it difficult for me to insert comments and suggestions). Appended is my additional "old-school" low-tech review. Wernli and Gray have produced an excellent and comprehensive overview paper on the dynamics of extratropical weather systems that will be a "must read" document once it is published. One area that could benefit from some additional discussion is the problem of tropical transition (TT). I have appended some additional material on the TT problem for consideration by the authors and the editors. Thank you.

Lance Bosart

We are most grateful to the reviewer for the nice words and for suggesting several papers to potentially include in the review. Below, we explain for each of them either where we included the reference or why we decided to not include it.

Some possible additional Tropical Transition references for possible inclusion in the Wernli and Gray review paper:

1. Bosart, L. F. and J. A. Bartlo, 1991: Tropical Storm Formation in a Baroclinic Environment: DOI: https://doi.org/10.1175/1520-0493(1991)119<1979:TSFIAB>2.0.CO;2

This was the very first paper on tropical transition (TT) to my knowledge. Only one problem: I did not propose a "catchy name" for this phenomenon. That would not come until 13 years later when Chris Davis and I published a paper in the AMS Bulletin of the American Meteorological Society (BAMS) entitled: "The TT Problem: Forecasting the Tropical Transition of Cyclones": (reference #2 below)

2. Christopher A. Davis and Lance F. Bosart (2004): The TT Problem: Forecasting the Tropical Transition of Cyclones: Published Online: 01 Nov 2004, Bull. American Meteor. Soc., DOI:https://doi.org/10.1175/BAMS-85-11-1657

The paper by Davis and Bosart (2004) was already included in the last version on L1930 (now L1964). The paper by Bosart and Bartlo (1991) is now included in the new paragraph about the role of diabatic processes for tropical transition in Sect. 5.3.2 (L1965).

Backstory: I was at NCAR the previous summer (2003). Chris Davis showed me a numerical simulation of a Gulf of Mexico tropical cyclone (TC) that left a lot to be desired (in more ways than one). I remembered my paper with Bartlo from 1991 on TC Diana (1984). I showed Chris how TC Diana formed from a baroclinic system and suggested that he run MM4 on TC Diana (1984). We got an excellent simulation … which prompted us to write our 2004 BAMS paper referenced above. Our full science paper on TC Diana was published one year earlier and can be found here:

Davis, C. A. and L. F. Bosart (2003): Baroclinically Induced Tropical Cyclogenesis; DOI: https://doi.org/10.1175/1520-0493(2003)131<2730:BITC>2.0.CO;2

It seems that the detailed paper about TC Diana is the paper by Davis and Bosart (2001): Numerical simulations of the genesis of Hurricane Diana. Part I: Control simulation. Mon. Wea. Rev., 129, 1859–1881. We included this paper in the new paragraph (L1967). The study by Davis and Bosart (2003) was also worth including (L1970), as it discusses the important role of diabatic processes for reducing vertical shear during tropical transition.

3. Galarneau, T. J., L. F. Bosart, C. A. Davis, and R. McTaggart-Cowan (2009): Baroclinic Transition of a Long-Lived Mesoscale Convective Vortex

We regard MCV dynamics as beyond the scope of this paper and therefore did not reference this study.

4. Bosart, L. F., and G. M. Lackmann, 1995: Post landfall tropical cyclone reintensification in a weakly baroclinic environment: A case study of Hurricane David (September 1979). Mon. Wea. Rev., 123, 3268–3291. DOI: https://doi.org/10.1175/1520-0493(1995)123<3268:PTCRIA>2.0.CO;2

As explained in our previous reply document, we decided to be comparatively brief about the topic of extratropical transition (ET) because of the recently published excellent two-part review paper about ET written by the experts (which we are not). Therefore, we prefer not to add more papers about ET. Thanks for your understanding.

5. McTaggart-Cowan et al., 2010: Development and Tropical Transition of an Alpine Lee Cyclone. Part I: Case Analysis and Evaluation of Numerical Guidance, Mon. Wea. Rev., DOI: https://doi.org/10.1175/2009MWR3147.1

We included a reference to this paper in L1974.

6. McTaggart-Cowan et al. (2006): Analysis of Hurricane Catarina (2004): DOI: https://doi.org/10.1175/MWR3330.1

Backstory: Initially, we met a lot of resistance from some folks in Brazil about whether Catarina was a legitimate TC due to domestic political reasons (e.g., the unspoken implication that the Brazilian meteorological service was asleep at the wheel). Fortunately for us, the

science … and the MWR reviewers of our paper … were on our side (if it looks like a duck, quacks like a duck, then it is a duck) and our paper sailed through the review process.

This is certainly a very interesting event and study, but since it does not focus particularly on the role of diabatic processes, we decided to not include it in our paper.

7. McTaggart-Cowan et al., 2007: Hurricane Katrina (2005). Part II: Evolution and Hemispheric Impacts of a Diabatically Generated Warm Pool. DOI: https://doi.org/10.1175/2007MWR1875.1

See response about ET studies above.

*Additional suggestion sent to the first author via Email:*

I left off a relevant tropical transition reference in my second review of your excellent paper with Gray; see:

Bentley et al. (2017): A Dynamically Based Climatology of Subtropical Cyclones that Undergo Tropical Transition in the North Atlantic Basin. DOI: https://doi.org/10.1175/MWR-D-15-0251.1

This paper appeared in 2016 and was already included in the previous version on L1939 (now L1951).